# On the Convergence of Adam-Type Algorithms for Bilevel Optimization under Unbounded Smoothness

## Abstract

Adam has become one of the most popular optimizers for training modern deep neural networks, such as transformers. However, its applicability is largely restricted to single-level optimization problems. In this paper, we aim to extend vanilla Adam to tackle bilevel optimization problems, which have important applications in machine learning, such as meta-learning. In particular, we study stochastic bilevel optimization problems where the lower-level function is strongly convex and the upper-level objective is nonconvex with potentially unbounded smoothness. This unbounded smooth objective function covers a broad class of neural networks, including transformers, which may exhibit non-Lipschitz gradients. In this work, we first introduce AdamBO, a single-loop Adam-type method that achieves $\widetilde{O}(\epsilon^{-4})$ oracle complexity to find $\epsilon$-stationary points, where the oracle calls involve stochastic gradient or Hessian/Jacobian-vector product evaluations. The key to our analysis is a novel randomness decoupling lemma that provides refined control over the lower-level variable. Additionally, we propose VR-AdamBO, a variance-reduced version with an improved oracle complexity of $\widetilde{O}(\epsilon^{-3})$. The improved analysis is based on a novel stopping time approach and a careful treatment of the lower-level error. We conduct extensive experiments on various machine learning tasks involving bilevel formulations with recurrent neural networks (RNNs) and transformers, demonstrating the effectiveness of our proposed Adam-type algorithms.

## 1 Introduction

The Adam algorithm (Kingma & Ba, 2014) is one of the most popular optimizers for training modern deep neural networks due to their computational efficiency and minimal need for hyperparameter tuning. For example, Adam has become the default choice for training transformers (Vaswani et al., 2017; Devlin et al., 2018) and vision transformers (ViT) (Dosovitskiy et al., 2021). Practitioners favor Adam and adaptive gradient methods in general because they significantly outperform stochastic gradient descent (SGD) for certain models, such as transformers (Zhang et al., 2019; Crawshaw et al., 2022; Kunstner et al., 2023; Ahn et al., 2023). Recently, there is a line of work analyzing the convergence of Adam under various assumptions (Guo et al., 2021b; Défossez et al., 2020; Wang et al., 2022; Zhang et al., 2022; Li et al., 2023a).

Despite the empirical and theoretical advances of Adam, it is only applicable for single-level optimization problems such as the empirical risk minimization. However, there is a huge class of machine learning problems which are inherently bilevel optimization problems (Bracken & McGill, 1973; Dempe, 2002), including meta-learning (Franceschi et al., 2018; Rajeswaran et al., 2019), reinforcement learning (Konda & Tsitsiklis, 2000), hyperparameter optimization (Franceschi et al., 2018; Feurer & Hutter, 2019) and continual learning (Borsos et al., 2020; Hao et al., 2023). Therefore, an important question arises: **How can we extend the applicability of vanilla Adam to solve bilevel optimization problems, while ensuring both provable theoretical convergence guarantees and strong empirical performance for machine learning applications?**

In this paper, we provide a positive answer to this question, under the setting of bilevel optimization under unbounded smoothness (Hao et al., 2024; Gong et al., 2024a). In particular, the bilevel optimization in this setting has the following form:

$$\min_{x \in \mathbb{R}^{d_x}} \Phi(x) := f(x, y^*(x)), \quad \text{s.t.,} \quad y^*(x) \in \arg\min_{y \in \mathbb{R}^{d_y}} g(x, y), \tag{1}$$

where $f$ and $g$ are upper- and lower-level functions respectively, and $f$ satisfies a unbounded smoothness condition (see Definition 3.1) and $g$ is a strongly-convex function in $y$. One example satisfying this particular setting is meta-learning (Finn et al., 2017; Franceschi et al., 2018) with certain machine learning models such as RNNs (Elman, 1990) or transformers (Vaswani et al., 2017), where $x$ represents all layers except for the prediction head, $y$ represents the prediction head, and the goal is to learn the shared model parameter $x$ to find a common representation such that it can quickly adapt to various tasks by simply adjusting the task-specific prediction head $y$. The unbounded smoothness condition for the upper-level function $f$ is particularly relevant in this paper for two main reasons. First, recent studies have demonstrated that the gradient's Lipschitz constant (i.e., the smoothness constant) is unbounded in various modern neural networks, including RNNs and transformers (Zhang et al., 2020b; Crawshaw et al., 2022; Hao et al., 2024). Second, Adam is empirically successful on training these neural networks (Vaswani et al., 2017; Kunstner et al., 2023) and its convergence under unbounded smoothness was recently proved within the single-level optimization framework (Li et al., 2023a). Therefore it is natural and imperative to design new Adam-type algorithms, building on the vanilla Adam approach, to solve bilevel optimization problems in the unbounded smoothness setting.

We introduce two Adam-type algorithms for such bilevel optimization problems with provable convergence guarantees. The first algorithm is called Adam for Bilevel Optimization (AdamBO). AdamBO begins by running a few iterations of SGD to warm-start the lower-level variable, after which it simultaneously applies vanilla Adam updates to the upper-level variable and SGD updates to the lower-level variable. The primary challenge for the convergence analysis of AdamBO is tackling the complicated dependency between the upper-level hypergradient bias and the lower-level estimation error when the upper-level performs the vanilla Adam update. The convergence analysis of AdamBO for unbounded smooth upper-level functions builds upon the insight of regarding bilevel optimization as a stochastic optimization problem under distributional drift (Gong et al., 2024a), but with a few important differences. First, our analysis incorporates a novel randomness decoupling lemma for lower-level error control, which arises from using Adam updates for the upper-level variable. Second, unlike (Hao et al., 2024; Gong et al., 2024a), the lower-level error in our setting is not necessarily small across iterations, requiring a more refined analysis to handle the hypergradient bias and establish convergence guarantees. In addition, we also introduce another Adam-type algorithm, namely VR-AdamBO, by incorporating the variance reduction techniques (Cutkosky & Orabona, 2019) along with Adam for the upper-level variable and the lower-level acceleration techniques (Gong et al., 2024b) to further improve the convergence rate. The analysis for VR-AdamBO relies on a novel stopping-time analysis in the context of bilevel optimization and a careful treatment for the lower-level error, which is different from the techniques used in single-level variance-reduced Adam (Li et al., 2023a). Our main contributions are summarized as follows.

- We design a variant of Adam, called AdamBO, for solving bilevel optimization problems under the unbounded smoothness setting. We prove that AdamBO converges to $\epsilon$-stationary points with $\widetilde{O}(\epsilon^{-4})$ oracle complexity. To achieve this result, we develop a novel randomness decoupling lemma for lower-level error control and a refined analysis for the hypergradient bias, which are of independent interest and could be applied to analyzing the convergence of other adaptive optimizers in bilevel optimization.

- We propose a variance-reduced variant of AdamBO, named VR-AdamBO, with an improved oracle complexity of $\widetilde{O}(\epsilon^{-3})$. The proof relies on a novel stopping time analysis in the context of bilevel optimization and a careful treatment for the lower-level error.

- We conduct experiments on deep AUC maximization and meta-learning for text classification tasks with RNNs and transformers to verify the effectiveness of the proposed Adam-type algorithms. We show that both AdamBO and VR-AdamBO consistently outperform other bilevel algorithms during the training process. Notably, for the transformer model, they improve the training (testing) AUC by at least 14% (7%) over other baselines. The running time results indicate that our algorithms converge much faster than baselines.

## 2 RELATED WORK

**Convergence Analysis of Adam.** Adam was proposed by (Kingma & Ba, 2014) and the convergence guarantee was established under the framework of online convex optimization. Reddi et al. (2019) identified a divergence example of Adam under fixed hyperparameters and designed new

variants to fix the divergence issue of Adam. Recently, there is a line of work analyzing the convergence of Adam under various assumptions and problem-dependent hyperparameter choices (Zhou et al., 2018; Guo et al., 2021b; Défossez et al., 2020; Wang et al., 2022; Zhang et al., 2022; Li et al., 2023a). The most related work to our paper is (Li et al., 2023a), which studied the convergence of Adam under relaxed assumptions (i.e., generalized smoothness as defined by (Li et al., 2023a)). However, all of these works only consider Adam within the single-level optimization framework and are not applicable for bilevel optimization problems.

**Bilevel Optimization.** Bilevel optimization was extensively studied in the literature, most of which focus on asymptotic convergence guarantees (Bracken & McGill, 1973; Vicente et al., 1994; Anandalingam & White, 1990; White & Anandalingam, 1993). Ghadimi & Wang (2018) studied bilevel optimization algorithms with non-asymptotic convergence guarantees when the lower-level function is strongly convex. The complexity results were later improved by a series of work (Hong et al., 2023; Ji et al., 2021; Chen et al., 2021; Dagréou et al., 2022; Kwon et al., 2023; Chen et al., 2023a). When each realization of the functions has a Lipschitz stochastic gradient, several works incorporate momentum-based variance reduction techniques (Cutkosky & Orabona, 2019) to further improve the convergence rate (Khanduri et al., 2021; Guo et al., 2021a; Yang et al., 2021). Recently, (Hao et al., 2024; Gong et al., 2024a;b) considered bilevel optimization with unbounded smoothness for the upper-level function and designed stochastic algorithms with convergence guarantees. However, none of these works use the Adam update under the bilevel optimization setting.

**Relaxed Smoothness.** Zhang et al. (2020b) initiated the convergence analysis of the gradient clipping algorithms under the relaxed smoothness condition, which was motivated by the loss landscape of RNNs and LSTMs. The work of (Zhang et al., 2020b) inspired a line of work focusing on designing various algorithms under the relaxed smoothness condition (Zhang et al., 2020a; Jin et al., 2021; Liu et al., 2022; Crawshaw et al., 2023a;b; Faw et al., 2023; Wang et al., 2023; Li et al., 2023a;b), some of them achieved improved convergence rates (Liu et al., 2023; Reisizadeh et al., 2023; Li et al., 2023a). Several variants of relaxed smoothness were considered in (Crawshaw et al., 2022; Chen et al., 2023b; Hao et al., 2024; Gong et al., 2024a;b). This work considered the same problem setting as in (Hao et al., 2024; Gong et al., 2024a;b), focusing on designing Adam-type algorithms for bilevel optimization with unbounded smooth upper-level functions.

## 3 PRELIMINARIES, NOTATIONS AND PROBLEM SETUP

Denote $\langle \cdot, \cdot \rangle$ and $\| \cdot \|$ as the inner product and Euclidean norm of a vector or spectral norm of a matrix. For any vectors $x$ and $y$, denote $x^2, \sqrt{x}, |x|, x \odot y, x/y$ as the coordinate-wise square, square root, absolute value, product and quotient, respectively. We write $x \preceq y$ to denote the coordinate-wise inequality between $x$ and $y$. We use $\widetilde{O}(\cdot), \widetilde{\Theta}(\cdot), \widetilde{\Omega}(\cdot)$ to denote asymptotic notations that hide polylogarithmic factors of $1/\epsilon$. Define $f, g : \mathbb{R}^{d_x} \times \mathbb{R}^{d_y} \to \mathbb{R}$ as the upper- and lower-level functions, where $f(x, y) = \mathbb{E}_{\xi \sim \mathcal{D}_f}[F(x, y; \xi)]$ and $g(x, y) = \mathbb{E}_{\zeta \sim \mathcal{D}_g}[G(x, y; \zeta)]$, with $\mathcal{D}_f$ and $\mathcal{D}_g$ being the underlying data distributions, respectively. When the lower-level function is strongly convex, the hypergradient has the following form (Ghadimi & Wang, 2018):

$$\nabla \Phi(x) = \nabla_x f(x, y^*(x)) - \nabla_{xy}^2 g(x, y^*(x))[\nabla_{yy}^2 g(x, y^*(x))]^{-1} \nabla_y f(x, y^*(x)).$$

The goal of this paper is to design Adam-type algorithms that can find $\epsilon$-stationary points of function $\Phi$ (i.e., finding an $x$ such that $\|\nabla \Phi(x)\| \leq \epsilon$). For a given $(x, y)$, we estimate the hypergradient $\nabla \Phi(x)$ using Neumann series approach (Ghadimi & Wang, 2018) with the following formulation:

$$\hat{\nabla} \phi(x, y; \bar{\xi}) = \nabla_x F(x, y; \xi) - \nabla_{xy}^2 G(x, y; \zeta^{(0)}) \left[ \frac{1}{l_{g,1}} \sum_{q=0}^{Q-1} \prod_{j=1}^{q} \left( I - \frac{\nabla_{yy}^2 G(x, y; \zeta^{(q,j)})}{l_{g,1}} \right) \right] \nabla_y F(x, y; \xi),$$

where $\bar{\xi} := \{\xi, \zeta^{(0)}, \bar{\zeta}^{(0)}, \ldots, \bar{\zeta}^{(Q-1)}\}$ and $\bar{\zeta}^{(q)} := \{\zeta^{(q,1)}, \ldots, \zeta^{(q,q)}\}$ for $q \geq 0$.

Now we start to state the main assumptions for our analysis.

**Definition 3.1** $((L_{x,0}, L_{x,1}, L_{y,0}, L_{y,1})$-smoothness (Hao et al., 2024, Assumption 1))**. *Let $z = (x, y)$ and $z' = (x', y')$, there exists $L_{x,0}, L_{x,1}, L_{y,0}, L_{y,1} > 0$ such that for all $z, z'$, if $\|z - z'\| \leq 1/\sqrt{L_{x,1}^2 + L_{y,1}^2}$, then $\|\nabla_x f(z) - \nabla_x f(z')\| \leq (L_{x,0} + L_{x,1}\|\nabla_x f(z)\|)\|z - z'\|$ and $\|\nabla_y f(z) - \nabla_y f(z')\| \leq (L_{y,0} + L_{y,1}\|\nabla_y f(z)\|)\|z - z'\|$.*

**Remark**: This definition characterizes the unbounded smoothness of the upper-level function $f$ and has also been used in previous works (Hao et al., 2024; Gong et al., 2024a;b). It can be regarded

as a generalization of the relaxed smooth assumption in (Zhang et al., 2020b) and the coordinate-wise relaxed smoothness assumption in (Crawshaw et al., 2022). Moreover, it has been empirically verified for bilevel formulations with RNNs (Hao et al., 2024).

**Assumption 3.2.** *Suppose the followings hold for functions $f$ and $g$: (i) $f$ is continuously differentiable and $(L_{x,0}, L_{x,1}, L_{y,0}, L_{y,1})$-smooth in $(x, y)$; (ii) For every $x$, $\|\nabla_y f(x, y^*(x))\| \leq l_{f,0}$; (iii) For every $x$, $g(x, y)$ is $\mu$-strongly convex in $y$ for $\mu > 0$; (iv) $g$ is continuously differentiable and $l_{g,1}$-smooth jointly in $(x, y)$; (v) $g$ is twice continuously differentiable, and $\nabla_{xy}^2 g$, $\nabla_{yy}^2 g$ are $l_{g,2}$-Lipschitz jointly in $(x, y)$; (vi) Objective function $\Phi$ is bounded from below by $\Phi^*$.*

**Remark:** Assumption 3.2 is standard in the bilevel optimization literature (Kwon et al., 2023; Ghadimi & Wang, 2018; Hao et al., 2024). Under this assumption, the objective function $\Phi$ is $(L_0, L_1)$-smooth, see Lemma B.10 in Appendix B for definitions of $L_0, L_1$ and more details.

**Assumption 3.3.** *Suppose the following stochastic estimators are unbiased and satisfy: (i) $\|\nabla_x F(x, y; \xi) - \nabla_x f(x, y)\| \leq \sigma_f$; (ii) $\|\nabla_y F(x, y; \xi) - \nabla_y f(x, y)\| \leq \sigma_f$; (iii) $\Pr(\|\nabla_y G(x, y; \zeta) - \nabla_y g(x, y)\| \geq s) \leq 2 \exp(-2s^2/\sigma_{g,1}^2)$; (iv) $\|\nabla_{xy}^2 G(x, y; \zeta) - \nabla_{xy}^2 g(x, y)\| \leq \sigma_{g,2}$; (v) $\|\nabla_{yy}^2 G(x, y; \zeta) - \nabla_{yy}^2 g(x, y)\| \leq \sigma_{g,2}$.*

**Remark:** Assumption 3.3 assumes the noise in the stochastic gradient and Hessian/Jacobian is almost-surely bounded or light-tailed. This is an standard assumption in the literature of optimization for single-level relaxed smooth functions (Zhang et al., 2020b;a), as well as for bilevel optimization under unbounded smooth upper-level functions (Hao et al., 2024; Gong et al., 2024a;b).

**Assumption 3.4.** *(i) Let $z = (x, y)$ and $z' = (x', y')$, if $\|z - z'\| \leq 1/\sqrt{L_{x,1}^2 + L_{y,1}^2}$, then for every $\xi$, $\|\nabla_y F(z; \xi) - \nabla_y F(z'; \xi)\| \leq (L_{y,0} + L_{y,1}\|\nabla_y f(z)\|)\|z - z'\|$; (ii) For every $\xi$ and $\zeta$, $G(x, y; \zeta)$ satisfy Assumption 3.2 (iv) and (v).*

**Remark**: Assumption 3.4 (i) requires that certain properties of the second argument (i.e., the lower-level variable $y$) in the upper-level function at the population level also hold almost surely for each random realization. Assumption 3.4 (ii) requires each random realization of the lower-level function satisfies the same property as in the population level. Similar assumptions were made implicitly in the bilevel optimization literature (Ghadimi & Wang, 2018). Note that this assumption does not assume any properties in terms of the upper-level variable $x$ under each random realization.

**Assumption 3.5.** $F(x, y; \xi)$ and $G(x, y; \zeta)$ satisfy Assumption 3.2 for every $\xi$ and $\zeta$ almost surely.

**Remark**: Assumption 3.5 poses a strictly stronger requirement than Assumption 3.4: it assumes each random realization for both upper-level and lower-level function has the same property as in the population level. This assumption has been shown to be necessary to obtain the improved oracle complexity $O(\epsilon^{-3})$ for both single-level problems (Arjevani et al., 2023; Cutkosky & Orabona, 2019) and bilevel problems (Khanduri et al., 2021; Yang et al., 2021; Gong et al., 2024b).

# 4 ADAMBO AND CONVERGENCE ANALYSIS

## 4.1 ALGORITHM DESIGN, MAIN CHALLENGES, AND TECHNIQUE OVERVIEW

**Algorithm Design.** Our first Adam-type algorithm AdamBO is presented in Algorithm 1. It consists of the following components. First, the algorithm requires several warm-start steps for updating the lower-level variable $y$ for a given initialization of the upper-level variable $x_0$ (line 2), which is designed to obtain a good estimate of the optimal lower-level variable at the very beginning and shares the same spirit of the bilevel algorithms introduced in (Hao et al., 2024; Gong et al., 2024a;b). Second, the algorithm updates both the upper- and lower-level variables simultaneously: the lower-level variable $y$ is updated by SGD, and the upper-level variable $x$ is updated by the vanilla Adam algorithm (lines $3 \sim 9$). Therefore, the upper-level update benefits from the coordinate-wise adaptive learning rate. In contrast, the existing bilevel optimization algorithms under the unbounded smoothness setting use normalized SGD with momentum to update the upper-level variable (Hao et al., 2024; Gong et al., 2024a;b), which use a universal learning rate for every coordinate.

**Main Challenges.** The main challenges for the convergence analysis of AdamBO are listed as follows. First, the analysis of vanilla Adam in the single-level generalized smooth optimization setting (Li et al., 2023a) is not directly applicable for bilevel problems. This is because the hypergradient estimator in bilevel optimization may have a non-negligible bias due to inaccurate estimation

---

**Algorithm 1** ADAMBO (all operations on vectors are element-wise)

1: **Input:** $\beta, \beta_{\mathrm{sq}}, \eta, \gamma, \lambda, T_0, T, x_1, y_0$
2: **Initialize** $y_1 = \mathrm{SGD}(x_1, y_0, \gamma, T_0)$, $\hat{m}_1 = \hat{\nabla}\phi(x_1, y_1; \bar{\xi}_1)$ and $\hat{v}_1 = (\hat{\nabla}\phi(x_1, y_1; \bar{\xi}_1))^2$
3: **for** $t = 1, \ldots, T$ **do**
4:     $y_{t+1} = y_t - \gamma \nabla_y G(x_t, y_t; \zeta_t)$
5:     $m_t = (1 - \beta)m_{t-1} + \beta\hat{\nabla}\phi(x_t, y_t; \bar{\xi}_t)$
6:     $v_t = (1 - \beta_{\mathrm{sq}})v_{t-1} + \beta_{\mathrm{sq}}(\hat{\nabla}\phi(x_t, y_t; \bar{\xi}_t))^2$
7:     $\hat{m}_t = \frac{m_t}{1-(1-\beta)^t}$
8:     $\hat{v}_t = \frac{v_t}{1-(1-\beta_{\mathrm{sq}})^t}$
9:     $x_{t+1} = x_t - \frac{\eta}{\sqrt{\hat{v}_t}+\lambda} \odot \hat{m}_t$
10: **end for**

---

of the lower-level variable, whereas the single-level analysis in (Li et al., 2023a) does not need to account for this issue. Second, the existing algorithms and analyses for bilevel optimization with unbounded smooth upper-level functions require the lower-level error to be small (Hao et al., 2024; Gong et al., 2024a;b), which may not hold for AdamBO. In particular, the existing analysis crucially relies on a fixed update length for the upper-level variable at every iteration (due to normalization): the analysis in (Hao et al., 2024; Gong et al., 2024a;b) views the update of the upper-level variable as a fixed distributional drift for the lower-level function, which is crucial to show that the lower-level error is small and the hypergradient bias is negligible. However, such an argument is not true for AdamBO: the Adam update for the lower-level variable does not have a fixed update size and it depends on randomness from both upper-level and lower-level random variables in the stochastic setting, which make the lower-level error control more challenging.

**Technique Overview.** To address these challenges, one of our main technical contributions is the introduction of a novel randomness decoupling lemma for controlling the lower-level error when the upper-level variable is updated by Adam, as illustrated in Section 4.3.2. This lemma provide a high probability guarantee for the lower-level error control when the upper-level update rule satisfies certain conditions (which are satisfied by the vanilla Adam update rule for the upper-level variable). The key novelty of this lemma lies in the randomness-decoupling fact: the high-probability bound depends solely on the randomness $\{\zeta_t\}_{t=1}^T$ from the lower-level random variables, and it holds for any fixed sequence of upper-level variables $\{x_t\}_{t=1}^T$ and any fixed upper-level random variables $\{\bar{\xi}_t\}_{t=1}^T$ that respect the Adam updates. To describe the condition that Adam satisfies and to prove this lemma, we introduce an auxiliary sequence (defined in (3)) that separates the randomness in the upper- and lower-level random variables, which is new and has not been leveraged in previous bilevel optimization literature.

## 4.2 MAIN RESULTS

We first introduce some notations and technical definitions. Denote $\sigma(\cdot)$ as the $\sigma$-algebra generated by the random variables within the argument. Let $\mathcal{F}_{\mathrm{init}}$ be the filtration for updating $y_1$ (see Algorithm 3): $\mathcal{F}_{\mathrm{init}} = \sigma(\pi_0, \ldots, \pi_{T_0-1})$. For any $t \geq 2$, define $\mathcal{F}_t^x, \mathcal{F}_t^y$ and $\mathcal{F}_t$ as $\mathcal{F}_t^x = \sigma(\bar{\xi}_1, \ldots, \bar{\xi}_{t-1})$, $\mathcal{F}_t^y = \sigma(\zeta_1, \ldots, \zeta_{t-1})$ and $\mathcal{F}_t = \sigma(\mathcal{F}_{\mathrm{init}} \cup \mathcal{F}_t^x \cup \mathcal{F}_t^y)$. We use $\mathbb{E}_t[\cdot]$ to denote the conditional expectation $\mathbb{E}[\cdot \mid \mathcal{F}_t]$. We also use $c_1, c_2, c_3$ to denote small enough constants and $C_1, C_2$ to denote large enough constants, all of which are independent of $\epsilon$ and $\delta$, where $\epsilon$ denotes the target gradient norm and $\delta$ denotes the failure probability. The definitions of problem-dependent constants $\sigma_\phi, C_{\phi,0}, C_{\phi,1}, \Delta_1, L_0, L_1, L, C_\beta$ are comprehensively listed in Appendix D.1.

**Theorem 4.1.** *Suppose Assumptions 3.2 to 3.4 hold. Let $G$ be a constant satisfying $G \geq \max\left\{4\lambda, 2\sigma_\phi, 4C_{\phi,0}, \frac{C_{\phi,1}}{L_1}, \sqrt{\frac{C_1\Delta_1 L_0}{C_L}}, \frac{C_1\Delta_1 L_1}{C_L}\right\}$. Given any $\epsilon > 0$ and $\delta \in (0,1)$, choose $0 \leq \beta_{\mathrm{sq}} \leq 1$, $\beta = \widetilde{\Theta}(\epsilon^2)$, $\gamma = \widetilde{\Theta}(\epsilon^2)$, $\eta = \widetilde{\Theta}(\epsilon^2)$, $Q = \widetilde{\Theta}(1)$, $T_0 = \widetilde{\Theta}(\epsilon^{-2})$. Run Algorithm 1 for $T = \max\left\{\frac{1}{\beta^2}, \frac{C_2\Delta_1 G}{\eta\epsilon^2}\right\} = \widetilde{O}(\epsilon^{-4})$ iterations. Then with probability at least $1 - \delta$ over the randomness in $\mathcal{F}_{T+1}$, we have $\|\nabla\Phi(x_t)\| \leq G$ for all $t \in [T]$, and $\frac{1}{T}\sum_{t=1}^T \|\nabla\Phi(x_t)\| \leq \epsilon^2$.*

**Remark**: The full statement of Theorem 4.1 with detailed parameter choices is deferred to Theorem D.12 in Appendix D. Theorem 4.1 provides the convergence guarantee for Algorithm 1:

AdamBO converges to $\epsilon$-stationary points with $T_0 + QT = \widetilde{O}(\epsilon^{-4})$ oracle complexity. This complexity result matches that of non-adaptive bilevel optimization algorithms in (Hao et al., 2024; Gong et al., 2024a) when the upper-level function exhibits unbounded smoothness, as well as the complexity of Adam for single-level optimization with generalized smooth functions (Li et al., 2023a). It is also worth noting that we choose a larger learning rate $\eta = \widetilde{\Theta}(\epsilon^2)$ for the upper-level updates, compared to $\eta = \widetilde{\Theta}(\epsilon^3)$ used in the SLIP algorithm (Gong et al., 2024a).

## 4.3 PROOF SKETCH

In this section, we provide a proof sketch for Theorem 4.1. The detailed proof can be found in Appendix D. Let $y_t^* = y^*(x_t)$. The key idea is to provide a high probability bound of lower-level estimation error $\|y_t - y_t^*\|$ when the upper-level variable $x$ is updated by the vanilla Adam. Lemma 4.4 provides such a guarantee: the lower-level error $\|y_t - y_t^*\|$ is bounded by a function of the initial estimation error $\|y_1 - y_1^*\|$, the variance term $\sigma_{g,1}^2$, and an auxiliary momentum estimator of the hypergradient $\|\hat{u}_t\|$ (see definition of $\hat{u}_t$ in (6)). Based on Lemma 4.4, we introduce Lemma 4.5 and 4.6, which incorporate the lower-level error into the upper-level problems and adapt the stopping time technique of Adam (Li et al., 2023a) to prove the convergence. The proof of Lemma 4.4 is a direct application of the randomness decoupling lemma (i.e., Lemma 4.2 in Section 4.3.2). All of the proofs in this section are based on Assumptions 3.2 to 3.4.

### 4.3.1 EQUIVALENT UPDATE RULE OF ADAMBO

Let $\alpha_t = \frac{\beta}{1-(1-\beta)^t}$ and $\alpha_t^{\text{sq}} = \frac{\beta_{\text{sq}}}{1-(1-\beta_{\text{sq}})^t}$. Inspired by (Li et al., 2023a), we provide an equivalent yet simpler update rule of lines 5-8 of Algorithm 1 (see Proposition A.1 for more details):

$$\hat{m}_t = (1-\alpha_t)\hat{m}_{t-1} + \alpha_t \hat{\nabla}\phi(x_t, y_t; \bar{\xi}_t), \quad \hat{v}_t = (1-\alpha_t^{\text{sq}})\hat{v}_{t-1} + \alpha_t^{\text{sq}}(\hat{\nabla}\phi(x_t, y_t; \bar{\xi}_t))^2.$$

### 4.3.2 RANDOM DECOUPLING LEMMA FOR LOWER-LEVEL ERROR CONTROL

In this section, we introduce the random decoupling lemma (Lemma 4.2) for the lower-level error control. The rationale is as follows: for any given upper-level variable sequence and any given randomness from the upper-level updates that satisfy certain conditions and are consistent with the AdamBO updates, we can bound the lower-level error with high probability, where the randomness is taken solely from lower-level random variables. Specifically, for any given sequence $\{\tilde{x}_t\}$, define $\tilde{\zeta}_t$ and $\hat{\xi}_t$ as the random variables from the lower-level and upper-level, respectively, at the $t$-th iteration (see (26) for definition). We consider the following update rule for $\{\tilde{y}_t\}$, which is exactly SGD and corresponds to line 5 of Algorithm 1:

$$\tilde{y}_{t+1} = \tilde{y}_t - \gamma \nabla_y G(\tilde{x}_t, \tilde{y}_t; \tilde{\zeta}_t). \tag{2}$$

Let $\tilde{y}_t^* = y^*(\tilde{x}_t)$ and $\tilde{\mathcal{F}}_t^y = \sigma(\tilde{\zeta}_1, \ldots, \tilde{\zeta}_{t-1})$. Denote $\tilde{G}_t := \max_{k \leq t} \|\nabla\Phi(\tilde{x}_k)\|$, $\tilde{L}_t := L_0 + L_1\tilde{G}_t$. We also introduce the following auxiliary sequences $\{\tilde{m}_t\}$ and $\{\tilde{u}_t\}$ for our analysis:

$$\tilde{m}_t = (1-\alpha_t)\tilde{m}_{t-1} + \alpha_t \hat{\nabla}\phi(\tilde{x}_t, \tilde{y}_t; \hat{\xi}_t), \quad \tilde{u}_t = (1-\alpha_t)\tilde{u}_{t-1} + \alpha_t \hat{\nabla}\phi(\tilde{x}_t, \tilde{y}_t^*; \hat{\xi}_t). \tag{3}$$

**Lemma 4.2** (Randomness Decoupling Lemma). *Given any sequence $\{\tilde{x}_t\}$ and any randomness $\{\hat{\xi}_t\}$ such that*

$$\|\tilde{x}_{t+1} - \tilde{x}_t\|^2 \leq \frac{2\eta^2}{\lambda^2} \left( \|\tilde{u}_t\|^2 + \tilde{L}_t^2 \sum_{j=1}^t d_{t,j} \|\tilde{y}_j - \tilde{y}_j^*\|^2 \right), \tag{4}$$

*where $\{d_{t,j}\}_{j=1}^t$ is defined in (10). Let $\{\tilde{y}_t\}$ be the iterates generated by the update rule (2) with $\gamma \leq 1/2l_{g,1}$ and choose $\gamma = 2\beta/\mu$. For any given $\delta \in (0,1)$ and all $t \geq 1$, the following holds with probability at least $1 - \delta$ over the randomness in $\tilde{\mathcal{F}}_{T+1}^y$:*

$$\|\tilde{y}_t - \tilde{y}_t^*\|^2 \leq \left(1 - \frac{\mu\gamma}{2}\right)^{t-1} \|\tilde{y}_1 - \tilde{y}_1^*\|^2 + \frac{8\gamma\sigma_{g,1}^2}{\mu} \ln \frac{eT}{\delta} \quad \text{(Variance)}$$

$$+ \left(\frac{4\eta^2 l_{g,1}^2}{\lambda^2\mu^3\gamma} \|\tilde{y}_1 - \tilde{y}_1^*\|^2 + \frac{16\eta^2 l_{g,1}^2\sigma_{g,1}^2}{\lambda^2\mu^4}\right) \sum_{i=1}^{t-1} \left(1 - \frac{\mu\gamma}{2}\right)^{t-1-i} \tilde{L}_i^2 \quad \text{(Drift)} \tag{5}$$

$$+ \frac{4\eta^2 l_{g,1}^2}{\lambda^2\mu^3\gamma} \sum_{i=1}^{t-1} \left(1 - \frac{\mu\gamma}{2}\right)^{t-1-i} \|\tilde{u}_i\|^2 + \frac{64\eta^4 l_{g,1}^4}{\lambda^4\mu^8\gamma^4} \sum_{i=1}^{t-1} \left(1 - \frac{\mu\gamma}{2}\right)^{t-1-i} \alpha_i\tilde{L}_i^2 \|\tilde{u}_i\|^2. \quad \text{(Drift)}$$

**Remark**: Lemma 4.2 shows that, when (4) holds for any sequence $\{\tilde{x}_t\}$ and any $\{\hat{\xi}_t\}$ (as satisfied by the vanilla Adam update for the upper-level variable), the lower-level error can be controlled with high probability as in (5). In addition, the high probability is taken over the randomness solely from the lower-level filtration $\tilde{\mathcal{F}}_{T+1}^y$. This lemma provides a technical tool to control the lower-level error without concerns about the dependency issues from the upper-level randomness. In particular, the right-hand side of (5) consists of two parts: the standard variance term, which does not involve the update of $\{\tilde{x}_t\}$ over $t$; and the drift terms, which account for the update of $\{\tilde{x}_t\}$ over time.

### 4.3.3 Applications of the Random Decoupling Lemma and Remaining Proof

Given a large enough constant $G$, denote $L = L_0 + L_1 G$ and $\psi = C_L G^2 / 2L$, where $G$ is defined in Theorem 4.1 and $C_L$ is defined in (43). Now we formally define the stopping time $\tau$ as

$$\tau := \min\{t \mid \Phi(x_t) - \Phi^* > \psi\} \wedge (T+1).$$

Based on Lemma D.1, we know that if $t < \tau$, we have both $\Phi(x_t) - \Phi^* \leq \psi$ and $\|\nabla\Phi(x_t)\| \leq G$. Similar to Section 4.3.2, we introduce the following auxiliary sequence $\{\hat{u}_t\}$ for our analysis:

$$\hat{u}_t = (1 - \alpha_t)\hat{u}_{t-1} + \alpha_t \hat{\nabla}\phi(x_t, y_t^*; \bar{\xi}_t). \tag{6}$$

**Lemma 4.3** (Warm-Start). *Choose $\gamma \leq 1/2l_{g,1}$. With probability at least $1 - \delta/4$ over the randomness in $\mathcal{F}_{init}$ (denote this event as $\mathcal{E}_0$) that: $\|y_1 - y_1^*\|^2 \leq \left(1 - \frac{\mu\gamma}{2}\right)^{T_0} \|y_0 - y_0^*\|^2 + \frac{8\gamma\sigma_{g,1}^2}{\mu} \ln \frac{4e}{\delta}$.*

**Lemma 4.4.** *Under the parameter choices in Lemma D.4, apply Lemma 4.2 with $\{\tilde{x}_t\} = \{x_t\}$, $\{\tilde{y}_t\} = \{y_t\}, \{\tilde{u}_t\} = \{\hat{u}_t\}$ and $\{\tilde{L}_t\} = \{\hat{L}_t\}$, then (5) holds with probability at least $1 - \delta/4$ over the randomness in $\mathcal{F}_{T+1}^y$ (denote this event as $\mathcal{E}_y$).*

**Remark**: Lemma 4.3 and Lemma 4.4 together provide a high probability bound for the lower-level error, where the randomness is taken only from the lower-level filtrations $\mathcal{F}_{init}$ and $\mathcal{F}_{T+1}^y$. Lemma 4.4 is a direct application of Lemma 4.2 to the actual sequence $\{x_t\}$ and $\{y_t\}$ in Algorithm 1.

**Lemma 4.5.** *If $t < \tau$, we have $\|\nabla\Phi(x_t)\| \leq G$, $\|\hat{u}_t\| \leq C_{u,0}$; under event $\mathcal{E}_0 \cap \mathcal{E}_y$, if $t < \tau$, we have $\|\hat{m}_t\| \leq C_{u,0} + C_{u,1}\varrho$, $\hat{v}_t \preceq (C_{u,0} + C_{u,1}\varrho)^2$, where constants $C_{u,0}, C_{u,1}, \varrho$ are defined in* (43) *and* (53)*, respectively.*

**Remark**: Lemma 4.5 generalizes the stopping time analysis from the single-level setting (Li et al., 2023a) to the bilevel setting and is useful for upper-level analysis. It shows that the momentum estimators of the hypergradient remains bounded when $t < \tau$ and $\mathcal{E}_0 \cap \mathcal{E}_y$ holds. This implies that $x_{t+1}$ and $x_t$ remains close for small enough $\eta$, allowing us to apply Lemmas B.10 and B.11.

**Lemma 4.6.** *Under event $\mathcal{E}_0 \cap \mathcal{E}_y$ and the parameter choices in Lemma D.4, we have*

$$\sum_{t=1}^{\tau-1} \|\hat{m}_t - \hat{u}_t\|^2 \leq TL^2 \left( \left(1 + \frac{8\eta^2 l_{g,1}^2 L^2}{\lambda^2 \mu^4 \gamma^2}\right) \|y_1 - y_1^*\|^2 + \left(\frac{8\gamma}{\mu} \ln \frac{4eT}{\delta} + \frac{32\eta^2 l_{g,1}^2 L^2}{\lambda^2 \mu^5 \gamma}\right) \sigma_{g,1}^2 \right)$$

$$+ L^2 \left( \frac{8\eta^2 l_{g,1}^2}{\lambda^2 \mu^4 \gamma^2} + \frac{2048\eta^4 l_{g,1}^4 L^2}{\lambda^4 \mu^8 \gamma^4} \left(2 + \ln \frac{1}{\beta}\right) \right) \sum_{t=1}^{\tau-1} \|\epsilon_t\|^2 + 2\|\nabla\Phi(x_t)\|^2 + \frac{2l_{g,1}^2 l_{f,0}^2}{\mu^2} \left(1 - \frac{\mu}{l_{g,1}}\right)^{2Q}.$$

**Remark**: Lemma 4.6 provides a bound for the difference between the actual momentum $\hat{m}_t$ versus the virtual momentum $\hat{u}_t$ under the good event $\mathcal{E}_0 \cap \mathcal{E}_y$, which is essential for establishing the convergence guarantees for AdamBO.

## 5 VR-AdamBO and Convergence Analysis

### 5.1 Algorithm Design

In this section, we propose a variance-reduced version of AdamBO, called VR-AdamBO, as shown in Algorithm 2. Similar to AdamBO, the VR-AdamBO algorithm also includes a warm-start phase for the lower-level variable (line 2): it runs stochastic Nesterov accelerated gradient (SNAG) method on the lower-level variable $y$ for $T_0$ iterations, with the initial upper-level variable $x_0$ fixed. After that, VR-AdamBO updates the upper-level variable by VRAdam (Li et al., 2023a) (i.e., the Adam algorithm (Kingma & Ba, 2014) with recursive momentum (Cutkosky & Orabona, 2019), lines

---

**Algorithm 2** VR-ADAMBO (all operations on vectors are element-wise, see Algorithm 5 for SNAG)

1: **Input:** $\beta, \beta_{sq}, \eta, \lambda, T, S_1, x_0, y_0$
2: **Initialize** $y_1 = y_2 = \hat{y}_1 = \hat{y}_2 = \text{SNAG}(x_0, y_0, \gamma, T_0)$, $m_1 = \hat{\nabla}\phi(x_1, y_1; S_1)$, $v_1 = \beta_{sq}m_1^2$,
  $x_1 = x_0$ and $x_2 = x_1 - \frac{\eta m_1}{|m_1| + \lambda}$, where $S_1$ is a batch of samples with size $S_1$.
3: **for** $t = 2, \cdots, T$ **do**
4:   **if** $t$ is a multiple of $I$ **then**
5:     Set $y_t^0 = y_t^{-1} = y_t$
6:     **for** $j = 0, \ldots, N-1$ **do**
7:       $z_t^j = y_t^j + \alpha(y_t^j - y_t^{j-1})$
8:       $y_t^{j+1} = z_t^j - \gamma\nabla_y G(x_t, z_t^j; \pi_t^j)$, where $\pi_t^j \sim \mathcal{D}_g$
9:     **end for**
10:     $y_{t+1} = y_t^N$
11:   **else**
12:     $y_{t+1} = y_t$
13:   **end if**
14:   $\hat{y}_{t+1} = (1 - \nu)\hat{y}_t + \nu y_{t+1}$
15:   $m_t = (1 - \beta)m_{t-1} + \beta\hat{\nabla}\phi(x_t, y_t; \bar{\xi}_t) + (1 - \beta)(\hat{\nabla}\phi(x_t, \hat{y}_t; \bar{\xi}_t) - \hat{\nabla}\phi(x_{t-1}, \hat{y}_{t-1}; \bar{\xi}_t))$
16:   $v_t = (1 - \beta_{sq})v_{t-1} + \beta_{sq}(\hat{\nabla}\phi(x_t, \hat{y}_t; \bar{\xi}_t))^2$
17:   $\hat{v}_t = \frac{v_t}{1 - (1 - \beta_{sq})^t}$
18:   $x_{t+1} = x_t - \frac{\eta}{\sqrt{\hat{v}_t} + \lambda} \odot m_t$
19: **end for**

---

$15 \sim 18$) and periodically updates the lower-level variable by SNAG (lines $4 \sim 14$). In particular, the lower-level variable is updated by $N$ steps of SNAG for a fixed upper-level variable (lines $6 \sim 8$) after every $I$ iterations of the upper-level updates via VRAdam (lines $4 \sim 13$), and the moving average of the lower-level variable is used to estimate the hypergradient (line 14). Note that the periodic updates for the lower-level variable have been widely used in bilevel optimization (Hao et al., 2024; Gong et al., 2024b). VR-AdamBO can be regarded as a generalization of the AccBO algorithm in (Gong et al., 2024b), with the key difference being that AccBO uses normalized SGD with momentum for the upper-level update, whereas VR-AdamBO employs VRAdam.

## 5.2 MAIN RESULTS

**Theorem 5.1.** *Suppose that Assumptions 3.2, 3.3 and 3.5 hold. Let $G$ be a constant satisfying $G \geq$ $\max\left\{4\lambda, 2\sigma_\phi, 4C_{\phi,0}, \frac{C_{\phi,1}}{L_1}, \sqrt{\frac{C_1\Delta_1 L_0}{C_L\delta}}, \frac{C_1\Delta_1 L_1}{C_L\delta}\right\}$. Choose $0 \leq \beta_{sq} \leq 1$, $\beta = \Theta(\epsilon^2)$, $\gamma = \widetilde{\Theta}(\epsilon^2)$, $\eta = \Theta(\epsilon)$, $\alpha = O(1)$, $\nu = \Theta(\epsilon)$, $I = \Theta(\epsilon^{-1})$, $Q = \widetilde{\Theta}(1)$, $N = \widetilde{\Theta}(\epsilon^{-1})$, $T_0 = \widetilde{\Theta}(\epsilon^{-1})$. Run Algorithm 2 for $T = \frac{64G\Delta_1}{\eta\delta\epsilon^2} = O(\epsilon^{-3})$ iterations. Then with probability at least $1 - \delta$ over the randomness in $\mathcal{F}_{T+1}$, we have $\|\nabla\Phi(x_t)\| \leq G$ for all $t \in [T]$, and $\frac{1}{T}\sum_{t=1}^T \|\nabla\Phi(x_t)\| \leq \epsilon^2$.*

**Remark**: The full statement of Theorem 5.1, including detailed parameter choices, is deferred to Theorem E.14 in Appendix E. Theorem 5.1 establishes an improved oracle complexity of $T_0 + TN/I + TQ = \widetilde{O}(\epsilon^{-3})$ for VR-AdamBO. This complexity result matches that of (Gong et al., 2024b) when the upper-level function is unbounded smooth, as well as the complexity of VRAdam for single-level optimization with generalized smooth objectives (Li et al., 2023a). Notably, we choose a larger learning rate $\eta = \widetilde{\Theta}(\epsilon)$ for the upper-level updates, compared to $\eta = \widetilde{\Theta}(\epsilon^2)$ used in the AccBO algorithm (Gong et al., 2024b).

## 5.3 PROOF SKETCH

In this section, we briefly discuss the challenges in analyzing VR-AdamBO and provide a roadmap for the proof. The detailed proofs can be found in Appendix E. Note that to apply relaxed smoothness property and descent inequality, as listed in Lemmas B.10 and B.11, one requirement is that $x_{t+1}$ and $x_t$ should remain close since Definition 3.1 is a local condition rather than a global one. For AdamBO, this requirement is not hard to satisfy with sufficiently small $\eta$, based on Lemma 4.5 and its remark below. However, VR-AdamBO may not satisfy such a almost sure bound for $\|m_t\|$ due

to the STORM-type update (Cutkosky & Orabona, 2019) for the upper-level variable. To overcome this difficulty, we introduce a novel stopping time approach in the context of bilevel optimization. Specifically, we define $\epsilon_t := m_t - \mathbb{E}_t[\hat{\nabla}\phi(x_t, \hat{y}_t; \bar{\xi}_t)]$ and the new stopping time $\tau$ as

$$\tau := \min\{t \mid \|\nabla\Phi(x_t)\| \le G\} \wedge \min\{t \mid \|\epsilon_t\| \ge G\} \wedge (T+1), \tag{7}$$

where $G$ is specified in Theorem 5.1. It is worth noting that our definition of $\tau$ for the analysis of VR-AdamBO differs from that in (Li et al., 2023a) for VRAdam, as our $\epsilon_t$ is not defined as the difference between $m_t$ and $\nabla\Phi(x_t)$. At this point, we may still fail to guarantee the boundedness of $\|m_t\|$ before time $\tau$, unless the hypergradient bias introduced by the lower-level estimation error can be effectively controlled. Fortunately, by leveraging the lower-level acceleration technique (Gong et al., 2024b) with periodic updates and averaging, we develop a new induction argument (i.e., Lemmas E.10 to E.12) to show that under $t < \tau$ and some good event $\mathcal{E}_y$, both $\|\hat{y}_t - y_t^*\|$ and $\|m_t\|$ are bounded. We then show the averaged lower-level error is small under the parameter choices in Theorem 5.1 (see Lemma E.4), which shares an similar spirit as Lemma 4.6. Combining aforementioned lemmas with the technique developed in (Li et al., 2023a) for the upper-level analysis, we obtain the improved complexity result. One can refer to Appendix E for more details.

## 6 EXPERIMENTS

**Deep AUC Maximization with RNNs/Transformers.** The Area Under the ROC Curve (AUC) (Hanley & McNeil, 1983) is a widely used metric for evaluating the effectiveness of binary classification models, especially in the imbalanced data scenarios. It is defined as the probability that the prediction score of a positive example is higher than that of a negative example (Hanley & McNeil, 1982). Deep AUC maximization (Liu et al., 2020; Ying et al., 2016) can be formulated as a min-max optimization problem (Liu et al., 2020): $\min_{\boldsymbol{w}\in\mathbb{R}^d,(a,b)\in\mathbb{R}^2}\max_{\alpha\in\mathbb{R}} f(\boldsymbol{w},a,b,\alpha) := \mathbb{E}_{\boldsymbol{z}}[F(\boldsymbol{w},a,b,\alpha;\boldsymbol{z})]$, where $F(\boldsymbol{w},a,b,\alpha;\boldsymbol{z}) = (1-p)(h(\boldsymbol{w};\boldsymbol{x})-a)^2\mathbb{I}_{[c=1]} + p(h(\boldsymbol{w};\boldsymbol{x})-b)^2\mathbb{I}_{[c=-1]} + 2(1+\alpha)(ph(\boldsymbol{w};\boldsymbol{x})\mathbb{I}_{[c=-1]} - (1-p)h(\boldsymbol{w};\boldsymbol{x})\mathbb{I}_{[c=1]}) - p(1-p)\alpha^2$, $\boldsymbol{w}$ denotes the model parameter of a deep neural network, and $\boldsymbol{z} = (\boldsymbol{x},c)$ represents a random training data sample ($\boldsymbol{x}$ represents the feature vector and $c \in \{+1,-1\}$ represents the class label), the function $h(\boldsymbol{w},\boldsymbol{x})$ is a scoring function for the sample with feature $\boldsymbol{x}$, and $p = \Pr(c=1)$ indicates the proportion of positive samples in the population. This min-max problem can be reformulated as the form of a bilevel optimization problem with lower-level objective function $g = -f$:

$$\min_{\boldsymbol{w}\in\mathbb{R}^d,(a,b)\in\mathbb{R}^2} \mathbb{E}_{\boldsymbol{z}}[F(\boldsymbol{w},a,b,\alpha^*(\boldsymbol{w},a,b);\boldsymbol{z})] \quad \text{s.t.,} \quad \alpha^*(\boldsymbol{w},a,b) \in \arg\min_{\alpha\in\mathbb{R}} -\mathbb{E}_{\boldsymbol{z}}[F(\boldsymbol{w},a,b,\alpha;\boldsymbol{z})].$$

In above, $(\boldsymbol{w},a,b)$ is the upper-level variable, and $\alpha$ is the lower-level variable. The lower-level problem is a strongly convex one-dimensional quadratic function with respect to $\alpha$, while the upper-level objective is non-convex and can exhibit unbounded smoothness when using a recurrent neural network or a transformer as the predictive model (Crawshaw et al., 2022; Zhang et al., 2020b).

In our experiment, we focus on tackling an imbalanced text classification task by maximizing the AUC metric. Specifically, we conduct experiments using deep AUC maximization on the imbalanced Sentiment140 dataset (Go et al., 2009), a binary text classification benchmark. Following the approach in (Yuan et al., 2021), we introduce imbalance in the training set using a pre-specified imbalance ratio ($p$) while keeping the test set distribution unchanged. For a given $p$, we randomly remove positive samples (labeled as 1) from the training set until the desired proportion of positive examples is achieved. In our experiment, we set $p$ to 0.8 (0.9), meaning that 80% (90%) of the training samples are positive examples. We run the experiment using two different models, a two-layer transformer, and a two-layer recurrent neural network (RNN) with the same input dimension of 300, hidden dimension of 4096, and an output dimension of 2.

To evaluate the effectiveness of our proposed bilevel optimization algorithm, we compare with recent bilevel optimization baselines, including StocBio (Ji et al., 2021), TTSA (Hong et al., 2023), SABA (Dagréou et al., 2022), MA-SOBA (Chen et al., 2023a), SUSTAIN (Khanduri et al., 2021), VRBO (Yang et al., 2021), BO-REP (Hao et al., 2024), SLIP (Gong et al., 2024a), and AccBO (Gong et al., 2024b). The training and testing results of the transformer model over 50 epochs are presented in Figure 3 (a) and (b), while the corresponding running times are shown in Figure 3 (c) and (d). Our proposed Adam-type algorithms, AdamBO and VR-AdamBO, show the faster convergence rate and significantly outperform other baselines. In particular, the performance on the training AUC (testing AUC) is better by at least 14% (7%) over other baselines. The running time results indicate

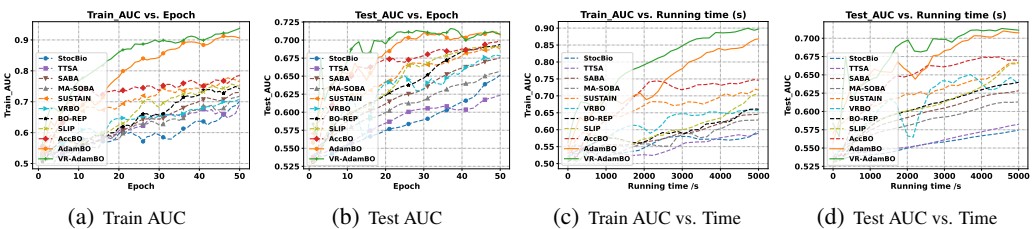

(a) Train AUC      (b) Test AUC      (c) Train AUC vs. Time      (d) Test AUC vs. Time

Figure 1: Transformer for AUC maximization on Sentiment140 dataset with imbalance ratio of 0.9.

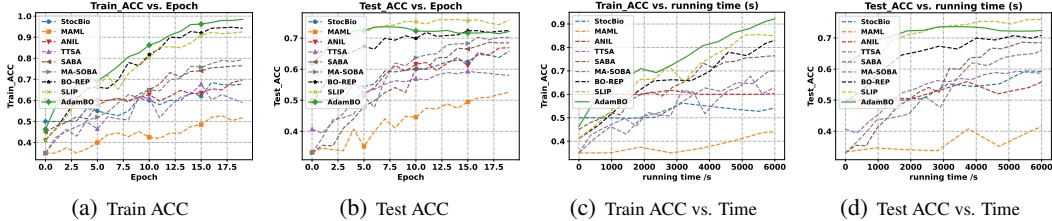

(a) Train ACC      (b) Test ACC      (c) Train ACC vs. Time      (d) Test ACC vs. Time

Figure 2: Comparison with bilevel optimization baselines on Hyper-representation.

that AdamBO and VR-AdamBO converge much faster to a high AUC value compared to the other baselines. We also perform the AUC maximization on a RNN model with imbalance rario of 0.8, and the results are presented in Appendix F.1. More detailed parameter tuning and selection can be found in Appendix F.

**Hyper-representation Learning.** Hyper-representation learning, i.e., meta-learning (Finn et al., 2017), aims to find a good meta learner parameterized by $x$, such that it can quickly adapt to a new task $i$ by fine-tuning the corresponding adapter $y_i$. Consider a meta-learning task consisting of $K$ tasks with the training set $\{\mathcal{D}_i^{tr} \mid i = 1, \ldots, K\}$ and validation set $\{\mathcal{D}_i^{val} \mid i = 1, \ldots, K\}$. Each task has a loss function $\mathcal{L}(x, y_i; \xi_i)$ over each sample $\xi_i$. This meta-learning problem can be reformulated as a bilevel optimization, where the lower-level objective function tries to find an optimal task-specific adapter $y_i^*(x)$ on training data $\mathcal{D}_i^{tr}$, and the upper-level minimizes the objective function on validation data $\mathcal{D}_i^{val}$ by finding the optimal meta-learner $x$ with a set of adapters $y = \{y_1^*(x), y_2^*(x), \ldots, y_K^*(x)\}$. We have the following formulation:

$$\min_x \frac{1}{K} \sum_{i=1}^{K} \frac{1}{|\mathcal{D}_i^{val}|} \sum_{\xi \in \mathcal{D}_i^{val}} \mathcal{L}(x, y^*(x); \xi), \text{ s.t., } y^*(x) = \arg\min_y \frac{1}{K} \sum_{i=1}^{K} \mathcal{L}_{\mathcal{D}_i^{tr}}(x, y_i; \zeta) + \frac{\mu}{2}\|y_i\|^2,$$

where $\mathcal{L}_{\mathcal{D}_i^{tr}}(x, y_i; \zeta) = \frac{1}{|\mathcal{D}_i^{tr}|} \sum_{\zeta \in \mathcal{D}_i^{tr}} \mathcal{L}(x, y_i; \zeta)$. The adapter (parameterized by $y_i$) is typically instantiated as the last linear layer, and the meta learner (parameterized by $x$) is the remaining layers of model, which guarantees that the lower-level function to be strongly-convex when $\mu > 0$.

We conduct the meta-learning experiments for the text classification on dataset Stanford Natural Language Inference (SNLI) (Bowman et al., 2015), which consists of 570k pairs of sentences with 3 classes. We construct $K = 500$ tasks, where each task $\mathcal{D}_i^{tr}$ and $\mathcal{D}_i^{val}$ randomly sample two disjoint categories from the original data, respectively. Empirically, we use mini-batches of meta-tasks for training, with a task batch size of 25. A 3-layer recurrent network is used as representation layers and a fully-connected layer as an adapter. The input dimension, hidden dimension and output dimension are set to be 300, 4096, and 3, respectively.

We compare with typical meta-learning algorithms, MAML (Rajeswaran et al., 2019) and ANIL (Raghu et al., 2019), and recent bilevel optimization algorithms, StocBio (Ji et al., 2021), TTSA (Hong et al., 2023), SABA (Dagréou et al., 2022), MA-SOBA (Chen et al., 2023a), BO-REP (Hao et al., 2024), SLIP (Gong et al., 2024a). The comparison results of training and testing accuracy are shown in Figure 2. AdamBO outperforms other baselines on training set, and exhibits faster convergence rate. One can refer to Appendix F for detailed hyper-parameter choices and experimental settings. All the experiments are run on an single NVIDIA A6000 (48GB memory) GPU and a AMD EPYC 7513 32-Core CPU.

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

CONTENTS

---

**Algorithm 3** SGD

---

1: **Input:** $x, y_0, \gamma, T_0$          # $\text{SGD}(x, y_0, \gamma, T_0)$
2: **Initialize** $y_0^{\text{init}} = y_0$
3: **for** $t = 0, 1, \ldots, T_0 - 1$ **do**
4:     Sample $\pi_t$ from distribution $\mathcal{D}_g$
5:     $y_{t+1}^{\text{init}} = y_t^{\text{init}} - \gamma \nabla_y G(x, y_t^{\text{init}}; \pi_t)$
6: **end for**

---

**Algorithm 4** ADAMBO (Equivalent update rule of Algorithm 1)

---

1: **Input:** $\beta, \beta_{\text{sq}}, \eta, \gamma, \lambda, T_0, T, x_1, y_0$
2: **Initialize** $y_1 = \text{SGD}(x_1, y_0, \gamma, T_0)$, $\hat{m}_1 = \hat{\nabla}\phi(x_1, y_1; \bar{\xi}_1)$ and $\hat{v}_1 = (\hat{\nabla}\phi(x_1, y_1; \bar{\xi}_1))^2$
3: **for** $t = 1, \ldots, T$ **do**
4:     $\alpha_t = \frac{\beta}{1-(1-\beta)^t}, \alpha_t^{\text{sq}} = \frac{\beta_{\text{sq}}}{1-(1-\beta_{\text{sq}})^t}$
5:     Draw new samples and perform the following updates
6:     $y_{t+1} = y_t - \gamma \nabla_y G(x_t, y_t; \zeta_t)$
7:     $\hat{m}_t = (1 - \alpha_t)\hat{m}_{t-1} + \alpha_t \hat{\nabla}\phi(x_t, y_t; \bar{\xi}_t)$
8:     $\hat{v}_t = (1 - \alpha_t^{\text{sq}})\hat{v}_{t-1} + \alpha_t^{\text{sq}}(\hat{\nabla}\phi(x_t, y_t; \bar{\xi}_t))^2$
9:     $x_{t+1} = x_t - \frac{\eta}{\sqrt{\hat{v}_t}+\lambda} \odot \hat{m}_t$
10: **end for**

---

**Algorithm 5** STOCHASTIC NESTEROV ACCELERATED GRADIENT METHOD (SNAG)

---

1: **Input:** $\tilde{x}, \tilde{y}_{-1}, \gamma, T_0$         # $\text{SNAG}(x, y_0, \gamma, T_0)$
2: **Initialize** $\tilde{y}_0 = \tilde{y}_{-1}$
3: **for** $t = 0, 1, \ldots, T_0 - 1$ **do**
4:     Sample $\tilde{\pi}_t$ from distribution $\mathcal{D}_g$
5:     $\tilde{z}_t = \tilde{y}_t + \alpha(\tilde{y}_t - \tilde{y}_{t-1})$
6:     $\tilde{y}_{t+1} = \tilde{z}_t - \gamma \nabla_y G(\tilde{x}, \tilde{z}_t; \tilde{\pi}_t)$
7: **end for**

---

## A    EQUIVALENT UPDATE RULE OF ADAMBO (ALGORITHM 1)

In this section, we aim to provide a simplified version of the bias correction steps (lines 7-8) of Algorithm 1. Inspired by (Li et al., 2023a, Appendix C.1), we present an equivalent yet simpler update rule of Algorithm 1 in the following Proposition A.1. The detailed equivalent framework is also outlined in Algorithm 4.

**Proposition A.1.** *Let $\alpha_t = \frac{\beta}{1-(1-\beta)^t}$ and $\alpha_t^{\text{sq}} = \frac{\beta_{\text{sq}}}{1-(1-\beta_{\text{sq}})^t}$. Then the update rule in Bi-Adam (Algorithm 1) is equivalent to that in Algorithm 4:*

$$
\begin{aligned}
y_{t+1} &= y_t - \gamma \nabla_y G(x_t, y_t; \zeta_t), \\
\hat{m}_t &= (1 - \alpha_t)\hat{m}_{t-1} + \alpha_t \hat{\nabla}\phi(x_t, y_t; \bar{\xi}_t), \\
\hat{v}_t &= (1 - \alpha_t^{\text{sq}})\hat{v}_{t-1} + \alpha_t^{\text{sq}}(\hat{\nabla}\phi(x_t, y_t; \bar{\xi}_t))^2, \\
x_{t+1} &= x_t - \frac{\eta}{\sqrt{\hat{v}_t} + \lambda} \odot \hat{m}_t,
\end{aligned}
\tag{8}
$$

*where initially we set $\hat{m}_1 = \hat{\nabla}\phi(x_1, y_1; \bar{\xi}_1)$ and $\hat{v}_1 = (\hat{\nabla}\phi(x_1, y_1; \bar{\xi}_1))^2$. There is no need to define $\hat{m}_0$ and $\hat{v}_0$ since $1 - \alpha_1 = 1 - \alpha_1^{\text{sq}} = 0$.*

*Proof of Proposition A.1.* We follow the same proof as in (Li et al., 2023a, Proposition E.1), but replace the stochastic gradient $\nabla f(x_t, \xi_t)$ in (Li et al., 2023a) with the stochastic hypergradient estimator $\hat{\nabla}\phi(x_t, y_t; \bar{\xi}_t)$ in our setting. We still provide the proof here for completeness.

Let $Z_t = 1 - (1-\beta)^t$. Then we know that $\alpha_t = \beta/Z_t$ and $m_t = Z_t \hat{m}_t$. By line 6 of Algorithm 1 (the momentum update rule for $m_t$), we have

$$
Z_t \hat{m}_t = (1 - \beta)Z_{t-1}\hat{m}_{t-1} + \beta \hat{\nabla}\phi(x_t, y_t; \bar{\xi}_t).
$$

Note that $Z_t$ satisfies the following property

$$(1 - \beta)Z_{t-1} = 1 - \beta - (1 - \beta)^t = Z_t - \beta.$$

Then we have

$$\hat{m}_t = \frac{Z_t - \beta}{Z_t}\hat{m}_{t-1} + \frac{\beta}{Z_t}\hat{\nabla}\phi(x_t, y_t; \bar{\xi}_t)$$

$$= (1 - \alpha_t)\hat{m}_{t-1} + \alpha_t\hat{\nabla}\phi(x_t, y_t; \bar{\xi}_t).$$

Next, we verify the initial condition. By Algorithm 1, since we set $m_0 = 0$, then we have $m_1 = \beta\hat{\nabla}\phi(x_1, y_1; \bar{\xi}_1)$. Therefore, we have $\hat{m}_1 = m_1/Z_1 = \hat{\nabla}\phi(x_1, y_1; \bar{\xi}_1)$ since $Z_1 = \beta$. Then the proof is completed by applying the same analysis on $v_t$ and $\hat{v}_t$. $\qquad\square$

## B  TECHNICAL LEMMAS

In this section, we present several useful algebraic facts (Appendix B.1), probabilistic lemmas (Appendix B.2), and auxiliary lemmas for bilevel optimization under the unbounded smoothness setting (Appendix B.3).

### B.1  USEFUL ALGEBRAIC FACTS

In this section, we will frequently use $\alpha_t$ and $\alpha_t^{\mathrm{sq}}$, so we restate their definitions here for the reader's convenience:

$$\alpha_t = \frac{\beta}{1 - (1 - \beta)^t} \qquad \text{and} \qquad \alpha_t^{\mathrm{sq}} = \frac{\beta_{\mathrm{sq}}}{1 - (1 - \beta_{\mathrm{sq}})^t}. \tag{9}$$

The following two lemmas, i.e., Lemmas B.1 and B.2, are useful for bounding the norm of the difference between Neumann series approximation matrices in Appendix B.3.

**Lemma B.1.** *For any matrix sequences $\{A_i\}_{i=1}^k$ and $\{B_i\}_{i=1}^k$ (where $k \geq 1$), it holds that*

$$\left\| \prod_{i=1}^k A_i - \prod_{i=1}^k B_i \right\| = \sum_{i=1}^k \|B_1\| \cdots \|B_{i-1}\|\|A_i - B_i\|\|A_{i+1}\| \cdots \|A_k\|,$$

*where we use the convention $A_{k+1} = B_0 = I$.*

*Proof of Lemma B.1.* It is easy to check that

$$\prod_{i=1}^k A_i - \prod_{i=1}^k B_i = A_1 \cdots A_k - B_1 \cdots B_k$$

$$= (A_1 - B_1)A_2 \cdots A_k + B_1(A_2 - B_2)A_3 \cdots A_k + \cdots + B_1 \cdots B_{k-1}(A_k - B_k)$$

$$= \sum_{i=1}^k B_1 \cdots B_{i-1}(A_i - B_i)A_{i+1} \cdots A_k,$$

where we set $A_{k+1} = B_0 = I$ in the last equality. The result follows by noting that the operator norm is submultiplicative. $\qquad\square$

**Lemma B.2.** *For any $Q \geq 1$ and $a \in (0, 1)$, we have*

$$\sum_{q=0}^{Q-1} q \cdot a^{q-1} \leq \frac{1}{(1-a)^2}.$$

*Proof of Lemma B.2.* We obtain the result by simple calculation:

$$\sum_{q=0}^{Q-1} q \cdot a^{q-1} = \frac{1 - Qa^{Q-1} + (Q-1)a^Q}{(1-a)^2} \leq \frac{1 - Qa^{Q-1} + (Q-1)a^{Q-1}}{(1-a)^2}$$

$$= \frac{1 - a^{Q-1}}{(1-a)^2} \leq \frac{1}{(1-a)^2}.$$

$\qquad\square$

The next four lemmas, Lemmas B.3 to B.6, are useful for controlling the lower-level estimation error and for proving the randomness decoupling lemma (i.e., Lemma 4.2) in Appendix C.

**Lemma B.3.** *For any $t \geq 1$, define $\{d_{t,j}\}_{j=0}^{t}$ as the following:*

$$d_{t,j} = \begin{cases} \prod_{i=1}^{t}(1 - \alpha_i), & j = 0 \\ \alpha_j \prod_{i=j+1}^{t}(1 - \alpha_i), & 1 \leq j \leq t - 1 \\ \alpha_t, & j = t. \end{cases} \tag{10}$$

*Then $\{d_{t,j}\}_{j=0}^{t}$ has the following properties:*

- *For $j = 0$, $d_{t,j} = 0$.*

- *For $1 \leq j \leq t$, $d_{t,j} = \alpha_t(1 - \beta)^{t-j}$.*

- *$\sum_{j=0}^{t} d_{t,j} = \sum_{j=1}^{t} d_{t,j} = 1$.*

*Proof of Lemma B.3.* Recall the definition of $\alpha_t$ in Algorithm 4, we have

$$\alpha_t = \frac{\beta}{1 - (1 - \beta)^t} \qquad \text{and} \qquad 1 - \alpha_t = \frac{1 - (1 - \beta)^{t-1}}{1 - (1 - \beta)^t}(1 - \beta).$$

It is obvious to see $\alpha_1 = 1$, then for $j = 0$ we have

$$d_{t,0} = \prod_{i=1}^{t}(1 - \alpha_i) = (1 - \alpha_t) \cdots (1 - \alpha_1) = 0.$$

For $1 \leq j \leq t - 1$ we have

$$d_{t,j} = \alpha_j \prod_{i=j+1}^{t}(1 - \alpha_i) = \frac{\beta}{1 - (1 - \beta)^j} \prod_{i=j+1}^{t} \frac{1 - (1 - \beta)^{i-1}}{1 - (1 - \beta)^i}(1 - \beta) = \alpha_t(1 - \beta)^{t-j}.$$

For $j = t$ we have

$$d_{t,t} = \alpha_t = \frac{\beta}{1 - (1 - \beta)^t} = \alpha_t(1 - \beta)^{t-t}.$$

For the last result of the lemma, we have

$$\sum_{j=0}^{t} d_{t,j} = \sum_{j=1}^{t} d_{t,j} = \sum_{j=1}^{t} \alpha_t(1 - \beta)^{t-j} = \frac{1 - (1 - \beta)^t}{\beta}\alpha_t = 1,$$

where we use $d_{t,0} = 0$ in the first equality. $\qquad \square$

**Lemma B.4.** *For any $x \in (0, 1]$, we have*

$$1 - \frac{1}{x} \leq \ln x \leq x - 1.$$

*Consequently, for any $\beta \in [0, 1)$ we have*

$$-\frac{\beta}{1 - \beta} \leq \ln(1 - \beta) \leq -\beta \qquad \text{and} \qquad \beta \leq -\ln(1 - \beta) \leq \frac{\beta}{1 - \beta}.$$

*Proof of Lemma B.4.* This is a well-known logarithm inequality, so we omit the proof here. $\qquad \square$

**Lemma B.5.** *For any $t \geq 1$, we have*

$$t\alpha_t(1 - \beta)^{t-1} \leq 1.$$

*Proof of Lemma B.5.* By definition of $\alpha_t$, we have

$$t\alpha_t(1-\beta)^{t-1} = \frac{\beta t(1-\beta)^{t-1}}{1-(1-\beta)^t}.$$

Let $f : \mathbb{R} \to \mathbb{R}$ be

$$f(t) = \frac{\beta t(1-\beta)^{t-1}}{1-(1-\beta)^t}.$$

Then we have

$$f'(t) = \frac{\beta(1-\beta)^{t-1}}{(1-(1-\beta)^t)^2}(1-(1-\beta)^t + t\ln(1-\beta)).$$

Let $g : \mathbb{R} \to \mathbb{R}$ be

$$g(t) = 1-(1-\beta)^t + t\ln(1-\beta).$$

Then we have

$$g'(t) = (1-(1-\beta)^t)\ln(1-\beta) \le 0.$$

Note that Lemma B.4 gives $g(1) = \beta + \ln(1-\beta) \le 0$, then for any $t \ge 1$ we have $g(t) \le g(1) \le 0$, and

$$f'(t) = \frac{\beta(1-\beta)^{t-1}}{(1-(1-\beta)^t)^2}g(t) \le 0.$$

Therefore, for any $t \ge 1$ we conclude that

$$t\alpha_t(1-\beta)^{t-1} = f(t) \le f(1) = 1.$$

$\square$

**Lemma B.6.** *For any $t \ge 1$ and $0 < \beta \le 1/2$, we have*

$$\sum_{i=1}^{t}(1-\beta)^{t-i}\alpha_i \le 32 + 16\ln\frac{1}{\beta}.$$

*Proof of Lemma B.6.* We split the summation as the following:

$$\sum_{i=1}^{t}(1-\beta)^{t-i}\alpha_i = \beta\sum_{i=1}^{t}\frac{(1-\beta)^{t-i}}{1-(1-\beta)^i} = \beta(1-\beta)^t\sum_{i=1}^{t}\frac{(1-\beta)^{-i}}{1-(1-\beta)^i}$$

$$= \beta(1-\beta)^t\left(\sum_{1\le i<1/\beta}\frac{(1-\beta)^{-i}}{1-(1-\beta)^i} + \sum_{1/\beta\le i\le t}\frac{(1-\beta)^{-i}}{1-(1-\beta)^i}\right).$$

Note that when $i < 1/\beta$, we have

$$(1-\beta)^i \le 1 - \frac{1}{2}\beta i \quad\Longrightarrow\quad 1-(1-\beta)^i \ge \frac{1}{2}\beta i \quad\Longrightarrow\quad \frac{1}{1-(1-\beta)^i} \le \frac{2}{\beta i},$$

and by Lemma B.4 and $\beta \le 1/2$ we know that

$$(1-\beta)^{-i} = \exp(-i\ln(1-\beta)) \le \exp\left(\frac{i\beta}{1-\beta}\right) \le \exp\left(\frac{1}{1-\beta}\right) \le e^2.$$

Then for the first part of the summation we have

$$\sum_{1\le i<1/\beta}\frac{(1-\beta)^{-i}}{1-(1-\beta)^i} \le \frac{2e^2}{\beta}\sum_{1\le i<1/\beta}\frac{1}{i} \le \frac{2e^2}{\beta}\left(1+\ln\frac{1}{\beta}\right). \tag{11}$$

Also note that when $i \ge 1/\beta$, we have

$$(1-\beta)^i \le \frac{1}{e} \quad\Longrightarrow\quad 1-(1-\beta)^i \ge 1-\frac{1}{e} \quad\Longrightarrow\quad \frac{1}{1-(1-\beta)^i} \le \frac{e}{e-1}.$$

Then for the second part of the summation we have

$$\sum_{1/\beta \le i \le t} \frac{(1-\beta)^{-i}}{1-(1-\beta)^i} \le \frac{e}{e-1} \sum_{1/\beta \le i \le t} (1-\beta)^{-i} \le \frac{e}{e-1} \sum_{1 \le i \le t} (1-\beta)^{-i} \le \frac{e(1-\beta)^{-t}}{(e-1)\beta}.$$

(12)

Combining (11) and (12) we obtain that

$$\sum_{i=1}^{t} (1-\beta)^{t-i} \alpha_i \le \beta(1-\beta)^t \left( \sum_{1 \le i < 1/\beta} \frac{(1-\beta)^{-i}}{1-(1-\beta)^i} + \sum_{1/\beta \le i \le t} \frac{(1-\beta)^{-i}}{1-(1-\beta)^i} \right)$$

$$\le \beta(1-\beta)^t \left( \frac{2e^2}{\beta}\left(1 + \ln\frac{1}{\beta}\right) + \frac{e(1-\beta)^{-t}}{(e-1)\beta} \right)$$

$$= 2e^2(1-\beta)^t \left(1 + \ln\frac{1}{\beta}\right) + \frac{e}{e-1}$$

$$\le 2e^2 \left(1 + \ln\frac{1}{\beta}\right) + \frac{e}{e-1}$$

$$\le 32 + 16\ln\frac{1}{\beta}.$$

□

Finally, we provide a useful lemma regarding the time-dependent re-scaled momentum parameters in (8) and Algorithm 4 for upper-level analysis.

**Lemma B.7** ((Li et al., 2023a, Lemma C.3)). *Let $\alpha_t = \frac{\beta}{1-(1-\beta)^t}$, then for all $T \ge 2$, we have*

$$\sum_{t=2}^{T} \alpha_t^2 \le 3(1 + \beta^2 T).$$

### B.2 Probabilistic Lemmas

In this section, we provide a well-known probabilistic lemma without proof.

**Lemma B.8** (Optional Stopping Theorem). *Let $\{Z_t\}_{t \ge 1}$ be a martingale with respect to a filtration $\{\mathcal{F}_t\}_{t \ge 0}$. Let $\tau$ be a bounded stopping time with respect to the same filtration. Then we have $\mathbb{E}[Z_\tau] = \mathbb{E}[Z_0]$.*

### B.3 Auxiliary Lemmas for Bilevel Optimization

In this section, we provide several useful lemmas for bilevel optimization under the unbounded smoothness setting, including the properties of the objective function $\Phi$ (Appendix B.3.1), the Neumann series approximation error (Appendix B.3.2), and the hypergradient estimation error (Appendix B.3.3).

#### B.3.1 Properties of the Objective Function

**Lemma B.9** ((Hao et al., 2024, Lemma 8)). *Under Assumption 3.2, we have*

*(I) $y^*(x)$ is $(l_{g,1}/\mu)$-Lipschitz continuous.*

*(II) $\|\nabla_x f(x, y^*(x))\| \le \|\nabla\Phi(x)\| + l_{g,1}l_{f,0}/\mu$.*

**Lemma B.10** (($L_0, L_1$)-smoothness (Hao et al., 2024, Lemma 9)). *Under Assumption 3.2, for any $x, x' \in \mathbb{R}^{d_x}$ we have*

$$\|\nabla\Phi(x) - \nabla\Phi(x')\| \le (L_0 + L_1\|\nabla\Phi(x')\|)\|x - x'\|$$

$$if \quad \|x - x'\| \le r := \frac{1}{\sqrt{(1 + l_{g,1}^2/\mu^2)(L_{x,1}^2 + L_{y,1}^2)}},$$

(13)

*where the $(L_0, L_1)$-smoothness constants $L_0$ and $L_1$ are defined as*

$$L_0 = \sqrt{1 + \frac{l_{g,1}^2}{\mu^2}} \left( L_{x,0} + L_{x,1} \frac{l_{g,1} l_{f,0}}{\mu} + \frac{l_{g,1}}{\mu} (L_{y,0} + L_{y,1} l_{f,0}) + l_{f,0} \frac{l_{g,1} l_{g,2} + \mu l_{g,2}}{\mu^2} \right),$$

$$L_1 = \sqrt{1 + \frac{l_{g,1}^2}{\mu^2}} L_{x,1}. \tag{14}$$

**Lemma B.11** (Descent Inequality (Hao et al., 2024, Lemma 10))**.** *Under Assumption 3.2, for any* $x, x' \in \mathbb{R}^{d_x}$ *we have*

$$\Phi(x) \leq \Phi(x') + \langle \nabla \Phi(x'), x - x' \rangle + \frac{L_0 + L_1 \|\nabla \Phi(x')\|}{2} \|x - x'\|^2$$

$$if \quad \|x - x'\| \leq r = \frac{1}{\sqrt{(1 + l_{g,1}^2/\mu^2)(L_{x,1}^2 + L_{y,1}^2)}}.$$

### B.3.2 NEUMANN SERIES APPROXIMATION

Throughout the paper, for given $(x, y) \in \mathbb{R}^{d_x} \times \mathbb{R}^{d_y}$, we estimate the hypergradient $\nabla \Phi(x)$ using Neumann series approach and the following formulation:

$$\hat{\nabla}\phi(x, y; \bar{\xi}) = \nabla_x F(x, y; \xi) - \nabla_{xy}^2 G(x, y; \zeta^{(0)}) \left[ \frac{1}{l_{g,1}} \sum_{q=0}^{Q-1} \prod_{j=1}^{q} \left( I - \frac{\nabla_{yy}^2 G(x, y; \zeta^{(q,j)})}{l_{g,1}} \right) \right] \nabla_y F(x, y; \xi),$$

where the randomness $\bar{\xi}$ is defined as

$$\bar{\xi} := \{\xi, \zeta^{(0)}, \bar{\zeta}^{(0)}, \dots, \bar{\zeta}^{(Q-1)}\}, \qquad \text{with} \quad \bar{\zeta}^{(q)} := \{\zeta^{(q,1)}, \dots, \zeta^{(q,q)}\}.$$

For simplicity, denote $P$ as the Neumann series approximation matrix for the Hessian inverse, then $P$ and $\mathbb{E}_{\bar{\xi}}[P]$ can be written as:

$$P = \frac{1}{l_{g,1}} \sum_{q=0}^{Q-1} \prod_{j=1}^{q} \left( I - \frac{\nabla_{yy}^2 G(x, y; \zeta^{(q,j)})}{l_{g,1}} \right) \quad \text{and} \quad \mathbb{E}_{\bar{\xi}}[P] = \frac{1}{l_{g,1}} \sum_{q=0}^{Q-1} \left( I - \frac{\nabla_{yy}^2 G(x, y)}{l_{g,1}} \right)^q. \tag{15}$$

Hence the simplified version of the hypergradient estimator and its expectation are

$$\hat{\nabla}\phi(x, y; \bar{\xi}) = \nabla_x F(x, y; \xi) - \nabla_{xy}^2 G(x, y; \zeta^{(0)}) P \nabla_y F(x, y; \xi),$$

$$\mathbb{E}_{\bar{\xi}}[\hat{\nabla}\phi(x, y; \bar{\xi})] = \nabla_x f(x, y) - \nabla_{xy}^2 g(x, y) \mathbb{E}_{\bar{\xi}}[P] \nabla_y f(x, y). \tag{16}$$

Also, we define $\bar{\nabla} f(x, y)$ as

$$\bar{\nabla} f(x, y) = \nabla_x f(x, y) - \nabla_{xy}^2 g(x, y) [\nabla_{yy}^2 g(x, y)]^{-1} \nabla_y f(x, y),$$

which is useful for the following analysis.

The following lemma bounds the norm of the Neumann series approximation matrix $P$ and characterizes the approximation error for the Hessian inverse in expectation.

**Lemma B.12.** *Under Assumptions 3.2 to 3.4, we have*

$$\|\mathbb{E}_{\bar{\xi}}[P]\| \leq \|P\| \leq \frac{1}{\mu} \qquad and \qquad \|\mathbb{E}_{\bar{\xi}}[P] - [\nabla_{yy}^2 g(x, y)]^{-1}\| \leq \frac{1}{\mu} \left( 1 - \frac{\mu}{l_{g,1}} \right)^Q.$$

*Proof of Lemma B.12.* We follow the similar proof as in (Ghadimi & Wang, 2018, Lemma 3.2). By Assumption 3.4 and definition of $P$ in (15), for any $Q \geq 1$ we have

$$\|\mathbb{E}_{\bar{\xi}}[P]\| \leq \|P\| = \left\| \frac{1}{l_{g,1}} \sum_{q=0}^{Q-1} \prod_{j=1}^{q} \left( I - \frac{\nabla_{yy}^2 G(x, y; \zeta^{(q,j)})}{l_{g,1}} \right) \right\| \leq \frac{1}{l_{g,1}} \sum_{q=0}^{Q-1} \left( 1 - \frac{\mu}{l_{g,1}} \right)^q \leq \frac{1}{\mu}.$$

As for the second result, we have

$$\|\mathbb{E}_{\bar{\xi}}[P] - [\nabla_{yy}^2 g(x,y)]^{-1}\| \leq \frac{1}{l_{g,1}} \left\| \sum_{q=Q}^{\infty} \left( I - \frac{\nabla_{yy}^2 G(x,y)}{l_{g,1}} \right)^q \right\|$$

$$\leq \frac{1}{l_{g,1}} \sum_{q=Q}^{\infty} \left\| \left( I - \frac{\nabla_{yy}^2 G(x,y)}{l_{g,1}} \right) \right\|^q \leq \frac{1}{\mu} \left( 1 - \frac{\mu}{l_{g,1}} \right)^Q.$$

$\square$

### B.3.3 HYPERGRADIENT ESTIMATION ERROR

**Lemma B.13.** *Under Assumptions 3.2 to 3.4, if $\|y - y^*(x)\| \leq r$, we have*

$$\|\hat{\nabla}\phi(x,y;\bar{\xi}) - \mathbb{E}_{\bar{\xi}}[\hat{\nabla}\phi(x,y;\bar{\xi})]\|$$
$$\leq \frac{\mu + 3l_{g,1} + \sigma_{g,2}}{\mu}\sigma_f + \frac{2l_{g,1} + \sigma_{g,2}}{\mu}l_{f,0} + \frac{2l_{g,1} + \sigma_{g,2}}{\mu}(L_{y,0} + L_{y,1}l_{f,0})\|y - y^*(x)\|.$$

*Proof of Lemma B.13.* We will use a short hand $y^* = y^*(x)$. By triangle inequality, we have

$$\|\hat{\nabla}\phi(x,y;\bar{\xi}) - \mathbb{E}_{\bar{\xi}}[\hat{\nabla}\phi(x,y;\bar{\xi})]\|$$
$$= \|(\nabla_x F(x,y;\xi) - \nabla_{xy}^2 G(x,y;\zeta^{(0)})P\nabla_y F(x,y;\xi)) - (\nabla_x f(x,y) - \nabla_{xy}^2 g(x,y)\mathbb{E}_{\bar{\xi}}[P]\nabla_y f(x,y))\|$$
$$\leq \underbrace{\|\nabla_x F(x,y;\xi) - \nabla_x f(x,y)\|}_{(A_1)} + \underbrace{\|(\nabla_{xy}^2 G(x,y;\zeta^{(0)}) - \nabla_{xy}^2 g(x,y))P\nabla_y F(x,y;\xi)\|}_{(A_2)}$$
$$+ \underbrace{\|\nabla_{xy}^2 g(x,y)(P - \mathbb{E}_{\bar{\xi}}[P])\nabla_y F(x,y;\xi)\|}_{(A_3)} + \underbrace{\|\nabla_{xy}^2 g(x,y)\mathbb{E}_{\bar{\xi}}[P](\nabla_y F(x,y;\xi) - \nabla_y f(x,y))\|}_{(A_4)}$$

**Bounding $(A_1)$.** By Assumption 3.3, we have

$$(A_1) = \|\nabla_x F(x,y;\xi) - \nabla_x f(x,y)\| \leq \sigma_f.$$

**Bounding $(A_2)$.** By Assumptions 3.2 and 3.3 and Lemma B.12, we have

$$(A_2) = \|(\nabla_{xy}^2 G(x,y;\zeta^{(0)}) - \nabla_{xy}^2 g(x,y))P\nabla_y F(x,y;\xi)\|$$
$$\leq \|\nabla_{xy}^2 G(x,y;\zeta^{(0)}) - \nabla_{xy}^2 g(x,y)\|\|P\|\|\nabla_y F(x,y;\xi)\|$$
$$\leq \frac{\sigma_{g,2}}{\mu}(\|\nabla_y F(x,y;\xi) - \nabla_y f(x,y)\| + \|\nabla_y f(x,y) - \nabla_y f(x,y^*)\| + \|\nabla_y f(x,y^*)\|)$$
$$\leq \frac{\sigma_{g,2}}{\mu}(\sigma_f + (L_{y,0} + L_{y,1}l_{f,0})\|y - y^*\| + l_{f,0})$$
$$= \frac{\sigma_{g,2}}{\mu}(\sigma_f + l_{f,0}) + \frac{\sigma_{g,2}}{\mu}(L_{y,0} + L_{y,1}l_{f,0})\|y - y^*\|.$$

**Bounding $(A_3)$.** By Assumptions 3.2 and 3.3 and Lemma B.12, we have

$$(A_3) = \|\nabla_{xy}^2 g(x,y)(P - \mathbb{E}_{\bar{\xi}}[P])\nabla_y F(x,y;\xi)\|$$
$$\leq \|\nabla_{xy}^2 g(x,y)\|\|(P - \mathbb{E}_{\bar{\xi}}[P])\|\|\nabla_y F(x,y;\xi)\|$$
$$\leq \frac{2l_{g,1}}{\mu}(\sigma_f + (L_{y,0} + L_{y,1}l_{f,0})\|y - y^*\| + l_{f,0})$$
$$= \frac{2l_{g,1}}{\mu}(\sigma_f + l_{f,0}) + \frac{2l_{g,1}}{\mu}(L_{y,0} + L_{y,1}l_{f,0})\|y - y^*\|,$$

where the second inequality uses the same step (the third inequality above) as in bounding $(A_2)$.

**Bounding** $(A_4)$.   By Assumptions 3.2 and 3.3 and Lemma B.12, we have

$$(A_4) = \|\nabla_{xy}^2 g(x,y)\mathbb{E}_{\bar{\xi}}[P](\nabla_y F(x,y;\xi) - \nabla_y f(x,y))\| \le \frac{l_{g,1}}{\mu}\sigma_f.$$

Then we obtain the final bound

$$\|\hat{\nabla}\phi(x,y;\bar{\xi}) - \mathbb{E}_{\bar{\xi}}[\hat{\nabla}\phi(x,y;\bar{\xi})]\| \le (A_1) + (A_2) + (A_3) + (A_4)$$
$$\le \frac{\mu + 3l_{g,1} + \sigma_{g,2}}{\mu}\sigma_f + \frac{2l_{g,1} + \sigma_{g,2}}{\mu}l_{f,0} + \frac{2l_{g,1} + \sigma_{g,2}}{\mu}(L_{y,0} + L_{y,1}l_{f,0})\|y - y^*\|.$$

$\square$

**Lemma B.14.** *Under Assumptions 3.2 to 3.4, if $\|y - y^*(x)\| \le r$, we have*

$$\|\hat{\nabla}\phi(x,y;\bar{\xi}) - \nabla\Phi(x)\| \le C_{\phi,0} + (C_{\phi,1} + L_1\|\nabla\Phi(x)\|)\|y - y^*(x)\|,$$

*where $L_1$ is defined in* (14) *and constants $C_{\phi,0}$ and $C_{\phi,1}$ are defined as*

$$C_{\phi,0} = \frac{\mu + 3l_{g,1} + \sigma_{g,2}}{\mu}\sigma_f + \frac{2l_{g,1} + \sigma_{g,2}}{\mu}l_{f,0} + \frac{l_{g,1}l_{f,0}}{\mu},$$
$$C_{\phi,1} = \frac{2l_{g,1} + \sigma_{g,2}}{\mu}(L_{y,0} + L_{y,1}l_{f,0}) + \frac{l_{g,1}}{\mu}(L_{y,0} + L_{y,1}l_{f,0}) + L_0.$$

$$(17)$$

*Proof of Lemma B.14.* We have the following decomposition:

$$\|\hat{\nabla}\phi(x,y;\bar{\xi}) - \nabla\Phi(x)\| \le \|\hat{\nabla}\phi(x,y;\bar{\xi}) - \mathbb{E}_{\bar{\xi}}[\hat{\nabla}\phi(x,y;\bar{\xi})]\|$$
$$+ \|\mathbb{E}_{\bar{\xi}}[\hat{\nabla}\phi(x,y;\bar{\xi})] - \bar{\nabla}f(x,y)\| + \|\bar{\nabla}f(x,y) - \nabla\Phi(x)\|,$$

For the first term, by Lemma B.13 we have

$$\|\hat{\nabla}\phi(x,y;\bar{\xi}) - \mathbb{E}_{\bar{\xi}}[\hat{\nabla}\phi(x,y;\bar{\xi})]\|$$
$$\le \frac{\mu + 3l_{g,1} + \sigma_{g,2}}{\mu}\sigma_f + \frac{2l_{g,1} + \sigma_{g,2}}{\mu}l_{f,0} + \frac{2l_{g,1} + \sigma_{g,2}}{\mu}(L_{y,0} + L_{y,1}l_{f,0})\|y - y^*\|.$$

$$(18)$$

For the second term, by Assumption 3.2 and Lemma B.12 we have

$$\|\mathbb{E}_{\bar{\xi}}[\hat{\nabla}\phi(x,y;\bar{\xi})] - \bar{\nabla}f(x,y)\|$$
$$= \|(\nabla_x f(x,y) - \nabla_{xy}^2 g(x,y)\mathbb{E}_{\bar{\xi}}[P]\nabla_y f(x,y))$$
$$- (\nabla_x f(x,y) - \nabla_{xy}^2 g(x,y)[\nabla_{yy}^2 g(x,y)]^{-1}\nabla_y f(x,y))\|$$
$$= \|\nabla_{xy}^2 g(x,y)(\mathbb{E}_{\bar{\xi}}[P] - [\nabla_{yy}^2 g(x,y)]^{-1})\nabla_y f(x,y)\|$$
$$\le \frac{l_{g,1}}{\mu}\left(1 - \frac{\mu}{l_{g,1}}\right)^Q (\|\nabla_y f(x,y) - \nabla_y f(x,y^*)\| + \|\nabla_y f(x,y)\|)$$
$$\le \frac{l_{g,1}}{\mu}\left(1 - \frac{\mu}{l_{g,1}}\right)^Q ((L_{y,0} + L_{y,1}l_{f,0})\|y - y^*\| + l_{f,0})$$
$$= \frac{l_{g,1}l_{f,0}}{\mu}\left(1 - \frac{\mu}{l_{g,1}}\right)^Q + \frac{l_{g,1}}{\mu}\left(1 - \frac{\mu}{l_{g,1}}\right)^Q (L_{y,0} + L_{y,1}l_{f,0})\|y - y^*\|.$$

$$(19)$$

For the third term, by Assumption 3.2 and Lemma B.9 we have

$$\|\bar{\nabla} f(x,y) - \nabla\Phi(x)\|$$

$$\leq \|\nabla_x f(x,y) - \nabla_x f(x,y^*)\|$$

$$\quad + \|\nabla_{xy}^2 g(x,y)[\nabla_{yy}^2 g(x,y)]^{-1}\nabla_y f(x,y) - \nabla_{xy}^2 g(x,y^*)[\nabla_{yy}^2 g(x,y^*)]^{-1}\nabla_y f(x,y^*)\|$$

$$\leq (L_{x,0} + L_{x,1}\|\nabla_x f(x,y^*)\|)\|y - y^*\|$$

$$\quad + \|\nabla_{xy}^2 g(x,y)[\nabla_{yy}^2 g(x,y)]^{-1}\nabla_y f(x,y) - \nabla_{xy}^2 g(x,y^*)[\nabla_{yy}^2 g(x,y)]^{-1}\nabla_y f(x,y)\|$$

$$\quad + \|\nabla_{xy}^2 g(x,y^*)[\nabla_{yy}^2 g(x,y)]^{-1}\nabla_y f(x,y) - \nabla_{xy}^2 g(x,y^*)[\nabla_{yy}^2 g(x,y^*)]^{-1}\nabla_y f(x,y)\|$$

$$\quad + \|\nabla_{xy}^2 g(x,y^*)[\nabla_{yy}^2 g(x,y^*)]^{-1}\nabla_y f(x,y) - \nabla_{xy}^2 g(x,y^*)[\nabla_{yy}^2 g(x,y^*)]^{-1}\nabla_y f(x,y^*)\|$$

$$\leq \left( L_{x,0} + L_{x,1}\left(\frac{l_{g,1}l_{f,0}}{\mu} + \|\nabla\Phi(x)\|\right)\right)\|y - y^*\|$$

$$\quad + \frac{l_{f,0}}{\mu}l_{g,2}\|y - y^*\| + \frac{l_{f,0}l_{g,1}}{\mu^2}l_{g,2}\|y - y^*\| + \frac{l_{g,1}}{\mu}(L_{y,0} + L_{y,1}\|\nabla_y f(x,y^*)\|)\|y - y^*\|$$

$$= \left( L_{x,0} + L_{x,1}\frac{l_{g,1}l_{f,0}}{\mu} + \frac{l_{g,1}}{\mu}(L_{y,0} + L_{y,1}l_{f,0}) + l_{f,0}\frac{\mu l_{g,2} + l_{g,1}l_{g,2}}{\mu^2} + L_{x,1}\|\nabla\Phi(x)\|\right)\|y - y^*\|$$

$$\leq (L_0 + L_1\|\nabla\Phi(x)\|)\|y - y^*\|,$$

$$(20)$$

where the last inequality uses the definition of $L_0$ and $L_1$ as in (14). Summing up $(18) + (19) + (20)$ gives the final bound

$$\|\hat{\nabla}\phi(x,y;\bar{\xi}) - \nabla\Phi(x)\| \leq \|\hat{\nabla}\phi(x,y;\bar{\xi}) - \mathbb{E}_{\bar{\xi}}[\hat{\nabla}\phi(x,y;\bar{\xi})]\|$$

$$\quad\quad\quad\quad\quad\quad\quad + \|\mathbb{E}_{\bar{\xi}}[\hat{\nabla}\phi(x,y;\bar{\xi})] - \bar{\nabla} f(x,y)\| + \|\bar{\nabla} f(x,y) - \nabla\Phi(x)\|$$

$$\leq \frac{\mu + 3l_{g,1} + \sigma_{g,2}}{\mu}\sigma_f + \frac{2l_{g,1} + \sigma_{g,2}}{\mu}l_{f,0} + \frac{l_{g,1}l_{f,0}}{\mu}\left(1 - \frac{\mu}{l_{g,1}}\right)^Q$$

$$\quad + \left(\frac{2l_{g,1} + \sigma_{g,2}}{\mu}(L_{y,0} + L_{y,1}l_{f,0}) + \frac{l_{g,1}}{\mu}\left(1 - \frac{\mu}{l_{g,1}}\right)^Q(L_{y,0} + L_{y,1}l_{f,0}) + L_0 + L_1\|\nabla\Phi(x)\|\right)\|y - y^*\|$$

$$\leq \frac{\mu + 3l_{g,1} + \sigma_{g,2}}{\mu}\sigma_f + \frac{2l_{g,1} + \sigma_{g,2}}{\mu}l_{f,0} + \frac{l_{g,1}l_{f,0}}{\mu}$$

$$\quad + \left(\frac{2l_{g,1} + \sigma_{g,2}}{\mu}(L_{y,0} + L_{y,1}l_{f,0}) + \frac{l_{g,1}}{\mu}(L_{y,0} + L_{y,1}l_{f,0}) + L_0 + L_1\|\nabla\Phi(x)\|\right)\|y - y^*\|$$

$$= C_{\phi,0} + (C_{\phi,1} + L_1\|\nabla\Phi(x)\|)\|y - y^*\|,$$

where the second and the third inequalities use $Q \geq 1$, and the last inequality is due to the definitions of $C_{\phi,0}$ and $C_{\phi,1}$ in (17). $\qquad\square$

### B.3.4 OTHER USEFUL LEMMAS

**Lemma B.15.** *Under Assumptions 3.2 to 3.4, if $\|y - y^*(x)\| \leq r$, we have* [1]

$$\|\hat{\nabla}\phi(x,y;\bar{\xi}) - \hat{\nabla}\phi(x,y^*(x);\bar{\xi})\| \leq (L_0 + L_1\|\nabla\Phi(x)\|)\|y - y^*(x)\|;$$

*if $\|x_1 - x_2\| \leq \mu r/(\mu + l_{g,1})$, we have*

$$\|\mathbb{E}_{\bar{\xi}_1}[\hat{\nabla}\phi(x_1,y_1^*;\bar{\xi}_1)] - \mathbb{E}_{\bar{\xi}_2}[\hat{\nabla}\phi(x_2,y_2^*;\bar{\xi}_2)]\| \leq (L_0 + L_1\|\nabla\Phi(x_1)\|)\|x_1 - x_2\|,$$

*where $y_i^* = y^*(x_i)$ for $i = 1, 2$, and constants $L_0$ and $L_1$ are defined in (14).*

*Proof of Lemma B.15.* We will use a short hand $y^* = y^*(x)$. Recall the definition of $\hat{\nabla}\phi(x,y;\bar{\xi})$ and $\hat{\nabla}\phi(x,y^*;\bar{\xi})$ in (16), we have

$$\hat{\nabla}\phi(x,y;\bar{\xi}) = \nabla_x F(x,y;\xi) - \nabla_{xy}^2 G(x,y;\zeta^{(0)}) P \nabla_y F(x,y;\xi),$$

$$\hat{\nabla}\phi(x,y^*;\bar{\xi}) = \nabla_x F(x,y^*;\xi) - \nabla_{xy}^2 G(x,y^*;\zeta^{(0)}) P^* \nabla_y F(x,y^*;\xi).$$

---

[1] Please note that $x_1$ and $x_2$ here are unrelated to Algorithm 1 and are deterministic.

where similar to (15), we define the Neumann series approximation matrix $P^*$ as

$$P^* = \frac{1}{l_{g,1}} \sum_{q=0}^{Q-1} \prod_{j=1}^{q} \left( I - \frac{\nabla_{yy}^2 G(x, y^*; \zeta^{(q,j)})}{l_{g,1}} \right). \tag{21}$$

Then by triangle inequality we have

$$\|\hat{\nabla}\phi(x, y; \bar{\xi}) - \hat{\nabla}\phi(x, y^*; \bar{\xi})\|$$
$$\leq \|\nabla_x F(x, y; \xi) - \nabla_x F(x, y^*; \xi)\|$$
$$\quad + \|\nabla_{xy}^2 G(x, y; \zeta^{(0)}) P \nabla_y F(x, y; \xi) - \nabla_{xy}^2 G(x, y^*; \zeta^{(0)}) P^* \nabla_y F(x, y^*; \xi)\|$$
$$\leq \underbrace{\|\nabla_x F(x, y; \xi) - \nabla_x F(x, y^*; \xi)\|}_{(A_1)} + \underbrace{\|\nabla_{xy}^2 G(x, y; \zeta^{(0)}) P (\nabla_y F(x, y; \xi) - \nabla_y F(x, y^*; \xi))\|}_{(A_2)}$$
$$\quad + \underbrace{\|\nabla_{xy}^2 G(x, y; \zeta^{(0)})(P - P^*)\nabla_y F(x, y^*; \xi)\|}_{(A_3)}$$
$$\quad + \underbrace{\|(\nabla_{xy}^2 G(x, y; \zeta^{(0)}) - \nabla_{xy}^2 G(x, y^*; \zeta^{(0)}))P^* \nabla_y F(x, y^*; \xi)\|}_{(A_4)}.$$

**Bounding $(A_1)$.** By Assumption 3.4 and Lemma B.9, we have

$$(A_1) = \|\nabla_x F(x, y; \xi) - \nabla_x F(x, y^*; \xi)\| \leq (L_{x,0} + L_{x,1}\|\nabla_x f(x, y^*)\|)\|y - y^*\|$$
$$\leq \left( L_{x,0} + L_{x,1} \left( \frac{l_{g,1} l_{f,0}}{\mu} + \|\nabla\Phi(x)\| \right) \right) \|y - y^*\|$$
$$= \left( L_{x,0} + \frac{L_{x,1} l_{g,1} l_{f,0}}{\mu} + L_{x,1}\|\nabla\Phi(x)\| \right) \|y - y^*\|.$$

**Bounding $(A_2)$.** By Assumption 3.4 and Lemma B.12, we have

$$(A_2) = \|\nabla_{xy}^2 G(x, y; \zeta^{(0)}) P (\nabla_y F(x, y; \xi) - \nabla_y F(x, y^*; \xi))\|$$
$$= \|\nabla_{xy}^2 G(x, y; \zeta^{(0)})\|\|P\|\|\nabla_y F(x, y; \xi) - \nabla_y F(x, y^*; \xi)\|$$
$$\leq \frac{l_{g,1}}{\mu}(L_{y,0} + L_{y,1}\|\nabla_y f(x, y^*)\|)\|y - y^*\| \leq \frac{l_{g,1}}{\mu}(L_{y,0} + L_{y,1} l_{f,0})\|y - y^*\|.$$

**Bounding $(A_3)$.** We first apply Lemma B.1 to obtain

$$\left\| \prod_{j=1}^{q} \left( I - \frac{\nabla_{yy}^2 G(x, y; \zeta^{(q,j)})}{l_{g,1}} \right) - \prod_{j=1}^{q} \left( I - \frac{\nabla_{yy}^2 G(x, y^*; \zeta^{(q,j)})}{l_{g,1}} \right) \right\|$$
$$\leq \sum_{j=1}^{q} \left( 1 - \frac{\mu}{l_{g,1}} \right)^{q-1} \frac{l_{g,2}}{l_{g,1}}\|y - y^*\| = q \left( 1 - \frac{\mu}{l_{g,1}} \right)^{q-1} \frac{l_{g,2}}{l_{g,1}}\|y - y^*\|.$$

Hence we can write

$$\|P - P^*\| \leq \frac{1}{l_{g,1}} \sum_{q=0}^{Q-1} q \left( 1 - \frac{\mu}{l_{g,1}} \right)^{q-1} \frac{l_{g,2}}{l_{g,1}}\|y - y^*\| \leq \frac{\mu^2 l_{g,2}}{l_{g,1}^4}\|y - y^*\| \leq \frac{l_{g,2}}{\mu^2}\|y - y^*\|,$$

where the second inequality uses Lemma B.2 with $a = \mu/l_{g,1}$, and the last inequality is due to $\mu < l_{g,1}$. Then by Assumption 3.4 we have

$$(A_3) = \|\nabla_{xy}^2 G(x, y; \zeta^{(0)})(P - P^*)\nabla_y F(x, y^*; \xi)\|$$
$$\leq \|\nabla_{xy}^2 G(x, y; \zeta^{(0)})\|\|(P - P^*)\|\|\nabla_y F(x, y^*; \xi)\| \leq \frac{l_{g,1} l_{g,2} l_{f,0}}{\mu^2}\|y - y^*\|.$$

**Bounding $(A_4)$.** By Assumption 3.4 and Lemma B.12, we have

$$(A_4) = \|(\nabla_{xy}^2 G(x,y;\zeta^{(0)}) - \nabla_{xy}^2 G(x,y^*;\zeta^{(0)}))P^* \nabla_y F(x,y^*;\xi)\|$$

$$\leq \|\nabla_{xy}^2 G(x,y;\zeta^{(0)}) - \nabla_{xy}^2 G(x,y^*;\zeta^{(0)})\|\|P^*\|\|\nabla_y F(x,y^*;\xi)\| \leq \frac{l_{g,2}l_{f,0}}{\mu}\|y - y^*\|.$$

**Final Bound.** Summing up $(A_1) + (A_2) + (A_3) + (A_4)$ yields the final bound

$$\|\hat{\nabla}\phi(x,y;\bar{\xi}) - \hat{\nabla}\phi(x,y^*;\bar{\xi})\| \leq (A_1) + (A_2) + (A_3) + (A_4)$$

$$\leq \left( L_{x,0} + L_{x,1}\frac{l_{g,1}l_{f,0}}{\mu} + \frac{l_{g,1}}{\mu}(L_{y,0} + L_{y,1}l_{f,0}) + l_{f,0}\frac{l_{g,1}l_{g,2} + \mu l_{g,2}}{\mu^2} + L_{x,1}\|\nabla\Phi(x)\| \right)\|y - y^*\|$$

$$\leq (L_0 + L_1\|\nabla\Phi(x)\|)\|y - y^*\|,$$

where the last inequality uses the definitions of $L_0$ and $L_1$ as in (14).

For the second result, we follow a similar procedure as above and obtain:

$$\|\mathbb{E}_{\bar{\xi}_1}[\hat{\nabla}\phi(x_1,y_1^*;\bar{\xi}_1)] - \mathbb{E}_{\bar{\xi}_2}[\hat{\nabla}\phi(x_2,y_2^*;\bar{\xi}_2)]\| \leq (A_1) + (A_2) + (A_3) + (A_4)$$

$$\leq \sqrt{1 + \frac{l_{g,1}^2}{\mu^2}} \left( L_{x,0} + L_{x,1}\frac{l_{g,1}l_{f,0}}{\mu} + \frac{l_{g,1}}{\mu}(L_{y,0} + L_{y,1}l_{f,0}) + l_{f,0}\frac{l_{g,1}l_{g,2} + \mu l_{g,2}}{\mu^2} + L_{x,1}\|\nabla\Phi(x_1)\| \right)\|x_1 - x_2\|$$

$$= (L_0 + L_1\|\nabla\Phi(x_1)\|)\|x_1 - x_2\|,$$

where the last inequality uses the definitions of $L_0$ and $L_1$ as in (14). $\qquad\square$

**Lemma B.16.** *Under Assumptions 3.2 to 3.4, we have*

$$\|\mathbb{E}_{\bar{\xi}}[\hat{\nabla}\phi(x,y^*(x);\bar{\xi})] - \nabla\Phi(x)\| \leq \frac{l_{g,1}l_{f,0}}{\mu}\left( 1 - \frac{\mu}{l_{g,1}} \right)^Q.$$

*Proof of Lemma B.16.* We will use a short hand $y^* = y^*(x)$. By definition of $\hat{\nabla}\phi(x,y;\bar{\xi})$ in (16) and the hypergradient formulation, we have

$$\mathbb{E}_{\bar{\xi}}[\hat{\nabla}\phi(x,y^*;\bar{\xi})] = \nabla_x f(x,y^*) - \nabla_{xy}^2 g(x,y^*)\mathbb{E}_{\bar{\xi}}[P]\nabla_y f(x,y^*),$$

$$\nabla\Phi(x) = \nabla_x f(x,y^*) - \nabla_{xy}^2 g(x,y^*)[\nabla_{yy}^2 g(x,y^*)]^{-1}\nabla_y f(x,y^*).$$

Then we obtain the conclusion by applying Assumption 3.2 and Lemma B.12:

$$\|\mathbb{E}_{\bar{\xi}}[\hat{\nabla}\phi(x,y;\bar{\xi})] - \nabla\Phi(x)\| = \|\nabla_{xy}^2 g(x,y^*)(\mathbb{E}_{\bar{\xi}}[P] - [\nabla_{yy}^2 g(x,y^*)]^{-1})\nabla_y f(x,y^*)\|$$

$$\leq \|\nabla_{xy}^2 g(x,y^*)\|\|\mathbb{E}_{\bar{\xi}}[P] - [\nabla_{yy}^2 g(x,y^*)]^{-1}\|\|\nabla_y f(x,y^*)\| \leq \frac{l_{g,1}l_{f,0}}{\mu}\left( 1 - \frac{\mu}{l_{g,1}} \right)^Q.$$

$\qquad\square$

# C PROOF OF THE RANDOM DECOUPLING LEMMA (LEMMA 4.2)

## C.1 RECURSIVE CONTROL ON MOMENT GENERATING FUNCTION

The following technical lemma on recursive control is crucial for establishing high probability guarantee for controlling the lower-level estimation error at anytime. We follow a similar argument as in (Cutler et al., 2023, Proposition 29) with a slight generalization.

**Proposition C.1** (Recursive control on MGF). *Consider scalar stochastic processes $(V_t)$, $(D_t)$, $(X_t)$ and $(Y_t)$ on a probability space with filtration $(\mathcal{H}_t)$, which are linked by the inequality*

$$V_{t+1} \leq \rho_t V_t + D_t\sqrt{V_t} + X_t + Y_t + \kappa_t \tag{22}$$

*for some deterministic constants $\rho_t \in (-\infty, 1]$ and $\kappa_t \in \mathbb{R}$. Suppose the following properties hold.*

- *$V_t$ and $Y_t$ are non-negative and $\mathcal{H}_t$-measurable.*

- $D_t$ is mean-zero sub-Gaussian conditioned on $\mathcal{H}_t$ with deterministic parameter $\sigma_t$:
$$\mathbb{E}[\exp(\theta D_t) \mid \mathcal{H}_t] \leq \exp(\theta^2 \sigma_t^2 / 2) \quad \text{for all} \quad \theta \in \mathbb{R}.$$

- $X_t$ is non-negative and sub-exponential conditioned on $\mathcal{H}_t$ with deterministic parameter $\nu_t$:
$$\mathbb{E}[\exp(\theta X_t) \mid \mathcal{H}_t] \leq \exp(\theta \nu_t) \quad \text{for all} \quad 0 \leq \theta \leq 1/\nu_t.$$

*Then the estimate*

$$\mathbb{E}[\exp(\theta V_{t+1})] \leq \exp(\theta(\nu_t + \kappa_t))\mathbb{E}[\exp(\theta((1 + \rho_t)V_t/2 + Y_t))]$$

*holds for any $\theta$ satisfying $0 \leq \theta \leq \min\left\{\frac{1-\rho_t}{2\sigma_t^2}, \frac{1}{2\nu_t}\right\}$.*

*Proof of Proposition C.1.* For any index $t \geq 0$ and any scalar $\theta \geq 0$, the law of total expectation implies

$$\mathbb{E}[\exp(\theta V_{t+1})] \leq \mathbb{E}\left[\exp\left(\theta\left(\rho_t V_t + D_t\sqrt{V_t} + X_t + Y_t + \kappa_t\right)\right)\right]$$
$$= \exp(\theta \kappa_t)\mathbb{E}\left[\exp(\theta(\rho_t V_t + Y_t))\mathbb{E}\left[\exp(\theta D_t\sqrt{V_t})\exp(\theta X_t) \mid \mathcal{H}_t\right]\right].$$

Hölder's inequality in turn yields

$$\mathbb{E}\left[\exp(\theta D_t\sqrt{V_t})\exp(\theta X_t) \mid \mathcal{H}_t\right] \leq \sqrt{\mathbb{E}\left[\exp(2\theta D_t\sqrt{V_t}) \mid \mathcal{H}_t\right] \cdot \mathbb{E}\left[\exp(2\theta X_t) \mid \mathcal{H}_t\right]}$$
$$\leq \sqrt{\exp(2\theta^2 \sigma_t^2 V_t)\exp(2\theta \nu_t)}$$
$$= \exp(\theta^2 \sigma_t^2 V_t)\exp(\theta \nu_t)$$

provided $0 \leq \theta \leq \frac{1}{2\nu_t}$. Therefore, if $\theta$ satisfies

$$0 \leq \theta \leq \min\left\{\frac{1-\rho_t}{2\sigma_t^2}, \frac{1}{2\nu_t}\right\},$$

then the following estimate holds for all $t \geq 0$:

$$\mathbb{E}[\exp(\theta V_{t+1})] \leq \exp(\theta \kappa_t)\mathbb{E}\left[\exp(\theta(\rho_t V_t + Y_t))\exp(\theta^2 \sigma_t^2 V_t)\exp(\theta \nu_t)\right]$$
$$= \exp(\theta(\nu_t + \kappa_t))\mathbb{E}\left[\exp(\theta((\rho_t + \theta \sigma_t^2)V_t + Y_t))\right]$$
$$\leq \exp(\theta(\nu_t + \kappa_t))\mathbb{E}\left[\exp(\theta((1 + \rho_t)V_t/2 + Y_t))\right],$$

where the last inequality uses the given range of $\theta$. Thus the proof is completed. $\qquad\square$

## C.2 PROOF OF LEMMA 4.2

In this section, we aim to provide a high-probability guarantee for the approximation error of the lower-level variable, namely $\|y_t - y_t^*\|$. Our main technical contribution is the any-sequence argument, which separates the randomness in the updates of the upper-level variable $x_t$ and the lower-level variable $y_t$. Specifically, for any given sequence $\{\tilde{x}_t\}$, we consider the following update rule for $\{\tilde{y}_t\}$ (which is the same as line 5 of Algorithm 1):

$$\tilde{y}_{t+1} = \tilde{y}_t - \gamma \nabla_y G(\tilde{x}_t, \tilde{y}_t; \tilde{\zeta}_t). \tag{23}$$

Before proceeding, we will first define (or restate) a few key concepts and useful notations.

**Filtration.** For any $t \geq 2$, define $\tilde{\mathcal{F}}_t^y$ as the filtration of the randomness used in updating $\tilde{y}_t$ before the $t$-th iteration:
$$\tilde{\mathcal{F}}_t^y = \sigma(\tilde{\zeta}_1, \ldots, \tilde{\zeta}_{t-1}), \tag{24}$$
where $\sigma(\cdot)$ denotes the $\sigma$-algebra generated by the random variables within the argument.

**Auxiliary Sequence.** We also introduce the following auxiliary sequence $\{\tilde{u}_t\}$ for our analysis:

$$\tilde{u}_t = (1 - \alpha_t)\tilde{u}_{t-1} + \alpha_t \hat{\nabla}\phi(\tilde{x}_t, \tilde{y}_t^*; \hat{\xi}_t) = \sum_{j=1}^{t} d_{t,j} \hat{\nabla}\phi(\tilde{x}_t, \tilde{y}_t^*; \hat{\xi}_t), \tag{25}$$

where the sequence $\{d_{t,j}\}_{j=1}^{t}$ is defined in (10) of Lemma B.3. Similar to (15), (16) and (21) in Appendix B, the hypergradient estimators $\hat{\nabla}\phi(\tilde{x}_t, \tilde{y}_t; \hat{\xi}_t)$ and $\hat{\nabla}\phi(\tilde{x}_t, \tilde{y}_t^*; \hat{\xi}_t)$ can be written as

$$\hat{\nabla}\phi(\tilde{x}_t, \tilde{y}_t; \hat{\xi}_t) = \nabla_x F(\tilde{x}_t, \tilde{y}_t; \tilde{\xi}_t) - \nabla_{xy}^2 G(\tilde{x}_t, \tilde{y}_t; \tilde{\zeta}_t^{(0)}) \tilde{P}_t \nabla_y F(\tilde{x}_t, \tilde{y}_t; \tilde{\xi}_t),$$

$$\hat{\nabla}\phi(\tilde{x}_t, \tilde{y}_t^*; \hat{\xi}_t) = \nabla_x F(\tilde{x}_t, \tilde{y}_t^*; \tilde{\xi}_t) - \nabla_{xy}^2 G(\tilde{x}_t, \tilde{y}_t^*; \tilde{\zeta}_t^{(0)}) \tilde{P}_t^* \nabla_y F(\tilde{x}_t, \tilde{y}_t^*; \tilde{\xi}_t),$$

where the randomness $\hat{\xi}_t$ is defined as

$$\hat{\xi}_t := \{\tilde{\xi}_t, \tilde{\zeta}_t^{(0)}, \tilde{\bar{\zeta}}^{(0)}, \dots, \tilde{\bar{\zeta}}^{(Q-1)}\}, \qquad \text{where} \quad \tilde{\bar{\zeta}}^{(q)} := \{\tilde{\zeta}^{(q,1)}, \dots, \tilde{\zeta}^{(q,q)}\}; \tag{26}$$

and the Neumann series approximation matrices $\tilde{P}_t$ and $\tilde{P}_t^*$ are defined as

$$\tilde{P}_t = \frac{1}{l_{g,1}} \sum_{q=0}^{Q-1} \prod_{j=1}^{q} \left( I - \frac{\nabla_{yy}^2 G(\tilde{x}_t, \tilde{y}_t; \tilde{\zeta}_t^{(q,j)})}{l_{g,1}} \right) \quad \text{and} \quad \tilde{P}_t^* = \frac{1}{l_{g,1}} \sum_{q=0}^{Q-1} \prod_{j=1}^{q} \left( I - \frac{\nabla_{yy}^2 G(\tilde{x}_t, \tilde{y}_t^*; \tilde{\zeta}_t^{(q,j)})}{l_{g,1}} \right).$$

**Constants.** We define the following constants, which will be useful for analysis. Given any sequence $\{\tilde{x}_t\}$, denote $\tilde{G}_t$ and $\tilde{L}_t$ as

$$\tilde{G}_t := \max_{1 \leq k \leq t} \|\nabla\Phi(\tilde{x}_k)\|, \quad \tilde{L}_t := L_0 + L_1 \tilde{G}_t, \tag{27}$$

where constants $L_0$ and $L_1$ are defined in (14).

**Lemma C.2** (Distance recursion, (Cutler et al., 2023, Lemma 25)). *Suppose that Assumptions 3.2 and 3.3 hold. For any given sequence $\{\tilde{x}_t\}$, let $\{\tilde{y}_t\}$ be the iterates generated by the update rule (23) with constant learning rate $\gamma \leq 1/2l_{g,1}$. Then for any $t \geq 1$, we have the following recursion:*

$$\|\tilde{y}_{t+1} - \tilde{y}_{t+1}^*\|^2 \leq (1 - \mu\gamma)\|\tilde{y}_t - \tilde{y}_t^*\|^2 + 2\gamma\langle\tilde{\varepsilon}_t, \tilde{v}_t\rangle\|\tilde{y}_t - \tilde{y}_t^*\| + 2\gamma^2\|\tilde{\varepsilon}_t\|^2 + \frac{2}{\mu\gamma}D_t^2, \tag{28}$$

*where $\tilde{v}_t := \frac{\tilde{y}_t - \tilde{y}_t^*}{\|\tilde{y}_t - \tilde{y}_t^*\|}$ if $\tilde{y}_t$ is distinct from $\tilde{y}_t^*$ and zero otherwise, $\tilde{\varepsilon}_t = \nabla_y g(\tilde{x}_t, \tilde{y}_t) - \nabla_y G(\tilde{x}_t, \tilde{y}_t; \tilde{\zeta}_t)$ denotes the noise, and $D_t := \|\tilde{y}_t^* - \tilde{y}_{t+1}^*\|$ is the minimizer drift at time $t$.*

**Lemma C.3** (Restatement of Lemma 4.2). *Suppose that Assumptions 3.2 and 3.3 hold. Given any sequence $\{\tilde{x}_t\}$ and any randomness $\{\hat{\xi}_t\}$ (see (26) for definition) such that*

$$\|\tilde{x}_{t+1} - \tilde{x}_t\|^2 \leq \frac{2\eta^2}{\lambda^2} \left( \|\tilde{u}_t\|^2 + \tilde{L}_t^2 \sum_{j=1}^{t} d_{t,j}\|\tilde{y}_j - \tilde{y}_j^*\|^2 \right), \tag{29}$$

*where $\tilde{u}_t$, $\{d_{t,j}\}_{j=1}^{t}$ and $\tilde{L}_t$ are defined in (25), (10) and (27), respectively. Let $\{\tilde{y}_t\}$ be the iterates generated by the update rule (23) with constant learning rate $\gamma \leq 1/2l_{g,1}$, and choose $\gamma = 2\beta/\mu$. Then for any given $\delta \in (0,1)$ and all $t \geq 1$, the following estimate holds with probability at least $1 - \delta$ over the randomness in $\tilde{\mathcal{F}}_{T+1}^y$:*

$$\|\tilde{y}_t - \tilde{y}_t^*\|^2 \leq \left( \left(1 - \frac{\mu\gamma}{2}\right)^{t-1} + \frac{4\eta^2 l_{g,1}^2}{\lambda^2\mu^3\gamma} \sum_{i=1}^{t-1} \left(1 - \frac{\mu\gamma}{2}\right)^{t-1-i} \tilde{L}_i^2 \right) \|\tilde{y}_1 - \tilde{y}_1^*\|^2$$

$$+ \left( \frac{8\gamma}{\mu} \ln\frac{eT}{\delta} + \frac{16\eta^2 l_{g,1}^2}{\lambda^2\mu^4} \sum_{i=1}^{t-1} \left(1 - \frac{\mu\gamma}{2}\right)^{t-1-i} \tilde{L}_i^2 \right) \sigma_{g,1}^2 \tag{30}$$

$$+ \frac{4\eta^2 l_{g,1}^2}{\lambda^2\mu^3\gamma} \sum_{i=1}^{t-1} \left(1 - \frac{\mu\gamma}{2}\right)^{t-1-i} \|\tilde{u}_i\|^2 + \frac{64\eta^4 l_{g,1}^4}{\lambda^4\mu^8\gamma^4} \sum_{i=1}^{t-1} \left(1 - \frac{\mu\gamma}{2}\right)^{t-1-i} \alpha_i \tilde{L}_i^2 \|\tilde{u}_i\|^2.$$

*Proof of Lemma C.3.* By Lemma C.2 and Lemma B.9, we have

$$\|\tilde{y}_{t+1} - \tilde{y}_{t+1}^*\|^2 \leq (1 - \mu\gamma)\|\tilde{y}_t - \tilde{y}_t^*\|^2 + 2\gamma\langle\tilde{\varepsilon}_t, \tilde{v}_t\rangle\|\tilde{y}_t - \tilde{y}_t^*\| + 2\gamma^2\|\tilde{\varepsilon}_t\|^2 + \frac{2}{\mu\gamma}D_t^2$$

$$\leq (1 - \mu\gamma)\|\tilde{y}_t - \tilde{y}_t^*\|^2 + 2\gamma\langle\tilde{\varepsilon}_t, \tilde{v}_t\rangle\|\tilde{y}_t - \tilde{y}_t^*\| + 2\gamma^2\|\tilde{\varepsilon}_t\|^2 + \frac{2l_{g,1}^2}{\mu^3\gamma}\|\tilde{x}_{t+1} - \tilde{x}_t\|^2$$

$$\leq (1 - \mu\gamma)\|\tilde{y}_t - \tilde{y}_t^*\|^2 + 2\gamma\langle\tilde{\varepsilon}_t, \tilde{v}_t\rangle\|\tilde{y}_t - \tilde{y}_t^*\| + 2\gamma^2\|\tilde{\varepsilon}_t\|^2 \tag{31}$$

$$+ \frac{4\eta^2 l_{g,1}^2}{\lambda^2\mu^3\gamma}\left(\|\tilde{u}_t\|^2 + \tilde{L}_t^2\sum_{j=1}^t d_{t,j}\|\tilde{y}_j - \tilde{y}_j^*\|^2\right),$$

where the last inequality uses (29). Note that under Assumption 3.3, there exists an absolute constant $c \geq 1$ such that for all $t \geq 1$, $\|\tilde{\varepsilon}_t\|^2$ is sub-exponential conditioned on $\tilde{\mathcal{F}}_t^y$ with parameter $c\sigma_{g,1}^2$, and $\tilde{\varepsilon}_t$ is mean-zero sub-Gaussian conditioned on $\tilde{\mathcal{F}}_t^y$ with parameter $c\sigma_{g,1}$ (Cutler et al., 2023, Theorem 30). For simplicity we set $c = 1$ here. Thus $\langle\tilde{\varepsilon}_t, u_t\rangle$ is mean-zero sub-Gaussian conditioned on $\tilde{\mathcal{F}}_t^y$ with parameter $\sigma_{g,1}$. Hence, in light of (31), we apply Proposition C.1 with

$$\mathcal{H}_t = \tilde{\mathcal{F}}_t^y, \quad V_t = \|\tilde{y}_t - \tilde{y}_t^*\|^2, \quad D_t = 2\eta\langle\tilde{\varepsilon}_t, \tilde{v}_t\rangle, \quad X_t = 2\gamma^2\|\tilde{\varepsilon}_t\|^2,$$

$$Y_t = \frac{4\eta^2 l_{g,1}^2}{\lambda^2\mu^3\gamma}\tilde{L}_t^2\sum_{j=1}^t d_{t,j}\|\tilde{y}_j - \tilde{y}_j^*\|^2,$$

$$\rho_t = 1 - \mu\gamma, \quad \kappa_t = \frac{4\eta^2 l_{g,1}^2}{\lambda^2\mu^3\gamma}\|\tilde{u}_t\|^2, \quad \sigma_t = 2\gamma\sigma_{g,1}, \quad \nu_t = 2\gamma^2\sigma_{g,1}^2,$$

yielding the following recursion

$$\mathbb{E}\left[\exp(\theta\tilde{V}_{t+1})\right] \leq \mathbb{E}\left[\exp\left\{\theta\left[\left(1 - \frac{\mu\gamma}{2}\right)\tilde{V}_t + 2\gamma^2\sigma_{g,1}^2 + \frac{4\eta^2 l_{g,1}^2}{\lambda^2\mu^3\gamma}\|\tilde{u}_t\|^2 + \frac{4\eta^2 l_{g,1}^2}{\lambda^2\mu^3\gamma}\tilde{L}_t^2\sum_{j=1}^t d_{t,j}\tilde{V}_j\right]\right\}\right] \tag{32}$$

for all $\theta$ satisfying

$$0 \leq \theta \leq \min\left\{\frac{\mu}{8\gamma\sigma_{g,1}^2}, \frac{1}{4\gamma^2\sigma_{g,1}^2}\right\} \leq \frac{\mu}{8\gamma\sigma_{g,1}^2}, \tag{33}$$

where in (32) we denote $\tilde{V}_t := \|\tilde{y}_t - \tilde{y}_t^*\|^2$, and the last inequality of (33) uses $\gamma \leq 1/2l_{g,1} \leq 1/2\mu$. By Lemma C.4 we use induction to show that for any $t \geq 1$ and $\lambda$ satisfying (33), it holds that

$$\mathbb{E}\left[\exp(\theta\tilde{V}_t)\right] \leq \mathbb{E}\left[\exp\left\{\theta\left[\left(1 - \frac{\mu\gamma}{2}\right)^{t-1}\tilde{V}_1 + \frac{4\gamma\sigma_{g,1}^2}{\mu} + \frac{4\eta^2 l_{g,1}^2}{\lambda^2\mu^3\gamma}\sum_{i=1}^{t-1}\left(1 - \frac{\mu\gamma}{2}\right)^{t-1-i}\|\tilde{u}_i\|^2\right.\right.\right.$$

$$+ \frac{4\eta^2 l_{g,1}^2}{\lambda^2\mu^3\gamma}\tilde{V}_1\sum_{i=1}^{t-1}\left(1 - \frac{\mu\gamma}{2}\right)^{t-1-i}\tilde{L}_i^2 + \frac{16\eta^2 l_{g,1}^2}{\lambda^2\mu^4}\sigma_{g,1}^2\sum_{i=1}^{t-1}\left(1 - \frac{\mu\gamma}{2}\right)^{t-1-i}\tilde{L}_i^2$$

$$\left.\left.\left. + \frac{64\eta^4 l_{g,1}^4}{\lambda^4\mu^8\gamma^4}\sum_{i=1}^{t-1}\left(1 - \frac{\mu\gamma}{2}\right)^{t-1-i}\alpha_i\tilde{L}_i^2\|\tilde{u}_i\|^2\right]\right\}\right],$$

where the first and the last lines use the sum of geometric series, and the second line is due to Lemma B.5:

$$\sum_{i=1}^{t-1}\left(1 - \frac{\mu\gamma}{2}\right)^{i-1} \leq \frac{2}{\mu\gamma}, \qquad i\alpha_i(1 - \beta)^{i-1} \leq 1,$$

$$\sum_{i=1}^{t-1}\left(1 - \frac{\mu\gamma}{2}\right)^{t-1-i}\alpha_i\tilde{L}_i^2\sum_{j=1}^i\left(1 - \frac{\mu\gamma}{2}\right)^{i-j}\|\tilde{u}_j\|^2 \leq \frac{2}{\mu\gamma}\sum_{i=1}^{t-1}\left(1 - \frac{\mu\gamma}{2}\right)^{t-1-i}\alpha_i\tilde{L}_i^2\|\tilde{u}_i\|^2.$$

Moreover, by setting $\vartheta$ as follows, we have

$$\vartheta := \frac{8\gamma\sigma_{g,1}^2}{\mu} \implies \frac{4\gamma\sigma_{g,1}^2}{\mu} \leq \vartheta \quad \text{and} \quad \frac{1}{\vartheta} = \frac{\mu}{8\gamma\sigma_{g,1}^2}.$$

Hence for any $t \geq 1$ we obtain

$$
\mathbb{E}\left[\exp\left\{\theta\left[\tilde{V}_t - \left(1 - \frac{\mu\gamma}{2}\right)^{t-1}\tilde{V}_1 - \frac{4\eta^2 l_{g,1}^2}{\lambda^2\mu^3\gamma}\sum_{i=1}^{t-1}\left(1 - \frac{\mu\gamma}{2}\right)^{t-1-i}\|\tilde{u}_i\|^2\right.\right.\right.
$$

$$
-\frac{4\eta^2 l_{g,1}^2}{\lambda^2\mu^3\gamma}\tilde{V}_1\sum_{i=1}^{t-1}\left(1 - \frac{\mu\gamma}{2}\right)^{t-1-i}\tilde{L}_i^2 - \frac{16\eta^2 l_{g,1}^2}{\lambda^2\mu^4}\sigma_{g,1}^2\sum_{i=1}^{t-1}\left(1 - \frac{\mu\gamma}{2}\right)^{t-1-i}\tilde{L}_i^2
$$

$$
\left.\left.\left.-\frac{64\eta^4 l_{g,1}^4}{\lambda^4\mu^8\gamma^4}\sum_{i=1}^{t-1}\left(1 - \frac{\mu\gamma}{2}\right)^{t-1-i}\alpha_i\tilde{L}_i^2\|\tilde{u}_i\|^2\right]\right\}\right] \leq \exp(\theta\vartheta) \quad \text{for all} \quad 0 \leq \theta \leq 1/\vartheta.
$$

Taking $\theta = 1/\vartheta$ and applying Markov's inequality and union bound completes the proof. $\qquad\square$

**Lemma C.4.** *Suppose* (32) *holds, where* $\tilde{u}_t$, $\{d_{t,j}\}_{j=1}^t$ *and* $\tilde{L}_t$ *are defined in* (25), (10) *and* (27), *respectively. Choosing* $\gamma = 2\beta/\mu$, *then for any* $t \geq 1$ *we have*

$$
\mathbb{E}\left[\exp(\theta\tilde{V}_t)\right] \leq \mathbb{E}\left[\exp\left\{\theta\left[\left(1 - \frac{\mu\gamma}{2}\right)^{t-1}\tilde{V}_1 + 2\gamma^2\sigma_{g,1}^2\sum_{i=1}^{t-1}\left(1 - \frac{\mu\gamma}{2}\right)^{i-1} + \frac{4\eta^2 l_{g,1}^2}{\lambda^2\mu^3\gamma}\sum_{i=1}^{t-1}\left(1 - \frac{\mu\gamma}{2}\right)^{t-1-i}\|\tilde{u}_i\|^2\right.\right.\right.
$$

$$
+\frac{4\eta^2 l_{g,1}^2}{\lambda^2\mu^3\gamma}\tilde{V}_1\sum_{i=1}^{t-1}\left(1 - \frac{\mu\gamma}{2}\right)^{t-1-i}i\alpha_i(1-\beta)^{i-1}\tilde{L}_i^2 + \frac{16\eta^2 l_{g,1}^2}{\lambda^2\mu^4}\sigma_{g,1}^2\sum_{i=1}^{t-1}\left(1 - \frac{\mu\gamma}{2}\right)^{t-1-i}\tilde{L}_i^2
$$

$$
\left.\left.\left.+\frac{32\eta^4 l_{g,1}^4}{\lambda^4\mu^7\gamma^3}\sum_{i=1}^{t-1}\left(1 - \frac{\mu\gamma}{2}\right)^{t-1-i}\alpha_i\tilde{L}_i^2\sum_{j=1}^i\left(1 - \frac{\mu\gamma}{2}\right)^{i-j}\|\tilde{u}_j\|^2\right]\right\}\right].
$$

$$
(34)
$$

*Proof of Lemma C.4.* We use induction to show that (34) holds for any $t \geq 1$ and $\lambda$ satisfying (33).

**Base Case.** For the base case $t = 1$, it is easy to check that
$$
\mathbb{E}[\exp(\theta\tilde{V}_1)] \leq \mathbb{E}[\exp(\theta\tilde{V}_1)].
$$

**Induction Step.** Now we assume that the induction hypothesis (34) holds for $1 \leq k \leq t$, then for $k = t + 1$ we have
$$
\mathbb{E}[\exp(\theta\tilde{V}_{t+1})] \leq \mathbb{E}[\exp(\theta[(A_1) + (A_2) + (A_3) + (A_4) + (A_5) + (A_6)])],
$$
where $(A_1), (A_2), (A_3), (A_4), (A_5)$ and $(A_6)$ are defined as

$$
(A_1) = \left(1 - \frac{\mu\gamma}{2}\right)\left(1 - \frac{\mu\gamma}{2}\right)^{t-1}\tilde{V}_1,
$$

$$
(A_2) = 2\gamma^2\sigma_{g,1}^2 + 2\gamma^2\sigma_{g,1}^2\left(1 - \frac{\mu\gamma}{2}\right)\sum_{i=1}^{t-1}\left(1 - \frac{\mu\gamma}{2}\right)^{i-1},
$$

$$
(A_3) = \frac{4\eta^2 l_{g,1}^2}{\lambda^2\mu^3\gamma}\|\tilde{u}_t\|^2 + \frac{4\eta^2 l_{g,1}^2}{\lambda^2\mu^3\gamma}\left(1 - \frac{\mu\gamma}{2}\right)\sum_{i=1}^{t-1}\left(1 - \frac{\mu\gamma}{2}\right)^{t-1-i}\|\tilde{u}_i\|^2,
$$

$$
(A_4) = \frac{4\eta^2 l_{g,1}^2}{\lambda^2\mu^3\gamma}\tilde{L}_t^2\sum_{j=1}^t d_{t,j}\left(1 - \frac{\mu\gamma}{2}\right)^{j-1}\tilde{V}_1 + \frac{4\eta^2 l_{g,1}^2}{\lambda^2\mu^3\gamma}\tilde{V}_1\left(1 - \frac{\mu\gamma}{2}\right)\sum_{i=1}^{t-1}\left(1 - \frac{\mu\gamma}{2}\right)^{t-1-i}i\alpha_i(1-\beta)^{i-1}\tilde{L}_i^2,
$$

$$
(A_5) = \frac{4\eta^2 l_{g,1}^2}{\lambda^2\mu^3\gamma}\tilde{L}_t^2\sum_{j=1}^t d_{t,j}\cdot 2\gamma^2\sigma_{g,1}^2\sum_{i=1}^{j-1}\left(1 - \frac{\mu\gamma}{2}\right)^{i-1} + \frac{16\eta^2 l_{g,1}^2}{\lambda^2\mu^4}\sigma_{g,1}^2\left(1 - \frac{\mu\gamma}{2}\right)\sum_{i=1}^{t-1}\left(1 - \frac{\mu\gamma}{2}\right)^{t-1-i}\tilde{L}_i^2,
$$

$$
(A_6) = \frac{4\eta^2 l_{g,1}^2}{\lambda^2\mu^3\gamma}\tilde{L}_t^2\sum_{j=1}^t d_{t,j}\frac{4\eta^2 l_{g,1}^2}{\lambda^2\mu^3\gamma}\sum_{i=1}^{j-1}\left(1 - \frac{\mu\gamma}{2}\right)^{j-1-i}\|\tilde{u}_i\|^2
$$

$$
+\frac{32\eta^4 l_{g,1}^4}{\lambda^4\mu^7\gamma^3}\left(1 - \frac{\mu\gamma}{2}\right)\sum_{i=1}^{t-1}\left(1 - \frac{\mu\gamma}{2}\right)^{t-1-i}\alpha_i\tilde{L}_i^2\sum_{j=1}^i\left(1 - \frac{\mu\gamma}{2}\right)^{i-j}\|\tilde{u}_j\|^2.
$$

We continue to bound each term individually.

**Bounding $(A_1)$.**

$$(A_1) = \left(1 - \frac{\mu\gamma}{2}\right)\left(1 - \frac{\mu\gamma}{2}\right)^{t-1}\tilde{V}_1 = \left(1 - \frac{\mu\gamma}{2}\right)^t \tilde{V}_1.$$

**Bounding $(A_2)$.**

$$(A_2) = 2\gamma^2\sigma_{g,1}^2 + 2\gamma^2\sigma_{g,1}^2\left(1 - \frac{\mu\gamma}{2}\right)\sum_{i=1}^{t-1}\left(1 - \frac{\mu\gamma}{2}\right)^{i-1} = 2\gamma^2\sigma_{g,1}^2\left(1 - \frac{\mu\gamma}{2}\right)\sum_{i=1}^{t}\left(1 - \frac{\mu\gamma}{2}\right)^{i-1}.$$

**Bounding $(A_3)$.**

$$(A_3) = \frac{4\eta^2 l_{g,1}^2}{\lambda^2\mu^3\gamma}\|\tilde{u}_t\|^2 + \frac{4\eta^2 l_{g,1}^2}{\lambda^2\mu^3\gamma}\left(1 - \frac{\mu\gamma}{2}\right)\sum_{i=1}^{t-1}\left(1 - \frac{\mu\gamma}{2}\right)^{t-1-i}\|\tilde{u}_i\|^2 = \frac{4\eta^2 l_{g,1}^2}{\lambda^2\mu^3\gamma}\sum_{i=1}^{t}\left(1 - \frac{\mu\gamma}{2}\right)^{t-i}\|\tilde{u}_i\|^2.$$

**Bounding $(A_4)$.** By Lemma B.3 and the choice of $\gamma = 2\beta/\mu$, we have

$$\frac{4\eta^2 l_{g,1}^2}{\lambda^2\mu^3\gamma}\tilde{L}_t^2\sum_{j=1}^{t}d_{t,j}\left(1 - \frac{\mu\gamma}{2}\right)^{j-1}\tilde{V}_1 = \frac{4\eta^2 l_{g,1}^2}{\lambda^2\mu^3\gamma}\tilde{L}_t^2\sum_{j=1}^{t}\alpha_t(1-\beta)^{t-j}(1-\beta)^{j-1}\tilde{V}_1$$

$$= \frac{4\eta^2 l_{g,1}^2}{\lambda^2\mu^3\gamma}\tilde{L}_t^2\sum_{j=1}^{t}\alpha_t(1-\beta)^{t-1}\tilde{V}_1$$

$$= \frac{4\eta^2 l_{g,1}^2}{\lambda^2\mu^3\gamma}t\alpha_t(1-\beta)^{t-1}\tilde{L}_t^2\tilde{V}_1.$$

Then we obtain

$$(A_4) = \frac{4\eta^2 l_{g,1}^2}{\lambda^2\mu^3\gamma}\tilde{L}_t^2\sum_{j=1}^{t}d_{t,j}\left(1 - \frac{\mu\gamma}{2}\right)^{j-1}\tilde{V}_1 + \frac{4\eta^2 l_{g,1}^2}{\lambda^2\mu^3\gamma}\tilde{V}_1\left(1 - \frac{\mu\gamma}{2}\right)\sum_{i=1}^{t-1}\left(1 - \frac{\mu\gamma}{2}\right)^{t-1-i}i\alpha_i(1-\beta)^{i-1}\tilde{L}_i^2$$

$$= \frac{4\eta^2 l_{g,1}^2}{\lambda^2\mu^3\gamma}t\alpha_t(1-\beta)^{t-1}\tilde{L}_t^2\tilde{V}_1 + \frac{4\eta^2 l_{g,1}^2}{\lambda^2\mu^3\gamma}\tilde{V}_1\left(1 - \frac{\mu\gamma}{2}\right)\sum_{i=1}^{t-1}\left(1 - \frac{\mu\gamma}{2}\right)^{t-1-i}i\alpha_i(1-\beta)^{i-1}\tilde{L}_i^2$$

$$= \frac{4\eta^2 l_{g,1}^2}{\lambda^2\mu^3\gamma}\tilde{V}_1\sum_{i=1}^{t}\left(1 - \frac{\mu\gamma}{2}\right)^{t-i}i\alpha_i(1-\beta)^{i-1}\tilde{L}_i^2.$$

**Bounding $(A_5)$.** By Lemma B.3 and the choice of $\gamma = 2\beta/\mu$, we have

$$\frac{4\eta^2 l_{g,1}^2}{\lambda^2\mu^3\gamma}\tilde{L}_t^2\sum_{j=1}^{t}d_{t,j}\cdot 2\gamma^2\sigma_{g,1}^2\sum_{i=1}^{j-1}\left(1 - \frac{\mu\gamma}{2}\right)^{i-1} \le \frac{8\eta^2 l_{g,1}^2}{\lambda^2\mu^4\gamma^2}2\gamma^2\sigma_{g,1}^2\tilde{L}_t^2\sum_{j=1}^{t}d_{t,j}$$

$$= \frac{16\eta^2 l_{g,1}^2}{\lambda^2\mu^4}\sigma_{g,1}^2\tilde{L}_t^2.$$

Then we obtain

$$(A_5) = \frac{4\eta^2 l_{g,1}^2}{\lambda^2\mu^3\gamma}\tilde{L}_t^2\sum_{j=1}^{t}d_{t,j}\cdot 2\gamma^2\sigma_{g,1}^2\sum_{i=1}^{j-1}\left(1 - \frac{\mu\gamma}{2}\right)^{i-1} + \frac{16\eta^2 l_{g,1}^2}{\lambda^2\mu^4}\sigma_{g,1}^2\left(1 - \frac{\mu\gamma}{2}\right)\sum_{i=1}^{t-1}\left(1 - \frac{\mu\gamma}{2}\right)^{t-1-i}\tilde{L}_i^2$$

$$\le \frac{16\eta^2 l_{g,1}^2}{\lambda^2\mu^4}\sigma_{g,1}^2\tilde{L}_t^2 + \frac{16\eta^2 l_{g,1}^2}{\lambda^2\mu^4}\sigma_{g,1}^2\sum_{i=1}^{t-1}\left(1 - \frac{\mu\gamma}{2}\right)^{t-i}\tilde{L}_i^2$$

$$= \frac{16\eta^2 l_{g,1}^2}{\lambda^2\mu^4}\sigma_{g,1}^2\sum_{i=1}^{t}\left(1 - \frac{\mu\gamma}{2}\right)^{t-i}\tilde{L}_i^2.$$

**Bounding** $(A_6)$. By Lemma B.3 and the choice of $\gamma = 2\beta/\mu$, we have

$$\frac{4\eta^2 l_{g,1}^2}{\lambda^2\mu^3\gamma}\tilde{L}_t^2\sum_{j=1}^{t}d_{t,j}\frac{4\eta^2 l_{g,1}^2}{\lambda^2\mu^3\gamma}\sum_{i=1}^{j-1}\left(1-\frac{\mu\gamma}{2}\right)^{j-1-i}\|\tilde{u}_i\|^2 \leq \frac{4\eta^2 l_{g,1}^2}{\lambda^2\mu^3\gamma}\frac{8\eta^2 l_{g,1}^2}{\lambda^2\mu^4\gamma^2}\tilde{L}_t^2\sum_{j=1}^{t}d_{t,j}\|\tilde{u}_j\|^2$$

$$\leq \frac{4\eta^2 l_{g,1}^2}{\lambda^2\mu^3\gamma}\frac{8\eta^2 l_{g,1}^2}{\lambda^2\mu^4\gamma^2}\alpha_t\tilde{L}_t^2\sum_{j=1}^{t}(1-\beta)^{t-j}\|\tilde{u}_j\|^2$$

$$= \frac{32\eta^4 l_{g,1}^4}{\lambda^4\mu^7\gamma^3}\alpha_t\tilde{L}_t^2\sum_{j=1}^{t}\left(1-\frac{\mu\gamma}{2}\right)^{t-j}\|\tilde{u}_j\|^2.$$

Then we obtain

$$(A_6) = \frac{4\eta^2 l_{g,1}^2}{\lambda^2\mu^3\gamma}\tilde{L}_t^2\sum_{j=1}^{t}d_{t,j}\frac{4\eta^2 l_{g,1}^2}{\lambda^2\mu^3\gamma}\sum_{i=1}^{j-1}\left(1-\frac{\mu\gamma}{2}\right)^{j-1-i}\|\tilde{u}_i\|^2$$

$$+ \frac{32\eta^4 l_{g,1}^4}{\lambda^4\mu^7\gamma^3}\left(1-\frac{\mu\gamma}{2}\right)\sum_{i=1}^{t-1}\left(1-\frac{\mu\gamma}{2}\right)^{t-1-i}\alpha_i\tilde{L}_i^2\sum_{j=1}^{i}\left(1-\frac{\mu\gamma}{2}\right)^{i-j}\|\tilde{u}_j\|^2$$

$$\leq \frac{32\eta^4 l_{g,1}^4}{\lambda^4\mu^7\gamma^3}\alpha_t\tilde{L}_t^2\sum_{j=1}^{t}\left(1-\frac{\mu\gamma}{2}\right)^{t-j}\|\tilde{u}_j\|^2 + \frac{32\eta^4 l_{g,1}^4}{\lambda^4\mu^7\gamma^3}\sum_{i=1}^{t-1}\left(1-\frac{\mu\gamma}{2}\right)^{t-i}\alpha_i\tilde{L}_i^2\sum_{j=1}^{i}\left(1-\frac{\mu\gamma}{2}\right)^{i-j}\|\tilde{u}_j\|^2$$

$$= \frac{32\eta^4 l_{g,1}^4}{\lambda^4\mu^7\gamma^3}\sum_{i=1}^{t}\left(1-\frac{\mu\gamma}{2}\right)^{t-i}\alpha_i\tilde{L}_i^2\sum_{j=1}^{i}\left(1-\frac{\mu\gamma}{2}\right)^{i-j}\|\tilde{u}_j\|^2.$$

**Final Bound for the Induction Step.** Putting these terms together and rearranging yields

$$\mathbb{E}\left[\exp(\theta\tilde{V}_{t+1})\right] \leq \mathbb{E}\left[\exp\left\{\theta\left[\left(1-\frac{\mu\gamma}{2}\right)^t\tilde{V}_1 + 2\gamma^2\sigma_{g,1}^2\sum_{i=1}^{t}\left(1-\frac{\mu\gamma}{2}\right)^{i-1} + \frac{4\eta^2 l_{g,1}^2}{\lambda^2\mu^3\gamma}\sum_{i=1}^{t}\left(1-\frac{\mu\gamma}{2}\right)^{t-i}\|\tilde{u}_i\|^2\right.\right.\right.$$

$$+ \frac{4\eta^2 l_{g,1}^2}{\lambda^2\mu^3\gamma}\tilde{V}_1\sum_{i=1}^{t}\left(1-\frac{\mu\gamma}{2}\right)^{t-i}i\alpha_i(1-\beta)^{i-1}\tilde{L}_i^2 + \frac{16\eta^2 l_{g,1}^2}{\lambda^2\mu^4}\sigma_{g,1}^2\sum_{i=1}^{t}\left(1-\frac{\mu\gamma}{2}\right)^{t-i}\tilde{L}_i^2$$

$$\left.\left.\left.+ \frac{32\eta^4 l_{g,1}^4}{\lambda^4\mu^7\gamma^3}\sum_{i=1}^{t}\left(1-\frac{\mu\gamma}{2}\right)^{t-i}\alpha_i\tilde{L}_i^2\sum_{j=1}^{i}\left(1-\frac{\mu\gamma}{2}\right)^{i-j}\|\tilde{u}_j\|^2\right]\right\}\right],$$

which aligns with (34) for $k = t + 1$. Thus, the induction step is complete, and (34) holds for any $t \geq 1$. $\qquad\square$

# D    CONVERGENCE ANALYSIS OF ADAMBO (ALGORITHM 1)

In this section, we provide detailed convergence analysis of Algorithm 1 (or equivalently, Algorithm 4). Before presenting the lemmas and the main theorem, we will first define (or restate) a few key concepts and useful notations.

## D.1    TECHNICAL DEFINITIONS AND USEFUL NOTATIONS

**Filtration.** Define $\mathcal{F}_{\text{init}}$ as the filtration for updating $y_1$ (i.e., the filtration of warm-start phase):

$$\mathcal{F}_t^{\text{init}} = \sigma(\pi_0, \ldots, \pi_{T_0-1}).$$

For any $t \geq 2$, define $\mathcal{F}_t^x$ and $\mathcal{F}_t^y$ as the filtrations of the randomness used in updating $x_t$ and $y_t$, respectively, before the $t$-th iteration:

$$\mathcal{F}_t^x = \sigma(\bar{\xi}_1, \ldots, \bar{\xi}_{t-1}), \quad \mathcal{F}_t^y = \sigma(\zeta_1, \ldots, \zeta_{t-1}),$$

where $\sigma(\cdot)$ denotes the $\sigma$-algebra generated by the random variables within the argument. Additionally, let $\mathcal{F}_t$ denote the filtration of all randomness before the $t$-th iteration:

$$\mathcal{F}_t = \sigma(\mathcal{F}_{\text{init}} \cup \mathcal{F}_t^x \cup \mathcal{F}_t^y).$$

**Expectation.** We use $\mathbb{E}_t[\cdot]$ to denote the conditional expectation $\mathbb{E}[\cdot \mid \mathcal{F}_t]$.

**Auxiliary Sequence.** Note that $\hat{m}_t$ (line 7 of Algorithm 4) can be written as

$$\hat{m}_t = (1 - \alpha_t)\hat{m}_{t-1} + \alpha_t \hat{\nabla}\phi(x_t, y_t; \bar{\xi}_t) = \sum_{j=1}^{t} d_{t,j} \hat{\nabla}\phi(x_t, y_t; \bar{\xi}_t). \tag{35}$$

Similar to Appendix C.2, we introduce the following auxiliary sequence $\{\hat{u}_t\}$ for our analysis:

$$\hat{u}_t = (1 - \alpha_t)\hat{u}_{t-1} + \alpha_t \hat{\nabla}\phi(x_t, y_t^*; \bar{\xi}_t) = \sum_{j=1}^{t} d_{t,j} \hat{\nabla}\phi(x_t, y_t^*; \bar{\xi}_t). \tag{36}$$

**Other Definitions.** We define the deviation of the rescaled auxiliary momentum from the conditional expectation of the hypergradient estimator as

$$\epsilon_t := \hat{u}_t - \mathbb{E}_t[\hat{\nabla}\phi(x_t, y_t^*; \bar{\xi}_t)]. \tag{37}$$

Also, let $h_t$ be the learning rate vector and $H_t$ be the learning rate matrix:

$$h_t := \frac{\eta}{\sqrt{\hat{v}_t} + \lambda} \qquad \text{and} \qquad H_t := \text{diag}(h_t). \tag{38}$$

Then the update rule for upper-level variable $x_t$ (line 10 of Algorithm 1) can be written as

$$x_{t+1} = x_t - h_t \odot \hat{m}_t = x_t - H_t \hat{m}_t. \tag{39}$$

**Stopping Time.** Given a large enough constant $G$ as defined in Theorem D.12, denote $L$ and $\psi$ as

$$L = L_0 + L_1 G \qquad \text{and} \qquad \psi = \frac{C_L G^2}{2L}, \tag{40}$$

where constants $L_0, L_1$ and $C_L$ are defined in (14) and (43). Now we formally define the stopping time $\tau$ as

$$\tau := \min\{t \mid \Phi(x_t) - \Phi^* > \psi\} \wedge (T + 1). \tag{41}$$

In other words, $\tau$ is the first time when the sub-optimality gap is strictly larger than $\psi$, truncated at $T + 1$ to make sure it is bounded. Based on Lemma D.1, we know that if $t < \tau$, we have both $\Phi(x_t) - \Phi^* \leq \psi$ and $\|\nabla\Phi(x_t)\| \leq G$.

**Constants.** We define the following constants, which will be useful for analysis.

$$G_t = \max_{1 \leq k \leq t} \|\nabla\Phi(x_k)\|, \quad \hat{L}_t = L_0 + L_1 G_t, \quad L = L_0 + L_1 G, \quad \Delta_1 = \Phi(x_1) - \Phi^*, \tag{42}$$

$$C_L = \frac{L_{x,1}}{\sqrt{L_{x,1}^2 + L_{y,1}^2}}, \quad C_{u,0} = C_{\phi,0} + G, \quad C_{u,1} = C_{\phi,1} + L_1 G, \tag{43}$$

$$\sigma_\phi = \frac{\mu + 3l_{g,1} + \sigma_{g,2}}{\mu} + \frac{2l_{g,1} + \sigma_{g,2}}{\mu} l_{f,0} + \frac{2l_{g,1} + \sigma_{g,2}}{\mu}(L_{y,0} + L_{y,1}l_{f,0})r. \tag{44}$$

$$C_\beta \geq \max\left\{ \frac{8e\sigma_\phi^4 G^2 \max\{1, \iota\}}{c_1^2 \delta \lambda^2 \epsilon^4}, \frac{8C_2 e\Delta_1 L\sigma_\phi G^3}{c_1 c_2 \delta \lambda^2 \epsilon^4}\left(1 + \frac{\sigma_\phi^2 G}{c_1 \lambda \epsilon^2}\right)\max\{1, \sqrt{\iota}, \iota\}, \right.$$

$$\left. \left(\frac{32e\sigma_\phi^4 G^2}{c_1^2 \delta \lambda^2 \epsilon^4}\right)^2, \left(\frac{48C_2 e\Delta_1 L\sigma_\phi G^3}{c_1 c_2 \delta \lambda^2 \epsilon^4}\left(1 + \frac{\sigma_\phi^2 G}{c_1 \lambda \epsilon^2}\right)\max\{1, \sqrt{\iota}, \iota\}\right)^2 \right\}. \tag{45}$$

Besides, constants $L_0, L_1$ are defined in (14), $C_{\phi,0}, C_{\phi,1}$ are defined in (17), and $r$ is defined in (13), respectively.

## D.2 AUXILIARY LEMMAS

We first introduce the following useful lemma, which is crucial for the subsequent stopping time analysis and for establishing the contradiction argument.

**Lemma D.1.** *Under Assumption 3.2, we have*

$$\|\nabla\Phi(x)\|^2 \le \frac{2}{C_L}(L_0 + L_1\|\nabla\Phi(x)\|)(\Phi(x) - \Phi^*),$$

*where constants $L_0$, $L_1$ and $C_L$ are defined in (14) and (43). Further, for any given constant $G > 0$, if we denote $\psi$ as in (40) and $\Phi(x) - \Phi^* \le \psi$, then we have $\|\nabla\Phi(x)\| \le G$.*

*Proof of Lemma D.1.* Let $x'$ be

$$x' = x - \frac{C_L\|\nabla\Phi(x)\|}{L_0 + L_1\|\nabla\Phi(x)\|},$$

then we have

$$\|x' - x\| = \frac{C_L\|\nabla\Phi(x)\|}{L_0 + L_1\|\nabla\Phi(x)\|} \le \frac{C_L}{L_1} = \frac{1}{\sqrt{(1 + l_{g,1}^2/\mu^2)(L_{x,1}^2 + L_{y,1}^2)}} = r,$$

where the inequality can be verified by considering both cases of $\|\nabla\Phi(x)\| \le L_0/L_1$ and $\|\nabla\Phi(x)\| \ge L_0/L_1$. By Lemma B.10, we have

$$\Phi^* - \Phi(x) \le \Phi(x') - \Phi(x) \le \langle\nabla\Phi(x), x' - x\rangle + \frac{L_0 + L_1\|\nabla\Phi(x)\|}{2}\|x' - x\|^2$$

$$= -\frac{C_L(2 - C_L)}{2(L_0 + L_1\|\nabla\Phi(x)\|)}\|\nabla\Phi(x)\|^2.$$

Rearranging the above inequality yields

$$\|\nabla\Phi(x)\|^2 \le \frac{2(L_0 + L_1\|\nabla\Phi(x)\|)}{C_L(2 - C_L)}(\Phi(x) - \Phi^*) \le \frac{2(L_0 + L_1\|\nabla\Phi(x)\|)}{C_L}(\Phi(x) - \Phi^*). \tag{46}$$

where the last inequality uses the definition of $C_L$ in (43) and $C_L \le 1$.

Now define the function $\varphi : \mathbb{R}_0^+ \to \mathbb{R}$ as

$$\varphi(u) := \frac{C_L u^2}{2(L_0 + L_1 u)}.$$

It is easy to verify $\varphi$ is increasing and $\varphi(u) \in [0, \infty)$. Thus, $\varphi$ is invertible and $\varphi^{-1}$ is also increasing. Then for any constant $G \ge 0$, denote $L$ and $\psi$ as in (40),

$$L = L_0 + L_1 G, \qquad \psi = \frac{C_L G^2}{2L} = \varphi(G).$$

The property of function $\varphi^{-1}$ and (46) imply that if $\Phi(x) - \Phi^* \le \psi$, we have

$$\|\nabla\Phi(x)\| \le \varphi^{-1}(\Phi(x) - \Phi^*) \le \varphi^{-1}(\psi) = G.$$

$\square$

Note that when $t < \tau$, some of the quantities in Algorithm 1 and Appendix D.1 are bounded almost surely. In particular, we have the following lemma.

**Lemma D.2.** *If $t < \tau$, we have*

$$\|\nabla\Phi(x_t)\| \le G, \quad \hat{L}_t \le L, \quad \|\hat{u}_t\| \le C_{u,0}, \quad h_t \preceq \frac{\eta}{\lambda}, \quad \|H_t\| \preceq \frac{\eta}{\lambda}.$$

*where $h_t$ is defined in (38), constants $\hat{L}_t$, $L$ and $C_{u,0}$ are defined in (42) and (43), respectively.*

*Proof of Lemma D.2.* By Lemma D.1 and definition of $\tau$, we have $\|\nabla\Phi(x_t)\| \leq G$ if $t < \tau$. Also, recall the definition of $G_t$, $\hat{L}_t$ and $L$ as in (42), we have $G_t = \max_{k\leq t}\|\nabla\Phi(x_k)\| \leq G$ if $t < \tau$, and hence gives $\hat{L}_t = L_0 + L_1 G_t \leq L_0 + L_1 G = L$. Before bounding $\|\hat{u}_t\|$, we first show $\|\hat{\nabla}\phi(x_t, y_t^*; \bar{\xi}_t)\| \leq C_{u,0}$. Lemma B.14 directly implies that if $t < \tau$, then

$$\|\hat{\nabla}\phi(x_t, y_t^*; \bar{\xi}_t)\| \leq C_{\phi,0} + (C_{\phi,1} + L_1\|\nabla\Phi(x_t)\|)\|y_t^* - y_t^*\| + \|\nabla\Phi(x_t)\| \leq C_{\phi,0} + G = C_{u,0},$$

where the last equality is due to the definition of $C_{u,0}$ in (43). Now $\|\hat{u}_t\|$ can be bounded by a standard induction argument as follows. First, for the base case $k = 1$, note that $\|\hat{\nabla}\phi(x_1, y_1^*; \bar{\xi}_1)\| \leq C_{u,0}$. Suppose $\|\hat{u}_{k-1}\| \leq C_{u,0}$ for some $k < \tau$, then by update rule of $\hat{u}_k$ in (36) we have

$$\|\hat{u}_k\| \leq (1 - \alpha_k)\|\hat{u}_{k-1}\| + \alpha_k\|\hat{\nabla}\phi(x_k, y_k; \bar{\xi}_k)\| \leq C_{u,0}.$$

Therefore, the induction is complete. The last two results directly follow from the definitions of $h_t$ and $H_t$ in (38). $\square$

## D.3 PROOF OF LEMMA 4.3

In the next lemma, we provide high probability bound for the warm-start phase.

**Lemma D.3** (Warm-Start, Restatement of Lemma 4.3). *Suppose that Assumptions 3.2 and 3.3 hold. Let $\{y_t^{init}\}$ be the iterates generated by Algorithm 3 with constant learning rate $\gamma \leq 1/2l_{g,1}$. Then for any given $\delta \in (0,1)$, the following estimate holds with probability at least $1 - \delta/4$ over the randomness in $\mathcal{F}_{init}$ (we denote this event as $\mathcal{E}_0$):*

$$\|y_1 - y_1^*\|^2 \leq \left(1 - \frac{\mu\gamma}{2}\right)^{T_0}\|y_0 - y_0^*\|^2 + \frac{8\gamma\sigma_{g,1}^2}{\mu}\ln\frac{4e}{\delta}. \tag{47}$$

*Proof of Lemma D.3.* For any given $\delta \in (0,1)$ and any fixed $t \geq 0$, we invoke (Cutler et al., 2023, Theorem 30) to obtain that

$$\|y_t^{init} - y_0^*\|^2 \leq \left(1 - \frac{\mu\gamma}{2}\right)^t\|y_0 - y_0^*\|^2 + \frac{8\gamma\sigma_{g,1}^2}{\mu}\ln\frac{4e}{\delta} \tag{48}$$

holds with probability at least $1 - \delta$ over the randomness in $\mathcal{F}_{init}$. Set $t = T_0$ and then we have

$$\|y_1 - y_1^*\|^2 = \|y_{T_0}^{init} - y_0^*\|^2 \leq \left(1 - \frac{\mu\gamma}{2}\right)^{T_0}\|y_0 - y_0^*\|^2 + \frac{8\gamma\sigma_{g,1}^2}{\mu}\ln\frac{4e}{\delta},$$

where the first equality is due to $y_1 = y_{T_0}^{init}$ and $y_1^* = y_0^*$ (since $x_1 = x_0$) by line 2 of Algorithm 1. $\square$

## D.4 PROOF OF LEMMA 4.4

The following Lemma D.4 (i.e., the complete version of Lemma 4.4) is a direct application of the randomness decoupling lemma (i.e., Lemma 4.2) to the actual sequences $\{x_t\}, \{y_t\}$ in Algorithm 1.

**Lemma D.4.** *Suppose that Assumptions 3.2 to 3.4 hold. Let $\{y_t\}$ be the iterates generated by Algorithm 1. Under the parameter choices in Theorem D.12, let $\eta$ further satisfy*

$$\eta \leq c_2 \min\left\{\frac{r\lambda}{G_T}, \frac{\lambda}{6L}, \frac{\sigma_\phi\lambda\beta}{\hat{L}_T G_T \max\{1, \sqrt{\iota}, \ln(1/\beta), \ln(C_\beta)\}}, \frac{\lambda^{3/2}\beta}{\hat{L}_T\sqrt{G_T}}\right\}, \tag{49}$$

*then for any given $\delta \in (0,1)$ and all $t \geq 1$, the following estimate holds with probability at least $1 - \delta/4$ over the randomness in $\mathcal{F}_{T+1}^y$ (we denote this event as $\mathcal{E}_y$):*

$$\|y_t - y_t^*\|^2 \leq \left(\left(1 - \frac{\mu\gamma}{2}\right)^{t-1} + \frac{4\eta^2 l_{g,1}^2}{\lambda^2\mu^3\gamma}\sum_{i=1}^{t-1}\left(1 - \frac{\mu\gamma}{2}\right)^{t-1-i}\hat{L}_i^2\right)\|y_1 - y_1^*\|^2$$

$$+ \left(\frac{8\gamma}{\mu}\ln\frac{4eT}{\delta} + \frac{16\eta^2 l_{g,1}^2}{\lambda^2\mu^4}\sum_{i=1}^{t-1}\left(1 - \frac{\mu\gamma}{2}\right)^{t-1-i}\hat{L}_i^2\right)\sigma_{g,1}^2$$

$$+ \frac{4\eta^2 l_{g,1}^2}{\lambda^2\mu^3\gamma}\sum_{i=1}^{t-1}\left(1 - \frac{\mu\gamma}{2}\right)^{t-1-i}\|\hat{u}_i\|^2 + \frac{64\eta^4 l_{g,1}^4}{\lambda^4\mu^8\gamma^4}\sum_{i=1}^{t-1}\left(1 - \frac{\mu\gamma}{2}\right)^{t-1-i}\alpha_i\hat{L}_i^2\|\hat{u}_i\|^2, \tag{50}$$

*where constant $\hat{L}_i$ and sequence $\{\hat{u}_i\}$ are defined in (42) and (36), respectively.*

*Proof of Lemma D.4.* First, with the parameter choices in Theorem D.12 and the additional choice for $\eta$ as in (49), we can follow the same procedure as Lemma D.13 (see "Verification for $\varrho \leq \min\{r, 1/4L_1\}$") to show that $\|y_t - y_t^*\| \leq r$ for all $t \in [T]$. Thus, the condition for applying Lemma B.15 is satisfied. Recall the definitions of $\hat{m}_t$ and $\hat{u}_t$ in (35) and (36), we have

$$
\begin{aligned}
\|\hat{m}_t - \hat{u}_t\|^2 &\leq \left\| \sum_{j=1}^{t} d_{t,j} (\hat{\nabla}\phi(x_j, y_j; \bar{\xi}_j) - \hat{\nabla}\phi(x_j, y_j^*; \bar{\xi}_j)) \right\|^2 \\
&\leq \sum_{j=1}^{t} d_{t,j} \|\hat{\nabla}\phi(x_j, y_j; \bar{\xi}_j) - \hat{\nabla}\phi(x_j, y_j^*; \bar{\xi}_j)\|^2 \\
&\leq \sum_{j=1}^{t} d_{t,j} (L_0 + L_1 \|\nabla\Phi(x_j)\|)^2 \|y_j - y_j^*\|^2 \leq \hat{L}_t^2 \sum_{j=1}^{t} d_{t,j} \|y_j - y_j^*\|^2,
\end{aligned}
\tag{51}
$$

where the second inequality uses Jensen's inequality, the third inequality is due to Lemma B.15, and the last inequality uses the definition of $\hat{L}_t$ in (42). By the update rule in Algorithm 4, we have

$$
\begin{aligned}
\|x_{t+1} - x_t\|^2 &\leq \|H_t\|^2 \|\hat{m}_t\|^2 \leq \frac{\eta^2}{\lambda^2} \|\hat{m}_t\|^2 \leq \frac{2\eta^2}{\lambda^2} \left( \|\hat{u}_t\|^2 + \|\hat{m}_t - \hat{u}_t\|^2 \right) \\
&\leq \frac{2\eta^2}{\lambda^2} \left( \|\hat{u}_t\|^2 + \hat{L}_t^2 \sum_{j=1}^{t} d_{t,j} \|y_j - y_j^*\|^2 \right),
\end{aligned}
$$

where the first inequality uses (38); the second inequality is due to Lemma D.2; the third inequality uses Young's inequality; and the last inequality is due to (51). This implies that the sequence $\{x_t\}$ and the randomness $\{\bar{\xi}_t\}$ generated by Algorithm 1 satisfy the condition (29) in Lemma C.3. Therefore, the result follows by applying Lemma C.3 with $\{\tilde{x}_t\} = \{x_t\}$ and $\{\hat{\xi}_t\} = \{\bar{\xi}_t\}$. □

**Remark.** In the end, we will show $\tau = T+1$ in the proof of Theorem D.12 (i.e., the complete version of Theorem 4.1), thus we can apply Lemma D.2 to obtain $G_T \leq G$ and $\hat{L}_T \leq L$. This suggests that under event $\mathcal{E}_0 \cap \mathcal{E}_y$, the additional requirement (49) is actually included in the parameter choices of Theorem D.12. Therefore, there is no need to worry about this temporary iterate-dependent requirement for the choice of $\eta$.

### D.5 PROOF OF LEMMA 4.5

Before proving Lemma 4.5, first note that when $t < \tau$ and $\mathcal{E}_0 \cap \mathcal{E}_y$ holds, some of the time-dependent quantities (such as $\hat{L}_t$ and $\|\hat{u}_t\|$) in Lemma D.4 can be well bounded by Lemma D.2. In particular, we have the following two high probability bounds for the lower-level approximation error $\|y_t - y_t^*\|$: the first one, (52), is useful for the convergence analysis; and the second one, (53), is crucial for proving Lemmas D.6 and D.8.

**Lemma D.5.** *Under event $\mathcal{E}_0 \cap \mathcal{E}_y$ and the parameter choices in Lemma D.4, if $t \leq \tau$, we have*

$$
\begin{aligned}
\|y_t - y_t^*\|^2 &\leq \left( \left(1 - \frac{\mu\gamma}{2}\right)^{t-1} + \frac{8\eta^2 l_{g,1}^2 L^2}{\lambda^2 \mu^4 \gamma^2} \right) \|y_1 - y_1^*\|^2 + \left( \frac{8\gamma}{\mu} \ln \frac{4eT}{\delta} + \frac{32\eta^2 l_{g,1}^2 L^2}{\lambda^2 \mu^5 \gamma} \right) \sigma_{g,1}^2 \\
&\quad + \frac{4\eta^2 l_{g,1}^2}{\lambda^2 \mu^3 \gamma} \sum_{i=1}^{t-1} \left(1 - \frac{\mu\gamma}{2}\right)^{t-1-i} \|\hat{u}_i\|^2 + \frac{64\eta^4 l_{g,1}^4 L^2}{\lambda^4 \mu^8 \gamma^4} \sum_{i=1}^{t-1} \left(1 - \frac{\mu\gamma}{2}\right)^{t-1-i} \alpha_i \|\hat{u}_i\|^2
\end{aligned}
\tag{52}
$$

*and*

$$
\begin{aligned}
\|y_t - y_t^*\|^2 &\leq \left(1 + \frac{8\eta^2 l_{g,1}^2 L^2}{\lambda^2 \mu^4 \gamma^2}\right) \|y_1 - y_1^*\|^2 + \left( \frac{8\gamma}{\mu} \ln \frac{4eT}{\delta} + \frac{32\eta^2 l_{g,1}^2 L^2}{\lambda^2 \mu^5 \gamma} \right) \sigma_{g,1}^2 \\
&\quad + \frac{8\eta^2 l_{g,1}^2 C_{u,0}^2}{\lambda^2 \mu^4 \gamma^2} + \frac{1024\eta^4 l_{g,1}^4 L^2 C_{u,0}^2}{\lambda^4 \mu^8 \gamma^4} \left(2 + \ln \frac{1}{\beta}\right) =: \varrho^2,
\end{aligned}
\tag{53}
$$

*where constants $L$ and sequence $\{\hat{u}_i\}$ are defined in (42) and (36), respectively.*

*Proof of Lemma D.5.* By Lemma D.2, we know that $\hat{L}_t \leq L$ and $\|\hat{u}_t\| \leq C_{u,0}$ if $t < \tau$. Then under event $\mathcal{E}_0 \cap \mathcal{E}_y$, (52) is obtained by replacing $\hat{L}_i$ with $L$, and (53) is obtained by substituting both $\hat{L}_i$ and $\|\hat{u}_i\|$ with $L$ and $C_{u,0}$, respectively. $\qquad\square$

With Lemma D.5 in place, we now formally present the statement of Lemma 4.5 below.

**Lemma D.6** (Complete version of Lemma 4.5). *Under event $\mathcal{E}_0 \cap \mathcal{E}_y$ and the parameter choices in Lemma D.4, if $t < \tau$, we have*

$$\|\hat{m}_t\| \leq C_{u,0} + C_{u,1}\varrho, \quad \hat{v}_t \preceq (C_{u,0} + C_{u,1}\varrho)^2, \quad \frac{\eta}{C_{u,0} + C_{u,1}\varrho + \lambda} \preceq h_t \preceq \frac{\eta}{\lambda};$$

*if $t \leq \tau$, we have*

$$\|\hat{\nabla}\phi(x_t, y_t; \bar{\xi}_t) - \mathbb{E}_t[\hat{\nabla}\phi(x_t, y_t; \bar{\xi}_t)]\| \leq \sigma_\phi,$$

$$\|\mathbb{E}_t[\hat{\nabla}\phi(x_t, y_t^*; \bar{\xi}_t)] - \mathbb{E}_{t-1}[\hat{\nabla}\phi(x_{t-1}, y_{t-1}^*; \bar{\xi}_{t-1})]\| \leq L\|x_t - x_{t-1}\|;$$

*where constants $C_{u,0}, C_{u,1}, \sigma_\phi, L$ and $\varrho$ are defined in* (43), (42) *and* (53), *respectively.*

*Proof of Lemma D.6.* By Lemma B.14, under event $\mathcal{E}_0 \cap \mathcal{E}_y$, if $t < \tau$, we have

$$\|\hat{\nabla}\phi(x_t, y_t; \bar{\xi}_t)\| \leq C_{\phi,0} + (C_{\phi,1} + L_1\|\nabla\Phi(x_t)\|)\|y_t - y_t^*\| + \|\nabla\Phi(x_t)\|$$
$$\leq C_{\phi,0} + G + (C_{\phi,1} + L_1 G)\varrho = C_{u,0} + C_{u,1}\varrho,$$

where the second inequality is due to Lemma D.2 and (53) in Lemma D.5, and the last equality uses the definitions in (43). We can bound $\|\hat{m}_t\|$ by a standard induction argument as follows. First, for the base case $k = 1$, note that

$$\|\hat{m}_1\| = \|\hat{\nabla}\phi(x_1, y_1; \bar{\xi}_1)\| \leq C_{u,0} + C_{u,1}\varrho.$$

Suppose $\|\hat{m}_{k-1}\| \leq C_{u,0} + C_{u,1}\varrho$ for some $k < \tau$, then we have

$$\|\hat{m}_k\| \leq (1 - \alpha_k)\|\hat{m}_{k-1}\| + \alpha_k\|\hat{\nabla}\phi(x_k, y_k; \bar{\xi}_k)\| \leq C_{u,0} + C_{u,1}\varrho.$$

Then we can show $\hat{v}_t \preceq (C_{u,0} + C_{u,1}\varrho)^2$ in a similar way (by induction argument) by noting that

$$(\hat{\nabla}\phi(x_t, y_t; \bar{\xi}_t))^2 \preceq \|\hat{\nabla}\phi(x_t, y_t; \bar{\xi}_t)\|^2 \leq (C_{u,0} + C_{u,1}\varrho)^2.$$

Given the bound on $\hat{v}_t$, it is straight forward to bound the learning rate $h_t$. As for the second last bound, by Lemma B.13 and (53) of Lemma D.5, under event $\mathcal{E}_0 \cap \mathcal{E}_y$, if $t \leq \tau$, we have

$$\|\hat{\nabla}\phi(x_t, y_t; \bar{\xi}_t) - \mathbb{E}_t[\hat{\nabla}\phi(x_t, y_t; \bar{\xi}_t)]\|$$
$$\leq \frac{\mu + 3l_{g,1} + \sigma_{g,2}}{\mu} + \frac{2l_{g,1} + \sigma_{g,2}}{\mu}l_{f,0} + \frac{2l_{g,1} + \sigma_{g,2}}{\mu}(L_{y,0} + L_{y,1}l_{f,0})\|y_t - y_t^*\|$$
$$\leq \frac{\mu + 3l_{g,1} + \sigma_{g,2}}{\mu} + \frac{2l_{g,1} + \sigma_{g,2}}{\mu}l_{f,0} + \frac{2l_{g,1} + \sigma_{g,2}}{\mu}(L_{y,0} + L_{y,1}l_{f,0})\varrho$$
$$\leq \sigma_\phi,$$

where the last equality uses $\varrho \leq r$ by Lemma D.13 and the definition of $\sigma_\phi$ in (44). The last bound can be obtained by applying Lemmas B.15 and D.2:

$$\|\mathbb{E}_t[\hat{\nabla}\phi(x_t, y_t^*; \bar{\xi}_t)] - \mathbb{E}_{t-1}[\hat{\nabla}\phi(x_{t-1}, y_{t-1}^*; \bar{\xi}_{t-1})]\| \leq (L_0 + L_1\|\nabla\Phi(x_{t-1})\|)\|x_t - x_{t-1}\|$$
$$\leq (L_0 + L_1 G)\|x_t - x_{t-1}\|$$
$$= L\|x_t - x_{t-1}\|,$$

where the last inequality uses the definition of $L$ in (42). $\qquad\square$

## D.6 PROOF OF LEMMA 4.6

The following lemma provides a bound for the difference between the actual momentum $\hat{m}_t$ versus the auxiliary momentum $\hat{u}_t$ under the good event $\mathcal{E}_0 \cap \mathcal{E}_y$, which is crucial for establishing the convergence guarantees for Algorithm 1.

**Lemma D.7.** *Under event $\mathcal{E}_0 \cap \mathcal{E}_y$ and the parameter choices in Lemma D.4, we have*

$$
\sum_{t=1}^{\tau-1} \|\hat{m}_t - \hat{u}_t\|^2 \le TL^2 \left( \left(1 + \frac{8\eta^2 l_{g,1}^2 L^2}{\lambda^2 \mu^4 \gamma^2}\right) \|y_1 - y_1^*\|^2 + \left(\frac{8\gamma}{\mu} \ln \frac{4eT}{\delta} + \frac{32\eta^2 l_{g,1}^2 L^2}{\lambda^2 \mu^5 \gamma}\right) \sigma_{g,1}^2 \right)
$$

$$
+ L^2 \left( \frac{8\eta^2 l_{g,1}^2}{\lambda^2 \mu^4 \gamma^2} + \frac{2048\eta^4 l_{g,1}^4 L^2}{\lambda^4 \mu^8 \gamma^4} \left(2 + \ln \frac{1}{\beta}\right) \right) \sum_{t=1}^{\tau-1} \|\epsilon_t\|^2 + 2\|\mathbb{E}_t[\hat{\nabla}\phi(x_t, y_t^*; \bar{\xi}_t)] - \nabla\Phi(x_t)\|^2
$$

$$
+ 2L^2 \left( \frac{8\eta^2 l_{g,1}^2}{\lambda^2 \mu^4 \gamma^2} + \frac{2048\eta^4 l_{g,1}^4 L^2}{\lambda^4 \mu^8 \gamma^4} \left(2 + \ln \frac{1}{\beta}\right) \right) \sum_{t=1}^{\tau-1} \|\nabla\Phi(x_t)\|^2.
$$

*Proof of Lemma D.7.* Under event $\mathcal{E}_0 \cap \mathcal{E}_y$, if $t < \tau$, by Lemma D.2 and (51) in Lemma D.4 we have

$$
\|\hat{m}_t - \hat{u}_t\|^2 \le \hat{L}_t^2 \sum_{j=1}^t d_{t,j} \|y_j - y_j^*\|^2 \le L^2 \sum_{j=1}^t d_{t,j} \|y_j - y_j^*\|^2.
$$

Now we apply (52) of Lemma D.5 and take summation to obtain

$$
\sum_{t=1}^{\tau-1} \sum_{j=1}^t d_{t,j} \|y_j - y_j^*\|^2
$$

$$
\le \sum_{t=1}^{\tau-1} \sum_{j=1}^t d_{t,j} \left( \left( \left(1 - \frac{\mu\gamma}{2}\right)^{j-1} + \frac{8\eta^2 l_{g,1}^2 L^2}{\lambda^2 \mu^4 \gamma^2} \right) \|y_1 - y_1^*\|^2 + \left(\frac{8\gamma}{\mu} \ln \frac{4eT}{\delta} + \frac{32\eta^2 l_{g,1}^2 L^2}{\lambda^2 \mu^5 \gamma}\right) \sigma_{g,1}^2 \right)
$$

$$
\tag{$A_1$}
$$

$$
+ \sum_{t=1}^{\tau-1} \sum_{j=1}^t d_{t,j} \left( \frac{4\eta^2 l_{g,1}^2}{\lambda^2 \mu^3 \gamma} \sum_{i=1}^{j-1} \left(1 - \frac{\mu\gamma}{2}\right)^{j-1-i} \|\hat{u}_i\|^2 + \frac{64\eta^4 l_{g,1}^4 L^2}{\lambda^4 \mu^8 \gamma^4} \sum_{i=1}^{j-1} \left(1 - \frac{\mu\gamma}{2}\right)^{j-1-i} \alpha_i \|\hat{u}_i\|^2 \right)
$$

$$
\tag{$A_2$}
$$

We continue to bound each term individually.

**Bounding** $(A_1)$. By Lemmas B.3 and B.5 and choice of $\gamma = 2\beta/\mu$, we have

$$
(A_1) = \sum_{t=1}^{\tau-1} \sum_{j=1}^t d_{t,j} \left( \left( \left(1 - \frac{\mu\gamma}{2}\right)^{j-1} + \frac{8\eta^2 l_{g,1}^2 L^2}{\lambda^2 \mu^4 \gamma^2} \right) \|y_1 - y_1^*\|^2 + \left(\frac{8\gamma}{\mu} \ln \frac{4eT}{\delta} + \frac{32\eta^2 l_{g,1}^2 L^2}{\lambda^2 \mu^5 \gamma}\right) \sigma_{g,1}^2 \right)
$$

$$
= \sum_{t=1}^{\tau-1} \sum_{j=1}^t d_{t,j} \left(1 - \frac{\mu\gamma}{2}\right)^{j-1} \|y_1 - y_1^*\|^2
$$

$$
+ \sum_{t=1}^{\tau-1} \sum_{j=1}^t d_{t,j} \left( \frac{8\eta^2 l_{g,1}^2 L^2}{\lambda^2 \mu^4 \gamma^2} \|y_1 - y_1^*\|^2 + \left(\frac{8\gamma}{\mu} \ln \frac{4eT}{\delta} + \frac{32\eta^2 l_{g,1}^2 L^2}{\lambda^2 \mu^5 \gamma}\right) \sigma_{g,1}^2 \right)
$$

$$
= \sum_{t=1}^{\tau-1} t\alpha_t (1-\beta)^{t-1} \|y_1 - y_1^*\|^2 + \sum_{t=1}^{\tau-1} \left( \frac{8\eta^2 l_{g,1}^2 L^2}{\lambda^2 \mu^4 \gamma^2} \|y_1 - y_1^*\|^2 + \left(\frac{8\gamma}{\mu} \ln \frac{4eT}{\delta} + \frac{32\eta^2 l_{g,1}^2 L^2}{\lambda^2 \mu^5 \gamma}\right) \sigma_{g,1}^2 \right)
$$

$$
\le T \left( \left(1 + \frac{8\eta^2 l_{g,1}^2 L^2}{\lambda^2 \mu^4 \gamma^2}\right) \|y_1 - y_1^*\|^2 + \left(\frac{8\gamma}{\mu} \ln \frac{4eT}{\delta} + \frac{32\eta^2 l_{g,1}^2 L^2}{\lambda^2 \mu^5 \gamma}\right) \sigma_{g,1}^2 \right),
$$

$$
\tag{54}
$$

where the last inequality uses $\tau \le T + 1$ by definition of $\tau$.

**Bounding** $(A_2)$. By Lemmas B.3 and B.6 and choice of $\gamma = 2\beta/\mu$, we have

$$
(A_2) = \sum_{t=1}^{\tau-1} \sum_{j=1}^{t} d_{t,j} \left( \frac{4\eta^2 l_{g,1}^2}{\lambda^2 \mu^3 \gamma} \sum_{i=1}^{j-1} \left(1 - \frac{\mu\gamma}{2}\right)^{j-1-i} \|\hat{u}_i\|^2 + \frac{64\eta^4 l_{g,1}^4 L^2}{\lambda^4 \mu^8 \gamma^4} \sum_{i=1}^{j-1} \left(1 - \frac{\mu\gamma}{2}\right)^{j-1-i} \alpha_i \|\hat{u}_i\|^2 \right)
$$

$$
\leq \frac{4\eta^2 l_{g,1}^2}{\lambda^2 \mu^4 \gamma^2} \sum_{t=1}^{\tau-1} \sum_{j=1}^{t} d_{t,j} \|\hat{u}_j\|^2 + \frac{64\eta^4 l_{g,1}^4 L^2}{\lambda^4 \mu^8 \gamma^4} \left(32 + 16\ln\frac{1}{\beta}\right) \sum_{t=1}^{\tau-1} \sum_{j=1}^{t} d_{t,j} \|\hat{u}_j\|^2
$$

$$
\leq \left( \frac{4\eta^2 l_{g,1}^2}{\lambda^2 \mu^4 \gamma^2} + \frac{1024\eta^4 l_{g,1}^4 L^2}{\lambda^4 \mu^8 \gamma^4} \left(2 + \ln\frac{1}{\beta}\right) \right) \sum_{t=1}^{\tau-1} \|\hat{u}_t\|^2.
$$

$$(55)$$

**Final Bound.** Combining (54) and (55) yields

$$
\sum_{t=1}^{\tau-1} \sum_{j=1}^{t} d_{t,j} \|y_j - y_j^*\|^2 \leq T \left( \left(1 + \frac{8\eta^2 l_{g,1}^2 L^2}{\lambda^2 \mu^4 \gamma^2}\right) \|y_1 - y_1^*\|^2 + \left(\frac{8\gamma}{\mu} \ln\frac{4eT}{\delta} + \frac{32\eta^2 l_{g,1}^2 L^2}{\lambda^2 \mu^5 \gamma}\right) \sigma_{g,1}^2 \right)
$$

$$
+ \left( \frac{4\eta^2 l_{g,1}^2}{\lambda^2 \mu^4 \gamma^2} + \frac{1024\eta^4 l_{g,1}^4 L^2}{\lambda^4 \mu^8 \gamma^4} \left(2 + \ln\frac{1}{\beta}\right) \right) \sum_{t=1}^{\tau-1} \|\hat{u}_t\|^2.
$$

In addition, recall the definition of $\hat{u}_t$ and $\epsilon_t$ in (36) and (37), by Young's inequality we have

$$
\|\hat{u}_t\|^2 \leq 2\|\epsilon_t\|^2 + 4\|\mathbb{E}_t[\hat{\nabla}\phi(x_t, y_t^*; \bar{\xi}_t)] - \nabla\Phi(x_t)\|^2 + 4\|\nabla\Phi(x_t)\|^2.
$$

Therefore, we conclude that

$$
\sum_{t=1}^{\tau-1} \|\hat{m}_t - \hat{u}_t\|^2 \leq L^2 \sum_{t=1}^{\tau-1} \sum_{j=1}^{t} d_{t,j} \|y_j - y_j^*\|^2
$$

$$
\leq TL^2 \left( \left(1 + \frac{8\eta^2 l_{g,1}^2 L^2}{\lambda^2 \mu^4 \gamma^2}\right) \|y_1 - y_1^*\|^2 + \left(\frac{8\gamma}{\mu} \ln\frac{4eT}{\delta} + \frac{32\eta^2 l_{g,1}^2 L^2}{\lambda^2 \mu^5 \gamma}\right) \sigma_{g,1}^2 \right)
$$

$$
+ L^2 \left( \frac{8\eta^2 l_{g,1}^2}{\lambda^2 \mu^4 \gamma^2} + \frac{2048\eta^4 l_{g,1}^4 L^2}{\lambda^4 \mu^8 \gamma^4} \left(2 + \ln\frac{1}{\beta}\right) \right) \sum_{t=1}^{\tau-1} \|\epsilon_t\|^2 + 2\|\mathbb{E}_t[\hat{\nabla}\phi(x_t, y_t^*; \bar{\xi}_t)] - \nabla\Phi(x_t)\|^2
$$

$$
+ 2L^2 \left( \frac{8\eta^2 l_{g,1}^2}{\lambda^2 \mu^4 \gamma^2} + \frac{2048\eta^4 l_{g,1}^4 L^2}{\lambda^4 \mu^8 \gamma^4} \left(2 + \ln\frac{1}{\beta}\right) \right) \sum_{t=1}^{\tau-1} \|\nabla\Phi(x_t)\|^2.
$$

$$\square$$

### D.7 PROOF OF THEOREM 4.1

The following lemma ensures that $x_{t+1}$ and $x_t$ remain close for sufficiently small $\eta$, allowing us to apply Lemma B.11 in Lemma D.9.

**Lemma D.8.** *Under event $\mathcal{E}_0 \cap \mathcal{E}_y$ and the parameter choices in Lemma D.4, if $t < \tau$, then we have* $\|x_{t+1} - x_t\| \leq \eta D$ *where* $D := 2G/\lambda$.

*Proof of Lemma D.8.* Under event $\mathcal{E}_0 \cap \mathcal{E}_y$, if $t < \tau$, then we have

$$
\|x_{t+1} - x_t\| \leq \|H_t\| \|\hat{m}_t\| \leq \frac{\eta}{\lambda} \|\hat{m}_t\| \leq \frac{\eta(C_{u,0} + C_{u,1}\varrho)}{\lambda} \leq \frac{2\eta G}{\lambda} = \eta D,
$$

where the first inequality uses (38), the second inequality is due to Lemma D.2, the third inequality uses Lemma D.6, the fourth inequality is due to Lemma D.13, and the last equality uses the definition of $D$. $\square$

Next, we provide a descent lemma for AdamBO.

**Lemma D.9.** *Under event $\mathcal{E}_0 \cap \mathcal{E}_y$ and the parameter choices in Lemma D.4, if $t < \tau$, we have*

$$
\Phi(x_{t+1}) - \Phi(x_t) \leq -\frac{\eta}{4G}\|\nabla\Phi(x_t)\|^2 + \frac{2\eta}{\lambda}\|\hat{m}_t - \hat{u}_t\|^2
$$
$$
+ \frac{4\eta}{\lambda}\|\epsilon_t\|^2 + \frac{4\eta}{\lambda}\|\mathbb{E}_t[\hat{\nabla}\phi(x_t, y_t^*; \bar{\xi}_t)] - \nabla\Phi(x_t)\|^2. \tag{56}
$$

*Proof of Lemma D.9.* By Lemmas D.6 and D.13 and choice of $G$, if $t < \tau$, we have

$$
\frac{\eta I}{2G} \preceq \frac{\eta}{C_{u,0} + C_{u,1}\varrho + \lambda} \preceq H_t \preceq \frac{\eta I}{\lambda}. \tag{57}
$$

Since we choose $\eta \leq r/D$, then by Lemma D.8 we have $\|x_{t+1} - x_t\| \leq r$ if $t < \tau$. Define $\hat{\epsilon}_t$ and $\epsilon_t$ as

$$
\hat{\epsilon}_t = \hat{m}_t - \nabla\Phi(x_t) \qquad \text{and} \qquad \epsilon_t = \hat{u}_t - \mathbb{E}_t[\hat{\nabla}\phi(x_t, y_t^*; \bar{\xi}_t)]. \tag{58}
$$

For any $t < \tau$, we apply Lemma B.11 to obtain that

$$
\Phi(x_{t+1}) - \Phi(x_t) \leq \langle\nabla\Phi(x_t), x_{t+1} - x_t\rangle + \frac{L_0 + L_1\|\nabla\Phi(x_t)\|}{2}\|x_{t+1} - x_t\|^2
$$
$$
\leq \langle\nabla\Phi(x_t), x_{t+1} - x_t\rangle + \frac{L}{2}\|x_{t+1} - x_t\|^2
$$
$$
= -\nabla\Phi(x_t)^\top H_t \hat{m}_t + \frac{L}{2}\hat{m}_t^\top H_t^2 \hat{m}_t
$$
$$
\leq -\|\nabla\Phi(x_t)\|_{H_t}^2 - \nabla\Phi(x_t)^\top H_t \hat{\epsilon}_t + \frac{\eta L}{2\lambda}\|\hat{m}_t\|_{H_t}^2
$$
$$
\leq -\frac{2}{3}\|\nabla\Phi(x_t)\|_{H_t}^2 + \frac{3}{4}\|\hat{\epsilon}_t\|_{H_t}^2 + \frac{\eta L}{\lambda}\left(\|\nabla\Phi(x_t)\|_{H_t}^2 + \|\hat{\epsilon}_t\|_{H_t}^2\right)
$$
$$
\leq -\frac{1}{2}\|\nabla\Phi(x_t)\|_{H_t}^2 + \|\hat{\epsilon}_t\|_{H_t}^2
$$
$$
\leq -\frac{\eta}{4G}\|\nabla\Phi(x_t)\|^2 + \frac{\eta}{\lambda}\|\hat{\epsilon}_t\|^2
$$
$$
\leq -\frac{\eta}{4G}\|\nabla\Phi(x_t)\|^2 + \frac{2\eta}{\lambda}\|\hat{m}_t - \hat{u}_t\|^2 + \frac{4\eta}{\lambda}\|\epsilon_t\|^2 + \frac{4\eta}{\lambda}\|\mathbb{E}_t[\hat{\nabla}\phi(x_t, y_t^*; \bar{\xi}_t)] - \nabla\Phi(x_t)\|^2,
$$

where the second inequality is due to Lemma D.2 and definition of $L$ in (42); the third inequality uses (58) and (57); the fourth inequality is due to Young's inequality $a^\top Ab \leq \frac{1}{3}\|a\|_A^2 + \frac{3}{4}\|b\|_A^2$ and $\|a+b\|^2 \leq 2\|a\|_A^2 + 2\|b\|_A^2$ for any PSD matrix $A$; the fifth inequality uses the choice of $\eta \leq \lambda/6L$; the second last inequality is due to (57); and the last inequality uses (58) and Young's inequality. $\square$

The following lemma is essential for bounding the sum of the error terms $\|\epsilon_t\|^2$ before time $\tau$. Since we introduce $\mathbb{E}_t[\hat{\nabla}\phi(x_t, y_t^*; \bar{\xi}_t)]$ as part of the definition of $\epsilon_t$ (see (58)), we can directly invoke (Li et al., 2023a, Lemma C.10) to obtain the high probability bound.

**Lemma D.10** ((Li et al., 2023a, Lemma C.10)). *Denote $w_t$ as*

$$
w_{t-1} = (1 - \alpha_t)(\epsilon_{t-1} + \mathbb{E}_{t-1}[\hat{\nabla}\phi(x_{t-1}, y_{t-1}^*; \bar{\xi}_{t-1})] - \mathbb{E}_t[\hat{\nabla}\phi(x_t, y_t^*; \bar{\xi}_t)]).
$$

*Under the parameter choices in Theorem D.12, for any given $\delta \in (0, 1)$, the following holds with probability at least $1 - \delta/4$ over the randomness in $\mathcal{F}_{T+1}$ (we denote this event as $\mathcal{E}_x$):*

$$
\sum_{t=2}^{\tau} \alpha_t\langle w_{t-1}, \hat{\nabla}\phi(x_t, y_t^*; \bar{\xi}_t) - \mathbb{E}_t[\hat{\nabla}\phi(x_t, y_t^*; \bar{\xi}_t)]\rangle \leq 5\sigma_\phi^2\sqrt{(1 + \beta^2 T)\ln(4/\delta)}.
$$

The next lemma bounds the sum of the error terms $\|\epsilon_t\|^2$ before time $\tau$.

**Lemma D.11.** *Under event $\mathcal{E}_0 \cap \mathcal{E}_y \cap \mathcal{E}_x$ and the parameter choices in Lemma D.4, we have*

$$
\sum_{t=1}^{\tau-1}\|\epsilon_t\|^2 - \frac{\lambda}{128G}\|\nabla\Phi(x_t)\|^2 \leq 8\sigma_\phi^2(1/\beta + \beta T) + 20\sigma_\phi^2\sqrt{(1/\beta^2 + T)\ln(4/\delta)}
$$
$$
+ \frac{\lambda}{128G}\sum_{t=1}^{\tau-1}\|\hat{m}_t - \hat{u}_t\|^2 + \|\mathbb{E}_t[\hat{\nabla}\phi(x_t, y_t^*; \bar{\xi}_t)] - \nabla\Phi(x_t)\|^2. \tag{59}
$$

*Proof of Lemma D.11.* We first denote $w_t$ as

$$w_{t-1} = (1 - \alpha_t)(\epsilon_{t-1} + \mathbb{E}_{t-1}[\hat{\nabla}\phi(x_{t-1}, y_{t-1}^*; \bar{\xi}_{t-1})] - \mathbb{E}_t[\hat{\nabla}\phi(x_t, y_t^*; \bar{\xi}_t)]).$$

By definition of $\epsilon_t$ and the update rule (8), we have

$$\begin{aligned}
\epsilon_t &= (1 - \alpha_t)(\epsilon_{t-1} + \mathbb{E}_{t-1}[\hat{\nabla}\phi(x_{t-1}, y_{t-1}^*; \bar{\xi}_{t-1})] - \mathbb{E}_t[\hat{\nabla}\phi(x_t, y_t^*; \bar{\xi}_t)]) \\
&\quad + \alpha_t(\hat{\nabla}\phi(x_t, y_t^*; \bar{\xi}_t) - \mathbb{E}_t[\hat{\nabla}\phi(x_t, y_t^*; \bar{\xi}_t)]) \\
&= w_{t-1} + \alpha_t(\hat{\nabla}\phi(x_t, y_t^*; \bar{\xi}_t) - \mathbb{E}_t[\hat{\nabla}\phi(x_t, y_t^*; \bar{\xi}_t)]).
\end{aligned} \tag{60}$$

By choice of $\eta$ we have

$$\eta \le \frac{c_2 r\lambda}{G} \le \frac{2c_2 r}{D} \le \frac{r}{D},$$

where in the last inequality we choose small enough $c_2$. By Lemma D.8 we have $\|x_t - x_{t-1}\| \le r$ if $t \le \tau$. Then for $2 \le t \le \tau$, we apply Lemma D.6 to obtain

$$\begin{aligned}
&\|\mathbb{E}_{t-1}[\hat{\nabla}\phi(x_{t-1}, y_{t-1}^*; \bar{\xi}_{t-1})] - \mathbb{E}_t[\hat{\nabla}\phi(x_t, y_t^*; \bar{\xi}_t)]\| \\
&\le L\|x_{t-1} - x_t\| \le \frac{\eta L}{\lambda}\|\hat{m}_{t-1}\| \le \frac{\eta L}{\lambda}(\|\nabla\Phi(x_{t-1})\| + \|\hat{\epsilon}_{t-1}\|) \\
&\le \frac{\eta L}{\lambda}\left(\|\nabla\Phi(x_{t-1})\| + \|\hat{m}_{t-1} - \hat{u}_{t-1}\| + \|\epsilon_{t-1}\| + \|\mathbb{E}_{t-1}[\hat{\nabla}\phi(x_{t-1}, y_{t-1}^*; \bar{\xi}_{t-1})] - \nabla\Phi(x_{t-1})\|\right),
\end{aligned} \tag{61}$$

where the third inequality uses (58). Hence we have

$$\begin{aligned}
\|w_{t-1}\|^2 &= \|(1 - \alpha_t)(\epsilon_{t-1} + \mathbb{E}_{t-1}[\hat{\nabla}\phi(x_{t-1}, y_{t-1}^*; \bar{\xi}_{t-1})] - \mathbb{E}_t[\hat{\nabla}\phi(x_t, y_t^*; \bar{\xi}_t)])\|^2 \\
&\le (1 - \alpha_t)^2(1 + \alpha_t)\|\epsilon_{t-1}\|^2 \\
&\quad + (1 - \alpha_t)^2\left(1 + \frac{1}{\alpha_t}\right)\|\mathbb{E}_{t-1}[\hat{\nabla}\phi(x_{t-1}, y_{t-1}^*; \bar{\xi}_{t-1})] - \mathbb{E}_t[\hat{\nabla}\phi(x_t, y_t^*; \bar{\xi}_t)])\|^2 \\
&\le (1 - \alpha_t)\|\epsilon_{t-1}\|^2 + \frac{1}{\alpha_t}\|\mathbb{E}_{t-1}[\hat{\nabla}\phi(x_{t-1}, y_{t-1}^*; \bar{\xi}_{t-1})] - \mathbb{E}_t[\hat{\nabla}\phi(x_t, y_t^*; \bar{\xi}_t)])\|^2 \\
&\le (1 - \alpha_t)\|\epsilon_{t-1}\|^2 + \frac{4\eta^2 L^2}{\lambda^2 \beta}\left(\|\nabla\Phi(x_{t-1})\|^2 + \|\epsilon_{t-1}\|^2\right) \\
&\quad + \frac{4\eta^2 L^2}{\lambda^2 \beta}\left(\|\hat{m}_{t-1} - \hat{u}_{t-1}\|^2 + \|\mathbb{E}_{t-1}[\hat{\nabla}\phi(x_{t-1}, y_{t-1}^*; \bar{\xi}_{t-1})] - \nabla\Phi(x_{t-1})\|^2\right) \\
&\le \left(1 - \frac{\alpha_t}{2}\right)\|\epsilon_{t-1}\|^2 + \frac{\lambda\beta}{256G}\|\nabla\Phi(x_{t-1})\|^2 \\
&\quad + \frac{\lambda\beta}{256G}\left(\|\hat{m}_{t-1} - \hat{u}_{t-1}\|^2 + \|\mathbb{E}_{t-1}[\hat{\nabla}\phi(x_{t-1}, y_{t-1}^*; \bar{\xi}_{t-1})] - \nabla\Phi(x_{t-1})\|^2\right),
\end{aligned} \tag{62}$$

where the first inequality uses Young's inequality $\|a + b\|^2 \le (1 + c)\|a\|^2 + (1 + 1/c)\|b\|^2$ for any $c > 0$; the second inequality is due to

$$(1 - \alpha_t)^2(1 + \alpha_t) \le (1 - \alpha_t)(1 - \alpha_t^2) \le 1 - \alpha_t,$$

$$(1 - \alpha_t)^2\left(1 + \frac{1}{\alpha_t}\right) = \frac{1}{\alpha_t}(1 - \alpha_t)^2(1 + \alpha_t) \le \frac{1}{\alpha}(1 - \alpha_t) \le \frac{1}{\alpha_t};$$

the third inequality uses (61) and Young's inequality; and the last inequality is due to the choice of $G$ and $\eta$ with small enough $c_2$:

$$\eta \le \frac{c_2\lambda^{3/2}\beta}{L\sqrt{G}} \le \frac{\lambda^{3/2}\beta}{32L\sqrt{G}} \implies \frac{4\eta^2 L^2}{\lambda^2 \beta} \le \frac{\lambda\beta}{256G} \le \frac{\beta}{256} \le \frac{\beta}{2} \le \frac{\alpha_t}{2}.$$

Plugging (62) back into (60) gives

$$\begin{aligned}
\|\epsilon_t\|^2 &= \|w_{t-1}\|^2 + 2\alpha_t\langle w_{t-1}, \hat{\nabla}\phi(x_t, y_t^*; \bar{\xi}_t) - \mathbb{E}_t[\hat{\nabla}\phi(x_t, y_t^*; \bar{\xi}_t)]\rangle \\
&\quad + \alpha_t^2\|\hat{\nabla}\phi(x_t, y_t^*; \bar{\xi}_t) - \mathbb{E}_t[\hat{\nabla}\phi(x_t, y_t^*; \bar{\xi}_t)]\|^2 \\
&\le \left(1 - \frac{\alpha_t}{2}\right)\|\epsilon_{t-1}\|^2 + \frac{\lambda\beta}{256G}\|\nabla\Phi(x_{t-1})\|^2 + \alpha_t^2\sigma_\phi^2 \\
&\quad + 2\alpha_t\langle\nu_{t-1}, \hat{\nabla}\phi(x_t, y_t^*; \bar{\xi}_t) - \mathbb{E}_t[\hat{\nabla}\phi(x_t, y_t^*; \bar{\xi}_t)]\rangle \\
&\quad + \frac{\lambda\beta}{256G}\left(\|\hat{m}_{t-1} - \hat{u}_{t-1}\|^2 + \|\mathbb{E}_{t-1}[\hat{\nabla}\phi(x_{t-1}, y_{t-1}^*; \bar{\xi}_{t-1})] - \nabla\Phi(x_{t-1})\|^2\right).
\end{aligned}$$

Rearranging the above inequality, for any $2 \le t \le \tau$, we have

$$\frac{\beta}{2}\|\epsilon_{t-1}\|^2 \le \frac{\alpha_t}{2}\|\epsilon_{t-1}\|^2 \le \|\epsilon_{t-1}\|^2 - \|\epsilon_t\|^2 + \frac{\lambda\beta}{256G}\|\nabla\Phi(x_{t-1})\|^2$$
$$+ \alpha_t^2\sigma_\phi^2 + 2\alpha_t\langle\nu_{t-1}, \hat{\nabla}\phi(x_t, y_t^*; \bar{\xi}_t) - \mathbb{E}_t[\hat{\nabla}\phi(x_t, y_t^*; \bar{\xi}_t)]\rangle$$
$$+ \frac{\lambda\beta}{256G}\left(\|\hat{m}_{t-1} - \hat{u}_{t-1}\|^2 + \|\mathbb{E}_{t-1}[\hat{\nabla}\phi(x_{t-1}, y_{t-1}^*; \bar{\xi}_{t-1})] - \nabla\Phi(x_{t-1})\|^2\right).$$

Then taking summation over $t$ from 2 to $\tau$ we obtain that

$$\sum_{t=2}^{\tau}\frac{\beta}{2}\|\epsilon_{t-1}\|^2 - \frac{\lambda\beta}{256G}\|\nabla\Phi(x_{t-1})\|^2$$

$$\le \|\epsilon_1\|^2 - \|\epsilon_\tau\|^2 + \sigma_\phi^2\sum_{t=2}^{\tau}\alpha_t^2 + 2\sum_{t=2}^{\tau}\alpha_t\langle w_{t-1}, \hat{\nabla}\phi(x_t, y_t^*; \bar{\xi}_t) - \mathbb{E}_t[\hat{\nabla}\phi(x_t, y_t^*; \bar{\xi}_t)]\rangle$$

$$+ \frac{\lambda\beta}{256G}\sum_{t=2}^{\tau}\|\hat{m}_{t-1} - \hat{u}_{t-1}\|^2 + \|\mathbb{E}_{t-1}[\hat{\nabla}\phi(x_{t-1}, y_{t-1}^*; \bar{\xi}_{t-1})] - \nabla\Phi(x_{t-1})\|^2$$

$$\le 4\sigma_\phi^2(1 + \beta^2 T) + 10\sigma_\phi^2\sqrt{(1 + \beta^2 T)\ln(4/\delta)}$$

$$+ \frac{\lambda\beta}{256G}\sum_{t=2}^{\tau}\|\hat{m}_{t-1} - \hat{u}_{t-1}\|^2 + \|\mathbb{E}_{t-1}[\hat{\nabla}\phi(x_{t-1}, y_{t-1}^*; \bar{\xi}_{t-1})] - \nabla\Phi(x_{t-1})\|^2,$$

where the last inequality uses Lemmas B.7 and D.10 and the fact that $\|\epsilon_1\|^2 \le \sigma_\phi^2$. Then we complete the proof by multiplying both sides by $2/\beta$. $\qquad\square$

With Lemmas D.9 and D.11, we are ready to prove Theorem 4.1. Below is the full statement of Theorem 4.1 with detailed parameter choices, where we use $c_1, c_2, c_3$ to denote small enough constants and $C_1, C_2$ to denote large enough ones. The definitions of problem-dependent constants $\sigma_\phi, C_{\phi,0}, C_{\phi,1}, \Delta_1, L_0, L_1, L, C_\beta$ are provided in Appendix D.1.

**Theorem D.12** (Restatement of Theorem 4.1). *Suppose that Assumptions 3.2 to 3.4 hold. Let $G$ be a constant satisfying*

$$G \ge \max\left\{4\lambda, 2\sigma_\phi, 4C_{\phi,0}, \frac{C_{\phi,1}}{L_1}, \sqrt{\frac{C_1\Delta_1 L_0}{C_L}}, \frac{C_1\Delta_1 L_1}{C_L}\right\}, \tag{63}$$

*Given any $\epsilon > 0$ and $\delta \in (0, 1)$, denote $\iota := \ln(4/\delta)$, and choose*

$$0 \le \beta_{sq} \le 1, \quad \beta \le \min\left\{1, \frac{c_1\lambda\epsilon^2}{\sigma_\phi^2 G\max\{1, \sqrt{\iota}, \ln(C_\beta)\}}\right\}, \quad \gamma = \frac{2\beta}{\mu}, \tag{64}$$

$$\eta \le c_2\min\left\{\frac{r\lambda}{G}, \frac{\lambda}{6L}, \frac{\sigma_\phi\lambda\beta}{LG\max\{1, \sqrt{\iota}, \ln(1/\beta), \ln(C_\beta)\}}, \frac{\lambda^{3/2}\beta}{L\sqrt{G}}\right\}, \tag{65}$$

$$Q \ge \frac{1}{2}\max\left\{\ln\beta\Big/\ln\left(1 - \frac{\mu}{l_{g,1}}\right), \ln\left(\frac{c_3\lambda\mu^2\epsilon^2}{Gl_{g,1}^2 l_{f,0}^2}\right)\Big/\ln\left(1 - \frac{\mu}{l_{g,1}}\right)\right\}, \tag{66}$$

$$T_0 = \ln\left(\frac{\sigma_{g,1}^2\beta}{\mu^2\|y_0 - y_0^*\|^2}\right)\Big/\ln(1 - \beta), \quad T = \max\left\{\frac{1}{\beta^2}, \frac{C_2\Delta_1 G}{\eta\epsilon^2}\right\}, \tag{67}$$

*where constant $C_\beta$ is defined as*

$$C_\beta \ge \max\left\{\frac{8e\sigma_\phi^4 G^2\max\{1, \iota\}}{c_1^2\delta\lambda^2\epsilon^4}, \frac{8C_2 e\Delta_1 L\sigma_\phi G^3}{c_1 c_2\delta\lambda^2\epsilon^4}\left(1 + \frac{\sigma_\phi^2 G}{c_1\lambda\epsilon^2}\right)\max\{1, \sqrt{\iota}, \iota\},\right.$$

$$\left.\left(\frac{32e\sigma_\phi^4 G^2}{c_1^2\delta\lambda^2\epsilon^4}\right)^2, \left(\frac{48C_2 e\Delta_1 L\sigma_\phi G^3}{c_1 c_2\delta\lambda^2\epsilon^4}\left(1 + \frac{\sigma_\phi^2 G}{c_1\lambda\epsilon^2}\right)\max\{1, \sqrt{\iota}, \iota\}\right)^2\right\}.$$

*Run Algorithm 1 for $T$ iterations. Then with probability at least $1 - \delta$ over the randomness in $\mathcal{F}_{T+1}$, we have $\|\nabla\Phi(x_t)\| \le G$ for all $t \in [T]$, and $\frac{1}{T}\sum_{t=1}^{T}\|\nabla\Phi(x_t)\| \le \epsilon^2$.*

*Proof of Theorem D.12.* By Lemmas D.3, D.4 and D.10, we have $\Pr(\mathcal{E}_0 \cap \mathcal{E}_y \cap \mathcal{E}_x) \geq 1 - 3\delta/4 \geq 1 - \delta$. The following analysis is conditioned on the event $\mathcal{E}_0 \cap \mathcal{E}_y \cap \mathcal{E}_x$.

Rearranging (56) of Lemma D.9 and telescoping over $t$ from 1 to $\tau - 1$, we have

$$
\sum_{t=1}^{\tau-1} 4\|\nabla\Phi(x_t)\|^2 - \frac{64G}{\lambda}\|\epsilon_t\|^2 \leq \frac{16G}{\eta}[(\Phi(x_1) - \Phi^*) - (\Phi(x_\tau) - \Phi^*)]
$$

$$
+ \frac{32G}{\lambda}\sum_{t=1}^{\tau-1}\|\hat{m}_t - \hat{u}_t\|^2 + 2\|\mathbb{E}_t[\hat{\nabla}\phi(x_t, y_t^*; \bar{\xi}_t)] - \nabla\Phi(x_t)\|^2. \tag{68}
$$

Also, (59) of Lemma D.11 can be written as

$$
\sum_{t=1}^{\tau-1} \frac{128G}{\lambda}\|\epsilon_t\|^2 - \|\nabla\Phi(x_t)\|^2 \leq \frac{128G}{\lambda}\left(8\sigma_\phi^2(1/\beta + \beta T) + 20\sigma_\phi^2\sqrt{(1/\beta^2 + T)\ln(4/\delta)}\right)
$$

$$
+ \sum_{t=1}^{\tau-1}\|\hat{m}_t - \hat{u}_t\|^2 + \|\mathbb{E}_t[\hat{\nabla}\phi(x_t, y_t^*; \bar{\xi}_t)] - \nabla\Phi(x_t)\|^2. \tag{69}
$$

Summing (68) and (69) and rearranging gives

$$
\frac{16G}{\eta}(\Phi(x_\tau) - \Phi^*) + 3\sum_{t=1}^{\tau-1}\|\nabla\Phi(x_t)\|^2 + \frac{64G}{\lambda}\sum_{t=1}^{\tau-1}\|\epsilon_t\|^2
$$

$$
\leq \frac{16G}{\eta\lambda}\left(\lambda\Delta_1 + 64\sigma_\phi^2\left(\frac{\eta}{\beta} + \eta\beta T\right) + 160\eta\sigma_\phi^2\sqrt{(1/\beta^2 + T)\ln(4/\delta)}\right)
$$

$$
+ \left(1 + \frac{32G}{\lambda}\right)\sum_{t=1}^{\tau-1}\|\hat{m}_t - \hat{u}_t\|^2 + \left(1 + \frac{64G}{\lambda}\right)\sum_{t=1}^{\tau-1}\|\mathbb{E}_t[\hat{\nabla}\phi(x_t, y_t^*; \bar{\xi}_t)] - \nabla\Phi(x_t)\|^2
$$

$$
\leq \frac{16G}{\eta\lambda}\left(\lambda\Delta_1 + 64\sigma_\phi^2\left(\frac{\eta}{\beta} + \eta\beta T\right) + 160\eta\sigma_\phi^2\sqrt{(1/\beta^2 + T)\ln(4/\delta)}\right)
$$

$$
+ \frac{33G}{\lambda}\sum_{t=1}^{\tau-1}\|\hat{m}_t - \hat{u}_t\|^2 + \frac{65G}{\lambda}\sum_{t=1}^{\tau-1}\|\mathbb{E}_t[\hat{\nabla}\phi(x_t, y_t^*; \bar{\xi}_t)] - \nabla\Phi(x_t)\|^2,
$$

where the last inequality uses $G \geq \lambda$. By Lemma D.7, we further have

$$
\frac{16G}{\eta}(\Phi(x_\tau) - \Phi^*) + 3\sum_{t=1}^{\tau-1}\|\nabla\Phi(x_t)\|^2 + \frac{64G}{\lambda}\sum_{t=1}^{\tau-1}\|\epsilon_t\|^2
$$

$$
\leq \frac{16G}{\eta\lambda}\left(\lambda\Delta_1 + 64\sigma_\phi^2\left(\frac{\eta}{\beta} + \eta\beta T\right) + 160\eta\sigma_\phi^2\sqrt{(1/\beta^2 + T)\ln(4/\delta)}\right)
$$

$$
+ \frac{33L^2GT}{\lambda}\left(\left(1 + \frac{8\eta^2 l_{g,1}^2 L^2}{\lambda^2\mu^4\gamma^2}\right)\|y_1 - y_1^*\|^2 + \left(\frac{8\gamma}{\mu}\ln\frac{4eT}{\delta} + \frac{32\eta^2 l_{g,1}^2 L^2}{\lambda^2\mu^5\gamma}\right)\sigma_{g,1}^2\right)
$$

$$
+ \frac{33L^2G}{\lambda}\left(\frac{8\eta^2 l_{g,1}^2}{\lambda^2\mu^4\gamma^2} + \frac{2048\eta^4 l_{g,1}^4 L^2}{\lambda^4\mu^8\gamma^4}\left(2 + \ln\frac{1}{\beta}\right)\right)\sum_{t=1}^{\tau-1}\|\epsilon_t\|^2 + 2\|\mathbb{E}_t[\hat{\nabla}\phi(x_t, y_t^*; \bar{\xi}_t)] - \nabla\Phi(x_t)\|^2
$$

$$
+ \frac{66L^2G}{\lambda}\left(\frac{8\eta^2 l_{g,1}^2}{\lambda^2\mu^4\gamma^2} + \frac{2048\eta^4 l_{g,1}^4 L^2}{\lambda^4\mu^8\gamma^4}\left(2 + \ln\frac{1}{\beta}\right)\right)\sum_{t=1}^{\tau-1}\|\nabla\Phi(x_t)\|^2
$$

$$
+ \frac{65G}{\lambda}\sum_{t=1}^{\tau-1}\|\mathbb{E}_t[\hat{\nabla}\phi(x_t, y_t^*; \bar{\xi}_t)] - \nabla\Phi(x_t)\|^2. \tag{70}
$$

By Lemma D.13, we know that

$$
\frac{66L^2G}{\lambda}\left(\frac{8\eta^2 l_{g,1}^2}{\lambda^2\mu^4\gamma^2} + \frac{2048\eta^4 l_{g,1}^4 L^2}{\lambda^4\mu^8\gamma^4}\left(2 + \ln\frac{1}{\beta}\right)\right) \leq 2,
$$

Then with $G \geq \lambda$, (70) can be simplified as

$$
\frac{16G}{\eta}(\Phi(x_\tau) - \Phi^*) + \sum_{t=1}^{\tau-1} \|\nabla\Phi(x_t)\|^2
$$

$$
\leq \frac{16G}{\eta\lambda}\left(\lambda\Delta_1 + 64\sigma_\phi^2\left(\frac{\eta}{\beta} + \eta\beta T\right) + 160\eta\sigma_\phi^2\sqrt{(1/\beta^2 + T)\ln(4/\delta)}\right)
$$

$$
+ \frac{33L^2GT}{\lambda}\left(\left(1 + \frac{8\eta^2 l_{g,1}^2 L^2}{\lambda^2\mu^4\gamma^2}\right)\|y_1 - y_1^*\|^2 + \left(\frac{8\gamma}{\mu}\ln\frac{4eT}{\delta} + \frac{32\eta^2 l_{g,1}^2 L^2}{\lambda^2\mu^5\gamma}\right)\sigma_{g,1}^2\right)
$$

$$
+ \frac{67GT l_{g,1}^2 l_{f,0}^2}{\lambda\mu^2}\left(1 - \frac{\mu}{l_{g,1}}\right)^{2Q} =: I_1.
$$

(71)

By definition of $\tau$ in (41), we have

$$
\frac{16G}{\eta}(\Phi(x_\tau) - \Phi^*) > \frac{16G\psi}{\eta} = \frac{8C_L G^3}{\eta L} =: I_2.
$$

By Lemma D.14, we have $I_1 \leq I_2$, which leads to a contradiction. Thus, we must have $\tau = T + 1$ conditioned on $\mathcal{E}_0 \cap \mathcal{E}_y \cap \mathcal{E}_x$. Therefore, combining (71) and Lemma D.14 finally yields that under event $\mathcal{E}_0 \cap \mathcal{E}_y \cap \mathcal{E}_x$,

$$
\frac{1}{T}\sum_{t=1}^{T}\|\nabla\Phi(x_t)\|^2 \leq \epsilon^2.
$$

Moreover, since $\tau = T + 1$, then by Lemma D.2 we can replace $\hat{L}_T$ and $G_T$ with $L$ and $G$ respectively, in the additional requirement (49) for $\eta$. Therefore, (49) is now included in the parameter choices of Theorem D.12, which indicates that the current parameter choices are sufficient. □

## D.8 Parameter Choices for AdamBO (Theorem D.12)

The following two lemmas, Lemmas D.13 and D.14, hide complicate calculations and will be useful in the contradiction argument and upper-level convergence analysis.

**Lemma D.13.** *Under the parameter choices in Theorem D.12, we have the following facts:*

$$
\ln\frac{4eT}{\delta} \leq \ln(C_\beta), \quad \|y_1 - y_1^*\|^2 \leq \frac{17\beta\sigma_{g,1}^2}{\mu^2}\ln(C_\beta),
$$

(72)

$$
\varrho \leq \min\left\{r, \frac{1}{4L_1}\right\}, \quad C_{u,0} + C_{u,1}\varrho + \lambda \leq 2G,
$$

(73)

$$
\frac{66L^2G}{\lambda}\left(\frac{8\eta^2 l_{g,1}^2}{\lambda^2\mu^4\gamma^2} + \frac{2048\eta^4 l_{g,1}^4 L^2}{\lambda^4\mu^8\gamma^4}\left(2 + \ln\frac{1}{\beta}\right)\right) \leq 2
$$

(74)

*Proof of Lemma D.13.* We first list all the relevant parameter choices below for convenience:

$$
G \geq \max\left\{4\lambda, 2\sigma_\phi, 4C_{\phi,0}, \frac{C_{\phi,1}}{L_1}, \sqrt{\frac{C_1\Delta_1 L_0}{C_L}}, \frac{C_1\Delta_1 L_1}{C_L}\right\},
$$

$$
\beta \leq \min\left\{1, \frac{c_1\lambda\epsilon^2}{\sigma_\phi^2 G\max\{1, \sqrt{\iota}, \ln(C_\beta)\}}\right\}, \quad \gamma = \frac{2\beta}{\mu},
$$

$$
\eta \leq c_2\min\left\{\frac{r\lambda}{G}, \frac{\sigma_\phi\lambda\beta}{LG\max\{1, \sqrt{\iota}, \ln(1/\beta), \ln(C_\beta)\}}, \frac{\lambda^{3/2}\beta}{L\sqrt{G}}\right\},
$$

$$
Q \geq \frac{1}{2}\max\left\{\ln\beta \Big/ \ln\left(1 - \frac{\mu}{l_{g,1}}\right), \ln\left(\frac{c_3\lambda\mu^2\epsilon^2}{Gl_{g,1}^2 l_{f,0}^2}\right)\Big/\ln\left(1 - \frac{\mu}{l_{g,1}}\right)\right\},
$$

$$
T_0 = \ln\left(\frac{\sigma_{g,1}^2\beta}{\mu^2\|y_0 - y_0^*\|^2}\right)\Big/\ln(1 - \beta), \quad T = \max\left\{\frac{1}{\beta^2}, \frac{C_2\Delta_1 G}{\eta\epsilon^2}\right\},
$$

where $C_\beta$ is defined in (45). Now we verify the above listed facts one by one.

**Verification for** (72): $\ln(4eT/\delta) \le \ln(C_\beta)$. We focus on the dominant terms for each parameter choice when $\epsilon$ is sufficiently small. For the remaining cases, the result can be easily obtained by following the same procedure. Specifically, we consider the case where $\beta, \eta$ and $T$ are chosen as

$$\beta = \frac{c_1 \lambda \epsilon^2}{\sigma_\phi^2 G \max\{1, \sqrt{\iota}, \ln(C_\beta)\}}, \quad \eta = \frac{c_2 \sigma_\phi \lambda \beta}{LG \max\{1, \sqrt{\iota}, \ln(1/\beta), \ln(C_\beta)\}}, \quad T = \max\left\{\frac{1}{\beta^2}, \frac{C_2 \Delta_1 G}{\eta \epsilon^2}\right\}.$$

(Case 1) If $T = 1/\beta^2$, then we have

$$\ln \frac{4eT}{\delta} = \ln \frac{4e}{\delta \beta^2}$$

$$= \ln\left(\frac{4e\sigma_\phi^4 G^2 \max\{1, \iota, \ln^2(C_\beta)\}}{c_1^2 \delta \lambda^2 \epsilon^4}\right) \le \ln\left(\frac{4e\sigma_\phi^4 G^2 (\max\{1, \iota\} + \ln^2(C_\beta))}{c_1^2 \delta \lambda^2 \epsilon^4}\right)$$

$$\le \ln\left(\frac{4e\sigma_\phi^4 G^2 (\max\{1, \iota\} + 4C_\beta^{1/2})}{c_1^2 \delta \lambda^2 \epsilon^4}\right) \le \ln(C_\beta),$$

where the second inequality uses $\ln x \le 2x^{1/4}$ for $x > 0$, and the last inequality is due to

$$\frac{4e\sigma_\phi^4 G^2 \max\{1, \iota\}}{c_1^2 \delta \lambda^2 \epsilon^4} \le \frac{C_\beta}{2} \qquad \text{and} \qquad \frac{16e\sigma_\phi^4 G^2}{c_1^2 \delta \lambda^2 \epsilon^4} C_\beta^{1/2} \le \frac{C_\beta}{2}$$

since

$$C_\beta \ge \max\left\{\frac{8e\sigma_\phi^4 G^2 \max\{1, \iota\}}{c_1^2 \delta \lambda^2 \epsilon^4}, \left(\frac{32e\sigma_\phi^4 G^2}{c_1^2 \delta \lambda^2 \epsilon^4}\right)^2\right\}.$$

(Case 2) If $T = \frac{C_2 \Delta_1 G}{\eta \epsilon^2}$, then we have

$$\ln \frac{4eT}{\delta} = \ln\left(\frac{4C_2 e \Delta_1 L \sigma_\phi G^3 \max\{1, \sqrt{\iota}, \ln(1/\beta), \ln(C_\beta)\} \max\{1, \sqrt{\iota}, \ln(C_\beta)\}}{c_1 c_2 \delta \lambda^2 \epsilon^4}\right)$$

$$= \ln\left(\frac{4C_2 e \Delta_1 L \sigma_\phi G^3 (\max\{1, \sqrt{\iota}\} + \ln(1/\beta) + \ln(C_\beta))(\max\{1, \sqrt{\iota}\} + \ln(C_\beta))}{c_1 c_2 \delta \lambda^2 \epsilon^4}\right). \tag{75}$$

Also note that

$$\ln \frac{1}{\beta} = \ln\left(\frac{\sigma_\phi^2 G \max\{1, \sqrt{\iota}, \ln(C_\beta)\}}{c_1 \lambda \epsilon^2}\right) \le \ln\left(\frac{\sigma_\phi^2 G (\max\{1, \sqrt{\iota}\} + \ln(C_\beta))}{c_1 \lambda \epsilon^2}\right)$$

$$\le \frac{\sigma_\phi^2 G (\max\{1, \sqrt{\iota}\} + \ln(C_\beta))}{c_1 \lambda \epsilon^2}.$$

Then we obtain

$$(\max\{1, \sqrt{\iota}\} + \ln(1/\beta) + \ln(C_\beta))(\max\{1, \sqrt{\iota}\} + \ln(C_\beta))$$

$$\le \left(\max\{1, \sqrt{\iota}\} + \frac{\sigma_\phi^2 G (\max\{1, \sqrt{\iota}\} + \ln(C_\beta))}{c_1 \lambda \epsilon^2} + \ln(C_\beta)\right)(\max\{1, \sqrt{\iota}\} + \ln(C_\beta))$$

$$= \left(1 + \frac{\sigma_\phi^2 G}{c_1 \lambda \epsilon^2}\right)(\max\{1, \iota\} + 2\max\{1, \sqrt{\iota}\}\ln(C_\beta) + \ln^2(C_\beta))$$

$$\le \left(1 + \frac{\sigma_\phi^2 G}{c_1 \lambda \epsilon^2}\right)(\max\{1, \iota\} + 2\max\{1, \sqrt{\iota}\}C_\beta^{1/2} + 4C_\beta^{1/2}) \tag{76}$$

$$\le \left(1 + \frac{\sigma_\phi^2 G}{c_1 \lambda \epsilon^2}\right)(\max\{1, \iota\} + 6\max\{1, \sqrt{\iota}, \iota\}C_\beta^{1/2})$$

$$\le \left(1 + \frac{\sigma_\phi^2 G}{c_1 \lambda \epsilon^2}\right)\max\{1, \sqrt{\iota}, \iota\}\left(1 + 6C_\beta^{1/2}\right)$$

where the second inequality uses $\ln x \leq x^{1/2}$ and $\ln x \leq 2x^{1/4}$ for $x > 0$. Thus, plugging (76) back into (75) and we have

$$
\ln \frac{4eT}{\delta} = \ln \left( \frac{4C_2 e \Delta_1 L \sigma_\phi G^3 (\max\{1, \sqrt{\iota}\} + \ln(1/\beta))(\max\{1, \sqrt{\iota}\} + \ln(C_\beta))}{c_1 c_2 \delta \lambda^2 \epsilon^4} \right)
$$

$$
\leq \ln \left( \frac{4C_2 e \Delta_1 L \sigma_\phi G^3}{c_1 c_2 \delta \lambda^2 \epsilon^4} \left( 1 + \frac{\sigma_\phi^2 G}{c_1 \lambda \epsilon^2} \right) \max\{1, \sqrt{\iota}, \iota\} \left( 1 + 6C_\beta^{1/2} \right) \right)
$$

$$
\leq \ln(C_\beta),
$$

where the last inequality is due to

$$
\frac{4C_2 e \Delta_1 L \sigma_\phi G^3}{c_1 c_2 \delta \lambda^2 \epsilon^4} \left( 1 + \frac{\sigma_\phi^2 G}{c_1 \lambda \epsilon^2} \right) \max\{1, \sqrt{\iota}, \iota\} \leq \frac{C_\beta}{2}
$$

and

$$
\frac{24C_2 e \Delta_1 L \sigma_\phi G^3}{c_1 c_2 \delta \lambda^2 \epsilon^4} \left( 1 + \frac{\sigma_\phi^2 G}{c_1 \lambda \epsilon^2} \right) \max\{1, \sqrt{\iota}, \iota\} C_\beta^{1/2} \leq \frac{C_\beta}{2}
$$

since

$$
C_\beta \geq \max \left\{ \frac{8C_2 e \Delta_1 L \sigma_\phi G^3}{c_1 c_2 \delta \lambda^2 \epsilon^4} \left( 1 + \frac{\sigma_\phi^2 G}{c_1 \lambda \epsilon^2} \right) \max\{1, \sqrt{\iota}, \iota\}, \right.
$$

$$
\left. \left( \frac{48C_2 e \Delta_1 L \sigma_\phi G^3}{c_1 c_2 \delta \lambda^2 \epsilon^4} \left( 1 + \frac{\sigma_\phi^2 G}{c_1 \lambda \epsilon^2} \right) \max\{1, \sqrt{\iota}, \iota\} \right)^2 \right\}.
$$

**Verification for** (72): $\|y_1 - y_1^*\|^2 \leq 17\beta \sigma_{g,1}^2 \ln(C_\beta)/\mu^2$. By choice of $T_0$ and $\gamma$, we have

$$
\|y_1 - y_1^*\|^2 \leq \left( 1 - \frac{\mu\gamma}{2} \right)^{T_0} \|y_0 - y_0^*\|^2 + \frac{8\gamma \sigma_{g,1}^2}{\mu} \ln \frac{4e}{\delta}
$$

$$
\leq \frac{\beta \sigma_{g,1}^2}{\mu^2} + \frac{16\beta \sigma_{g,1}^2}{\mu^2} \ln \frac{4e}{\delta}
$$

$$
\leq \frac{17\beta \sigma_{g,1}^2}{\mu^2} \ln(C_\beta),
$$

where the last inequality uses $T \geq 1/\beta^2 \geq 1$ and $\ln(4eT/\delta) \leq \ln(C_\beta)$.

**Verification for** (73): $\varrho \leq \min\{r, 1/4L_1\}$. By Lemma D.5 and choices of $\eta, \gamma$ and $\beta$, we have

$$
\varrho^2 = \left( 1 + \frac{8\eta^2 l_{g,1}^2 L^2}{\lambda^2 \mu^4 \gamma^2} \right) \|y_1 - y_1^*\|^2 + \left( \frac{8\gamma}{\mu} \ln \frac{4eT}{\delta} + \frac{32\eta^2 l_{g,1}^2 L^2}{\lambda^2 \mu^5 \gamma} \right) \sigma_{g,1}^2
$$

$$
+ \frac{8\eta^2 l_{g,1}^2 C_{u,0}^2}{\lambda^2 \mu^4 \gamma^2} + \frac{1024\eta^4 l_{g,1}^4 L^2 C_{u,0}^2}{\lambda^4 \mu^8 \gamma^4} \left( 2 + \ln \frac{1}{\beta} \right)
$$

$$
\leq \left( 1 + \frac{2\eta^2 l_{g,1}^2 L^2}{\lambda^2 \mu^2 \beta^2} \right) \frac{17\beta \sigma_{g,1}^2}{\mu^2} \ln(C_\beta) + \left( \frac{16\beta}{\mu^2} \ln(C_\beta) + \frac{16\eta^2 l_{g,1}^2 L^2}{\lambda^2 \mu^4 \beta} \right) \sigma_{g,1}^2
$$

$$
+ \frac{2\eta^2 l_{g,1}^2 C_{u,0}^2}{\lambda^2 \mu^2 \beta^2} + \frac{64\eta^4 l_{g,1}^4 L^2}{\lambda^4 \mu^4 \beta^4} \left( 2 + \ln \frac{1}{\beta} \right)
$$

$$
\leq \left( 1 + \frac{2c_2^2 \sigma_\phi^2 l_{g,1}^2}{\mu^2 G^2} \right) \frac{17c_1 \lambda \sigma_{g,1}^2 \epsilon^2}{\mu^2 \sigma_\phi^2 G} + \left( \frac{16c_1 \lambda \epsilon^2}{\mu^2 \sigma_\phi^2 G} + \frac{16c_1 c_2^2 \lambda l_{g,1}^2 \epsilon^2}{\mu^4 G^3} \right) \sigma_{g,1}^2
$$

$$
+ \frac{2c_2^2 \sigma_\phi^2 l_{g,1}^2 C_{u,0}^2}{\mu^2 L^2 G^2} + \frac{192c_2^4 \sigma_\phi^4 l_{g,1}^4}{\mu^4 L^2 G^4} \leq \min \left\{ r, \frac{1}{4L_1} \right\},
$$

where in the last inequality we choose small enough $c_1$ and $c_2$.

**Verification of** (73): $C_{u,0} + C_{u,1}\varrho + \lambda \leq 2G$. By definitions of $C_{u,0}, C_{u,1}$ in (43) and choice of $G$, we have

$$C_{u,0} + C_{u,1}\varrho + \lambda = C_{\phi,0} + G + (C_{\phi,1} + L_1 G)\varrho + \lambda$$

$$\leq \frac{G}{4} + G + \frac{G}{2} + \frac{G}{4} = G.$$

**Verification for** (74). By choices of $\eta, \gamma$ and $\beta$, we have

$$\frac{66L^2 G}{\lambda} \left( \frac{8\eta^2 l_{g,1}^2}{\lambda^2 \mu^4 \gamma^2} + \frac{2048\eta^4 l_{g,1}^4 L^2}{\lambda^4 \mu^8 \gamma^4} \left( 2 + \ln \frac{1}{\beta} \right) \right)$$

$$\leq \frac{66L^2 G}{\lambda} \left( \frac{2\eta^2 l_{g,1}^2}{\lambda^2 \mu^2 \beta^2} + \frac{128\eta^4 l_{g,1}^4 L^2}{\lambda^4 \mu^4 \beta^4} \left( 2 + \ln \frac{1}{\beta} \right) \right) \leq \frac{66L^2 G}{\lambda} \left( \frac{2c_2^2 \sigma_\phi^2 l_{g,1}^2}{\mu^2 L^2 G^2} + \frac{384 c_2^4 \sigma_\phi^4 l_{g,1}^4}{\mu^4 L^2 G^4} \right) \leq 2,$$

where in the last inequality we choose small enough $c_2$. $\qquad\square$

**Lemma D.14.** *Denote $I_1$ and $I_2$ as*

$$I_1 := \frac{16G}{\eta\lambda} \left( \lambda\Delta_1 + 64\sigma_\phi^2 \left( \frac{\eta}{\beta} + \eta\beta T \right) + 160\eta\sigma_\phi^2 \sqrt{(1/\beta^2 + T)\ln(4/\delta)} \right)$$

$$+ \frac{33L^2 GT}{\lambda} \left( \left( 1 + \frac{8\eta^2 l_{g,1}^2 L^2}{\lambda^2 \mu^4 \gamma^2} \right) \|y_1 - y_1^*\|^2 + \left( \frac{8\gamma}{\mu} \ln \frac{4eT}{\delta} + \frac{32\eta^2 l_{g,1}^2 L^2}{\lambda^2 \mu^5 \gamma} \right) \sigma_{g,1}^2 \right) \tag{77}$$

$$+ \frac{67GT l_{g,1}^2 l_{f,0}^2}{\lambda\mu^2} \left( 1 - \frac{\mu}{l_{g,1}} \right)^{2Q},$$

$$I_2 := \frac{8 C_L G^3}{\eta L}.$$

*For any given $\epsilon > 0$, under the parameter choice in Theorem D.12, we have $I_1 \leq I_2$ and $I_1/T \leq \epsilon^2$.*

*Proof of Lemma D.14.* We first verify $I_1 \leq I_2$ and then verify $I_1/T \leq \epsilon^2$.

**Proof of $I_1 \leq I_2$.** We start to show $I_1/I_2 \leq 1$. We have

$$\frac{I_1}{I_2} \leq \frac{2L}{\lambda C_L G^2} \left( \lambda\Delta_1 + 64\sigma_\phi^2 \left( \frac{\eta}{\beta} + \eta\beta T \right) + 160\eta\sigma_\phi^2 \sqrt{(1/\beta^2 + T)\ln(4/\delta)} \right)$$

$$+ \frac{5L^3 \eta T}{\lambda C_L G^2} \left( \left( 1 + \frac{8\eta^2 l_{g,1}^2 L^2}{\lambda^2 \mu^4 \gamma^2} \right) \|y_1 - y_1^*\|^2 + \left( \frac{8\gamma}{\mu} \ln \frac{4eT}{\delta} + \frac{32\eta^2 l_{g,1}^2 L^2}{\lambda^2 \mu^5 \gamma} \right) \sigma_{g,1}^2 \right)$$

$$+ \frac{9 l_{g,1}^2 l_{f,0}^2 L \eta T}{\lambda\mu^2 C_L G^2} \left( 1 - \frac{\mu}{l_{g,1}} \right)^{2Q}$$

$$\leq \frac{\lambda\Delta_1}{8\lambda\Delta_1} + \frac{2L}{\lambda C_L G^2} \left( 64\sigma_\phi^2 \left( \frac{2\eta}{\beta} + \frac{C_2 \Delta_1 G\beta}{\epsilon^2} \right) + 160\sigma_\phi^2 \sqrt{\iota} \sqrt{\frac{5\eta^2}{2\beta^2} + \frac{1}{2} \left( \frac{C_2 \Delta_1 G\beta}{\epsilon^2} \right)^2} \right)$$

$$+ \frac{5L^3}{\lambda C_L G^2} \left( \frac{\eta}{\beta^2} + \frac{C_2 \Delta_1 G}{\epsilon^2} \right) \left( 1 + \frac{2\eta^2 l_{g,1}^2 L^2}{\lambda^2 \mu^2 \beta^2} \right) \frac{17\beta\sigma_{g,1}^2}{\mu^2} \ln(C_\beta)$$

$$+ \frac{5L^3}{\lambda C_L G^2} \left( \frac{\eta}{\beta^2} + \frac{C_2 \Delta_1 G}{\epsilon^2} \right) \left( \frac{16\beta}{\mu^2} \ln(C_\beta) + \frac{16\eta^2 l_{g,1}^2 L^2}{\lambda^2 \mu^4 \beta} \right) \sigma_{g,1}^2$$

$$+ \frac{9 l_{g,1}^2 l_{f,0}^2 L}{\lambda\mu^2 C_L G^2} \left( \frac{\eta}{\beta^2} + \frac{C_2 \Delta_1 G}{\epsilon^2} \right) \beta$$

$$\leq \frac{1}{8} + \frac{16L}{\lambda C_L G^2} \left( \frac{48 c_2 \lambda\sigma_\phi^3}{LG} + 24 c_1 C_2 \lambda\Delta_1 \right)$$

$$+ \frac{5L^3}{\lambda C_L G^2} \left( \frac{c_2 \sigma_\phi \lambda}{LG} + \frac{c_1 C_2 \lambda \Delta_1}{\sigma_\phi^2} \right) \left( 1 + \frac{2c_2^2 \sigma_\phi^2 l_{g,1}^2}{\mu^2 G^2} \right) \frac{17 \sigma_{g,1}^2}{\mu^2}$$

$$+ \frac{5L^3}{\lambda C_L G^2} \left( \frac{c_2 \sigma_\phi \lambda}{LG} + \frac{c_1 C_2 \lambda \Delta_1}{\sigma_\phi^2} \right) \left( \frac{16}{\mu^2} + \frac{2c_2^2 \sigma_\phi^2 l_{g,1}^2}{\mu^4 G^2} \right) \sigma_{g,1}^2$$

$$+ \frac{9 l_{g,1}^2 l_{f,0}^2 L}{\lambda \mu^2 C_L G^2} \left( \frac{c_2 \sigma_\phi \lambda}{LG} + \frac{c_1 C_2 \lambda \Delta_1}{\sigma_\phi^2} \right)$$

$$= \frac{1}{8} + \frac{384 L}{\lambda C_L G^2} \left( \frac{2c_2 \lambda \sigma_\phi^3}{LG} + c_1 C_2 \lambda \Delta_1 \right)$$

$$+ \left( \frac{c_2 \sigma_\phi \lambda}{LG} + \frac{c_1 C_2 \lambda \Delta_1}{\sigma_\phi^2} \right) \left( \frac{5L^3}{\lambda C_L G^2} \left( \frac{17}{\mu^2} + \frac{36 c_2^2 \sigma_\phi^2 l_{g,1}^2}{\mu^4 G^2} \right) \sigma_{g,1}^2 + \frac{9 l_{g,1}^2 l_{f,0}^2 L}{\lambda \mu^2 C_L G^2} \right)$$

$$\leq 1,$$

where the first inequality is due to (77); the second inequality uses large enough $C_1$ and (Li et al., 2023a, Lemma C.5), the fact that $\ln(4eT/\delta) \leq \ln(C_\beta)$ and $\ln(4eT/\delta) \leq \ln(C_\beta)$, and the choice of $\gamma, Q, T$ that

$$\eta T \leq \frac{\eta}{\beta^2} + \frac{C_2 \Delta_1 G}{\epsilon^2}, \quad \gamma = \frac{2\beta}{\mu}, \quad Q \geq \frac{1}{2} \ln \beta \Big/ \ln \left( 1 - \frac{\mu}{l_{g,1}} \right);$$

the third inequality uses (Li et al., 2023a, Lemma C.5), the choice of $\eta, \beta$ that

$$\frac{\eta}{\beta} \leq \frac{c_2 \sigma_\phi \lambda}{LG \max\{1, \sqrt{\iota}, \ln(C_\beta)\}}, \quad \frac{\beta}{\epsilon^2} \leq \frac{c_1 \lambda}{\sigma_\phi^2 G \max\{1, \sqrt{\iota}, \ln(C_\beta)\}};$$

and in the last inequality we choose small enough $c_1$ and $c_2$.

**Proof of $I_1/T \leq \epsilon^2$.** Last, we show $I_1/T \leq \epsilon^2$. We have

$$\frac{I_1}{T} = \frac{16 G \Delta_1}{\eta T} + \frac{1024 \sigma_\phi^2 G}{\lambda \beta T} + \frac{1024 \sigma_\phi^2 G}{\lambda} + \frac{2560 \sigma_\phi^2 G \sqrt{\iota}}{\lambda} \sqrt{\frac{1}{\beta^2 T^2} + \frac{1}{T}}$$

$$+ \frac{33 L^2 G}{\lambda} \left( \left( 1 + \frac{8 \eta^2 l_{g,1}^2 L^2}{\lambda^2 \mu^4 \gamma^2} \right) \| y_1 - y_1^* \|^2 + \left( \frac{8\gamma}{\mu} \ln \frac{4eT}{\delta} + \frac{32 \eta^2 l_{g,1}^2 L^2}{\lambda^2 \mu^5 \gamma} \right) \sigma_{g,1}^2 \right)$$

$$+ \frac{67 G l_{g,1}^2 l_{f,0}^2}{\lambda \mu^2} \left( 1 - \frac{\mu}{l_{g,1}} \right)^{2Q}$$

$$\leq \frac{16 \epsilon^2}{C_2} + \frac{3584 \sigma_\phi^2 G \max\{1, \sqrt{\iota}\}}{\lambda \beta T} + \frac{1024 \sigma_\phi^2 G}{\lambda} + \frac{2560 \sigma_\phi^2 G \sqrt{\iota}}{\lambda \sqrt{T}} + 67 c_3 \epsilon^2$$

$$+ \frac{33 L^2 G}{\lambda} \left( \left( 1 + \frac{2 \eta^2 l_{g,1}^2 L^2}{\lambda^2 \mu^2 \beta^2} \right) \frac{17 \beta \sigma_{g,1}^2}{\mu^2} \ln(C_\beta) + \left( \frac{16\beta}{\mu^2} \ln(C_\beta) + \frac{16 \eta^2 l_{g,1}^2 L^2}{\lambda^2 \mu^4 \beta} \right) \sigma_{g,1}^2 \right)$$

$$\leq \frac{16 \epsilon^2}{C_2} + \frac{7168 \sigma_\phi^2 G \max\{1, \sqrt{\iota}\} \beta}{\lambda} + 67 c_3 \epsilon^2$$

$$+ \frac{33 L^2 G}{\lambda} \left( \left( 1 + \frac{2 c_2^2 \sigma_\phi^2 l_{g,1}^2}{\mu^2 G^2} \right) \frac{17 c_1 \lambda \sigma_{g,1}^2 \epsilon^2}{\mu^2 \sigma_\phi^2 G} + \left( \frac{16 c_1 \lambda \epsilon^2}{\mu^2 \sigma_\phi^2 G} + \frac{16 c_1 c_2^2 \lambda l_{g,1}^2 \epsilon^2}{\mu^4 G^3} \right) \sigma_{g,1}^2 \right)$$

$$= \left( \frac{16}{C_2} + 7168 c_1 + 67 c_3 + \frac{33 c_1 L^2 G}{\lambda} \left( \left( 1 + \frac{2 c_2^2 \sigma_\phi^2 l_{g,1}^2}{\mu^2 G^2} \right) \frac{17 \lambda \sigma_{g,1}^2}{\mu^2 \sigma_\phi^2 G} + \left( \frac{16 \lambda}{\mu^2 \sigma_\phi^2 G} + \frac{16 c_2^2 \lambda l_{g,1}^2}{\mu^4 G^3} \right) \sigma_{g,1}^2 \right) \right) \epsilon^2$$

$$\leq \epsilon^2,$$

where the first inequality uses $\sqrt{a+b} \leq \sqrt{a} + \sqrt{b}$ for $a, b \geq 0$, the fact that $\ln(4eT/\delta) \leq \ln(C_\beta)$ and $\| y_1 - y_1^* \|^2 \leq 17 \beta \sigma_{g,1}^2 \ln(C_\beta)/\mu^2$, and the choice of $T, Q, \gamma$ that

$$T \geq \frac{C_2 \Delta_1 G}{\eta \epsilon^2}, \quad Q \geq \frac{1}{2} \ln \left( \frac{c_3 \lambda \mu^2 \epsilon^2}{G l_{g,1}^2 l_{f,0}^2} \right) \Big/ \ln \left( 1 - \frac{\mu}{l_{g,1}} \right), \quad \gamma = \frac{2\beta}{\mu};$$

the second inequality uses the choice of $T, \eta, \beta$ that

$$T \geq \frac{1}{\beta^2}, \quad \eta \leq \frac{c_2 \sigma_\phi \lambda \beta}{LG}, \quad \beta \leq \frac{c_1 \lambda \epsilon^2}{\sigma_\phi^2 G \max\{1, \sqrt{\iota}, \ln(C_\beta)\}};$$

and in the last inequality we choose small enough $c_1, c_2, c_3$ and large enough $C_2$. $\qquad \square$

# E CONVERGENCE ANALYSIS OF VR-ADAMBO (ALGORITHM 2)

In this section, we provide convergence analysis for VR-AdamBO (Algorithm 2). Before presenting the lemmas and the main theorem, we first define (or restate) a few key concepts and useful notations.

## E.1 TECHNICAL DEFINITIONS AND USEFUL NOTATIONS

**Filtration.** Let $\sigma(\cdot)$ be the $\sigma$-algebra generated by the random variables within the argument. Define $\mathcal{F}_{\text{init}}$ as the filtration for updating $y_1$ (i.e., the filtration of warm-start phase):

$$\mathcal{F}_{\text{init}} = \sigma(\tilde{\pi}_0, \dots, \tilde{\pi}_{T_0-1}),$$

For any $t \geq 2$, define $\mathcal{F}_t^x$ as the filtration of the randomness used in updating $x_t$ before the $t$-th iteration:

$$\mathcal{F}_t^x = \sigma(\mathcal{S}_1, \bar{\xi}_2, \dots, \bar{\xi}_{t-1}),$$

also define $\mathcal{F}_t^y$ as the filtration of the randomness used in updating $y_t$ when $t$ is a multiple of $I$:

$$\mathcal{F}_t^y = \sigma(\pi_t^0, \dots, \pi_t^{N-1}).$$

Additionally, let $\mathcal{F}_t$ denote the filtration of all randomness before the $t$-th iteration:

$$\mathcal{F}_t = \sigma(\mathcal{F}_{\text{init}} \cup \mathcal{F}_t^x \cup (\cup_{k<t} \mathcal{F}_k^y)).$$

**Expectation.** We use $\mathbb{E}_t[\cdot]$ to denote the conditional expectation $\mathbb{E}[\cdot \mid \mathcal{F}_t]$.

**Other Definitions.** We define the deviation of the momentum from the conditional expectation of the hypergradient estimator as

$$\epsilon_t := m_t - \mathbb{E}_t[\hat{\nabla}\phi(x_t, y_t; \bar{\xi}_t)]. \tag{78}$$

Also, let $h_t$ be the learning rate vector and $H_t$ be the learning rate matrix:

$$h_t := \frac{\eta}{\sqrt{\hat{v}_t} + \lambda} \qquad \text{and} \qquad H_t := \text{diag}(h_t).$$

Then the update rule for upper-level variable $x_t$ (line 18 of Algorithm 2) can be written as

$$x_{t+1} = x_t - h_t \odot \hat{m}_t = x_t - H_t \hat{m}_t.$$

**Stopping Time.** Given a large enough constant $G$ as defined in Theorem E.14, denote $L$ and $\psi$ as

$$L = L_0 + L_1 G \qquad \text{and} \qquad \psi = \frac{C_L G^2}{2L},$$

where constants $L_0, L_1$ and $C_L$ are defined in (14) and (43). Now we formally define the stopping time $\tau$ as

$$\tau := \min\{t \mid \Phi(x_t) - \Phi^* > \psi\} \wedge \min\{t \mid \|\epsilon_t\| > G\} \wedge (T+1). \tag{79}$$

Based on Lemma D.1, we know if $t < \tau$, we have $\Phi(x_t) - \Phi^* \leq \psi$, $\|\epsilon_t\| \leq G$ and $\|\nabla\Phi(x_t)\| \leq G$.

**Constants.** We define the following constants, which will be useful for analysis.

$$L = L_0 + L_1 G, \quad \Delta_1 = \Phi(x_1) - \Phi^*, \quad C_m = 2G + \frac{l_{g,1} l_{f,0}}{\mu} + Lr, \quad C_\eta = \frac{512e\Delta_1 \sigma_\phi LG^2}{c_1 \lambda^2 \delta^{3/2} \epsilon^3},$$

$$\varrho_{\max} = \max_{k \leq \lfloor T/I \rfloor} \|y_{kI+1} - y_{kI}^*\|, \quad \hat{\varrho}_{\max} = \max_{t \leq T} \|\hat{y}_t - y_t^*\|,$$

$$\hat{\varrho} = \left( \|\hat{y}_1 - y_0^*\| + \eta + \frac{\eta l_{g,1}}{\lambda\mu}\left(\frac{1-\nu}{\nu} + I\right)\left(2G + \frac{l_{g,1} l_{f,0}}{\mu}\right) \right) \Big/ \left(1 - \frac{\eta l_{g,1} L}{\lambda\mu}\left(\frac{1-\nu}{\nu} + I\right)\right).$$

Besides, constants $L_0, L_1$ are defined in (14), $r$ is defined in (13), and $\sigma_\phi$ is defined in Appendix D.1, respectively.

### E.2 AUXILIARY LEMMAS

First note that when $t < \tau$, some of the quantities in Algorithm 2 and Appendix E.1 are bounded almost surely. In particular, we have the following lemma.

**Lemma E.1.** *If $t < \tau$, we have*

$$\|\nabla\Phi(x_t)\| \leq G, \quad \|\epsilon_t\| \leq G, \quad h_t \preceq \frac{\eta}{\lambda};$$

*further, if $\|\hat{y}_t - y_t^*\| \leq r$, then we have*

$$\|m_t\| \leq \|\nabla\Phi(x_t)\| + \|\epsilon_t\| + L\|\hat{y}_t - y_t^*\| + \frac{l_{g,1}l_{f,0}}{\mu}\left(1 - \frac{\mu}{l_{g,1}}\right)^Q$$

*Proof of Lemma E.1.* For the first three results, the proof is the same as Lemma D.2. For the third one, if $t < \tau$ and $\|\hat{y}_t - y_t^*\| \leq r$, we have

$$\|m_t\| \leq \|m_t - \mathbb{E}_t[\hat{\nabla}\phi(x_t, \hat{y}_t; \bar{\xi}_t)]\| + \|\mathbb{E}_t[\hat{\nabla}\phi(x_t, \hat{y}_t; \bar{\xi}_t)] - \mathbb{E}_t[\hat{\nabla}\phi(x_t, y_t^*; \bar{\xi}_t)]\|$$
$$+ \|\mathbb{E}_t[\hat{\nabla}\phi(x_t, y_t^*; \bar{\xi}_t)] - \nabla\Phi(x_t)\| + \|\nabla\Phi(x_t)\|$$

$$\leq \|\nabla\Phi(x_t)\| + \|\epsilon_t\| + (L_0 + L_1\|\nabla\Phi(x_t)\|)\|\hat{y}_t - y_t^*\| + \frac{l_{g,1}l_{f,0}}{\mu}\left(1 - \frac{\mu}{l_{g,1}}\right)^Q$$

$$\leq \|\nabla\Phi(x_t)\| + \|\epsilon_t\| + L\|\hat{y}_t - y_t^*\| + \frac{l_{g,1}l_{f,0}}{\mu}\left(1 - \frac{\mu}{l_{g,1}}\right)^Q,$$

where the second inequality uses the definition of $\epsilon_t$, Lemmas B.15 and B.16, the third inequality is due to $\|\nabla\Phi(x_t)\| \leq G$ if $t < \tau$, and the definition of $\hat{\varrho}_{\max}$. $\square$

Next, we provide an upper bound for $\|y_t - y_t^*\|$ using the structure of periodic updates.

**Lemma E.2.** *For any $t \geq 1$, we have*

$$\|y_t - y_t^*\| \leq \varrho_{\max} + \frac{\eta l_{g,1}}{\lambda\mu}\sum_{i=k_t I}^{t-1}\|m_i\|, \quad where \quad \varrho_{\max} := \max_{k \leq \lfloor T/I \rfloor}\|y_{kI+1} - y_{kI}^*\|,$$

*where $k_t = \lfloor t/I \rfloor$ and we define $m_0 = 0$ for completeness.*

*Proof of Lemma E.2.* For any $k_t I + 1 \leq t \leq (k_t + 1)I$, we have

$$\|y_t - y_t^*\| \leq \|y_{kI+1} - y_{kI}^*\| + \sum_{i=k_t I}^{t-1}\|y_i^* - y_{i+1}^*\| \leq \varrho_{\max} + \sum_{i=k_t I}^{t-1}\|y_i^* - y_{i+1}^*\|$$

$$\leq \varrho_{\max} + \frac{l_{g,1}}{\mu}\sum_{i=k_t I}^{t-1}\|x_{i+1} - x_i\| \leq \varrho_{\max} + \frac{\eta l_{g,1}}{\lambda\mu}\sum_{i=k_t I}^{t-1}\|m_i\|,$$

where the second inequality uses the definition of $\varrho_{\max}$, the third inequality is due to (B.9), and the last inequality uses the update rule in Algorithm 2 and Lemma E.1. $\square$

The following lemma provides bound for the lower-level estimation error.

**Lemma E.3.** *Consider the averaging step (line 15) of Algorithm 2, for any $t \geq 1$ we have*

$$\|\hat{y}_t - y_t^*\| \leq (1-\nu)^{t-1}\|\hat{y}_1 - y_0^*\| + \frac{(1-\nu)\eta l_{g,1}}{\lambda\mu}\sum_{i=1}^{t}(1-\nu)^{t-i}\|m_{i-1}\| + \nu\sum_{i=1}^{t}(1-\nu)^{t-i}\|y_i - y_i^*\|.$$

*Proof of Lemma E.3.* Define $\hat{y}_0 = y_0$ for simplicity. By the update rule of $\hat{y}_t$, we have

$$\|\hat{y}_t - y_t^*\| = \|(1-\nu)(\hat{y}_{t-1} - y_t^*) + \nu(y_t - y_t^*)\|$$
$$= \|(1-\nu)(\hat{y}_{t-1} - y_{t-1}^*) + (1-\nu)(y_{t-1}^* - y_t^*) + \nu(y_t - y_t^*)\|$$
$$\leq (1-\nu)\|\hat{y}_{t-1} - y_{t-1}^*\| + (1-\nu)\|y_{t-1}^* - y_t^*\| + \nu\|y_t - y_t^*\|.$$

We apply the above inequality recursively to obtain

$$\|\hat{y}_t - y_t^*\| \leq (1-\nu)^{t-1}\|\hat{y}_1 - y_1^*\| + (1-\nu)\sum_{i=2}^{t}(1-\nu)^{t-i}\|y_{i-1}^* - y_i^*\| + \nu\sum_{i=2}^{t}(1-\nu)^{t-i}\|y_i - y_i^*\|$$

$$\leq (1-\nu)^{t-1}\|\hat{y}_1 - y_0^*\| + \frac{(1-\nu)\eta l_{g,1}}{\lambda\mu}\sum_{i=1}^{t}(1-\nu)^{t-i}\|m_{i-1}\| + \nu\sum_{i=1}^{t}(1-\nu)^{t-i}\|y_i - y_i^*\|,$$

where the last inequality uses $x_1 = x_0$ and Lemma B.9. $\qquad\square$

The following lemma characterizes the averaged lower-level estimation error.

**Lemma E.4.** *Under the parameter choices in Theorem E.14, if $\|\hat{y}_t - y_t^*\| \leq r$ holds for all $t$, then we have*

$$\sum_{t=1}^{\tau}\|\hat{y}_t - y_t^*\|^2 \leq \frac{6}{\nu}\|\hat{y}_1 - y_0^*\|^2 + 12\varrho_{\max}^2 T$$

$$+ \frac{72l_{g,1}^2}{\lambda^2\mu^2}\left(\frac{\eta^2}{\nu^2} + \eta^2 I^2\right)\sum_{t=1}^{\tau-1}\left(\|\nabla\Phi(x_t)\|^2 + \|\epsilon_t\|^2 + \frac{l_{g,1}^2 l_{f,0}^2}{\mu^2}\left(1 - \frac{\mu}{l_{g,1}}\right)^{2Q}\right)$$

*and*

$$\sum_{t=1}^{\tau}\|y_t - y_t^*\|^2 \leq \frac{48\eta^2 l_{g,1}^2 I^2 L^2}{\nu\lambda^2\mu^2}\|\hat{y}_1 - y_0^*\|^2 + \left(2 + \frac{96\eta^2 l_{g,1}^2 I^2 L^2}{\lambda^2\mu^2}\right)\varrho_{\max}^2 T$$

$$+ \frac{8\eta^2 l_{g,1}^2 I^2}{\lambda^2\mu^2}\left(1 + \frac{72l_{g,1}^2 L^2}{\lambda^2\mu^2}\left(\frac{\eta^2}{\nu^2} + \eta^2 I^2\right)\right)\sum_{t=1}^{\tau-1}\left(\|\nabla\Phi(x_t)\|^2 + \|\epsilon_t\|^2\right)$$

$$+ \frac{8\eta^2 l_{g,1}^2 I^2}{\lambda^2\mu^2}\left(1 + \frac{72l_{g,1}^2 L^2}{\lambda^2\mu^2}\left(\frac{\eta^2}{\nu^2} + \eta^2 I^2\right)\right)\frac{l_{g,1}^2 l_{f,0}^2}{\mu^2}\left(1 - \frac{\mu}{l_{g,1}}\right)^{2Q}.$$

*Proof of Lemma E.4.* If $t \leq \tau$, by Lemma E.2 we have

$$\|y_t - y_t^*\| \leq \varrho_{\max} + \frac{\eta l_{g,1}}{\lambda\mu}\sum_{i=k_t I}^{t-1}\|m_i\|.$$

Then we have

$$\|y_t - y_t^*\|^2 \leq \left(\varrho_{\max} + \frac{\eta l_{g,1}}{\lambda\mu}\sum_{i=k_t I}^{t-1}\|m_i\|\right)^2 \leq 2\varrho_{\max}^2 + \frac{2\eta^2 l_{g,1}^2}{\lambda^2\mu^2}\left(\sum_{i=k_t I}^{t-1}\|m_i\|\right)^2$$

$$\leq 2\varrho_{\max}^2 + \frac{2\eta^2 l_{g,1}^2 I}{\lambda^2\mu^2}\sum_{i=k_t I}^{t-1}\|m_i\|^2. \tag{80}$$

Taking summation on both sides of (80) over $t$ from 1 to $\nu$, we have

$$\sum_{t=1}^{\tau}\|y_t - y_t^*\|^2 = \sum_{k=0}^{\lfloor\nu/I\rfloor-1}\sum_{t=kI}^{\min\{(k+1)I,\nu\}}\|y_t - y_t^*\|^2$$

$$\leq \sum_{k=0}^{\lfloor\nu/I\rfloor-1}\sum_{t=kI}^{\min\{(k+1)I,\nu\}}\left(2\varrho_{\max}^2 + \frac{2\eta^2 l_{g,1}^2 I}{\lambda^2\mu^2}\sum_{i=k_t I}^{t-1}\|m_i\|^2\right) \tag{81}$$

$$\leq 2\varrho_{\max}^2(\nu-1) + \frac{2\eta^2 l_{g,1}^2 I^2}{\lambda^2\mu^2}\sum_{k=0}^{\lfloor\nu/I\rfloor-1}\sum_{t=kI}^{\min\{(k+1)I,\nu\}}\|m_t\|^2$$

$$\leq 2\varrho_{\max}^2 T + \frac{2\eta^2 l_{g,1}^2 I^2}{\lambda^2\mu^2}\sum_{t=1}^{\tau-1}\|m_t\|^2,$$

where the last inequality uses the definition of $\nu$ and $m_0 = 0$.

By Lemma E.3, we have

$$
\begin{aligned}
\sum_{t=1}^{\tau}\|\hat{y}_t - y_t^*\|^2 &\leq 3\sum_{t=1}^{\tau}(1-\nu)^{2(t-1)}\|\hat{y}_1 - y_0^*\|^2 + 3\sum_{t=1}^{\tau}\left(\frac{(1-\nu)\eta l_{g,1}}{\nu\lambda\mu}\nu\sum_{i=1}^{t}(1-\nu)^{t-i}\|m_{i-1}\|\right)^2 \\
&\quad + 3\sum_{t=1}^{\tau}\left(\nu\sum_{i=1}^{t}(1-\nu)^{t-i}\|y_i - y_i^*\|\right)^2 \\
&\leq \frac{3}{\nu}\|\hat{y}_1 - y_0^*\|^2 + \frac{3(1-\nu)^2\eta^2 l_{g,1}^2}{\nu^2\lambda^2\mu^2}\sum_{t=1}^{\tau}\nu\sum_{i=1}^{t}(1-\nu)^{t-i}\|m_{i-1}\|^2 \\
&\quad + 3\sum_{t=1}^{\tau}\nu\sum_{i=1}^{t}(1-\nu)^{t-i}\|y_i - y_i^*\|^2 \\
&\leq \frac{3}{\nu}\|\hat{y}_1 - y_0^*\|^2 + \frac{3(1-\nu)^2\eta^2 l_{g,1}^2}{\nu^2\lambda^2\mu^2}\sum_{t=1}^{\tau-1}\|m_t\|^2 + 3\sum_{t=1}^{\tau}\|y_t - y_t^*\|^2 \\
&\leq \frac{3}{\nu}\|\hat{y}_1 - y_0^*\|^2 + \frac{3(1-\nu)^2\eta^2 l_{g,1}^2}{\nu^2\lambda^2\mu^2}\sum_{t=1}^{\tau-1}\|m_t\|^2 + 3\left(2\varrho_{\max}^2 T + \frac{2\eta^2 l_{g,1}^2 I^2}{\lambda^2\mu^2}\sum_{t=1}^{\tau-1}\|m_t\|^2\right) \\
&\leq \frac{3}{\nu}\|\hat{y}_1 - y_0^*\|^2 + 6\varrho_{\max}^2 T + \frac{9l_{g,1}^2}{\lambda^2\mu^2}\left(\frac{\eta^2}{\nu^2} + \eta^2 I^2\right)\sum_{t=1}^{\tau-1}\|m_t\|^2,
\end{aligned}
$$

where the first inequality uses Young's inequality; the second inequality is due to Jensen's inequality; the third inequality uses the sum of geometric series; the fourth inequality is due to (81). By Lemma E.1, we know that

$$
\|m_t\| \leq \|\nabla\Phi(x_t)\| + \|\epsilon_t\| + L\|\hat{y}_t - y_t^*\| + \frac{l_{g,1}l_{f,0}}{\mu}\left(1 - \frac{\mu}{l_{g,1}}\right)^Q.
$$

Then we obtain

$$
\begin{aligned}
\sum_{t=1}^{\tau}\|\hat{y}_t - y_t^*\|^2 &\leq \frac{3}{\nu}\|\hat{y}_1 - y_0^*\|^2 + 6\varrho_{\max}^2 T \\
&\quad + \frac{9l_{g,1}^2}{\lambda^2\mu^2}\left(\frac{\eta^2}{\nu^2} + \eta^2 I^2\right)\sum_{t=1}^{\tau-1}\left(\|\nabla\Phi(x_t)\| + \|\epsilon_t\| + L\|\hat{y}_t - y_t^*\| + \frac{l_{g,1}l_{f,0}}{\mu}\left(1 - \frac{\mu}{l_{g,1}}\right)^Q\right)^2 \\
&\leq \frac{3}{\nu}\|\hat{y}_1 - y_0^*\|^2 + 6\varrho_{\max}^2 T \\
&\quad + \frac{36l_{g,1}^2}{\lambda^2\mu^2}\left(\frac{\eta^2}{\nu^2} + \eta^2 I^2\right)\sum_{t=1}^{\tau-1}\left(\|\nabla\Phi(x_t)\|^2 + \|\epsilon_t\|^2 + L^2\|\hat{y}_t - y_t^*\|^2 + \frac{l_{g,1}^2 l_{f,0}^2}{\mu^2}\left(1 - \frac{\mu}{l_{g,1}}\right)^{2Q}\right).
\end{aligned}
$$

Under the parameter choices in Theorem E.14, by Appendix E.5 we have

$$
\frac{36l_{g,1}^2}{\lambda^2\mu^2}\left(\frac{\eta^2}{\nu^2} + \eta^2 I^2\right)L^2 \leq \frac{1}{2}.
$$

Rearranging the above inequality yields

$$
\begin{aligned}
\sum_{t=1}^{\tau}\|\hat{y}_t - y_t^*\|^2 &\leq \frac{6}{\nu}\|\hat{y}_1 - y_0^*\|^2 + 12\varrho_{\max}^2 T \\
&\quad + \frac{72l_{g,1}^2}{\lambda^2\mu^2}\left(\frac{\eta^2}{\nu^2} + \eta^2 I^2\right)\sum_{t=1}^{\tau-1}\left(\|\nabla\Phi(x_t)\|^2 + \|\epsilon_t\|^2 + \frac{l_{g,1}^2 l_{f,0}^2}{\mu^2}\left(1 - \frac{\mu}{l_{g,1}}\right)^{2Q}\right).
\end{aligned}
\tag{82}
$$

Finally, using the previous results we conclude that

$$\sum_{t=1}^{\tau} \|y_t - y_t^*\|^2 \le 2\varrho_{\max}^2 T + \frac{2\eta^2 l_{g,1}^2 I^2}{\lambda^2 \mu^2} \sum_{t=1}^{\tau-1} \|m_t\|^2$$

$$\le 2\varrho_{\max}^2 T + \frac{2\eta^2 l_{g,1}^2 I^2}{\lambda^2 \mu^2} \sum_{t=1}^{\tau-1} \left( \|\nabla\Phi(x_t)\| + \|\epsilon_t\| + L\|\hat{y}_t - y_t^*\| + \frac{l_{g,1} l_{f,0}}{\mu} \left(1 - \frac{\mu}{l_{g,1}}\right)^Q \right)^2$$

$$\le 2\varrho_{\max}^2 T + \frac{8\eta^2 l_{g,1}^2 I^2}{\lambda^2 \mu^2} \sum_{t=1}^{\tau-1} \left( \|\nabla\Phi(x_t)\|^2 + \|\epsilon_t\|^2 + L^2\|\hat{y}_t - y_t^*\|^2 + \frac{l_{g,1}^2 l_{f,0}^2}{\mu^2} \left(1 - \frac{\mu}{l_{g,1}}\right)^{2Q} \right)$$

$$\le 2\varrho_{\max}^2 T + \frac{8\eta^2 l_{g,1}^2 I^2}{\lambda^2 \mu^2} \sum_{t=1}^{\tau-1} \left( \|\nabla\Phi(x_t)\|^2 + \|\epsilon_t\|^2 + \frac{l_{g,1}^2 l_{f,0}^2}{\mu^2} \left(1 - \frac{\mu}{l_{g,1}}\right)^{2Q} \right)$$

$$+ \frac{8\eta^2 l_{g,1}^2 I^2 L^2}{\lambda^2 \mu^2} \left( \frac{6}{\nu} \|\hat{y}_1 - y_0^*\|^2 + 12\varrho_{\max}^2 T \right.$$

$$+ \frac{72 l_{g,1}^2}{\lambda^2 \mu^2} \left( \frac{\eta^2}{\nu^2} + \eta^2 I^2 \right) \sum_{t=1}^{\tau-1} \left( \|\nabla\Phi(x_t)\|^2 + \|\epsilon_t\|^2 + \frac{l_{g,1}^2 l_{f,0}^2}{\mu^2} \left(1 - \frac{\mu}{l_{g,1}}\right)^{2Q} \right) \right)$$

$$= \frac{48\eta^2 l_{g,1}^2 I^2 L^2}{\nu \lambda^2 \mu^2} \|\hat{y}_1 - y_0^*\|^2 + \left( 2 + \frac{96\eta^2 l_{g,1}^2 I^2 L^2}{\lambda^2 \mu^2} \right) \varrho_{\max}^2 T$$

$$+ \frac{8\eta^2 l_{g,1}^2 I^2}{\lambda^2 \mu^2} \left( 1 + \frac{72 l_{g,1}^2 L^2}{\lambda^2 \mu^2} \left( \frac{\eta^2}{\nu^2} + \eta^2 I^2 \right) \right) \sum_{t=1}^{\tau-1} \left( \|\nabla\Phi(x_t)\|^2 + \|\epsilon_t\|^2 \right)$$

$$+ \frac{8\eta^2 l_{g,1}^2 I^2}{\lambda^2 \mu^2} \left( 1 + \frac{72 l_{g,1}^2 L^2}{\lambda^2 \mu^2} \left( \frac{\eta^2}{\nu^2} + \eta^2 I^2 \right) \right) \frac{l_{g,1}^2 l_{f,0}^2}{\mu^2} \left(1 - \frac{\mu}{l_{g,1}}\right)^{2Q},$$

where the first inequality uses (81), the third inequality is due to Young's inequality, and the last inequality uses (82). $\qquad \square$

The next lemma is a generalization of (Li et al., 2023a, Lemma D.4) under the bilevel optimization setting.

**Lemma E.5.** *Under the parameter choices in Theorem E.14, if $t \le \tau$, and $\eta, \nu, I, \varrho_{\max}, \hat{\varrho}_{\max}$ further satisfy*

$$\left( \left( 1 + \frac{\nu l_{g,1}}{\mu} \right) \frac{\eta}{\lambda} + \frac{\nu \eta l_{g,1} I}{\lambda \mu} \right) \max_{t \le T} \|m_t\| + \nu(\varrho_{\max} + \hat{\varrho}_{\max}) \le r, \quad \hat{\varrho}_{\max} := \max_{t \le T} \|\hat{y}_t - y_t^*\| \le r, \tag{83}$$

*then we have*

$$\|W_t\| \le \beta\sigma_\phi + \frac{2\eta L}{\lambda} \left( 1 + \frac{\nu l_{g,1}}{\mu} \right) (\|\nabla\Phi(x_{t-1})\| + \|\epsilon_{t-1}\|) + \left( 2\nu L + \frac{2\eta L^2}{\lambda} \left( 1 + \frac{\nu l_{g,1}}{\mu} \right) \right) \|\hat{y}_{t-1} - y_{t-1}^*\|$$

$$+ 2\nu L\|y_t - y_t^*\| + \frac{2\eta L l_{g,1} l_{f,0}}{\lambda\mu} \left( 1 + \frac{\nu l_{g,1}}{\mu} \right) \left( 1 - \frac{\mu}{l_{g,1}} \right)^Q.$$

*Proof of Lemma E.5.* By the definition of $W_t$, we know that

$$W_t = \beta(\hat{\nabla}\phi(x_t, \hat{y}_t; \bar{\xi}_t) - \mathbb{E}_t[\hat{\nabla}\phi(x_t, \hat{y}_t; \bar{\xi}_t)]) + (1 - \beta)\delta_t,$$

where $\delta_t$ is denoted as

$$\delta_t = \hat{\nabla}\phi(x_t, \hat{y}_t; \bar{\xi}_t) - \hat{\nabla}\phi(x_{t-1}, \hat{y}_{t-1}; \bar{\xi}_t) - \mathbb{E}_t[\hat{\nabla}\phi(x_t, \hat{y}_t; \bar{\xi}_t)] + \mathbb{E}_t[\hat{\nabla}\phi(x_{t-1}, \hat{y}_{t-1}; \bar{\xi}_t)].$$

By condition (83) and Lemma E.2, it is easy to verify that

$$\|x_t - x_{t-1}\| + \|\hat{y}_t - \hat{y}_{t-1}\| \le \left( 1 + \frac{\nu l_{g,1}}{\mu} \right) \frac{\eta}{\lambda} \|m_{t-1}\| + \nu\|y_t - y_t^*\| + \nu\|\hat{y}_{t-1} - y_{t-1}^*\|$$

$$\le \left( 1 + \frac{\nu l_{g,1}}{\mu} \right) \frac{\eta}{\lambda} \max_{t \le T} \|m_t\| + \nu \left( \varrho_{\max} + \frac{\eta l_{g,1} I}{\lambda\mu} \max_{t \le T} \|m_t\| \right) + \nu\hat{\varrho}_{\max}$$

$$\le r.$$

Then we have

$$\|\delta_t\| \leq \|\hat{\nabla}\phi(x_t, \hat{y}_t; \bar{\xi}_t) - \hat{\nabla}\phi(x_{t-1}, \hat{y}_{t-1}; \bar{\xi}_t)\| + \|\mathbb{E}_t[\hat{\nabla}\phi(x_t, \hat{y}_t; \bar{\xi}_t)] - \mathbb{E}_t[\hat{\nabla}\phi(x_{t-1}, \hat{y}_{t-1}; \bar{\xi}_t)]\|$$

$$\leq 2L(\|x_t - x_{t-1}\| + \|\hat{y}_t - \hat{y}_{t-1}\|)$$

$$\leq 2L\left(\left(1 + \frac{\nu l_{g,1}}{\mu}\right)\|x_t - x_{t-1}\| + \nu\|y_t - y_t^*\| + \nu\|\hat{y}_{t-1} - y_{t-1}^*\|\right)$$

$$\leq 2L\left(\left(1 + \frac{\nu l_{g,1}}{\mu}\right)\frac{\eta}{\lambda}\|m_{t-1}\| + \nu\|y_t - y_t^*\| + \nu\|\hat{y}_{t-1} - y_{t-1}^*\|\right)$$

$$\leq \frac{2\eta L}{\lambda}\left(1 + \frac{\nu l_{g,1}}{\mu}\right)\left(\|\nabla\Phi(x_{t-1})\| + \|\epsilon_{t-1}\| + L\|\hat{y}_{t-1} - y_{t-1}^*\| + \frac{l_{g,1}l_{f,0}}{\mu}\left(1 - \frac{\mu}{l_{g,1}}\right)^Q\right)$$

$$+ 2\nu L(\|y_t - y_t^*\| + \|\hat{y}_{t-1} - y_{t-1}^*\|)$$

$$= \frac{2\eta L}{\lambda}\left(1 + \frac{\nu l_{g,1}}{\mu}\right)(\|\nabla\Phi(x_{t-1})\| + \|\epsilon_{t-1}\|) + \left(2\nu L + \frac{2\eta L^2}{\lambda}\left(1 + \frac{\nu l_{g,1}}{\mu}\right)\right)\|\hat{y}_{t-1} - y_{t-1}^*\|$$

$$+ 2\nu L\|y_t - y_t^*\| + \frac{2\eta L l_{g,1}l_{f,0}}{\lambda\mu}\left(1 + \frac{\nu l_{g,1}}{\mu}\right)\left(1 - \frac{\mu}{l_{g,1}}\right)^Q,$$

where the second inequality uses Lemma B.15; the third inequality is due to

$$\|\hat{y}_t - \hat{y}_{t-1}\| \leq \nu\|y_t - y_t^*\| + \nu\|y_t^* - y_{t-1}^*\| + \nu\|y_{t-1}^* - \hat{y}_{t-1}\|$$

$$\leq \frac{\nu l_{g,1}}{\mu}\|x_t - x_{t-1}\| + \nu\|y_t - y_t^*\| + \nu\|y_{t-1}^* - \hat{y}_{t-1}\|;$$

the fourth inequality uses the update rule in Algorithm 2 and $h_t \preceq \eta/\lambda$ by Lemma E.13; the last inequality is again due to Lemma E.13. Thus we obtain

$$\|W_t\| \leq \beta\sigma_\phi + \frac{2\eta L}{\lambda}\left(1 + \frac{\nu l_{g,1}}{\mu}\right)(\|\nabla\Phi(x_{t-1})\| + \|\epsilon_{t-1}\|) + \left(2\nu L + \frac{2\eta L^2}{\lambda}\left(1 + \frac{\nu l_{g,1}}{\mu}\right)\right)\|\hat{y}_{t-1} - y_{t-1}^*\|$$

$$+ 2\nu L\|y_t - y_t^*\| + \frac{2\eta L l_{g,1}l_{f,0}}{\lambda\mu}\left(1 + \frac{\nu l_{g,1}}{\mu}\right)\left(1 - \frac{\mu}{l_{g,1}}\right)^Q.$$

$\square$

The following is the descent lemma for VR-AdamBO, whose proof is similar to that of Lemma D.9.

**Lemma E.6.** *Under the parameter choices in Theorem E.14, if $t < \tau$, and $\eta$, $\hat{\varrho}_{\max}$ further satisfy*

$$\eta \leq \frac{r\lambda}{\max_{t\leq T}\|m_t\|}, \quad \hat{\varrho}_{\max} \leq \min\left\{r, \frac{1}{4L_1}\right\}, \tag{84}$$

*then we have*

$$\Phi(x_{t+1}) - \Phi(x_t) \leq -\frac{\eta}{4G}\|\nabla\Phi(x_t)\|^2 + \frac{2\eta}{\lambda}\|\epsilon_t\|^2 + \frac{4\eta L^2}{\lambda}\|\hat{y}_t - y_t^*\|^2$$

$$+ \frac{4\eta}{\lambda}\|\mathbb{E}_t[\hat{\nabla}\phi(x_t, y_t^*; \bar{\xi}_t)] - \nabla\Phi(x_t)\|^2. \tag{85}$$

*Proof of Lemma E.6.* The proof is essentially the same as that of Lemma D.9, except for the last step. Define $\hat{\epsilon}_t$ and $\epsilon_t$ as

$$\hat{\epsilon}_t = m_t - \nabla\Phi(x_t) \quad \text{and} \quad \epsilon_t = m_t - \mathbb{E}_t[\hat{\nabla}\phi(x_t, \hat{y}_t; \bar{\xi}_t)]. \tag{86}$$

By choice of $\eta$ in (84), we have

$$\|x_{t+1} - x_t\| \leq \frac{\eta}{\lambda}\|m_t\| \leq \frac{r\lambda}{\lambda\max_{t\leq T}\|m_t\|}\|m_t\| \leq r.$$

Then for any $t < \tau$, by Lemma D.9 we have

$$\Phi(x_{t+1}) - \Phi(x_t) \leq -\frac{\eta}{4G}\|\nabla\Phi(x_t)\|^2 + \frac{\eta}{\lambda}\|\hat{\epsilon}_t\|^2$$

$$\leq -\frac{\eta}{4G}\|\nabla\Phi(x_t)\|^2 + \frac{2\eta}{\lambda}\|\epsilon_t\|^2 + \frac{4\eta}{\lambda}\|\mathbb{E}_t[\hat{\nabla}\phi(x_t,\hat{y}_t;\bar{\xi}_t)] - \mathbb{E}_t[\hat{\nabla}\phi(x_t,y_t^*;\bar{\xi}_t)]\|^2$$

$$+ \frac{4\eta}{\lambda}\|\mathbb{E}_t[\hat{\nabla}\phi(x_t,y_t^*;\bar{\xi}_t)] - \nabla\Phi(x_t)\|^2$$

$$\leq -\frac{\eta}{4G}\|\nabla\Phi(x_t)\|^2 + \frac{2\eta}{\lambda}\|\epsilon_t\|^2 + \frac{4\eta L^2}{\lambda}\|\hat{y}_t - y_t^*\|^2$$

$$+ \frac{4\eta}{\lambda}\|\mathbb{E}_t[\hat{\nabla}\phi(x_t,y_t^*;\bar{\xi}_t)] - \nabla\Phi(x_t)\|^2,$$

where the second inequality uses (86) and Young's inequality, the third inequality is due to Lemma B.15 and the definition of $\nu$. $\qquad\square$

The next lemma uses Optional Stopping Theorem () to bound the sum of the error terms $\|\epsilon_t\|^2$ before time $\tau$ in expectation.

**Lemma E.7.** *Under the parameter choices in Theorem E.14, if $\eta,\nu,I,\varrho_{\max},\hat{\varrho}_{\max}$ further satisfy (83) and (84), then we have*

$$\mathbb{E}\left[\sum_{t=1}^{\tau-1}\frac{3\beta}{4}\|\epsilon_t\|^2 - \frac{3\lambda\beta}{64G}\|\nabla\Phi(x_t)\|^2\right]$$

$$\leq 8\beta^2\sigma_\phi^2 T - \mathbb{E}[\|\epsilon_\tau\|^2] + 48L^2\left(\nu^2 + \frac{\lambda\beta}{400G}\right)\left(\frac{6}{\nu}\|\hat{y}_1 - y_0^*\|^2 + 12\varrho_{\max}^2 T\right)$$

$$+ 24\nu^2 L^2\left(\frac{48\eta^2 l_{g,1}^2 I^2 L^2}{\nu\lambda^2\mu^2}\|\hat{y}_1 - y_0^*\|^2 + \left(2 + \frac{96\eta^2 l_{g,1}^2 I^2 L^2}{\lambda^2\mu^2}\right)\varrho_{\max}^2 T\right)$$

$$+ \left(\frac{\lambda\beta}{32G} + \frac{24\eta^2 L^2 T}{\lambda^2}\left(1 + \frac{l_{g,1}}{\mu}\right)^2\right)\frac{l_{g,1}^2 l_{f,0}^2}{\mu^2}\left(1 - \frac{\mu}{l_{g,1}}\right)^{2Q},$$

*Proof of Lemma E.7.* By Lemma E.5 we have

$$\|W_t\|^2 \leq 6\beta^2\sigma_\phi^2 + \frac{24\eta^2 L^2}{\lambda^2}\left(1 + \frac{\nu l_{g,1}}{\mu}\right)^2(\|\nabla\Phi(x_{t-1})\|^2 + \|\epsilon_{t-1}\|^2)$$

$$+ 6\left(2\nu L + \frac{2\eta L^2}{\lambda}\left(1 + \frac{\nu l_{g,1}}{\mu}\right)\right)^2\|\hat{y}_{t-1} - y_{t-1}^*\|^2 + 24\nu^2 L^2\|y_t - y_t\|^2$$

$$+ \frac{24\eta^2 L^2 l_{g,1}^2 l_{f,0}^2}{\lambda^2\mu^2}\left(1 + \frac{\nu l_{g,1}}{\mu}\right)^2\left(1 - \frac{\mu}{l_{g,1}}\right)^{2Q}$$

$$\leq 6\beta^2\sigma_\phi^2 + \frac{\lambda\beta}{64G}(\|\nabla\Phi(x_{t-1})\|^2 + \|\epsilon_{t-1}\|^2) + 48L^2\left(\nu^2 + \frac{\lambda\beta}{400G}\right)\|\hat{y}_{t-1} - y_{t-1}^*\|^2$$

$$+ 24\nu^2 L^2\|y_t - y_t\|^2 + \frac{24\eta^2 L^2 l_{g,1}^2 l_{f,0}^2}{\lambda^2\mu^2}\left(1 + \frac{l_{g,1}}{\mu}\right)^2\left(1 - \frac{\mu}{l_{g,1}}\right)^{2Q}$$

where the first inequality uses Young's inequality; the second inequality is due to Lemma E.4, $\nu < 1$ and choices of $\eta,\beta$ such that

$$\eta = \frac{\sigma_{g,1}\sqrt{\beta}}{\sqrt{C_2}\mu} \leq \frac{\mu\lambda^{3/2}}{40L(\mu + l_{g,1})}\sqrt{\frac{\beta}{G}},$$

where in the last inequality we choose large enough $C_2$. Note that

$$\|\epsilon_t\| = (1-\beta)^2\|\epsilon_{t-1}\|^2 + \|W_t\|^2 + (1-\beta)\langle\epsilon_{t-1}, W_t\rangle.$$

Taking summation over $2 \le t \le \tau$, we obtain

$$\sum_{t=2}^{\tau} \|\epsilon_t\|^2 \le (1-\beta)^2 \sum_{t=2}^{\tau} \|\epsilon_{t-1}\|^2 + \sum_{t=2}^{\tau} \|W_t\|^2 + (1-\beta) \sum_{t=2}^{\tau} \langle \epsilon_{t-1}, W_t \rangle$$

$$\le (1-\beta)^2 \sum_{t=2}^{\tau} \|\epsilon_{t-1}\|^2 + (1-\beta) \sum_{t=2}^{\tau} \langle \epsilon_{t-1}, W_t \rangle$$

$$+ 6\beta^2 \sigma_\phi^2 (\nu-1) + \frac{\lambda\beta}{64G} \sum_{t=2}^{\tau} \|\nabla\Phi(x_{t-1})\|^2 + \|\epsilon_{t-1}\|^2$$

$$+ 48L^2 \left( \nu^2 + \frac{\lambda\beta}{400G} \right) \sum_{t=2}^{\tau} \|\hat{y}_{t-1} - y_{t-1}^*\|^2 + 24\nu^2 L^2 \sum_{t=2}^{\tau} \|y_t - y_t\|^2$$

$$+ \frac{24\eta^2 L^2 l_{g,1}^2 l_{f,0}^2 (\nu-1)}{\lambda^2 \mu^2} \left( 1 + \frac{l_{g,1}}{\mu} \right)^2 \left( 1 - \frac{\mu}{l_{g,1}} \right)^{2Q}$$

$$\le (1-\beta)^2 \sum_{t=2}^{\tau} \|\epsilon_{t-1}\|^2 + \frac{3\lambda\beta}{64G} \sum_{t=2}^{\tau} (\|\nabla\Phi(x_{t-1})\|^2 + \|\epsilon_{t-1}\|^2) + (1-\beta) \sum_{t=2}^{\tau} \langle \epsilon_{t-1}, W_t \rangle$$

$$+ 6\beta^2 \sigma_\phi^2 T + 48L^2 \left( \nu^2 + \frac{\lambda\beta}{400G} \right) \left( \frac{6}{\nu} \|\hat{y}_1 - y_0^*\|^2 + 12\varrho_{\max}^2 T \right)$$

$$+ 24\nu^2 L^2 \left( \frac{48\eta^2 l_{g,1}^2 I^2 L^2}{\nu\lambda^2\mu^2} \|\hat{y}_1 - y_0^*\|^2 + \left( 2 + \frac{96\eta^2 l_{g,1}^2 I^2 L^2}{\lambda^2\mu^2} \right) \varrho_{\max}^2 T \right)$$

$$+ \left( \frac{\lambda\beta}{32G} + \frac{24\eta^2 L^2 T}{\lambda^2} \left( 1 + \frac{l_{g,1}}{\mu} \right)^2 \right) \frac{l_{g,1}^2 l_{f,0}^2}{\mu^2} \left( 1 - \frac{\mu}{l_{g,1}} \right)^{2Q}$$

$$\le (1 - 3\beta/4) \sum_{t=2}^{\tau} \|\epsilon_{t-1}\|^2 + \frac{3\lambda\beta}{64G} \sum_{t=2}^{\tau} \|\nabla\Phi(x_{t-1})\|^2 + (1-\beta) \sum_{t=2}^{\tau} \langle \epsilon_{t-1}, W_t \rangle$$

$$+ 6\beta^2 \sigma_\phi^2 T + 48L^2 \left( \nu^2 + \frac{\lambda\beta}{400G} \right) \left( \frac{6}{\nu} \|\hat{y}_1 - y_0^*\|^2 + 12\varrho_{\max}^2 T \right)$$

$$+ 24\nu^2 L^2 \left( \frac{48\eta^2 l_{g,1}^2 I^2 L^2}{\nu\lambda^2\mu^2} \|\hat{y}_1 - y_0^*\|^2 + \left( 2 + \frac{96\eta^2 l_{g,1}^2 I^2 L^2}{\lambda^2\mu^2} \right) \varrho_{\max}^2 T \right)$$

$$+ \left( \frac{\lambda\beta}{32G} + \frac{24\eta^2 L^2 T}{\lambda^2} \left( 1 + \frac{l_{g,1}}{\mu} \right)^2 \right) \frac{l_{g,1}^2 l_{f,0}^2}{\mu^2} \left( 1 - \frac{\mu}{l_{g,1}} \right)^{2Q},$$

where the third inequality uses Lemma E.4 and the choices of $\eta$ and $\beta$, and the last inequality is due to $G \ge \lambda$. Taking expectations on both sides, rearranging the terms, and noting that

$$\mathbb{E}\left[ \sum_{t=2}^{\tau} \langle \epsilon_{t-1}, W_t \rangle \right] = 0$$

by the Optional Stopping Theorem (i.e., Lemma B.8), we have

$$\mathbb{E}\left[ \sum_{t=1}^{\tau-1} \frac{3\beta}{4} \|\epsilon_t\|^2 - \frac{3\lambda\beta}{64G} \|\nabla\Phi(x_t)\|^2 \right]$$

$$\le 8\beta^2 \sigma_\phi^2 T - \mathbb{E}[\|\epsilon_\tau\|^2] + 48L^2 \left( \nu^2 + \frac{\lambda\beta}{400G} \right) \left( \frac{6}{\nu} \|\hat{y}_1 - y_0^*\|^2 + 12\varrho_{\max}^2 T \right)$$

$$+ 24\nu^2 L^2 \left( \frac{48\eta^2 l_{g,1}^2 I^2 L^2}{\nu\lambda^2\mu^2} \|\hat{y}_1 - y_0^*\|^2 + \left( 2 + \frac{96\eta^2 l_{g,1}^2 I^2 L^2}{\lambda^2\mu^2} \right) \varrho_{\max}^2 T \right)$$

$$+ \left( \frac{\lambda\beta}{32G} + \frac{24\eta^2 L^2 T}{\lambda^2} \left( 1 + \frac{l_{g,1}}{\mu} \right)^2 \right) \frac{l_{g,1}^2 l_{f,0}^2}{\mu^2} \left( 1 - \frac{\mu}{l_{g,1}} \right)^{2Q},$$

where the inequality uses the choice of $S_1$ to derive $\mathbb{E}[\|\epsilon_1\|] \le \sigma_\phi^2/S_1 \le 2\beta^2 \sigma_\phi^2 T$. $\qquad\square$

Combing Lemmas E.6 and E.7, we obtain the following lemma.

**Lemma E.8.** *Under the parameter choices in Theorem E.14, if $\eta, \nu, I, \varrho_{\max}, \hat{\varrho}_{\max}$ further satisfy (83) and (84), then we have*

$$\mathbb{E}\left[\sum_{t=1}^{\tau-1} \|\nabla\Phi(x_t)\|^2\right] \leq \mathcal{I}, \quad \mathbb{E}[\Phi(x_\nu) - \Phi^*] \leq \frac{\eta\mathcal{I}}{8G}, \quad \mathbb{E}[\|\epsilon_\tau\|^2] \leq \frac{\lambda\beta\mathcal{I}}{16G},$$

*where $\mathcal{I}$ is defined in (89).*

*Proof of Lemma E.8.* By Lemma E.6, if $t < \tau$, then

$$\Phi(x_{t+1}) - \Phi(x_t) \leq -\frac{\eta}{4G}\|\nabla\Phi(x_t)\|^2 + \frac{2\eta}{\lambda}\|\epsilon_t\|^2 + \frac{4\eta L^2}{\lambda}\|\hat{y}_t - y_t^*\|^2$$
$$+ \frac{4\eta}{\lambda}\|\mathbb{E}_t[\hat{\nabla}\phi(x_t, y_t^*; \bar{\xi}_t)] - \nabla\Phi(x_t)\|^2.$$

Taking summation over $1 \leq t < \tau$, rearranging terms, multiplying both sides by $8G/\eta$, and taking expectation, we obtain

$$\mathbb{E}\left[\sum_{t=1}^{\tau-1} 2\|\nabla\Phi(x_t)\|^2 - \frac{8G}{\eta}\|\epsilon_t\|^2\right]$$

$$\leq \frac{8G}{\eta}\mathbb{E}[\Phi(x_1) - \Phi(x_\nu)] + \frac{32GL^2}{\lambda}\mathbb{E}\left[\sum_{t=1}^{\tau-1}\|\hat{y}_t - y_t^*\|^2\right] + \frac{32GTl_{g,1}^2 l_{f,0}^2}{\lambda\mu^2}\left(1 - \frac{\mu}{l_{g,1}}\right)^{2Q}$$

$$\leq \frac{8G}{\eta}\mathbb{E}[\Phi(x_1) - \Phi(x_\nu)] + \frac{32GL^2}{\lambda}\left(\frac{6}{\nu}\|\hat{y}_1 - y_0^*\|^2 + 12\varrho_{\max}^2 T\right)$$

$$+ \frac{32 \cdot 72Gl_{g,1}^2 L^2}{\lambda^3\mu^2}\left(\frac{\eta^2}{\nu^2} + \eta^2 I^2\right)\mathbb{E}\left[\sum_{t=1}^{\tau-1}\|\nabla\Phi(x_t)\|^2 + \|\epsilon_t\|^2\right]$$

$$+ \left(\frac{32GT}{\lambda} + \frac{32 \cdot 72GTl_{g,1}^2 L^2}{\lambda^3\mu^2}\left(\frac{\eta^2}{\nu^2} + \eta^2 I^2\right)\right)\frac{l_{g,1}^2 l_{f,0}^2}{\mu^2}\left(1 - \frac{\mu}{l_{g,1}}\right)^{2Q}$$

$$\leq \frac{8G}{\eta}\mathbb{E}[\Phi(x_1) - \Phi(x_\nu)] + \frac{32GL^2}{\lambda}\left(\frac{6}{\nu}\|\hat{y}_1 - y_0^*\|^2 + 12\varrho_{\max}^2 T\right)$$

$$+ \mathbb{E}\left[\sum_{t=1}^{\tau-1}\frac{1}{4}\|\nabla\Phi(x_t)\|^2 + \frac{4G}{\lambda}\|\epsilon_t\|^2\right] + \left(\frac{32GT}{\lambda} + \frac{32 \cdot 72GTl_{g,1}^2 L^2}{\lambda^3\mu^2}\left(\frac{\eta^2}{\nu^2} + \eta^2 I^2\right)\right)\frac{l_{g,1}^2 l_{f,0}^2}{\mu^2}\left(1 - \frac{\mu}{l_{g,1}}\right)^{2Q},$$
$$(87)$$

where the second inequality uses Lemma E.4, and the last inequality is due to the choices of $\eta$ and $\beta$. Also, by Lemma E.7 we have

$$\mathbb{E}\left[\sum_{t=1}^{\tau-1}\frac{12G}{\lambda}\|\epsilon_t\|^2 - \frac{3}{4}\|\nabla\Phi(x_t)\|^2\right]$$

$$\leq \frac{128G\sigma_\phi^2\beta T}{\lambda} - \frac{16G}{\lambda\beta}\mathbb{E}[\|\epsilon_\tau\|^2] + \frac{16 \cdot 48GL^2}{\lambda\beta}\left(\nu^2 + \frac{\lambda\beta}{400G}\right)\left(\frac{6}{\nu}\|\hat{y}_1 - y_0^*\|^2 + 12\varrho_{\max}^2 T\right)$$

$$+ \frac{16 \cdot 24\nu^2 GL^2}{\lambda\beta}\left(\frac{48\eta^2 l_{g,1}^2 I^2 L^2}{\nu\lambda^2\mu^2}\|\hat{y}_1 - y_0^*\|^2 + \left(2 + \frac{96\eta^2 l_{g,1}^2 I^2 L^2}{\lambda^2\mu^2}\right)\varrho_{\max}^2 T\right)$$

$$+ \frac{16G}{\lambda\beta}\left(\frac{\lambda\beta}{32G} + \frac{24\eta^2 L^2 T}{\lambda^2}\left(1 + \frac{l_{g,1}}{\mu}\right)^2\right)\frac{l_{g,1}^2 l_{f,0}^2}{\mu^2}\left(1 - \frac{\mu}{l_{g,1}}\right)^{2Q}.$$
$$(88)$$

Then summing (87) + (88) gives

$$\mathbb{E}\left[\sum_{t=1}^{\tau-1}\|\nabla\Phi(x_t)\|^2\right] + \frac{8G}{\eta}\mathbb{E}[\Phi(x_\nu)-\Phi^*] + \frac{16G}{\lambda\beta}\mathbb{E}[\|\epsilon_\tau\|^2]$$

$$\leq \frac{8G\Delta_1}{\eta} + \frac{128G\sigma_\phi^2\beta T}{\lambda} + \left(\frac{32GL^2}{\lambda} + \frac{16\cdot 48GL^2}{\lambda\beta}\left(\nu^2 + \frac{\lambda\beta}{400G}\right)\right)\left(\frac{6}{\nu}\|\hat{y}_1 - y_0^*\|^2 + 12\varrho_{\max}^2 T\right)$$

$$+ \frac{16\cdot 24\nu^2 GL^2}{\lambda\beta}\left(\frac{48\eta^2 l_{g,1}^2 I^2 L^2}{\nu\lambda^2\mu^2}\|\hat{y}_1 - y_0^*\|^2 + \left(2 + \frac{96\eta^2 l_{g,1}^2 I^2 L^2}{\lambda^2\mu^2}\right)\varrho_{\max}^2 T\right)$$

$$+ \left[\left(\frac{32GT}{\lambda} + \frac{32\cdot 72GT l_{g,1}^2 L^2}{\lambda^3\mu^2}\left(\frac{\eta^2}{\nu^2} + \eta^2 I^2\right)\right)\right.$$

$$\left. + \frac{16G}{\lambda\beta}\left(\frac{\lambda\beta}{32G} + \frac{24\eta^2 L^2 T}{\lambda^2}\left(1 + \frac{l_{g,1}}{\mu}\right)^2\right)\right]\frac{l_{g,1}^2 l_{f,0}^2}{\mu^2}\left(1 - \frac{\mu}{l_{g,1}}\right)^{2Q}$$

$$:= \mathcal{I},$$

(89)

which implies that

$$\mathbb{E}\left[\sum_{t=1}^{\tau-1}\|\nabla\Phi(x_t)\|^2\right] \leq \mathcal{I}, \quad \mathbb{E}[\Phi(x_\nu)-\Phi^*] \leq \frac{\eta\mathcal{I}}{8G}, \quad \mathbb{E}[\|\epsilon_\tau\|^2] \leq \frac{\lambda\beta\mathcal{I}}{16G}.$$

$\square$

### E.3 LOWER-LEVEL ERROR CONTROL

In this section, we aim to provide high probability bound for the lower-level estimation error. First, we present the following high probability guarantee for SNAG (Algorithm 5), as implied by (Gong et al., 2024b, Lemmas C.3 and C.6).

**Lemma E.9** (SNAG). *Suppose that Assumptions 3.2, 3.3 and 3.5 hold. Let $\{\tilde{y}_t\}$ be the iterates generated by Algorithm 5 with constant learning rate $\gamma \leq 1/2l_{g,1}$. Then for any given $\delta \in (0,1)$ and any fixed $t \geq 1$, the following holds with probability at least $1 - \delta$ over the randomness in $\tilde{\mathcal{F}}_{T_0}^y$:*

$$\|\tilde{y}_t - y^*(\tilde{x})\|^2 \leq \frac{3}{\mu\gamma}\left(1 - \frac{\sqrt{\mu\gamma}}{4}\right)^t\|\tilde{y}_0 - y^*(\tilde{x})\|^2 + \frac{4\gamma\sigma_{g,1}^2}{\mu}\ln\frac{e}{\delta}.$$

*Proof of Lemma E.9.* We will use a short hand $\tilde{y}^* = y^*(\tilde{x})$. By (Gong et al., 2024b, Lemmas C.6 and C.3), we have

$$V_t \leq \left(1 - \frac{\sqrt{\mu\gamma}}{4}\right)^t V_0 + \frac{4\gamma\sigma_{g,1}^2}{\mu}\ln\frac{eT_0}{\delta},$$

where $V_t$ is the notation defined in (Gong et al., 2024b, Lemmas C.3) which satisfy the following by noting that $\tilde{y}_{-1} = \tilde{y}_0$ and $\mu\gamma \leq 1$:

$$V_t \geq \frac{\mu}{2}\|\tilde{y}_t - \tilde{y}^*\|^2, \quad V_0 \leq \frac{1 + (1-\sqrt{\mu\gamma})^2}{2\gamma}\|\tilde{y}_0 - \tilde{y}^*\|^2 \leq \frac{3}{2\gamma}\|\tilde{y}_0 - \tilde{y}^*\|^2.$$

Therefore, we obtain

$$\frac{\mu}{2}\|\tilde{y}_t - \tilde{y}^*\|^2 \leq \frac{3}{2\gamma}\left(1 - \frac{\sqrt{\mu\gamma}}{4}\right)^t\|\tilde{y}_0 - \tilde{y}^*\|^2 + 2\gamma\sigma_{g,1}^2\ln\frac{eT_0}{\delta}.$$

Rearranging the above inequality yields the result. $\square$

**Lemma E.10.** *Under the parameter choices in Theorem E.14, for any given $\delta \in (0,1)$, the following holds with probability at least $1 - I\delta/8T$ over the randomness in $\sigma(\mathcal{F}_{init} \cup (\cup_{t\leq T}\mathcal{F}_t^y))$ (we denote this event as $\mathcal{E}_y$):*

$$\|y_t - y_t^*\| \leq \eta + \frac{\eta l_{g,1}}{\lambda\mu}\sum_{i=k_t I}^{t-1}\|m_i\|,$$

*where $k_t = \lfloor t/I \rfloor$ and we define $m_0 = 0$ for completeness.*

*Proof of Lemma E.10.* By line 3 of Algorithm 2, Lemma E.9 and Appendix E.5, with probability at least $1 - \delta/8T$ over the randomness in $\mathcal{F}_{\text{init}}$ we have

$$\|y_1 - y_0^*\|^2 \leq \frac{3}{\mu\gamma} \left(1 - \frac{\sqrt{\mu\gamma}}{4}\right)^{T_0} \|y_0 - y_0^*\|^2 + \frac{4\gamma\sigma_{g,1}^2}{\mu} \ln \frac{8eT}{\delta} \leq \frac{\eta^2}{2} + \frac{\eta^2}{2} = \eta^2,$$

which gives $\|y_1 - y_0^*\| \leq \eta$. Then for any $1 \leq t \leq I$, we have

$$\|y_t - y_t^*\| \leq \|y_1 - y_0^*\| + \sum_{i=0}^{t-1} \|y_i^* - y_{i+1}^*\| \leq \eta + \sum_{i=0}^{t-1} \|y_i^* - y_{i+1}^*\|$$

$$\leq \eta + \frac{l_{g,1}}{\mu} \sum_{i=0}^{t-1} \|x_{i+1} - x_i\| \leq \eta + \frac{\eta l_{g,1}}{\lambda\mu} \sum_{i=0}^{t-1} \|m_i\|.$$

Also, under the parameter choices in Theorem E.14, Lemma E.12 shows that $\|y_{kI+1} - y_{kI}^*\| \leq \eta$ for all $k$. Similarly, for $k_t I + 1 \leq t \leq (k_t + 1)I$ we have

$$\|y_t - y_t^*\| \leq \eta + \frac{\eta l_{g,1}}{\lambda\mu} \sum_{i=k_t I}^{t-1} \|m_i\|.$$

$\square$

**Lemma E.11.** *Consider Algorithm 2 for $1 \leq t \leq I$. Under event $\mathcal{E}_y$ and the parameter choices in Theorem E.14, if $t \leq \tau$, we have*

$$\|\hat{y}_t - y_t^*\| \leq \hat{\varrho} := \left(\|\hat{y}_1 - y_0^*\| + \eta + \frac{\eta l_{g,1}}{\lambda\mu} \left(\frac{1-\nu}{\nu} + I\right)\left(2G + \frac{l_{g,1}l_{f,0}}{\mu}\right)\right) \bigg/ \left(1 - \frac{\eta l_{g,1}L}{\lambda\mu}\left(\frac{1-\nu}{\nu} + I\right)\right); \tag{90}$$

*if $t < \tau$, we have*

$$\|m_t\| \leq C_m := 2G + \frac{l_{g,1}l_{f,0}}{\mu} + Lr. \tag{91}$$

*Proof of Lemma E.11.* For $0 \leq t \leq I$, we will use induction to show that if $t < \tau$, then

$$\|\hat{y}_t - y_t^*\| \leq \hat{\varrho}, \qquad \text{and} \qquad \|m_t\| \leq C_m.$$

**Base Case.** For $t = 1$, it is easy to check that

$$\|\hat{y}_1 - y_1^*\| = \|\hat{y}_1 - y_0^*\| \leq \hat{\varrho},$$

where the first equality uses $x_1 = x_0$. Also, since $\hat{\varrho} \leq r$ and thus $\|\hat{y}_1 - y_1^*\| \leq r$ by Appendix E.5, then we have

$$\|m_1\| \leq \|m_1 - \mathbb{E}_1[\hat{\nabla}\phi(x_1, \hat{y}_1; \bar{\xi}_1)]\| + \|\mathbb{E}_1[\hat{\nabla}\phi(x_1, \hat{y}_1; \bar{\xi}_1)] - \mathbb{E}_1[\hat{\nabla}\phi(x_1, y_1^*; \bar{\xi}_1)]\|$$

$$+ \|\mathbb{E}_1[\hat{\nabla}\phi(x_1, y_1^*; \bar{\xi}_1)] - \nabla\Phi(x_1)\| + \|\nabla\Phi(x_1)\|$$

$$\leq \|\epsilon_1\| + (L_0 + L\|\nabla\Phi(x_1)\|)\|\hat{y}_1 - y_1^*\| + \frac{l_{g,1}l_{f,0}}{\mu}\left(1 - \frac{\mu}{l_{g,1}}\right)^Q + \|\nabla\Phi(x_1)\|$$

$$\leq 2G + \frac{l_{g,1}l_{f,0}}{\mu} + L\hat{\varrho} \leq 2G + \frac{l_{g,1}l_{f,0}}{\mu} + Lr,$$

where the second inequality uses Lemmas B.15 and B.16, the third inequality is due to the induction hypothesis and the definition of $\tau$, and the last inequality again uses $\hat{\varrho} \leq r$ by Appendix E.5. Thus, the base case $t = 0$ holds.

**Induction Step.** Suppose the induction hypothesis holds for $t \leq k-1$ with $k < \tau$, then for $t = k$ we have

$$\|\hat{y}_k - y_k^*\| \leq (1-\nu)^k \|\hat{y}_1 - y_0^*\| + \frac{(1-\nu)\eta l_{g,1}}{\lambda\mu} \sum_{i=1}^{k} (1-\nu)^{k-i} \|m_{i-1}\| + \nu \sum_{i=1}^{k} (1-\nu)^{k-i} \|y_i - y_i^*\|$$

$$\leq (1-\nu)^k \|\hat{y}_1 - y_0^*\| + \frac{(1-\nu)\eta l_{g,1}}{\lambda\mu} \sum_{i=1}^{k} (1-\nu)^{k-i} \|m_{i-1}\|$$

$$+ \nu \sum_{i=1}^{k} (1-\nu)^{k-i} \left( \eta + \frac{\eta l_{g,1}}{\lambda\mu} \sum_{j=0}^{i-1} \|m_j\| \right)$$

$$\leq \left( \|\hat{y}_1 - y_0^*\| + \eta + \frac{\eta l_{g,1}}{\lambda\mu} \left( \frac{1-\nu}{\nu} + I \right) \left( 2G + \frac{l_{g,1} l_{f,0}}{\mu} \right) \right) \bigg/ \left( 1 - \frac{\eta l_{g,1} L}{\lambda\mu} \left( \frac{1-\nu}{\nu} + I \right) \right) = \hat{\varrho},$$

where the first inequality uses Lemma E.3, the second inequality is due to Lemma E.10 and $k \leq I$, and the last inequality uses the induction hypothesis, the sum of geometric series, and the definition of $\hat{\varrho}$. Also, we have

$$\|m_k\| \leq \|m_k - \mathbb{E}_k[\hat{\nabla}\phi(x_k, \hat{y}_k; \bar{\xi}_k)]\| + \|\mathbb{E}_k[\hat{\nabla}\phi(x_k, \hat{y}_k; \bar{\xi}_k)] - \mathbb{E}_k[\hat{\nabla}\phi(x_k, y_k^*; \bar{\xi}_k)]\|$$

$$+ \|\mathbb{E}_k[\hat{\nabla}\phi(x_k, y_k^*; \bar{\xi}_k)] - \nabla\Phi(x_k)\| + \|\nabla\Phi(x_k)\|$$

$$\leq \|\epsilon_k\| + (L_0 + L\|\nabla\Phi(x_k)\|)\|\hat{y}_k - y_k^*\| + \frac{l_{g,1} l_{f,0}}{\mu} \left( 1 - \frac{\mu}{l_{g,1}} \right)^Q + \|\nabla\Phi(x_k)\|$$

$$\leq 2G + \frac{l_{g,1} l_{f,0}}{\mu} + L\hat{\varrho} \leq 2G + \frac{l_{g,1} l_{f,0}}{\mu} + Lr,$$

where the second inequality uses Lemmas B.15 and B.16, the third inequality is due to the induction hypothesis and the definition of $\tau$, and the last inequality again uses $\hat{\varrho} \leq r$ by Appendix E.5. Therefore, the induction step is complete.

$\square$

**Lemma E.12.** *Under event $\mathcal{E}_y$ and the parameter choices in Theorem E.14, if $t \leq \tau$, then $\|\hat{y}_t - y_t^*\| \leq \hat{\varrho}$; and if $t < \tau$, then $\|m_t\| \leq C_m$; where $\hat{\varrho}$ and $C_m$ are defined in* (90) *and* (91).

*Proof of Lemma E.12.* By Lemma E.11, we know that the statement of Lemma E.12 holds true for $0 \leq t \leq I$. Now we consider the case for $I + 1 \leq t \leq 2I$. First, by Lemma E.10 we know that

$$\|y_I - y_I^*\| \leq \eta + \frac{\eta l_{g,1}}{\lambda\mu} \sum_{i=0}^{I-1} \|m_i\| \leq \eta + \frac{\eta l_{g,1} I}{\lambda\mu} C_m.$$

Then by choice of $N$ and Appendix E.5 we have

$$\|y_{I+1} - y_I^*\|^2 \leq \frac{3}{\mu\gamma} \left( 1 - \frac{\sqrt{\mu\gamma}}{4} \right)^N \|y_I - y_I^*\|^2 + \frac{4\gamma\sigma_{g,1}^2}{\mu} \ln\frac{8eT}{\delta} \leq \frac{\eta^2}{2} + \frac{\eta^2}{2} = \eta^2,$$

which gives $\|y_{I+1} - y_I^*\| \leq \eta$. Now we follow the same procedure as Lemmas E.10 and E.11 to obtain the result for $I + 1 \leq t \leq 2I$. For the general case $k_t I + 1 \leq t \leq \min\{(k_t+1)I, T\}$, where $1 \leq k_t \leq \lfloor T/I \rfloor$, we can easily obtain the results by repeating the previous steps. $\square$

With Lemma E.12, some of the important quantities in Algorithm 2 are well bounded before time $\tau$.

**Lemma E.13.** *Under event $\mathcal{E}_y$ and the parameter choices in Theorem E.14, if $t < \tau$, we have*

$$\hat{v}_t \preceq (C_{u,0} + C_{u,1}\hat{\varrho})^2, \quad \frac{\eta}{C_{u,0} + C_{u,1}\hat{\varrho} + \lambda} \preceq h_t \preceq \frac{\eta}{\lambda},$$

$$\|m_t\| \leq \|\nabla\Phi(x_t)\| + \|\epsilon_t\| + L\|\hat{y}_t - y_t^*\| + \frac{l_{g,1} l_{f,0}}{\mu} \left( 1 - \frac{\mu}{l_{g,1}} \right)^Q \leq C_m;$$

*if $t \leq \tau$, we have*

$$\|\hat{y}_t - y_t^*\| \leq \hat{\varrho}, \quad \|\hat{\nabla}\phi(x_t, \hat{y}_t; \bar{\xi}_t) - \mathbb{E}_t[\hat{\nabla}\phi(x_t, \hat{y}_t; \bar{\xi}_t)]\| \leq \sigma_\phi.$$

*Proof of Lemma E.13.* Note that if $t < \tau$, by Lemma E.1 we have

$$\|m_t\| \le \|\nabla\Phi(x_t)\| + \|\epsilon_t\| + L\|\hat{y}_t - y_t^*\| + \frac{l_{g,1}l_{f,0}}{\mu}\left(1 - \frac{\mu}{l_{g,1}}\right)^Q$$

$$\le 2G + L\hat{\varrho} + \frac{l_{g,1}l_{f,0}}{\mu} \le 2G + Lr + \frac{l_{g,1}l_{f,0}}{\mu} = C_m,$$

where the second inequality uses Lemma E.12, the third inequality is due to $\hat{\varrho} \le r$ by Appendix E.5, and the last inequality uses the definition of $C_m$. The remaining terms can be bounded in a similar way as Lemma D.6. $\square$

### E.4 PROOF OF THEOREM 5.1

With Lemmas E.8 and E.13, we are ready to prove Theorem E.14. Below is the full statement of Theorem 5.1 with detailed parameter choices, where we use $c_1, c_2$ to denote small enough constants and $C_1, C_2$ to denote large enough ones. The definitions of problem-dependent constants $\sigma_\phi, C_{\phi,0}, C_{\phi,1}, \Delta_1, L_0, L_1, L, C_m, C_\eta$ are provided in Appendices D.1 and E.1.

**Theorem E.14.** *Suppose that Assumptions 3.2, 3.3 and 3.5 hold. Let $G$ be a constant satisfying*

$$G \ge \max\left\{4\lambda, 2\sigma_\phi, 4C_{\phi,0}, \frac{C_{\phi,1}}{L_1}, \sqrt{\frac{C_1\Delta_1 L_0}{C_L\delta}}, \frac{C_1\Delta_1 L_1}{C_L\delta}\right\}, \tag{92}$$

*Given any $\epsilon > 0$ and $\delta \in (0,1)$, choose*

$$\eta \le c_1 \cdot \min\left\{\frac{r\lambda}{G}, \frac{\lambda}{L}, \frac{\sigma_{g,1}^2 G^2\delta}{\lambda\Delta_1\mu^2}, \frac{\lambda^2\sqrt{\delta}\epsilon}{\sigma_\phi GL}\right\}, \quad 0 \le \beta_{\text{sq}} \le 1, \quad \beta = \frac{C_2\mu^2\eta^2}{\sigma_{g,1}^2},$$

$$\gamma = \frac{c_2\beta}{\mu\max\{1, \ln(C_\eta)\}}, \quad \nu = \sqrt{\beta}, \quad I = \frac{1}{\sqrt{\beta}}, \quad T = \frac{64G\Delta_1}{\eta\delta\epsilon^2}, \quad S_1 \ge \frac{1}{2\beta^2 T},$$

$$T_0 \ge \ln\left(\frac{\mu\gamma\eta^2}{6\|y_0 - y_0^*\|^2}\right) \Big/ \ln\left(1 - \frac{\sqrt{\mu\gamma}}{4}\right), \quad N \ge \ln\left(\frac{\mu\gamma\eta^2}{6(\eta + \eta l_{g,1}IC_m/\lambda\mu)^2}\right) \Big/ \ln\left(1 - \frac{\sqrt{\mu\gamma}}{4}\right),$$

$$Q \ge \frac{\ln\left(\frac{2\mu^2 G\Delta_1}{\eta l_{g,1}^2 l_{f,0}^2}\right)}{2\ln\left[\left(\frac{32GT}{\lambda} + \frac{32\cdot72GTl_{g,1}^2 L^2}{\lambda^3\mu^2}\left(\frac{\eta^2}{\nu^2} + \eta^2 I^2\right)\right) + \frac{16G}{\lambda\beta}\left(\frac{\lambda\beta}{32G} + \frac{24\eta^2 L^2 T}{\lambda^2}\left(1 + \frac{l_{g,1}}{\mu}\right)^2\right)\right]},$$

*where $C_\eta$ is defined as*

$$C_\eta = \frac{512e\Delta_1\sigma_\phi LG^2}{c_1\lambda^2\delta^{3/2}\epsilon^3}.$$

*Run Algorithm 2 for $T$ iterations. Then with probability at least $1 - \delta$ over the randomness in $\mathcal{F}_{T+1}$, we have $\|\nabla\Phi(x_t)\| \le G$ for all $t \in [T]$, and $\frac{1}{T}\sum_{t=1}^T\|\nabla\Phi(x_t)\| \le \epsilon^2$.*

*Proof of Theorem E.14.* By Lemma E.8, we have

$$\mathbb{E}\left[\sum_{t=1}^{\tau-1}\|\nabla\Phi(x_t)\|^2\right] \le \mathcal{I}, \quad \mathbb{E}[\Phi(x_\nu) - \Phi^*] \le \frac{\eta\mathcal{I}}{8G}, \quad \mathbb{E}[\|\epsilon_\tau\|^2] \le \frac{\lambda\beta\mathcal{I}}{16G}.$$

First, note that if $\tau = \tau_1 \le T$, we know $\Phi(x_\tau) - \Phi^* > \psi$ by the definition of $\tau$. Then we have

$$\Pr(\tau = \tau_1 \le T) \le \Pr(\Phi(x_\tau) - \Phi^* > \psi) \le \frac{\mathbb{E}[\Phi(x_\tau) - \Phi^*]}{\psi} \le \frac{\eta\mathcal{I}}{8G\psi} = \frac{\eta L\mathcal{I}}{4C_L G^3},$$

where the second inequality uses Markov's inequality. (40)

Similarly, if $\tau_2 = \tau \le T$, we know $\|\epsilon_t\| > G$ by the definition of $\tau$. Then we also have

$$\Pr(\tau = \tau_2 \le T) \le \Pr(\|\epsilon_\tau\| > G) = \Pr(\|\epsilon_\tau\|^2 > G^2) \le \frac{\mathbb{E}[\|\epsilon_\tau\|^2]}{G^2} \le \frac{\lambda\beta\mathcal{I}}{16G^3}.$$

Under event $\mathcal{E}_y$, we further have

$$\mathcal{I} \leq \frac{16G\Delta_1}{\eta}, \tag{93}$$

which implies that

$$\frac{\eta L\mathcal{I}}{4C_L G^3} \leq \frac{4\Delta_1 L}{C_L G^2} = \frac{4\Delta_1 L_0}{C_L G^2} + \frac{4\Delta_1 L_1}{C_L G} \leq \frac{8\delta}{C_1} \leq \frac{\delta}{16},$$

where the first equality use the definition of $L$, the second inequality is due to the choice of $G$, and in the last inequality we choose large enough $C_1$; (93) also implies that

$$\frac{\lambda\beta\mathcal{I}}{16G^3} \leq \frac{\lambda\beta\Delta_1}{\eta G^2} = \frac{C_2\lambda\mu^2\Delta_1\eta}{\sigma_{g,1}^2 G^2} \leq c_1 C_2\delta \leq \frac{\delta}{16},$$

where the first equality uses the choice of $\beta$, the second inequality is due to the choice of $\eta$, and in the last inequality we choose small enough $c_1$. Thus, we obtain for $i = 1, 2$ that

$$\Pr(\tau = \tau_i \leq T) = \Pr(\tau = \tau_i \leq T \mid \mathcal{E}_y)\Pr(\mathcal{E}_y) + \Pr(\tau = \tau_i \leq T \mid \mathcal{E}_y^c)\Pr(\mathcal{E}_y^c)$$

$$\leq \frac{\delta}{16}\left(1 - \frac{\delta}{8}\right) + \frac{\delta}{8} \leq \frac{3\delta}{16}.$$

Therefore, we have

$$\Pr(\tau \leq T) \leq \Pr(\tau = \tau_1 \leq T) + \Pr(\tau = \tau_2 \leq T) \leq \frac{3\delta}{8}.$$

Also, note that by Lemma E.8 we have

$$\mathcal{I} \geq \mathbb{E}\left[\sum_{t=1}^{\tau-1} \|\nabla\Phi(x_t)\|^2\right]$$

$$\geq \Pr(\{\tau = T + 1\} \cap \mathcal{E}_y)\mathbb{E}\left[\sum_{t=1}^{T} \|\nabla\Phi(x_t)\|^2 \mid \{\tau = T + 1\} \cap \mathcal{E}_y\right]$$

$$\geq \frac{1}{2}\mathbb{E}\left[\sum_{t=1}^{T} \|\nabla\Phi(x_t)\|^2 \mid \{\tau = T + 1\} \cap \mathcal{E}_y\right],$$

where the last inequality uses

$$\Pr(\{\tau = T + 1\} \cap \mathcal{E}_y) = 1 - \Pr(\{\tau \leq T\} \cup \mathcal{E}_y^c)$$

$$\geq 1 - \Pr(\tau \leq T) - \Pr(\mathcal{E}_y^c) \geq 1 - \frac{3\delta}{8} - \frac{\delta}{8} = 1 - \frac{\delta}{2} \geq \frac{1}{2}.$$

Then we obtain

$$\mathbb{E}\left[\frac{1}{T}\sum_{t=1}^{T} \|\nabla\Phi(x_t)\|^2 \mid \{\tau = T + 1\} \cap \mathcal{E}_y\right] \leq 2\mathcal{I} \leq \frac{32G\Delta_1}{\eta T} \leq \frac{\delta\epsilon^2}{2}, \tag{94}$$

where the second inequality uses (93), and the last inequality is due to the choice of $T$. Now we define $\mathcal{E}^c$ to be the event that Algorithm 2 does not converge to $\epsilon$-stationary points:

$$\mathcal{E}^c := \left\{\frac{1}{T}\sum_{t=1}^{T} \|\nabla\Phi(x_t)\|^2 > \epsilon^2\right\}.$$

By (94) and Markov's inequality, we have

$$\Pr(\mathcal{E}^c \mid \{\tau = T + 1\} \cap \mathcal{E}_y) \leq \frac{\delta\epsilon^2}{2\epsilon^2} = \frac{\delta}{2}.$$

Then we have

$$\Pr(\mathcal{E}^c \cup \{\tau \leq T\} \cup \mathcal{E}_y^c) \leq \Pr(\{\tau \leq T\} \cup \mathcal{E}_y^c) + \Pr(\mathcal{E}^c \mid \{\tau = T + 1\} \cap \mathcal{E}_y)$$

$$\leq \Pr(\{\tau \leq T\}) + \Pr(\mathcal{E}_y^c) + \Pr(\mathcal{E}^c \mid \{\tau = T + 1\} \cap \mathcal{E}_y)$$

$$\leq \frac{3\delta}{8} + \frac{\delta}{8} + \frac{\delta}{2} = \delta,$$

which yields

$$\Pr(\mathcal{E} \cap \{\tau = T + 1\} \cap \mathcal{E}_y) = 1 - \Pr(\mathcal{E}^c \cup \{\tau \leq T\} \cup \mathcal{E}_y^c) \geq 1 - \delta.$$

Therefore, we conclude that with probability at least $1 - \delta$, we have $\tau = T + 1$ and $\|\nabla\Phi(x_t)\| \leq G$ for all $t \in [T]$, and $\frac{1}{T}\sum_{t=1}^{T} \|\nabla\Phi(x_t)\|^2 \leq \epsilon^2$. $\qquad\square$

### E.5 PARAMETER CHOICES FOR VR-ADAMBO (THEOREM E.14)

We first list all the relevant parameter choices below for convenience:

$$G \geq \max\left\{4\lambda, 2\sigma_\phi, 4C_{\phi,0}, \frac{C_{\phi,1}}{L_1}, \sqrt{\frac{C_1\Delta_1 L_0}{C_L\delta}}, \frac{C_1\Delta_1 L_1}{C_L\delta}\right\}, \quad C_\eta = \frac{512C_3e\Delta_1\sigma_\phi LG^2}{c_1\lambda^2\delta^{3/2}\epsilon^3},$$

$$\eta \leq c_1 \cdot \min\left\{\frac{r\lambda}{G}, \frac{\lambda}{L}, \frac{\lambda^2\delta}{\Delta_1 L^2}, \frac{\lambda^2\sqrt{\delta}\epsilon}{\sigma_\phi GL}\right\}, \quad 0 \leq \beta_{\mathrm{sq}} \leq 1, \quad \beta = \frac{C_2\mu^2\eta^2}{\sigma_{g,1}^2},$$

$$\gamma = \frac{c_2\beta}{\mu\max\{1, \ln(C_\eta)\}}, \quad \nu = \sqrt{\beta}, \quad I = \frac{1}{\sqrt{\beta}}, \quad T = \frac{C_3 G\Delta_1}{\eta\delta\epsilon^2}, \quad S_1 \geq \frac{1}{2\beta^2 T},$$

$$T_0 \geq \ln\left(\frac{\mu\gamma\eta^2}{6\|y_0 - y_0^*\|^2}\right) \Big/ \ln\left(1 - \frac{\sqrt{\mu\gamma}}{4}\right), \quad N \geq \ln\left(\frac{\mu\gamma\eta^2}{6(\eta + \eta l_{g,1}IC_m/\lambda\mu)^2}\right) \Big/ \ln\left(1 - \frac{\sqrt{\mu\gamma}}{4}\right),$$

**Verification for Lemma E.4.** By choices of $\eta, \beta, \nu$ and $I$, we have

$$\frac{36l_{g,1}^2 L^2}{\lambda^2\mu^2}\left(\frac{\eta^2}{\nu^2} + \eta^2 I^2\right) = \frac{72\eta^2 l_{g,1}^2 L^2}{\lambda^2\mu^2\beta} = \frac{72\sigma_{g,1}^2 l_{g,1}^2 L^2}{C_2\lambda^2\mu^4} \leq \frac{1}{2},$$

where in the last inequality we choose large enough $C_2$.

**Verification for Lemma E.8.** By choices of $\eta, \beta, \nu$ and $I$, we have

$$\frac{32 \cdot 72 G l_{g,1}^2 L^2}{\lambda^3\mu^2}\left(\frac{\eta^2}{\nu^2} + \eta^2 I^2\right) = \frac{32 \cdot 72\eta^2 G l_{g,1}^2 L^2}{\lambda^3\mu^2\beta} = \frac{32 \cdot 72 G\sigma_{g,1}^2 l_{g,1}^2 L^2}{C_2\lambda^3\mu^4} \leq \min\left\{\frac{1}{4}, \frac{4G}{\lambda}\right\},$$

where in the last inequality we choose large enough $C_2$.

**Verification for Lemmas E.10 and E.12.** Similar to Appendix D, we focus on the dominant terms for each parameter choice when $\epsilon$ is sufficiently small. For the remaining cases, the result can be easily obtained by following the same procedure. Specifically, we consider the case where $\eta$ is chose as

$$\eta = \frac{c_1\lambda^2\sqrt{\delta}\epsilon}{\sigma_\phi GL}.$$

Under event $\mathcal{E}_y$, by Lemma E.10 we have

$$\|y_1 - y_0^*\|^2 \leq \frac{3}{\mu\gamma}\left(1 - \frac{\sqrt{\mu\gamma}}{4}\right)^{T_0}\|y_0 - y_0^*\|^2 + \frac{4\gamma\sigma_{g,1}^2}{\mu}\ln\frac{8eT}{\delta} \leq \frac{\eta^2}{2} + \frac{4c_2\beta\sigma_{g,1}^2}{\mu^2\max\{1, \ln(C_\eta)\}}\ln(C_\eta)$$

$$= \frac{\eta^2}{2} + \frac{4c_2 C_2\eta^2}{\ln(C_\eta)}\ln(C_\eta) \leq \frac{\eta^2}{2} + \frac{\eta^2}{2} = \eta^2,$$

where the second inequality uses the choices of $T_0$ and $\gamma$, the third inequality is due to the choice of $\beta$, and in the last inequality we choose small enough $c_2$.

Under event $\mathcal{E}_y$, by Lemma E.12, for $k \geq 1$ we have

$$\|y_{kI+1} - y_{kI}^*\|^2 \leq \frac{3}{\mu\gamma}\left(1 - \frac{\sqrt{\mu\gamma}}{4}\right)^N\|y_{kI} - y_{kI}^*\|^2 + \frac{4\gamma\sigma_{g,1}^2}{\mu}\ln\frac{8eT}{\delta}$$

$$\leq \frac{3}{\mu\gamma}\left(1 - \frac{\sqrt{\mu\gamma}}{4}\right)^N\left(\eta + \frac{\eta l_{g,1}I}{\lambda\mu}C_m\right)^2 + \frac{\eta^2}{2} \leq \frac{\eta^2}{2} + \frac{\eta^2}{2} = \eta^2,$$

where the third inequality uses the choice of $N$.

**Verification for** $\|\hat{y}_t - y_t^*\| \leq \hat{\varrho} \leq \min\{r, 1/4L_1\}$. First, by choices of $\eta, \nu$ and $I$, we have

$$\frac{\eta l_{g,1} L}{\lambda \mu} \left(\frac{1-\nu}{\nu} + I\right) \leq \frac{l_{g,1} L}{\lambda \mu} \left(\frac{\eta}{\nu} + \eta I\right) = \frac{2\eta l_{g,1} L}{\lambda \mu \sqrt{\beta}}$$

$$\leq \frac{2\sigma_{g,1} l_{g,1} L}{\sqrt{C_2} \lambda \mu^2} \leq \frac{1}{2},$$

where in the last inequality we choose large enough $C_2$. Under event $\mathcal{E}_y$, by Lemma E.12 we have

$$\|\hat{y}_t - y_t^*\| \leq \hat{\varrho}$$

$$= \left(\|\hat{y}_1 - y_0^*\| + \eta + \frac{\eta l_{g,1}}{\lambda \mu} \left(\frac{1-\nu}{\nu} + I\right) \left(2G + \frac{l_{g,1} l_{f,0}}{\mu}\right)\right) \bigg/ \left(1 - \frac{\eta l_{g,1} L}{\lambda \mu} \left(\frac{1-\nu}{\nu} + I\right)\right)$$

$$\leq 2 \left(2\eta + \frac{\eta l_{g,1}}{\lambda \mu} \left(\frac{1-\nu}{\nu} + I\right) \left(2G + \frac{l_{g,1} l_{f,0}}{\mu}\right)\right)$$

$$\leq 2 \left(\frac{2c_1 r \lambda}{G} + \frac{2\sigma_{g,1} l_{g,1} L}{\sqrt{C_2} \lambda \mu^2} \left(2G + \frac{l_{g,1} l_{f,0}}{\mu}\right)\right) \leq \min\left\{r, \frac{1}{4L_1}\right\},$$

where the second inequality uses $\|\hat{y}_1 - y_0^*\| \leq \eta$, the third inequality is due to the choices of $\eta, \nu$ and $I$, and in the last inequality we choose small enough $c_1$ and large enough $C_2$. Therefore, we also have

$$C_m = 2G + \frac{l_{g,1} l_{f,0}}{\mu} + L\hat{\varrho} \leq 2G + \frac{l_{g,1} l_{f,0}}{\mu} + Lr.$$

**Verification for Lemma E.7.** By choices of $\eta, \beta, \nu$ and $I$, we have

$$48L^2 \left(\nu^2 + \frac{\lambda \beta}{400G}\right) \cdot \frac{72 l_{g,1}^2 L^2}{\lambda^2 \mu^2} \left(\frac{\eta^2}{\nu^2} + \eta^2 I^2\right) = \frac{48 \cdot 144 \sigma_{g,1}^2 l_{g,1}^2 L^4}{C_2 \lambda^2 \mu^4} \left(1 + \frac{\lambda}{400G}\right) \beta \leq \frac{\lambda \beta}{64G},$$

where in the last inequality we choose large enough $C_2$. We also have

$$24\nu^2 L^2 \cdot \frac{8\eta^2 l_{g,1}^2 I^2}{\lambda^2 \mu^2} \left(1 + \frac{72 l_{g,1}^2 L^2}{\lambda^2 \mu^2} \left(\frac{\eta^2}{\nu^2} + \eta^2 I^2\right)\right) \leq \frac{192 \sigma_{g,1}^2 l_{g,1}^2 L^2}{C_2 \lambda^2 \mu^4} \left(1 + \frac{72 \sigma_{g,1}^2 l_{g,1}^2 L^2}{C_2 \lambda^2 \mu^4}\right) \beta \leq \frac{\lambda \beta}{64G},$$

where in the last inequality we choose large enough $C_2$. In addition, under event $\mathcal{E}_y$, we have

$$\left(\left(1 + \frac{\nu l_{g,1}}{\mu}\right) \frac{\eta}{\lambda} + \frac{\nu \eta l_{g,1} I}{\lambda \mu}\right) \max_{t \leq T} \|m_t\| + \nu(\varrho_{\max} + \hat{\varrho}_{\max})$$

$$\leq \left(\frac{1}{\lambda} + \frac{2l_{g,1}}{\lambda \mu}\right) C_m \eta + \sqrt{\beta}(\eta + \varrho)$$

$$\leq \left(\frac{1}{\lambda} + \frac{2l_{g,1}}{\lambda \mu}\right) \left(2G + \frac{l_{g,1} l_{f,0}}{\mu} + Lr\right) \eta + \frac{\sqrt{C_2} \mu \eta}{\sigma_{g,1}} \left(1 + \frac{c_1 \lambda}{G}\right) r$$

$$\leq \left(\frac{1}{\lambda} + \frac{2l_{g,1}}{\lambda \mu}\right) \left(2G + \frac{l_{g,1} l_{f,0}}{\mu} + Lr\right) \frac{c_1 \lambda}{L} + \frac{c_1 \sqrt{C_2} \mu \lambda}{\sigma_{g,1} L} \left(1 + \frac{c_1 \lambda}{G}\right) r \leq r,$$

where the first inequality uses Lemmas E.10 and E.12 such that $\max_{t \leq T} \|m_t\| \leq C_m$, $\varrho_{\max} \leq \eta$ and $\hat{\varrho}_{\max} \leq \hat{\varrho}$; the second inequality is due to the choices of $\eta$ and $\beta$, and $\hat{\varrho} \leq r$; and in the last inequality we use the choice of $\eta$ again and choose small enough $c_1$.

**Verification for Theorem E.14.** Under event $\mathcal{E}_y$, by Lemma E.8 we have

$$
\mathcal{I} = \frac{8G\Delta_1}{\eta} + \frac{128G\sigma_\phi^2\beta T}{\lambda} + \left( \frac{32GL^2}{\lambda} + \frac{16 \cdot 48GL^2}{\lambda\beta}\left(\nu^2 + \frac{\lambda\beta}{400G}\right)\right)\left(\frac{6}{\nu}\|\hat{y}_1 - y_0^*\|^2 + 12\varrho_{\max}^2 T\right)
$$

$$
+ \frac{16 \cdot 24\nu^2 GL^2}{\lambda\beta}\left(\frac{48\eta^2 l_{g,1}^2 I^2 L^2}{\nu\lambda^2\mu^2}\|\hat{y}_1 - y_0^*\|^2 + \left(2 + \frac{96\eta^2 l_{g,1}^2 I^2 L^2}{\lambda^2\mu^2}\right)\varrho_{\max}^2 T\right)
$$

$$
+ \left[\left(\frac{32GT}{\lambda} + \frac{32 \cdot 72GT l_{g,1}^2 L^2}{\lambda^3\mu^2}\left(\frac{\eta^2}{\nu^2} + \eta^2 I^2\right)\right)\right.
$$

$$
\left. + \frac{16G}{\lambda\beta}\left(\frac{\lambda\beta}{32G} + \frac{24\eta^2 L^2 T}{\lambda^2}\left(1 + \frac{l_{g,1}}{\mu}\right)^2\right)\right]\frac{l_{g,1}^2 l_{f,0}^2}{\mu^2}\left(1 - \frac{\mu}{l_{g,1}}\right)^{2Q} \le \frac{16G\Delta_1}{\eta}
$$

$$
\le \frac{8G\Delta_1}{\eta} + \frac{128G\sigma_\phi^2\beta T}{\lambda} + \left( \frac{32GL^2}{\lambda} + \frac{16 \cdot 48GL^2}{\lambda\beta}\left(\nu^2 + \frac{\lambda\beta}{400G}\right)\right)\left(\frac{6\eta^2}{\nu} + 12\eta^2 T\right)
$$

$$
+ \frac{16 \cdot 24\nu^2 GL^2}{\lambda\beta}\left(\frac{48\eta^4 l_{g,1}^2 I^2 L^2}{\nu\lambda^2\mu^2} + \left(2 + \frac{96\eta^2 l_{g,1}^2 I^2 L^2}{\lambda^2\mu^2}\right)\eta^2 T\right) + \frac{2G\Delta_1}{\eta}
$$

$$
\le \frac{16G\Delta_1}{\eta},
$$

where the first inequality uses the choice of $Q$ and the fact that under event $\mathcal{E}_y$, $\varrho_{\max} \le \eta$ by Lemma E.10, and in the last inequality we plug in the choices of $\beta, \eta, \nu, I, T$ and choose small enough $c_1, c_2$ and large enough $C_1, C_2$ to obtain the final bound for $\mathcal{I}$.

# F  MORE EXPERIMENTAL DETAILS

## F.1  RNN RESULTS ON AUC MAXIMIZATION

The results with RNNs for both training and testing over 25 epochs are presented in Figure 3 (a) and (b), while the corresponding running times are shown in Figure 3 (c) and (d). Our proposed Adam-type algorithms, AdamBO and VR-AdamBO, show the faster convergence rate and significantly outperform other baselines during training process.

## F.2  HYERPARAMETER SETTINGS FOR DEEP AUC MAXMIZATION

We tune the best hyperparameters for each algorithm, including upper-/lower-level step size, the number of inner loops, momentum parameters, etc. The upper-level learning rate $\eta$ and the lower-level learning rate $\gamma$ are tuned in a wide range of $[1.0 \times 10^{-6}, 0.1]$ for all the baselines on experiments of AUC maximization.

**AUC maximization on transformer.** The best learning rates $(\eta, \gamma)$ are summarized as follows: Stocbio: $(0.005, 0.0001)$, TTSA: $(0.0005, 0.001)$, SABA: $(0.001, 0.005)$, MA-SOBA: $(0.0005, 0.005)$, SUSTAIN: $(0.005, 0.001)$, VRBO: $(0.005, 0.0005)$, BO-REP: $(0.0001, 0.0001)$, SLIP: $(0.0001, 0.001)$, AccBO: $(0.0005, 0.0001)$, AdamBO: $(5.0 \times 10^{-6}, 0.005)$, VR-AdamBO: $(5.0 \times 10^{-6}, 0.005)$. Note that SUSTAIN decays its upper-/lower-level step size with epoch $(t)$ by $\eta = \eta/(t+2)^{1/3}, \eta_{low} = \gamma/(t+2)^{1/3}$. Other algorithms use a constant learning rate.

**AUC maximization on RNN.** The best learning rates $(\eta, \gamma)$ are summarized as follows: StocBio: $(0.01, 0.001)$, TTSA: $(0.005, 0.01)$, SABA: $(0.01, 0.005)$, MA-SOBA: $(0.01, 0.005)$, SUSTAIN: $(0.03, 0.01)$, VRBO: $(0.05, 0.01)$, BO-REP: $(0.001, 0.001)$, SLIP: $(0.001, 0.001)$, AccBO: $(0.005, 0.005)$, AdamBO: $(1.0 \times 10^{-5}, 0.001)$, VR-AdamBO: $(1.0 \times 10^{-5}, 0.001)$.

Other hyper-parameter settings are summarized as follows. The steps for neumann series estimation in StocBiO, VRBO, AdamBO, and VR-AdamBO is set to 3, while it is uniformly sampled from $\{1, 2, 3\}$ in TTSA, SUSTAIN, and AccBO. AccBO and VR-AdamBO use the Nesterov accelerated gradient descent for lower-level update, the momentum parameter $\alpha = 0.5$ for AccBO and $\alpha = 0.1$ for VR-AdamBO, the averaging parameter $\nu = 0.5$ for AccBO and $\nu = 0.1$ for VR-AdamBO. The batch size is set to 32 for all algorithms except VRBO, which uses a larger batch size of 64

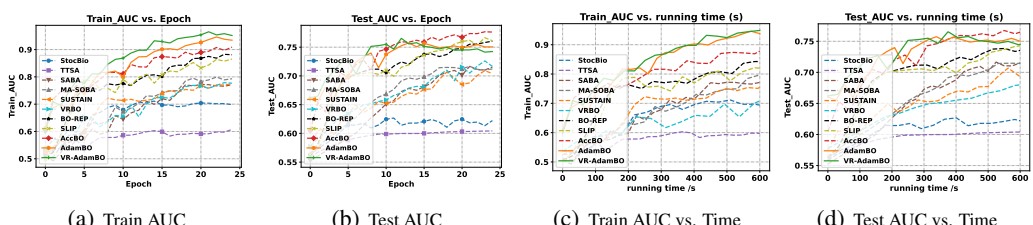

(a) Train AUC     (b) Test AUC     (c) Train AUC vs. Time     (d) Test AUC vs. Time

Figure 3: RNN for AUC maximization on Sentiment140 dataset with imbalance ratio of 0.8. Figures (a), (b) are the results over epochs, Figures (c), (d) are the results over running time.

(tuned in the range of $\{32, 64, 128, 256, 512\}$) at the checkpoint (snapshot) step and 32 otherwise. The momentum parameter $\beta = 0.1$ is fixed in SLIP, AccBO, MA-SOBA, BO-REP, AdamBO and VR-AdamBO, and $\beta_{\mathrm{sq}} = 0.001$ in AdamBO and VR-AdamBO . The warm start steps for the lower level variable in BO-REP, SLIP, AccBO, AdamBO and VR-AdamBO are set to 3. The number of inner loops for StocBio is set to 3. BO-REP uses the periodic updates for low-level variable, and sets the iterations $N = 3$ and the update interval $I = 2$. The hyperparameter $\lambda$ in the Adam update is fixed as $1.0 \times 10^{-8}$ in AdamBO and VR-AdamBO.

### F.3 HYERPARAMETER SETTINGS FOR HYPER-REPRESENTATION

The upper-level learning rate $\eta$ and the lower-level learning rate $\gamma$ are tuned in a range of $[1.0 \times 10^{-4}, 0.1]$ for all the baselines. The optimal learning rate pairs are listed as follows, $(0.01, 0.01)$ for MAML, $(0.01, 0.05)$ for ANIL, $(0.01, 0.01)$ for StocBio, $(0.02, 0.05)$ for TTSA, $(0.01, 0.05)$ for SABA, $(0.05, 0.05)$ for MA-SOBA, and $(0.1, 0.05)$ for both BO-REP and SLIP, $(1.0 \times 10^{-4}, 1.0 \times 10^{-3})$ for AdamBO.

Other hyper-parameter settings are summarized as follows. The steps for neumann series estimation in StocBiO, AdamBO is set to 3, while it is uniformly sampled from $\{1, 2, 3\}$ in TTSA. The momentum parameter $\beta = 0.1$ is fixed in SLIP, MA-SOBA, BO-REP, AdamBO, and $\beta_{\mathrm{sq}} = 0.001$ in AdamBO. The warm start steps for the lower level variable in BO-REP, SLIP, AdamBO are set to 3. The number of inner loops for StocBio is set to 3. BO-REP uses the periodic update for the low-level variable, and sets the iterations $N = 3$ and the update interval $I = 2$. The hyperparameter $\lambda$ in the Adam update is fixed as $1.0 \times 10^{-8}$ in AdamBO and VR-AdamBO.

## G COMPARISON TABLES

**Assumption G.1.** *Consider the following smoothness assumptions:*

*(A) The objective function is L-smooth.*

*(B) The objective function is $(L_0, L_1)$-smooth (Zhang et al., 2020a, Definition 1.1, Remark 2.3).*

*(C) The objective function is $(\rho, L_0, L_\rho)$-smooth with $0 \le \rho < 2$ (Li et al., 2023a, Definition 3.2).*

*The above assumptions satisfy: Assumption G.1(A) $\Longrightarrow$ Assumption G.1(B) $\Longrightarrow$ Assumption G.1(C). In other words, Assumption G.1(A) is the strongest, and Assumption G.1(C) is the weakest.*

**Assumption G.2.** *The (stochastic) gradient norm of the objective function is (almost surely) bounded.*

**Assumption G.3.** *Suppose the following stochastic estimators are unbiased and satisfy:*

$$\mathbb{E}_{\xi \sim \mathcal{D}_f}[\|\nabla_x F(x, y; \xi) - \nabla_x f(x, y)\|^2] \le \sigma_{f,1}^2, \quad \mathbb{E}_{\xi \sim \mathcal{D}_f}[\|\nabla_y F(x, y; \xi) - \nabla_y f(x, y)\|^2] \le \sigma_{f,1}^2,$$

$$\Pr\{\|\nabla_y G(x, y; \xi) - \nabla_y g(x, y)\| \ge \lambda\} \le 2 \exp(-2\lambda^2/\sigma_{g,1}^2) \quad \forall \lambda > 0,$$

$$\mathbb{E}_{\zeta \sim \mathcal{D}_g}[\|\nabla_{xy}^2 G(x, y; \zeta) - \nabla_{xy}^2 g(x, y)\|^2] \le \sigma_{g,2}^2, \quad \mathbb{E}_{\zeta \sim \mathcal{D}_g}[\|\nabla_{yy}^2 G(x, y; \zeta) - \nabla_{yy}^2 g(x, y)\|^2] \le \sigma_{g,2}^2.$$

Table 1: Comparison of Adam-related papers under different settings and assumptions. ✓ represents dropping the bias correction term for the first-order momentum while keeping it for the second-order momentum. $d$ denotes the dimension. Only the key assumptions are listed here.

| Adam Paper | Problem | Stochastic Setting | Assumptions | Bias Correction | Complexity |
|---|---|---|---|---|---|
| De et al. (2018) | Single-Level | Deterministic | G.1(A) + G.2 | ✗ | $O(\epsilon^{-6})$ |
| Défossez et al. (2020) | Single-Level | Stochastic (Expectation) | G.1(A) + G.2 | ✓ | $\widetilde{O}(d\epsilon^{-4})$ |
| Guo et al. (2021b) | Single-Level | Stochastic (Expectation) | G.1(A) + G.2 [2] | ✗ | $O(\epsilon^{-4})$ |
| Zhang et al. (2022) | Single-Level | Stochastic (Finite Sum) | G.1(A) | ✓ (Randomly Reshuffled) | Not Converge [3] |
| Wang et al. (2022) | Single-Level | Stochastic (Finite Sum) | G.1(B) | ✗ (Randomly Reshuffled) | Not Converge |
| Li et al. (2023a) | Single-Level | Stochastic (Expectation) | G.1(C) | ✓ | $O(\epsilon^{-4})$ |
| AdamBO (This work, Theorem 4.1) | Bilevel | Stochastic (Expectation) | G.1(B) [4] | ✓ | $\widetilde{O}(\epsilon^{-4})$ |

| Variance-Reduced Adam Paper | Problem | Stochastic Setting | Assumptions | Bias Correction | Complexity |
|---|---|---|---|---|---|
| VR ADAM (Wang & Klabjan, 2022) | Single-Level | Stochastic (Expectation) | G.1(A) + G.2 | ✓ (Resetting) | Asymptotic Convergence |
| VRAdam (Li et al., 2023a) | Single-Level | Stochastic (Expectation) | G.1(C) | ✓ | $O(\epsilon^{-3})$ |
| VR-AdamBO (This work, Theorem 5.1) | Bilevel | Stochastic (Expectation) | G.1(B) | ✓ | $\widetilde{O}(\epsilon^{-3})$ |

Table 2: Comparison of bilevel optimization algorithms under the unbounded smoothness setting.

| Method | Problem | Stochastic Setting | Loop Style | Assumptions | Adam-Type | Learning Rate $\eta$ | Complexity |
|---|---|---|---|---|---|---|---|
| BO-REP (Hao et al., 2024) | Bilevel | Stochastic (Expectation) | Double | Assumptions 3.2 and G.3 | ✗ | $O(\epsilon^3)$ | $\widetilde{O}(\epsilon^{-4})$ |
| SLIP (Gong et al., 2024a) | Bilevel | Stochastic (Expectation) | Single | Assumptions 3.2 and G.3 | ✗ | $\widetilde{\Theta}(\epsilon^3)$ | $\widetilde{O}(\epsilon^{-4})$ |
| AdamBO (This work, Theorem 4.1) | Bilevel | Stochastic (Expectation) | Single | Assumptions 3.2 to 3.4 | ✓ | $\widetilde{\Theta}(\epsilon^2)$ | $\widetilde{O}(\epsilon^{-4})$ |

| Method (Variance-Reduction) | Problem | Stochastic Setting | Loop Style | Assumptions | Adam-Type | Learning Rate $\eta$ | Complexity |
|---|---|---|---|---|---|---|---|
| AccBO (Gong et al., 2024b) | Bilevel | Stochastic (Expectation) | Double [5] | Assumptions 3.2, 3.5 and G.3 | ✗ | $\widetilde{\Theta}(\epsilon^2)$ | $\widetilde{O}(\epsilon^{-3})$ |
| VR-AdamBO (This work, Theorem 5.1) | Bilevel | Stochastic (Expectation) | Double | Assumptions 3.2, 3.3 and 3.5 | ✓ | $O(\epsilon)$ | $\widetilde{O}(\epsilon^{-3})$ |

# H  ADDITIONAL EXPERIMENTS

## H.1  META-LEARNING ON BERT

We have conducted meta-learning experiments on a larger language model, specifically an 8-layer BERT (Devlin et al., 2018) model. The experiments are performed on a widely-used question classification dataset TREC (Li & Roth, 2002), which contains 6 coarse-grained categories. To evaluate our approach on meta-learning, we construct $K = 500$ meta tasks, where the training data $\mathcal{D}_i^{tr}$ and validation data $\mathcal{D}_i^{val}$ for the $i$-th task are randomly sampled from two disjoint categories, with 5 examples per category. A BERT model, with 8 self-attention layers and a fully-connected layer, is used in our experiment. The self-attention layers serve as representation layers (with their parameters treated as upper-level variables) and the fully-connected layer (with its parameters treated as lower-level variables) serves as an adapter, where each self-attention layer consists of 8 self-attention heads with the hidden size being 768. The fully-connected layer acts as a classifier, with the input dimension of 768 and the output dimension of 6 (corresponding to the 6 categories). Our bilevel optimization algorithm trains the representation layers and the adapter on the meta tasks ($\mathcal{D}^{tr}$ and $\mathcal{D}^{val}$) from scratch, and then evaluate it on the test set $\mathcal{D}^{te}$. During the evaluation phase, we fix the parameters of representation layers and just fine-tune the adapters. We train the models for 20 epochs and compare it with other bilevel optimization baseline algorithms. The training and testing comparison results are presented in Figure 4. As shown, the proposed algorithm AdamBO achieves fast convergence to the best training and test results among all baselines.

The best upper-level learning rate $\eta$ and lower-level learning rate $\gamma$ for all baselines are tuned. The detailed settings are summarized as following:

---

[2](Guo et al., 2021b, Assumption 2) can be implied by Assumption G.2, although it is weaker.

[3]Adam can converge with an additional strong growth condition (Zhang et al., 2022; Wang et al., 2022).

[4]Under Assumption 3.2, the objective function $\Phi$ is $(L_0, L_1)$-smooth, see Lemma B.10 for details.

[5]The single-loop version (Option I) of AccBO (Gong et al., 2024b) only works for one-dimensional quadratic lower-level function.

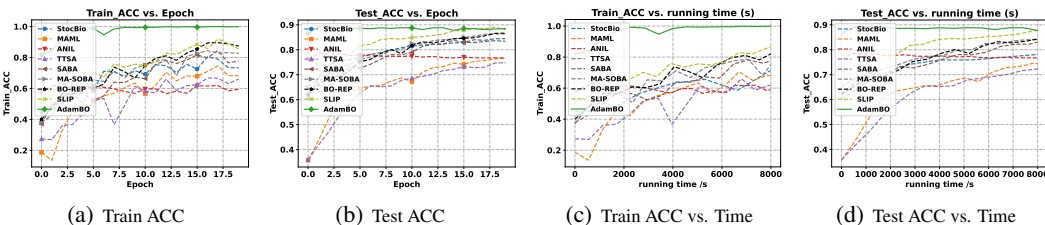

(a) Train ACC  (b) Test ACC  (c) Train ACC vs. Time  (d) Test ACC vs. Time

Figure 4: Comparison with bilevel optimization baselines on hyper-representation. The experiment is performed on a large language model BERT, which contains 8 transformer encoder layers acting as the representation layers and a fully-connected layer acting as the adapter.

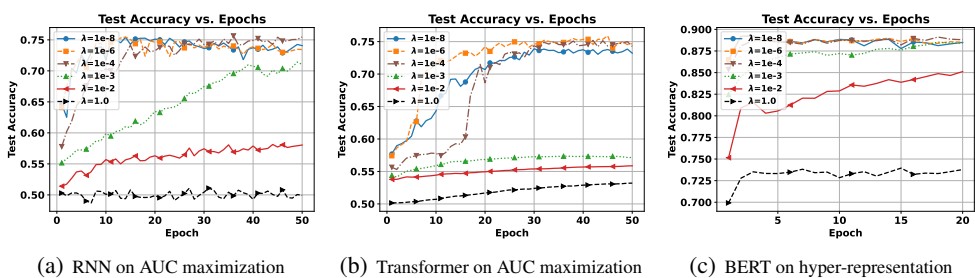

(a) RNN on AUC maximization  (b) Transformer on AUC maximization  (c) BERT on hyper-representation

Figure 5: Test accuracy of different models on AUC maximization and hyper-representaion using AdamBO with $\beta = 0.1$, $\beta_{\mathrm{sq}} = 0.001$ and different $\lambda$s. (a) a 2-layer RNN model on AUC maximization (data imbalanced ratio = 0.8); (b) a 2-layer Transformer model on AUC maximization (data imbalanced ratio = 0.9); (c) an 8-layer BERT model on hyper-representation.

The upper-level learning rate $\eta$ and the lower-level learning rate $\gamma$ are tuned in a range of $[1.0 \times 10^{-4}, 0.1]$ for all the baselines. The optimal learning rate pairs $(\eta, \gamma)$ are, $(0.01, 0.001)$ for MAML, $(0.01, 0.02)$ for ANIL, $(0.01, 0.002)$ for StocBio, $(0.01, 0.001)$ for TTSA, $(0.01, 0.01)$ for SABA, $(0.01, 0.01)$ for MA-SOBA, and $(0.1, 0.05)$ for both BO-REP and SLIP, $(1.0 \times 10^{-4}, 5.0 \times 10^{-3})$ for AdamBO. Please refer to our code https://anonymous.4open.science/r/AdamBO for more experimental details.

## H.2 SENSITIVITY TO THE CHOICE OF $\lambda$

In addition, we have conducted additional experiments in Figure 5 to show the empirical performance of our algorithm is not very sensitive to the choice of $\lambda$. Although the default choice of $\lambda$ is $10^{-8}$ (Kingma & Ba, 2014), increasing it up to $10^{-4}$ only causes minor differences in AUC maximization, and increasing it up to $10^{-3}$ leads to minor changes in hyper-representation performance with BERT (Devlin et al., 2018).

## I PROOF SKETCH FOR VR-ADAMBO (THEOREM 5.1)

For VR-AdamBO, we provide a more detailed proof sketch here due to space constraints in the main text. In particular, we present two main challenges and outline the proof roadmap to address them.

**Challenge 1: VR-AdamBO vs. VRAdam (Li et al., 2023a).** The analysis of VRAdam in the single-level generalized smooth optimization setting (Li et al., 2023a) is not directly applicable to bilevel problems. This is because the hypergradient estimator in bilevel optimization may have a non-negligible bias due to inaccurate estimation of the lower-level variable, whereas the single-level analysis in (Li et al., 2023a) does not need to account for this issue. To control the lower-level estimation error, we leverage the lower-level acceleration technique (Gong et al., 2024b) with periodic updates and averaging. While we largely adopt the framework of VRAdam for the upper-level

analysis, the main distinction lies in our incorporation of the hypergradient bias—arising from inaccurate estimates of the optimal lower-level variable at each iteration—into the upper-level analysis. This is detailed in Lemmas E.5 to E.8 of our paper, which correspond to Lemmas D.4, D.6, D.7, and D.8 in (Li et al., 2023a), respectively.

**Challenge 2: VR-AdamBO vs. AccBO (Gong et al., 2024b).** Although both VR-AdamBO and AccBO Gong et al. (2024b) use the same lower-level update (periodic SNAG with averaging) and adopt the same variance reduction technique STORM (Cutkosky & Orabona, 2019) for the first-order momentum update, the key difference between these two algorithms lies in the upper-level update: AccBO uses normalized SGD with momentum, while VR-AdamBO employs VRAdam. This distinction leads to significantly different theoretical analyses for VR-AdamBO and AccBO. In particular, for VR-AdamBO, we introduce a novel stopping time approach in the context of bilevel optimization (see equation (7) in Section 5.3), building on the VR-Adam analysis Li et al. (2023a). Base on the definition of stopping time $\tau$, we develop a new induction argument (i.e., Lemmas E.10 to E.12) to show that under $t < \tau$ and some good event $\mathcal{E}_y$ (see Lemma E.10 for definition), both $\|\hat{y}_t - y_t^*\|$ and $\|m_t\|$ are bounded. We then show the averaged lower-level error is small (see Lemma E.4) under the parameter choices in Theorem 5.1, which shares an similar spirit as Lemma 4.6 for AdamBO. Combining the aforementioned lemmas with the upper-level analysis mentioned above in Challenge 1 (i.e., Lemmas E.5 to E.8), we can obtain the improved $\widetilde{O}(\epsilon^{-3})$ complexity result.

