# OpenReview forum: "On the Convergence of Adam-Type Algorithms for Bilevel Optimization under Unbounded Smoothness"
_ICLR.cc/2025/Conference — Submitted to ICLR 2025_

### Official Review · Reviewer_x4qJ · 2024-11-04

**Soundness:** 3
**Presentation:** 2
**Contribution:** 3
**Rating:** 6
**Confidence:** 2

**Summary:**

The paper presents AdamBO, a novel extension of the Adam optimizer designed for bilevel optimization problems, particularly when the upper-level function exhibits unbounded smoothness. The authors address the challenges inherent in bilevel optimization, such as the dependency between the upper-level hypergradient and lower-level variable. The work also introduces VR-AdamBO, a variance-reduced variant, to enhance convergence efficiency. Both algorithms demonstrate provable theoretical convergence guarantees and are validated through experiments on tasks involving RNNs and transformers, showing significant performance improvements.

The authors also proves that AdamBO converges to $\varepsilon$-stationary points with $O(\varepsilon^{-4})$ oracle complexity. The variance-reduced variant VR-AdamBO has an improved oracle complexity of $O(\varepsilon^{-3})$.

**Strengths:**

1. The authors provide comprehensive and solid convergence proofs for both AdamBO and VR-AdamBO, incorporating novel elements such as a randomness decoupling lemma and stopping-time analysis for variance reduction. These theoretical insights are significant and may be of independent interest for future research.
2. The algorithms are tested on relevant machine learning tasks, including meta-learning and deep AUC maximization, showing strong empirical performance.

**Weaknesses:**

1. Instead of stacking all the assumptions and lemmas, I think a better way to present your theoretical results is that placing the detailed proof sketch in the main text and moving some assumptions and lemmas to the appendix. Also, it would be better if you can clearly state the difficult part of the proofs as well as the comparison and difference with other relevant works. The current version is quite difficult for the audience to understand the entire line of the proof.

**Questions:**

Please refer to the "weaknesses" part for the questions.

**Details Of Ethics Concerns:**

No ethics concerns for this paper.

---

> ### Author Response · Authors · 2024-11-17
> **Rebuttal by Authors**
>
> *Thank you for taking the time to review our paper. We have addressed each of your concerns below.*
>
> **Q1. Instead of stacking all the assumptions and lemmas, I think a better way to present your theoretical results is that placing the detailed proof sketch in the main text and moving some assumptions and lemmas to the appendix. Also, it would be better if you can clearly state the difficult part of the proofs as well as the comparison and difference with other relevant works. The current version is quite difficult for the audience to understand the entire line of the proof.**
>
> **A1.** Thank you for your suggestion. We have added **Table 2 in Appendix G** (on page 68), which outlines the assumptions and complexities of our proposed algorithms, AdamBO and VR-AdamBO, in comparison to other popular bilevel optimization algorithms from previous literature under the unbounded smoothness setting. Please refer to **Appendix G** for details.
>
> For AdamBO, we have actually provided a detailed description of the main challenges and technical overview in Section 4.1, along with a detailed proof sketch in Section 4.3 (see Lemmas 4.2 to 4.6). For VR-AdamBO, we have added a more detailed proof sketch in **Appendix I** (on pages 69-70) due to space constraints in the main text. In particular, we present two main challenges and outline the proof roadmap to address them.
>
> - **Challenge 1: VR-AdamBO vs. VRAdam ([Li et al., 2023](https://arxiv.org/pdf/2304.13972)).** The analysis of VRAdam in the single-level generalized smooth optimization setting ([Li et al., 2023](https://arxiv.org/pdf/2304.13972)) is not directly applicable to bilevel problems. This is because the hypergradient estimator in bilevel optimization may have a non-negligible bias due to inaccurate estimation of the lower-level variable, whereas the single-level analysis in ([Li et al., 2023](https://arxiv.org/pdf/2304.13972)) does not need to account for this issue. To control the lower-level estimation error, we leverage the lower-level acceleration technique ([Gong et al., 2024b](https://arxiv.org/pdf/2409.19212)) with periodic updates and averaging. While we largely adopt the framework of VRAdam for the upper-level analysis, the main distinction lies in our incorporation of the hypergradient bias----arising from inaccurate estimates of the optimal lower-level variable at each iteration----into the upper-level analysis. This is detailed in Lemmas E.5 to E.8 of our paper, which correspond to Lemmas D.4, D.6, D.7, and D.8 in ([Li et al., 2023](https://arxiv.org/pdf/2304.13972)), respectively.
>
> - **Challenge 2: VR-AdamBO vs. AccBO ([Gong et al., 2024b](https://arxiv.org/pdf/2409.19212)).** Although both VR-AdamBO and AccBO ([Gong et al., 2024b](https://arxiv.org/pdf/2409.19212)) use the same lower-level update (periodic SNAG with averaging) and adopt the same variance reduction technique STORM ([Cutkosky & Orabona, 2019](https://arxiv.org/pdf/1905.10018)) for the first-order momentum update, the key difference between these two algorithms lies in the upper-level update: AccBO uses normalized SGD with momentum, while VR-AdamBO employs VRAdam ([Li et al., 2023](https://arxiv.org/pdf/2304.13972)). This distinction leads to significantly different theoretical analyses for VR-AdamBO and AccBO. In particular, for VR-AdamBO, we introduce a novel stopping time approach in the context of bilevel optimization (see equation (7) in Section 5.3), building on the VR-Adam analysis ([Li et al., 2023](https://arxiv.org/pdf/2304.13972)). Base on the definition of stopping time $\tau$, we develop a new induction argument (i.e., Lemmas E.10 to E.12) to show that under $t<\tau$ and some good event $\mathcal{E}_y$ (see Lemma E.10 for definition of $\mathcal{E}_y$), both $\\|\hat{y}_t-y_t^*\\|$ and $\\|m_t\\|$ are bounded. We then show the averaged lower-level error is small (see Lemma E.4) under the parameter choices in Theorem 5.1, which shares an similar spirit as Lemma 4.6 for AdamBO. Combining the aforementioned lemmas with the upper-level analysis mentioned above in Challenge 1 (i.e., Lemmas E.5 to E.8), we can finally obtain the improved $\widetilde{O}(\epsilon^{-3})$ complexity result.

---

> > ### Author Response · Authors · 2024-11-22
> > **Looking forward to your feedback**
> >
> > Dear Reviewer x4qJ,
> >
> > Thank you for reviewing our paper. We have carefully addressed your concerns regarding the comparison with previous literature and provided a detailed proof sketch of the proposed algorithms.
> >
> > Please let us know if our responses address your concerns accurately. We appreciate your time and efforts and are open to discussing any further questions you may have.
> >
> > Best,
> >
> > Authors

---

> > ### Comment · Reviewer_x4qJ · 2024-11-25
> > **Official Comment**
> >
> > Thanks so much for your clear response. I think it answers my questions and I keep my original score.

---

### Official Review · Reviewer_NMMT · 2024-11-08

**Soundness:** 3
**Presentation:** 3
**Contribution:** 3
**Rating:** 6
**Confidence:** 2

**Summary:**

In this paper, the authors extend to Adam to solve bilevel optimization problems. The authors proposed two Adam-type algorithms, AdamBO and VR-AdamBO that is a variance-reduced version with an improved oracle complexity of the first one. The authors analysis a novel decoupling lemma that provides refined control over the lower-level variable. Finally, the authors test the effectiveness of their proposed algorithms on some machine learning tasks.

**Strengths:**

1. the authors provide solid theoretical guarantees for the convergence of their proposed Adam-type algorithms under the unbounded smoothness assumption and other standard assumptions in current bilevel optimization analysis.

2. A novel random decoupling lemma that is proposed by the authors provides a technical tool to control the lower-level error without concerns about the dependency issues from the upper-level randomness. This lemma can be extended to other analysis of Adam-type algorithms for bilevel optimizaition problems.

**Weaknesses:**

1. The author did not provide a clear comparison of the assumptions and the complexity of the newly proposed algorithms with popular algorithms that appeared in previous literature in the paper.

2. The theoretical convergence results provided by the author, whether in AdamBO or VR-AdamBO algorithms, cannot guarantee that the first-order momentum coefficient is a constant, that is, the coefficient depends on the accuracy level $\epsilon$.

**Questions:**

1. In your assumption 3.5: “….. almost surely ”, what does “almost surely” mean?

2. In the part of VR-AdamBO, you state that “VR-AdamBO can be regarded as a generalization of the AccBO….., with the key difference being that AccBO uses normalized SGD with momentum for the upper-level update,…”, so do you mean that you adopted the same variance reduction technique as AccBO, the only difference being the way you update the upper level variables?

---

> ### Author Response · Authors · 2024-11-17
> **Rebuttal by Authors**
>
> *Thank you for taking the time to review our paper. We have addressed each of your concerns below.*
>
> **Q1. The author did not provide a clear comparison of the assumptions and the complexity of the newly proposed algorithms with popular algorithms that appeared in previous literature in the paper.**
>
> **A1.** Thank you for your suggestion. We have added **Table 2 in Appendix G** (on page 68), which outlines the assumptions and complexities of our proposed algorithms, AdamBO and VR-AdamBO, in comparison to other popular bilevel optimization algorithms from previous literature under the unbounded smoothness setting. Please refer to **Appendix G** for more details.
>
> **Q2. The theoretical convergence results provided by the author, whether in AdamBO or VR-AdamBO algorithms, cannot guarantee that the first-order momentum coefficient is a constant, that is, the coefficient depends on the accuracy level $\epsilon$.**
>
> **A2.** Thank you for your insightful question. First, our paper considers stochastic bilevel optimization with the upper-level function being unbounded smooth and the main focus of our work is to extend vanilla Adam to solve bilevel optimization problems with theoretical convergence guarantees. For single-level stochastic optimization problem with a general expectation formulation, the analysis by ([Li et al., 2023](https://arxiv.org/pdf/2304.13972)) is the only convergence analysis of Adam under the unbounded smoothness setting (see **Table 1 in Appendix G** on page 68 for a detailed comparison). ([Li et al., 2023](https://arxiv.org/pdf/2304.13972)) requires $\beta = O(\epsilon^2)$ to establish Adam's convergence. Our paper considers the more challenging bilevel optimization problem and similarly requires $\beta = \widetilde{\Theta}(\epsilon^2)$ to be small.
>
> Second, we want to emphasize that small $\beta$ is indeed a common assumption in the literature. There are other works in convergence analysis of Adam that also require $\beta$ be to small (i.e., $\beta$ depends on target gradient norm $\epsilon$), for example, see ([Guo et al., 2021b](https://arxiv.org/pdf/2104.14840), Theorem 6), where they choose $\beta = O(\epsilon^2)$. Moreover, existing convergence analyses of Adam that do not need such choice of $\beta$ require other strong assumptions for the objective function, which is incompatible to our setting. They either rely on the bounded gradient assumption ([De et al., 2018](https://arxiv.org/pdf/1807.06766); [Défossez et al., 2020](https://arxiv.org/pdf/2003.02395)), or they only prove convergence to some neighborhood of stationary points with a constant radius unless assuming the strong growth condition under the finite sum setting ([Zhang et al., 2022](https://arxiv.org/pdf/2208.09632); [Wang et al., 2022](https://arxiv.org/pdf/2208.09900)). Please see our **Table 1 in Appendix G** and the related work section (convergence of Adam) of ([Li et al., 2023](https://arxiv.org/pdf/2304.13972)) for more details.
>
> **Q3. In your assumption 3.5: “... almost surely”, what does “almost surely” mean?**
>
> **A3.** Almost surely is a commonly used concept in probability theory (and measure theory).
> In probability theory, an event is said to happen almost surely (sometimes abbreviated as *a.s.*) if it happens with probability 1 (with respect to the probability measure). In other words, the set of outcomes on which the event does not occur has probability 0, even though the set might not be empty.
>
> In the context of Assumption 3.5, the statement "$F(x,y;\xi)$ and $G(x,y;\zeta)$ satisfy Assumption 3.2 for every $\xi$ and $\zeta$ almost surely" means that $F(x,y;\xi)$ and $G(x,y;\zeta)$ satisfy Assumption 3.2 for every $\xi$ and $\zeta$ with probability 1, according to the definition of almost surely provided above. This assumption has been shown to be necessary to obtain the improved oracle complexity $O(\epsilon^{-3})$ or $\widetilde{O}(\epsilon^{-3})$ for both single-level problems ([Cutkosky & Orabona, 2019](https://arxiv.org/pdf/1905.10018), Section 3) and bilevel problems ([Khanduri et al., 2021](https://arxiv.org/pdf/2102.07367), Assumption 3), ([Yang et al., 2021](https://arxiv.org/pdf/2106.04692), Assumption 2), ([Guo et al. 2021a](https://arxiv.org/pdf/2105.02266), Assumption 2), ([Gong et al., 2024b](https://arxiv.org/pdf/2409.19212), Assumption 3.4).

---

> > ### Author Response · Authors · 2024-11-17
> > **Rebuttal by Authors (Cont'd)**
> >
> > **Q4. In the part of VR-AdamBO, you state that “VR-AdamBO can be regarded as a generalization of the AccBO..., with the key difference being that AccBO uses normalized SGD with momentum for the upper-level update, ...”, so do you mean that you adopted the same variance reduction technique as AccBO, the only difference being the way you update the upper level variables?**
> >
> > **A4.** Yes, both VR-AdamBO and AccBO (along with VR-Adam ([Li et al., 2023](https://arxiv.org/pdf/2304.13972))) adopt the same variance reduction technique, STORM ([Cutkosky and Orabona, 2019](https://arxiv.org/pdf/1905.10018)). The key difference between these two algorithms lies in the upper-level update: AccBO uses normalized SGD with momentum, while VR-AdamBO employs VRAdam. This distinction leads to significantly different theoretical analyses for VR-AdamBO and AccBO. In particular, for VR-AdamBO, we introduce a novel stopping time approach in the context of bilevel optimization (see equation (7) in Section 5.3), building on the VR-Adam analysis ([Li et al., 2023](https://arxiv.org/pdf/2304.13972)). To our knowledge, this technique is new and has not been leveraged in previous bilevel optimization literature.

---

> > > ### Author Response · Authors · 2024-11-22
> > > **Looking forward to your feedback**
> > >
> > > Dear Reviewer NMMT,
> > >
> > > Thank you for reviewing our paper. We have carefully addressed your concerns regarding the comparison with previous literature, the requirement for a small $\beta$, the definition of 'almost surely', and the differences between VR-AdamBO and AccBO.
> > >
> > > Please let us know if our responses address your concerns accurately. We appreciate your time and efforts and are open to discussing any further questions you may have.
> > >
> > > Best,
> > >
> > > Authors

---

### Official Review · Reviewer_szE1 · 2024-11-09

**Soundness:** 2
**Presentation:** 2
**Contribution:** 2
**Rating:** 3
**Confidence:** 4

**Summary:**

This paper extends the Adam optimizer to address bilevel optimization problems. The authors introduce AdamBO, a single-loop, Adam-based algorithm for bilevel problems with a strongly convex lower-level function and a nonconvex upper-level function, achieving an iteration complexity of O(epsilon^{-4}). They also propose VR-AdamBO, a variance-reduced version that further improves oracle complexity to O(epsilon^{-3}). The proof utilized a randomness decoupling lemma and stopping time approach. The paper demonstrates the effectiveness of these algorithms on one meta-learning dataset.

**Strengths:**

Extend Adam to bilevel optimization problems, and provide some convergence anlysis.

**Weaknesses:**

i) The paper requires the coefficient of the learning rate to be 1/(sqrt{v} + lambda) where lambda >0 (it is often called epsilon in Adam papers).
  i.1) This makes the analysis rather weak. In fact, with a positive lambda, the analysis will avoid a major challenge of lower bounding sqrt{v}, making the analysis much easier.
  i.2) There are quite a few constants that depend on lambda >0:
   a) beta < O(lambda);
   b) stepsize eta < O(lambda^1.5 beta) = O(lambda^2.5).
  This makes the effective stepsize always super small (it can be proved that the effective stepsize eta/(sqrt{v} + beta) < eta/beta < lamda^1.5). Then it is somewhat easy to prove Adam converges with an upper bounded stepsize.

ii) The paper requires beta (which is 1 - beta1) to be very small, i.e. the typical beta1 is very close to 1, and the new gradient information comes very slowly. This is a very special variant of Adam, making the result not applicable to general Adam. This significantly weakens the claim of the paper that "Adam-based  bilevel algorithm is proved to converge".

iii) Experiments are relatively weak. AUC minimization of binary text classification is not a common task,and it only uses 3-layer RNN and transformers. Meta-learning on  SNLI with 3-layer RNN and transformers is not that convincing.

**Questions:**

i) Why do you require beta to be very small? Can you get around the assumption?

ii) Can you get around the asumption of lambda > 0, or remove the dependence of eta on lambda?

---

> ### Author Response · Authors · 2024-11-17
> **Rebuttal by Authors**
>
> *Thank you for taking the time to review our paper. We have addressed each of your concerns below.*
>
> **Q1. The paper requires the coefficient of the learning rate to be $\frac{\eta}{\sqrt{\hat{v}_t} + \lambda}$ where $\lambda > 0$ (it is often called $\epsilon$ in Adam papers). 1) This makes the analysis rather weak. In fact, with a positive lambda, the analysis will avoid a major challenge of lower bounding $\sqrt{\hat{v}_t}$, making the analysis much easier. 2) There are quite a few constants that depend on $\lambda > 0$: a) $\beta < O(\lambda)$; b) stepsize $\eta < O(\lambda^{1.5}\beta) = O(\lambda^{2.5})$. This makes the effective stepsize always super small (it can be proved that the effective stepsize $\frac{\eta}{\sqrt{\hat{v}_t} + \lambda} < \frac{\eta}{\beta} < \lambda^{1.5})$. Then it is somewhat easy to prove Adam converges with an upper bounded stepsize.**
>
> **A1.** Thank you for your insightful question. We would like to emphasize that our convergence analysis is not for Adam in the single-level optimization, but for the bilevel variants of Adam (AdamBO and VR-AdamBO). Analyzing the convergence of bilevel variants of Adam under the unbounded smoothness setting is significantly different and more challenging than the convergence analysis of vanilla Adam ([Li et al., 2023](https://arxiv.org/pdf/2304.13972)).
>
> First, $\lambda>0$ is a commonly used choice for the vanilla Adam ([Kingma & Ba, 2014](https://arxiv.org/pdf/1412.6980)) and the convergence analysis of Adam even in single-level stochastic optimization with a general expectation formulation under the unbounded smoothness setting requires $\lambda>0$ ([Li et al., 2023](https://arxiv.org/pdf/2304.13972)). To our knowledge, no existing work has removed the requirement of $\lambda > 0$ under the same setting. Please refer to **Table 1 in Appendix G** (on page 68) for a detailed comparison. Our paper considers the more challenging bilevel optimization problem and hence also requires $\lambda>0$.
>
> Second, we would like to clarify that our theoretical results hold for any $\lambda>0$ (we treat $\lambda$ as a constant), not just for small enough $\lambda$. Also, the upper-bounded stepsize due to $\lambda>0$ does not imply the analyses of bilevel variants of Adam are easy. For example, the analysis in ([Li et al., 2023](https://arxiv.org/pdf/2304.13972)) does not need to consider the hypergradient bias due to the inaccurate estimates for the optimal lower-level variable at every iteration, while our bilevel variants require new techniques to handle them. One significant challenge is the randomness dependency issue for both level variables when the upper-level performs the Adam update. To address this, we introduce a novel randomness decoupling lemma for lower-level error control when the upper-level variable is updated by Adam, as illustrated in Section 4.3.2. This technique is nontrivial and has not been previously explored in either single-level or bilevel optimization literature, making our analysis novel and distinguishing it from existing bilevel optimization algorithms under the same setting ([Hao et al., 2024](https://arxiv.org/pdf/2401.09587.pdf); [Gong et al., 2024a](https://openreview.net/pdf?id=36rWa8zVkh); [b](https://arxiv.org/pdf/2409.19212)). In fact, as shown in **Table 2 in Appendix G** (on page 68), our choices of learning rate $\eta$ are quite different from prior works under the same setting:
> - $\eta=\widetilde{\Theta}(\epsilon^2)$ for AdamBO, compared to $\eta=O(\epsilon^3)$ for BO-REP ([Hao et al., 2024](https://arxiv.org/pdf/2401.09587.pdf)) and $\eta=\widetilde{\Theta}(\epsilon^3)$ for SLIP ([Gong et al., 2024a](https://openreview.net/pdf?id=36rWa8zVkh));
> - $\eta=O(\epsilon)$ for VR-AdamBO, compared to $\eta=\widetilde{\Theta}(\epsilon^2)$ for AccBO ([Gong et al., 2024b](https://arxiv.org/pdf/2409.19212)).
>
> In addition, we have conducted additional experiments in **Figure 5 at Appendix H** (on page 69) to show the empirical performance of our algorithm is not very sensitive to the choice of $\lambda$. Although the default choice of $\lambda$ is $10^{-8}$ ([Kingma & Ba, 2014](https://arxiv.org/pdf/1412.6980)), increasing it up to $10^{-4}$ only causes minor differences in AUC maximization, and increasing it up to $10^{-3}$ leads to minor changes in hyper-representation performance with BERT ([Devlin et al., 2018](https://arxiv.org/pdf/1810.04805)).

---

> ### Author Response · Authors · 2024-11-17
> **Rebuttal by Authors (Cont'd)**
>
> **Q2. The paper requires $\beta$ (which is $1 - \beta_1$) to be very small, i.e. the typical $\beta_1$ is very close to 1, and the new gradient information comes very slowly. This is a very special variant of Adam, making the result not applicable to general Adam. This significantly weakens the claim of the paper that "Adam-based bilevel algorithm is proved to converge".**
>
> **A2.** Thank you for your insightful question. First, our paper considers stochastic bilevel optimization with the upper-level function being unbounded smooth and the main focus of our work is to extend vanilla Adam to solve bilevel optimization problems with theoretical convergence guarantees. For single-level stochastic optimization problem with a general expectation formulation, the analysis by ([Li et al., 2023](https://arxiv.org/pdf/2304.13972)) is the only convergence analysis of Adam under the unbounded smoothness setting (see **Table 1 in Appendix G** on page 68 for a detailed comparison). ([Li et al., 2023](https://arxiv.org/pdf/2304.13972)) requires $\beta = O(\epsilon^2)$ to establish Adam's convergence. Our paper considers the more challenging bilevel optimization problem and similarly requires $\beta = \widetilde{\Theta}(\epsilon^2)$ to be small.
>
> Second, we want to emphasize that small $\beta$ is indeed a common assumption in the literature. There are other works in convergence analysis of Adam that also require $\beta$ be to small (i.e., $\beta$ depends on target gradient norm $\epsilon$), for example, see ([Guo et al., 2021b](https://arxiv.org/pdf/2104.14840), Theorem 6), where they choose $\beta = O(\epsilon^2)$. Moreover, existing convergence analyses of Adam that do not need such choice of $\beta$ require other strong assumptions for the objective function, which is incompatible to our setting. They either rely on the bounded gradient assumption ([De et al., 2018](https://arxiv.org/pdf/1807.06766); [Défossez et al., 2020](https://arxiv.org/pdf/2003.02395)), or they only prove convergence to some neighborhood of stationary points with a constant radius unless assuming the strong growth condition under the finite sum setting ([Zhang et al., 2022](https://arxiv.org/pdf/2208.09632); [Wang et al., 2022](https://arxiv.org/pdf/2208.09900)). Please see our **Table 1 in Appendix G** and the related work section (convergence of Adam) of ([Li et al., 2023](https://arxiv.org/pdf/2304.13972)) for more details.

---

> ### Author Response · Authors · 2024-11-17
> **Rebuttal by Authors (Cont'd)**
>
> **Q3. Experiments are relatively weak. AUC minimization of binary text classification is not a common task,and it only uses 3-layer RNN and transformers. Meta-learning on SNLI with 3-layer RNN and transformers is not that convincing.**
>
> **A3.** We have conducted meta-learning experiments on a larger language model, specifically an 8-layer BERT ([Devlin et al., 2018](https://arxiv.org/pdf/1810.04805)) model. The experiments are performed on a widely-used question classification dataset TREC ([Li & Roth, 2002](https://aclanthology.org/C02-1150.pdf)), which contains 6 coarse-grained categories. To evaluate our approach on meta-learning, we construct $K=500$ meta tasks, where the training data $\mathcal{D}_i^{tr}$ and validation data $\mathcal{D}_i^{val}$ for the $i$-th task are randomly sampled from two disjoint categories, with $5$ examples per category. A BERT model, with 8 self-attention layers and a fully-connected layer, is used in our experiment. The self-attention layers serve as representation layers (with their parameters treated as upper-level variables) and the fully-connected layer (with its parameters treated as lower-level variables) serves as an adapter, where each self-attention layer consists of 8 self-attention heads with the hidden size being 768. The fully-connected layer acts as a classifier, with the input dimension of 768 and the output dimension of 6 (corresponding to the 6 categories). Our bilevel optimization algorithm trains the representation layers and the adapter on the meta tasks ($\mathcal{D}^{tr}$ and $\mathcal{D}^{val}$) from scratch, and then evaluate it on the test set $\mathcal{D}^{te}$. During the evaluation phase, we fix the parameters of representation layers and just fine-tune the adapters. We train the models for 20 epochs and compare it with other bilevel optimization baseline algorithms. The training and testing comparison results are presented in **Figure 4 at Appendix H** (on page 69). As shown, the proposed algorithm AdamBO achieves fast convergence to the best training and test results among all baselines.
>
> The best upper-level learning rate $\eta$ and lower-level learning rate $\gamma$ for all baselines are tuned. The detailed settings are summarized as following:
>
> The upper-level learning rate $\eta$ and the lower-level learning rate $\gamma$ are tuned in a range of $[1.0\times 10^{-4}, 0.1]$ for all the baselines. The optimal learning rate pairs $(\eta, \gamma)$ are, $(0.01, 0.001)$ for MAML, $(0.01, 0.02)$ for ANIL, $(0.01, 0.002)$ for StocBio, $(0.01, 0.001)$ for TTSA, $(0.01, 0.01)$ for SABA, $(0.01, 0.01)$ for MA-SOBA, and $(0.1, 0.05)$ for both BO-REP and SLIP, $(1.0\times 10^{-4}, 5.0\times 10^{-3})$ for AdamBO. Please refer to our code (https://anonymous.4open.science/r/AdamBO) for more experimental details.
>
> **Q4. Why do you require $\beta$ to be very small? Can you get around the assumption?**
>
> **A4.** The purpose of choosing small $\beta$ is to handle the hypergradient bias due to the inaccurate estimates of the optimal lower-level variable (at every iteration) in the stochastic bilevel optimization setting, which has widely been used in the literature ([Chen et al., 2023](https://arxiv.org/pdf/2306.12067); [Hao et al., 2024](https://arxiv.org/pdf/2401.09587.pdf); [Gong et al., 2024a](https://openreview.net/pdf?id=36rWa8zVkh); [b](https://arxiv.org/pdf/2409.19212)). If we do not choose small $\beta$, the algorithm uses less historical gradient information and the hypergradient bias of the current iterate will possibly be large unless we use large mini-batch size (i.e., batch size depends on $1/\epsilon$) such as in ([Ji et al., 2021](https://arxiv.org/pdf/2010.07962); [Arbel & Maira, 2022](https://arxiv.org/pdf/2111.14580)).
>
> In particular, our Lemma 4.6 controls the accumulated hypergradient bias, which grows with a sublinear rate in terms of $T$ (to be more precise, it grows with a rate of $\sqrt{T}$) if we choose $\beta=\widetilde{\Theta}(\epsilon^2)$ (and thus $\gamma=\widetilde{\Theta}(\epsilon^2)$, see exact parameter choices in Theorem D.12 at Appendix D for details).

---

> ### Author Response · Authors · 2024-11-17
> **Rebuttal by Authors (Cont'd)**
>
> **Q5. Can you get around the assumption of $\lambda > 0$, or remove the dependence of $\lambda$ on $\eta$?**
>
> **A5.** Thank you for your insightful question. First, we would like to clarify that $\lambda$ is part of the vanilla Adam algorithm ([Kingma & Ba, 2014](https://arxiv.org/pdf/1412.6980)), and it is set to $\lambda=10^{-8}>0$ as the default choice. In other words, it is not zero although its default choice is small. In addition, as mentioned above, we have conducted additional experiments in **Figure 5 at Appendix H** (on page 69) to show the empirical performance of our algorithm is not very sensitive to the choice of $\lambda$. While the default choice of $\lambda$ is $10^{-8}$ ([Kingma & Ba, 2014](https://arxiv.org/pdf/1412.6980)), increasing it up to $10^{-4}$ only causes minor differences in AUC maximization, and increasing it up to $10^{-3}$ leads to minor changes in hyper-representation performance with BERT ([Devlin et al., 2018](https://arxiv.org/pdf/1810.04805)).
>
> Second, we would like to emphasize that the focus of this paper is to extend the applicability of vanilla Adam to tackle bilevel optimization problems with theoretical convergence guarantees, rather than removing the dependence on $\lambda$ in the analysis of Adam for single-level optimization: our analysis works for any $\lambda>0$ (we treat $\lambda$ as a constant). One of our main technical contributions is the introduction of a novel randomness decoupling lemma for lower-level error control when the upper-level variable is updated by Adam, as illustrated in Section 4.3.2. In fact, we largely adopt the framework of ([Li et al., 2023](https://arxiv.org/pdf/2304.13972)) for the upper-level analysis and thus our $\eta$ also depends on $\lambda$, similar to the parameter choices in ([Li et al., 2023](https://arxiv.org/pdf/2304.13972), Theorems 4.1 and 6.2).

---

> ### Author Response · Authors · 2024-11-21
> **Looking forward to your feedback**
>
> Dear Reviewer szE1,
>
> Thank you for reviewing our paper. We have carefully addressed your concerns regarding the novelty of our analysis, the assumption of $\lambda > 0$, and the requirement for a small $\beta$. Additionally, we have included an 8-layer BERT meta-learning experiment and a sensitivity test for $\lambda$ in **Appendix H** (on page 69) of the revised paper.
>
> - First, $\lambda>0$ is also used in the original Adam paper and the proof of **bilevel variants of Adam** is nontrivial: previous analysis in single-level optimization does not need to consider the hypergradient bias due to the inaccurate estimates for the optimal lower-level variable at every iteration, while our bilevel variants require new techniques (randomness decoupling lemma, Lemma 4.2) to handle them.
> - Second, small $\beta$ (i.e., $\beta_1$ is close to $1$ in the original Adam paper ([Kingma & Ba, 2014](https://arxiv.org/pdf/1412.6980))) is indeed a reasonable choice and has been widely used in the literature, e.g.,  ([Li et al., 2023](https://arxiv.org/pdf/2304.13972); [Guo et al., 2021b](https://arxiv.org/pdf/2104.14840)).
> - Third, we have added a 8-layer BERT meta-learning experiment and a sensitivity test of $\lambda$ in Appendix H. The experimental results demonstrate that our proposed algorithm consistently outperforms other baselines and that its empirical performance is not very sensitive to the choice of $\lambda$.
>
> Please let us know if our responses address your concerns accurately. We appreciate your time and efforts and are open to discussing any further questions you may have.
>
> Best,
>
> Authors

---

> > ### Author Response · Authors · 2024-11-26
> >
> > Dear Reviewer szE1,
> >
> > We sincerely thank you for your valuable reviews. We have carefully addressed your concerns regarding the novelty of our analysis, the assumption of $\lambda > 0$, and the requirement for a small $\beta$. Additionally, we have included an 8-layer BERT meta-learning experiment and a sensitivity test for $\lambda$ in **Appendix H** (on page 69) of the revised paper.
> >
> > As we are approaching the end of the discussion period, please let us know if our responses address your concerns accurately. We appreciate your time and efforts and are open to discussing any further questions you may have.
> >
> > Best,
> >
> > Authors

---

> ### Comment · Reviewer_szE1 · 2024-12-03
> **not resolved my concern**
>
> The response does not address my concern directly.
>
> "the upper-bounded stepsize due to does not imply the analyses of bilevel variants of Adam are easy"
>
> I agree that bilevel optimization introduces extra challenges. However, what I was saying is that the upper bound on stepsize has removed a major challenge of analyzing Adam itself. This makes the claim "this work has analyzed Adam together with bilevel optimization" somewhat misleading, since one major challenge of Adam is removed. This paper may give an impression that "the paper has largely resolved the challenge of combining Adam and bilevel optimization"; however, my previous comment said that due to the very special stepsize, it might be far from resolving the issue of analyzing Adam and bilevel optimization together.
> No matter whether using upper-bounded-tiny-stepsize is difficult or not, it does not provide the first solid solution of Adam + bilevel.
>
> What the authors tried to argue is "bilevel is nontrivial"; but that does not address my cocern that the paper has not properly addressed "bilevel + Adam".
>
> Minor issues:
> "clarify that our theoretical results hold for any lambda (we treat as a constant), not just for small enough lambda"
> Not sure what this means. I did no say the result holds for small enough lambda in my comment.

---

> ### Comment · Reviewer_szE1 · 2024-12-03
> **small beta makes the result further away from a good attempt at "Adam + bilevel"**
>
> Thanks for the explanation.
> There are some papers for analyzing Adam. The arguement seems to be "there exist papers with small beta, so it is fine to use small beta for bilevel". I don't think this is a strong argument.  The authors said " existing convergence analyses of Adam that do not need such choice of beta require other strong assumptions".  I think this comment is misleading, as it implies that the use of tiny beta (i.e. beta1 being very close to 1) is weaker than or equal to other papers' assumption. There is no justification to this claim.

---

> ### Comment · Reviewer_szE1 · 2024-12-03
> **experiments and lambda; summary**
>
> Let me finish the repsonse.
>
> I thank the authors for adding the experiments. I think they are useful.
>
> For lambda:
> 1) I know the original design has a epsilon (or lambda in this paper). The tricky things is: first, the role of lambda in the analysis of Adam can be exaggerated, since it makes the effective stepsize to be easily controllable, avoiding a major challenge of analyzing Adam.
>    While this seem conceptual comment, a more concrete outcome is: if the bounds is based on lambda, then some term in the bound would be expectionally large as lambda --> 0, making the bound far from practice. If someone can have a controlled bound based on lambda, then having beta is fine; but this paper does not control the constant. It avoids a major difficulty of analyzing Adam.
>     I think one should acknowledge the weakeness of the bound directly, instead of saying "in practice there is a lambda" without pointing out the weakeness of the bound due to lambda.
>
> The authors re-empahsized this is for bilevel. Again, I know this is bilevel and there is extra challenge. But one should seek to tackle the true challenge of combining Adam and bilevel optimization.
>
> 2) I also offered a weaker alternative: remove the dependence of eta on lambda.
>  It is not just about lambda >0, but about having stepsiz eta dependent on lambda. This is artificial.
>  The authors did not respond to this comment.
>
> Overall, I think the authors spend too much time aruging that these assumptions appeared in other papers, but **not responding to my comments that they have extra assumptions on the dependence of beta, eta on lambda.**
>
> ---------------------------------------------
> Summary: The paper has a few limitations: i) beta1 almost close to 1, lamda >0 ; ii) and having stepsize eta and beta1 dependent on lambda. These limitations makes the paper a bit far from truly tackling the challenge of "Adam + bilevel".
>
> Furthermore, the authors did not admit these drawbacks in the rebuttal and the paper, making me worried that the paper will be misleading to the community.
>
> Thus I will keep my score.

---

> > ### Author Response · Authors · 2024-12-03
> > **Response**
> >
> > Thank you for your feedback. We can acknowledge these weaknesses explicitly in the revised version.
> >
> > We want to emphasize that the analysis of the upper-level variable is a direct adaptation of ([Li et al., 2023](https://arxiv.org/pdf/2304.13972)), which is already the state-of-the-art analysis of single-level Adam under relaxed assumptions. While there might be a room for improving the analysis of single-level Adam by removing $\lambda$ dependency, this is not our focus and we believe our contribution of bilevel Adam is still an important first step by building upon the results of  ([Li et al., 2023](https://arxiv.org/pdf/2304.13972)).

---

### Official Review · Reviewer_8c3X · 2024-11-09

**Soundness:** 4
**Presentation:** 3
**Contribution:** 3
**Rating:** 6
**Confidence:** 4

**Summary:**

In this paper, the authors propose a variant of Adam called AdamBO, specifically designed for bilevel optimization problems. AdamBO updates the parameters of both the lower-level and upper-level functions simultaneously within a single loop. The authors provide a theoretical convergence guarantee for the hypergradient under an unbounded smoothness condition, which os common in recent theoretical analyses. Additionally, the authors introduce a variance reduction version of AdamBO that achieves improved oracle complexity under stronger conditions.

**Strengths:**

- This paper gives the first study to investigate the convergence of Adam-type algorithms in the context of bilevel optimization problems. The assumptions made for the theoretical convergence analysis are mostly mild and standard. The results presented are convincing and well-supported by the explanations in the proof sketch.

- The writing in the introduction is relatively clear and comprehensive. Readers can easily follow the motivation behind this study, and the proposed algorithms are explained very clearly.

- In addition to the theoretical guarantees for gradient convergence, the authors provide convincing experimental results that support the performance of AdamBO in bilevel optimization problems.

**Weaknesses:**

- The condition in Theorems 4.1 and 5.1 that states $\beta=\tilde \Theta(\epsilon^2)$ is unusual. The default setting for $\beta$ of Adam is 0.1 in Pytorch, and this condition appears to imply poor convergence results for such a default choice of $\beta$.

- Some variables are not provided a definition when introduced, which causes difficulties in understanding. For instance, notations related to stochastic gradients are not explained clearly: $\zeta^{(q,j)}$'s at line 151 seems undefined and a formal definition for this notation needs to be added, particularly regarding the meanings of the superscripts $q$ and $j$.

**Questions:**

See weakness.

---

> ### Author Response · Authors · 2024-11-17
> **Rebuttal by Authors**
>
> *Thank you for taking the time to review our paper. We have addressed each of your concerns below.*
>
> **Q1. The condition in Theorems 4.1 and 5.1 that states $\beta = \widetilde{\Theta}(\epsilon^2)$ is unusual. The default setting for $\beta$ of Adam is 0.1 in Pytorch, and this condition appears to imply poor convergence results for such a default choice of $\beta$.**
>
> **A1.** Thank you for your insightful question. First, our paper considers stochastic bilevel optimization with the upper-level function being unbounded smooth and the main focus of our work is to extend vanilla Adam to solve bilevel optimization problems with theoretical convergence guarantees. For single-level stochastic optimization problem with a general expectation formulation, the analysis by ([Li et al., 2023](https://arxiv.org/pdf/2304.13972)) is the only convergence analysis of Adam under the unbounded smoothness setting (see **Table 1 in Appendix G** on page 68 for a detailed comparison). ([Li et al., 2023](https://arxiv.org/pdf/2304.13972)) requires $\beta = O(\epsilon^2)$ to establish Adam's convergence. Our paper considers the more challenging bilevel optimization problem and similarly requires $\beta = \widetilde{\Theta}(\epsilon^2)$ to be small.
>
> Second, we want to emphasize that small $\beta$ is indeed a common assumption in the literature. There are other works in convergence analysis of Adam that also require $\beta$ be to small (i.e., $\beta$ depends on target gradient norm $\epsilon$), for example, see ([Guo et al., 2021b](https://arxiv.org/pdf/2104.14840), Theorem 6), where they choose $\beta = O(\epsilon^2)$. Moreover, existing convergence analyses of Adam that do not need such choice of $\beta$ require other strong assumptions for the objective function, which is incompatible to our setting. They either rely on the bounded gradient assumption ([De et al., 2018](https://arxiv.org/pdf/1807.06766); [Défossez et al., 2020](https://arxiv.org/pdf/2003.02395)), or they only prove convergence to some neighborhood of stationary points with a constant radius unless assuming the strong growth condition under the finite sum setting ([Zhang et al., 2022](https://arxiv.org/pdf/2208.09632); [Wang et al., 2022](https://arxiv.org/pdf/2208.09900)). Please see our **Table 1 in Appendix G** and the related work section (convergence of Adam) of ([Li et al., 2023](https://arxiv.org/pdf/2304.13972)) for more details.
>
> **Q2. Some variables are not provided a definition when introduced, which causes difficulties in understanding. For instance, notations related to stochastic gradients are not explained clearly: $\zeta^{(q,j)}$'s at line 151 seems undefined and a formal definition for this notation needs to be added, particularly regarding the meanings of the superscripts $q$ and $j$.**
>
> **A2.** Thank you for your suggestion, and we apologize for any ambiguity. In line 151, $\zeta^{(q,j)} \sim \mathcal{D}g$ represents the i.i.d. data samples used in the Neumann series approach to estimate the Hessian inverse. Specifically, $q$ and $j$ (where $0 \leq q \leq Q-1$ and $1 \leq j \leq q$) denote the subscripts of the data samples, corresponding to $\sum_{q=0}^{Q-1}$ and $\prod_{j=1}^{q}$ in the formulation of the Neumann series estimation. Please let us know if there are any other undefined variables.

---

> > ### Author Response · Authors · 2024-11-22
> > **Looking forward to your feedback**
> >
> > Dear Reviewer 8c3X,
> >
> > Thank you for reviewing our paper. We have carefully addressed your concerns regarding the requirement for a small $\beta$ and the definitions of variables.
> >
> > Please let us know if our responses address your concerns accurately. We appreciate your time and efforts and are open to discussing any further questions you may have.
> >
> > Best,
> >
> > Authors

---

### Author Response · Authors · 2024-11-17
**General Response to All Reviewers**

Thank you to all the reviewers for taking the time to review our paper and provide valuable feedback. We have addressed each of your concerns individually and summarize the key changes made during the rebuttal phase below. Major modifications are highlighted in blue in Appendices G, H, and I of the revised paper.

1. We have explained why we need a small $\beta$ (i.e., $\beta$ depends on the target gradient norm $\epsilon$) in theory. First, small $\beta$ is indeed a common assumption for convergence analysis of Adam in the literature ([Li et al., 2023](https://arxiv.org/pdf/2304.13972); [Guo et al., 2021b](https://arxiv.org/pdf/2104.14840)). In addition, the purpose of using small $\beta$ is to control the hypergradient bias caused by the lower-level estimation error in the bilevel optimization setting as in ([Chen et al., 2023](https://arxiv.org/pdf/2306.12067); [Hao et al., 2024](https://arxiv.org/pdf/2401.09587.pdf); [Gong et al., 2024a](https://openreview.net/pdf?id=36rWa8zVkh); [b](https://arxiv.org/pdf/2409.19212)). Second, existing convergence analyses of Adam that do not require this choice of $\beta$ rely on other strong assumptions for the objective function, such as the bounded gradient assumption ([De et al., 2018](https://arxiv.org/pdf/1807.06766); [Défossez et al., 2020](https://arxiv.org/pdf/2003.02395)) or the strong growth condition ([Zhang et al., 2022](https://arxiv.org/pdf/2208.09632); [Wang et al., 2022](https://arxiv.org/pdf/2208.09900)). These assumptions are incompatible with our setting.

2. In Appendix G, we have added two tables: **Table 1** (on page 68) compares Adam-related papers under different settings and assumptions, and **Table 2** (on page 68) compares various bilevel optimization algorithms under the unbounded smoothness setting.

3. In Appendix H, we have conducted additional meta-learning experiments using an 8-layer BERT ([Devlin et al., 2018](https://arxiv.org/pdf/1810.04805)) on the widely-used question classification dataset TREC ([Li & Roth, 2002](https://aclanthology.org/C02-1150.pdf)). The results, shown in **Figure 4** (on page 69), demonstrate that our proposed algorithm AdamBO achieves fast convergence to the best training and testing performance among all baselines.

4. In Appendix H, we have also conducted experiments in **Figure 5** (on page 69) to show the empirical performance of our algorithm is not very sensitive to the choice of $\lambda$. Although the default choice of $\lambda$ is $10^{-8}$ ([Kingma & Ba, 2014](https://arxiv.org/pdf/1412.6980)), increasing it up to $10^{-4}$ only causes minor differences in AUC maximization, and increasing it up to $10^{-3}$ leads to minor changes in hyper-representation performance with BERT ([Devlin et al., 2018](https://arxiv.org/pdf/1810.04805)).

5. In Appendix I, we have provided a more detailed proof sketch for VR-AdamBO (Theorem 5.1).

Please let us know if you have additional concerns, and we are more than willing to address any further questions.

Best,

Authors

---

### Meta-Review · Area_Chair_vAFh · 2024-12-23

**Metareview:**

After careful consideration of the reviews and subsequent author responses, the consensus among the reviewers remains that this work, while presenting an attempt to extend Adam-type methods to bilevel optimization, does not fully address the core challenges of analyzing Adam in bilevel settings.

**Additional Comments On Reviewer Discussion:**

1.	Strong and Unusual Conditions on Parameters:
Multiple reviewers noted that the theoretical results hinge on highly restrictive conditions. In particular, the requirement that the effective step size and the first-order momentum parameter depend on the desired accuracy level \epsilon is seen as a strong and unconventional assumption. This makes the analysis closer to a special-case scenario rather than providing a genuinely general or practical convergence theory for Adam in bilevel problems.
	2.	Dependence on \lambda (or \epsilon in Adam’s denominator):
The analysis heavily relies on introducing an artificial positive constant in the Adam denominator (analogous to \epsilon in the original Adam paper), effectively bounding the step size away from challenging regimes where Adam’s adaptive step size is crucial. This approach sidesteps the key difficulty in analyzing Adam, diminishing the significance of the claimed contribution. Reviewers felt this does not truly capture the complexity of “Adam + bilevel” analysis and leaves open the real challenge of handling Adam’s adaptive nature under less restrictive assumptions.
	3.	Small \beta Assumptions:
The theoretical results require the momentum parameter \beta_1 to be extremely close to 1 (i.e., \beta = 1 - \beta_1 very small). While not entirely unprecedented, relying on such conditions places the results at odds with standard practical choices and does not convincingly generalize to commonly used Adam configurations.
	4.	Limited Empirical Scope and Weaknesses in Experiments:
Although the authors added more experiments (e.g., with an 8-layer BERT) during the rebuttal period, the reviewers remain unconvinced that these results suffice to compensate for the restrictive theoretical assumptions. The experiments do not conclusively demonstrate the necessity or broad utility of the proposed theoretical framework. Questions about the relevance of the chosen tasks and the limited model scales also surfaced.
	5.	Inadequate Addressing of Reviewers’ Core Concerns:
While the authors provided extensive rebuttals, some reviewers felt their core concerns—particularly regarding the strength and unusual nature of the assumptions—were not satisfactorily resolved. The authors’ attempts to justify these assumptions by pointing to other works still did not alleviate the perception that the analysis falls short of a practical and robust theoretical foundation for Adam in bilevel optimization.

---

### Decision · Program_Chairs · 2025-01-22

Reject